# Implicit Regularization for Tubal Tensor Factorizations via Gradient Descent

Santhosh Karnik [* 1]  Anna Veselovska [* 2 3]  Mark Iwen [4 5]  Felix Krahmer [2 3]

## Abstract

We provide a rigorous analysis of implicit regularization in an overparametrized tensor factorization problem beyond the lazy training regime. For matrix factorization problems, this phenomenon has been studied in a number of works. A particular challenge has been to design universal initialization strategies which provably lead to implicit regularization in gradient-descent methods. At the same time, it has been argued by (Cohen et al., 2016) that more general classes of neural networks can be captured by considering tensor factorizations. However, in the tensor case, implicit regularization has only been rigorously established for gradient flow or in the lazy training regime. In this paper, we prove the first tensor result of its kind for gradient descent rather than gradient flow. We focus on the tubal tensor product and the associated notion of low tubal rank, encouraged by the relevance of this model for image data. We establish that gradient descent in an overparametrized tensor factorization model with a small random initialization exhibits an implicit bias towards solutions of low tubal rank. Our theoretical findings are illustrated in an extensive set of numerical simulations show-casing the dynamics predicted by our theory as well as the crucial role of using a small random initialization.

## 1. Introduction

Analyzing implicit regularization during Neural Network (NN) training is considered crucial for understanding why

---
[*]Equal contribution [1]Department of Mathematics, Northeastern University, Boston, USA [2]Department of Mathematics and Munich Data Science Institute, Technical University of Munich, Munich, Germany [3]Munich Center for Machine Learning, Munich, Germany [4]Department of Mathematics, Michigan State University, East Lansing, USA [5]Department of Computational Mathematics Science and Engineering, Michigan State University, East Lansing, USA. Correspondence to: Anna Veselovska <anna.veselovska@tum.de>.

*Proceedings of the 42$^{nd}$ International Conference on Machine Learning*, Vancouver, Canada. PMLR 267, 2025. Copyright 2025 by the author(s).

overparametrization can give rise to superior generalization capability and lead to strong overall NN performance. Consequently, there has been a recent surge in research aimed at explaining how gradient-based methods interact with overparameterized models under nonconvex losses (see, e.g., (Ma et al., 2018; Ling & Strohmer, 2019)). Notably, recent empirical and theoretical studies have suggested that gradient-based methods with small random initializations exhibit a bias towards low-rank solutions in a variety of models.

For matrix factorization models which represent linear neural networks, a rigorous analysis of implicit bias is available for both gradient descent (Gunasekar et al., 2018; Stöger & Soltanolkotabi, 2021) and gradient flow (its asymptotic limit for small step size) (Bah et al., 2022; Chou et al., 2024). In contrast, for neural networks with nonlinear activation, there has been a good deal of work done showing that fully connected layers can be represented by, e.g., tensor train factorizations in (Novikov et al., 2015; Razin et al., 2021). As a consequence, it has been argued that tensor factorizations should be considered instead of matrix factorizations (see, e.g., (Cohen et al., 2016)). For tensor factorization models, however, results predating 2024 were only available for the asymptotic regime, i.e., gradient flow. This is perhaps due to the many additional complications in the tensor setting beyond those in the matrix setting including, e.g, that there are many different valid notions of tensor rank, each of which motivates its own equally valid class of tensor factorizations. For gradient descent applied to the tensor recovery problem, only a very recent partial analysis by (Liu et al., 2024) currently exists for the tubal factorization model. This analysis requires that the initialization already well approximates the solution, only after which the convergence of gradient descent toward a low tubal-rank solution is shown. Herein we also focus on the tubal factorization, but establish the corresponding implicit regularization result without needing such a strong initialization assumption.

Our work is motivated by recent research showing that the way neural networks are trained, especially with gradient descent, can lead to solutions with useful structure, even without adding explicit regularization terms. This phenomenon, known as implicit regularization, has been studied in contexts such as sparse recovery (Vaskevicius et al., 2019) and low-rank matrix completion (Li et al., 2020), where specific

network architectures are designed to encourage certain types of structure in the solutions. However, for tensor recovery problems, most existing work either focuses only on gradient flow or provides only partial analysis. To the best of our knowledge, our paper is the first to analyze implicit bias under gradient descent with small random initialization for a tensor recovery problem. We focus on the tubal rank model, which is particularly relevant for applications like video representation. This opens the door to a broader investigation into how implicit regularization can be used for structured tensor recovery, how network architectures influence this bias, and what conditions ensure convergence. We see this work as a starting point for a larger line of research on implicit regularization in tensor problems.

**Related work:** In deep learning it is common to use more network parameters than training points. In such overparameterized scenarios there are usually many networks that achieve zero training error so that the training algorithm effectively imposes an implicit regularization (bias) on the solution it computes. In practice, training networks with gradient descent is both common and tends to favor solutions that generalize well, offering the exploration of how gradient descent implicitly regularlizes in overparameterized regimes as one avenue for better understanding the success of deep learning more widely. As a result, a lot of recent work has been focussed on understanding the implicit regularization phenomena of gradient descent in multiple settings. The first theoretical works in this direction (Gunasekar et al., 2017; 2018; Geyer et al., 2020; Arora et al., 2019; Soudry et al., 2018) concentrated on training linear networks and suggested that during training (stochastic) gradient descent implicitly converges to a linear network (i.e., a linear function described by a matrix) that's low rank. Motivated by specific deep learning tasks, multiple works also investigated implicit bias phenomena in the special cases of sparse vector and low-rank matrix recovery from underdetermined measurements via an overparameterized square loss functional, where the vectors and matrices to be reconstructed were deeply factorized into several vector/matrix factors. In this setting, these works then showed that the dynamics of vanilla gradient descent are biased towards sparse/low-rank solutions, respectively (Chou et al., 2024; 2023; Li et al., 2022; Kolb et al., 2023).

In the realm of optimization, a substantial body of work has also emerged that provides guarantees for gradient descent's convergence in the nonconvex setting for different problems such as phase retrieval, matrix completion, and blind deconvolution. Broadly, these findings can be categorized into two main approaches: smart initialization coupled with local convergence (demonstrating, e.g., local convergence of descent techniques starting from carefully designed spectral initializations) (Ma et al., 2018; Tu et al., 2016; Ling &

Strohmer, 2019; Candes et al., 2015); and landscape analysis paired with saddle-escaping algorithms which show, e.g., that all local minima are global and that saddle points exhibit strict negative curvature so that (stochastic) gradient-based methods can effectively escape saddles and ensure convergence to global minimizers (Jin et al., 2017; Ge et al., 2015; Raginsky et al., 2017).

Notably, several studies (Woodworth et al., 2020; Ghorbani et al., 2020) have highlighted the importance of the scale of the training initialization for the generalization and test performance of modern machine learning architectures. In fact, a small random initialization followed by (stochastic) gradient descent is arguably the most widely used training algorithm in contemporary machine learning. And, stronger generalization performance is typically observed with smaller-scale initializations. Implicit bias for low-rank matrix recovery with small random initializations has been extensively studied in this setting as a result by, e.g., (Stöger & Soltanolkotabi, 2021; Soltanolkotabi et al., 2023; Wind, 2023; Kim & Chung, 2024). These studies have shown that a small random Gaussian initialization behaves similarly to a spectral initialization in overparameterized settings. Furthermore, they have shown that gradient descent algorithms with this initialization tend to converge towards low-rank solutions (i.e., that they demonstrate an implicit regularization towards low-rank solutions).

Recently, numerous connections between tensor decompositions and training neural networks have also been established by, e.g., (Novikov et al., 2015; Razin et al., 2021; 2022). These studies argue that low-rank tensor factorization helps explain implicit regularization in deep learning, as well as how properties of real-world data translate this regularization to generalization. Similar to how matrix factorization can be viewed as a linear neural network (i.e., a fully connected network with linear activation), tensor factorizations correspond to a specific type of shallow (depth-two) nonlinear convolutional neural network (Cohen et al., 2016; Razin et al., 2021). Additionally, (Novikov et al., 2015) demonstrated that the dense weight matrices of fully connected layers can be converted to tensor trains while preserving the layer's expressive power. These findings have positioned low-rank tensor factorizations as theoretical surrogates for various neural network learning settings, thereby enhancing our understanding of implicit regularization and overparameterization, and so further motivating investigation in this area.

Since no unique definition of tensor rank is available, related literature concerning implicit bias has naturally split with respect to the notion of tensor rank being considered: CP-rank, Tucker-rank, and tubal-rank, in analogy to the analysis of algorithms specifically designed for tensor recovery and completion by, e.g., (Zhang et al., 2019; Hou et al., 2021;

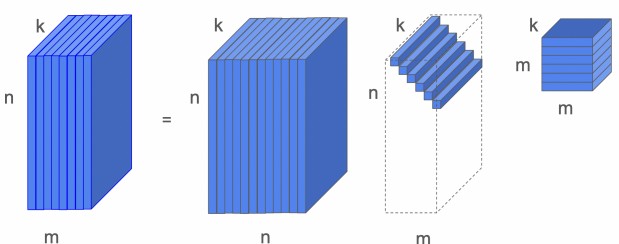

Figure 1: A low tubal-rank factorization of a three-dimensional tensor. Using the (reduced) tubal-SVD, each three-dimensional tensor $\mathcal{T} \in \mathbb{R}^{n \times m \times k}$ can be decomposed into a tubal product of three tensors $\mathcal{T} = \mathcal{V} * \mathbf{\Sigma} * \mathcal{W}^\top$ with $\mathcal{V} \in \mathbb{R}^{n \times n \times k}$, $\mathcal{W} \in \mathbb{R}^{m \times m \times k}$ and the frontal slice diagonal tensor $\mathbf{\Sigma} \in \mathbb{R}^{n \times m \times k}$. Here, the tubal rank of a tensor is the number of non-zero singular tubes in $\mathbf{\Sigma} \in \mathbb{R}^{n \times m \times k}$. For example, in the figure, the tubal rank of the tensor is equal to six.

Kong et al., 2018; Ahmed et al., 2020; Liu et al., 2019; 2020; Haselby et al., 2024). For the CP-tensor factorization, several results are available for gradient-based methods (Wang et al., 2020; Ge & Ma, 2017). The first theoretical analysis of implicit regularization towards low tensor rank under arbitrarily small initialization was provided considering gradient flow in (Razin et al., 2021). In (Ge et al., 2015), it has been shown for the orthogonal tensor decomposition problem a simple variant of the stochastic gradient algorithm is able to leverage a low-rank structure from an arbitrary starting point. In addition, (Wang et al., 2020) shows that using gradient descent on an over-parametrized objective for the CP-rank tensor decomposition problem one could go beyond the lazy training regime and utilize certain low-rank structures.

Perhaps most closely related to this paper, very recently (Liu et al., 2024) analyzed the convergence of factorized gradient descent for the low-tubal-rank sensing problem, showing that with carefully designed spectral initialization the gradient iterates converge to a low-tubal rank tensor. Although the authors in (Liu et al., 2024) allow for over-parametrization, they argue the minimal recovery error can be achieved when knowing the true rank, thereby leaving questions concerning the advantages of overparametrization and small random initializations open.

**Our contribution:** Motivated by connections between tensor rank and non-linear neural network representations, herein we study the implicit regularization phenomenon for low tubal-rank tensor recovery. Namely, our objective is to analyze the recovery process of a tensor with a low tubal-rank factorization (Kilmer & Martin, 2011) (see Fig 1) from a limited number of random linear measurements. More

specifically, we consider tensors of the form $\mathcal{X} * \mathcal{X}^\top$ and employ a non-convex method based on the tensor factorization, minimizing the loss function using gradient descent with a small random initialization. To the best of our knowledge, we are the first to investigate the implicit bias phenomenon for gradient descent with a small random initialization applied to a tensor factorization. Namely, we demonstrate that, irrespective of the degree of overparameterization, vanilla gradient descent with a small random initialization applied to a tubal tensor factorization will consistently converge to a low tubal-rank solution.

Inspired by recent results for the low-rank matrix sensing problem by (Stöger & Soltanolkotabi, 2021), we establish that gradient descent iterates with small random initializations can be closely approximated by power method iterations in (Gleich et al., 2013; Kilmer et al., 2013) modulo normalization, and deduce that after sufficient time the iterates approach a commonly used spectral initialization from the tubal-rank literature in (Liu et al., 2024). Along the way we must also overcome, e.g., a challenging intersection between the tensor slices during each gradient descent iterate which forces a non-trivial convergence analysis.

**Organization:** In Section 2, we define our notation and present a few basic facts regarding tubal tensors. In Section 3, we state our problem and our main result. In Section 4, we outline the steps of the proof in order to provide intuition. In Section 5, we show numerical experiments which demonstrate our theoretical findings. We conclude the paper in Section 6. The proof of our main result is broken up into several lemmas, which are stated and proven in the appendix.

## 2. Notation and Preliminaries

Every tensor in this paper will be an order-3 tensor whose third mode is length $k$. For such a tensor $\mathcal{T} \in \mathbb{R}^{m \times n \times k}$, we define a block-diagonal Fourier domain representation by

$$\overline{\mathcal{T}} = \text{blockdiag}(\overline{\mathcal{T}}^{(1)}, \dots, \overline{\mathcal{T}}^{(k)}) \in \mathbb{C}^{mk \times nk}$$

where the $j$-th block $\overline{\mathcal{T}}^{(j)} \in \mathbb{C}^{m \times n}$ is defined by $\overline{\mathcal{T}}^{(j)}(i, i') = \sum_{j'=1}^{k} \mathcal{T}(i, i', j') e^{-\sqrt{-1} 2\pi(j-1)(j'-1)/k}$. In other words, we take the FFT of each tube, and then arrange the resulting frontal slices into a block-diagonal matrix.

The tubal product (or t-product) of two tubal tensors $\mathcal{A} \in \mathbb{R}^{m \times q \times k}$ and $\mathcal{B} \in \mathbb{R}^{q \times n \times k}$ is a tubal tensor $\mathcal{A} * \mathcal{B} \in \mathbb{R}^{m \times n \times k}$ whose tubes are given by

$$(\mathcal{A} * \mathcal{B})(i, i', :) = \sum_{p=1}^{q} \mathcal{A}(i, p, :) * \mathcal{B}(p, i', :).$$

Here, $*$ denotes the circular convolution operation, i.e., $(\boldsymbol{x} *$

$\boldsymbol{y})_i = \sum_{j=1}^{k} x_j y_{i-j \pmod{k}}$. One can check that $\overline{\mathcal{A} * \mathcal{B}} = \overline{\mathcal{A}}\,\overline{\mathcal{B}}$.

For any tubal tensor $\mathcal{T} \in \mathbb{R}^{m \times n \times k}$, its tubal transpose $\mathcal{T}^\top \in \mathbb{R}^{n \times m \times k}$ is given by $(\mathcal{T}^\top)(i, i', 1) = \mathcal{T}(i', i, 1)$ and $(\mathcal{T}^\top)(i, i', j) = \mathcal{T}(i', i, k + 2 - j)$ for $j = 2, \dots, k$, i.e., we take the transpose of each face, and then reverse the order of frontal slices $j = 2, \dots, k$. This ensures that $\overline{\mathcal{T}^\top} = \overline{\mathcal{T}}^\top$.

For any $n$, the $n \times n \times k$ identity tensor $\mathcal{I} \in \mathbb{R}^{n \times n \times k}$ is defined by $\mathcal{I}(:, :, 1) = I_{n \times n}$ (identity matrix), and $\mathcal{I}(:, :, j) = 0_{n \times n}$ (zero matrix). An orthogonal tensor $\mathcal{Q} \in \mathbb{R}^{n \times n \times k}$ satisfies $\mathcal{Q} * \mathcal{Q}^\top = \mathcal{Q}^\top * \mathcal{Q} = \mathcal{I}$. An orthonormal tensor $\mathcal{W} \in \mathbb{R}^{m \times n \times k}$ with $m \geq n$ satisfies $\mathcal{W}^\top * \mathcal{W} = \mathcal{I}$.

The tubal-SVD (Kilmer & Martin, 2011) (or t-SVD) of a tubal tensor $\mathcal{T} \in \mathbb{R}^{m \times n \times k}$ is a factorization of the form

$$\mathcal{T} = \mathcal{U} * \mathbf{\Sigma} * \mathcal{V}^\top \qquad (2.1)$$

where $\mathcal{U} \in \mathbb{R}^{m \times m \times k}$ and $\mathcal{V} \in \mathbb{R}^{n \times n \times k}$ are orthogonal, and each frontal slice of $\mathbf{\Sigma} \in \mathbb{R}^{m \times n \times k}$ is diagonal. The t-SVD of a tensor $\mathcal{T} \in \mathbb{R}^{m \times n \times k}$ can be computed as follows: (1) compute the FFT of each tube of $\mathcal{T}$ to get the frontal slices $\overline{\mathcal{T}}^{(j)}, j = 1, \dots, k$, (2) compute the SVD of each resulting frontal slice $\overline{\mathcal{T}}^{(j)} = \overline{U}^{(j)} \overline{\Sigma}^{(j)} \overline{V}^{(j)\top}$, (3) concatenate the matrices $\{\overline{U}^{(j)}\}_{j=1}^{k}$ into a tubal tensor $\widetilde{\mathcal{U}} \in \mathbb{C}^{m \times m \times k}$ and take the inverse FFT along mode-3 to obtain $\mathcal{U} \in \mathbb{R}^{m \times m \times k}$ (and similarly to obtain $\mathbf{\Sigma} \in \mathbb{R}^{m \times n \times k}$ and $\mathcal{V} \in \mathbb{R}^{n \times n \times k}$). The tubal rank of a tensor $\mathcal{T} \in \mathbb{R}^{m \times n \times k}$ is the number of non-zero diagonal tubes in the $\mathbf{\Sigma}$ tensor of its t-SVD, i.e., $\mathrm{rank}(\mathcal{T}) = \#\{i : \mathbf{\Sigma}(i, i, :) \neq 0\}$. For an illustration of the t-SVD decomposition, see Figure 1. We also define the condition number $\kappa(\mathcal{T})$ of the tubal tensor $\mathcal{T} \in \mathbb{R}^{m \times n \times k}$ by

$$\kappa(\mathcal{T}) := \frac{\sigma_1(\overline{\mathcal{T}})}{\sigma_{\min\{m,n\}k}(\overline{\mathcal{T}})}.$$

Finally, for tubal tensors $\mathcal{T} \in \mathbb{R}^{m \times n \times k}$ we define the tensor spectral norm $\|\mathcal{T}\| := \|\overline{\mathcal{T}}\|$ and the tensor nuclear norm $\|\mathcal{T}\|_* := \|\overline{\mathcal{T}}\|_*$ as the spectral and nuclear norm respectively of the block-diagonal Fourier domain representation $\overline{\mathcal{T}}$, and the tensor Frobenius norm $\|\mathcal{T}\|_F^2 := \sum_{i=1}^{m} \sum_{j=1}^{n} \sum_{\ell=1}^{k} \mathcal{T}(i, j, \ell)^2 = \frac{1}{k} \|\overline{\mathcal{T}}\|_F^2$ as a scaled version of the Frobenius norm of the block-diagonal Fourier domain representation $\overline{\mathcal{T}}$.

## 3. Main Results

**Problem Formulation** Let $\mathcal{X} \in \mathbb{R}^{n \times r \times k}$ have tubal rank $r \leq n$ so that $\mathcal{X} * \mathcal{X}^\top \in S_+^{n \times n \times k}$ is a tubal positive semidefinite tensor with tubal rank $r$. Let $\kappa = \kappa(\mathcal{X})$ be the condition number of $\mathcal{X}$. Suppose we observe $m$ linear

measurements of $\mathcal{X} * \mathcal{X}^\top$, that is

$$y_i = \left\langle \mathcal{A}_i, \mathcal{X} * \mathcal{X}^\top \right\rangle \quad \text{for} \quad i = 1, \dots, m \qquad (3.1)$$

where each $\mathcal{A}_i \in S^{n \times n \times k}$ is a tubal-symmetric tensor. We can write this compactly as $\boldsymbol{y} = \mathcal{A}(\mathcal{X} * \mathcal{X}^\top)$ where $\mathcal{A} : S^{n \times n \times k} \to \mathbb{R}^m$ is the linear measurement operator. We aim to recover $\mathcal{X} * \mathcal{X}^\top$ from our measurements $\boldsymbol{y}$ by using gradient descent to learn an overparameterized factorization. Specifically, we fix an $R \geq r$ and try to find a $\mathcal{U} \in \mathbb{R}^{n \times R \times k}$ such that $\mathcal{U} * \mathcal{U}^\top = \mathcal{X} * \mathcal{X}^\top$ by using gradient descent to minimize the loss function

$$\ell(\mathcal{U}) := \left\| \mathcal{A}\left(\mathcal{U} * \mathcal{U}^\top\right) - \boldsymbol{y} \right\|_2^2 \qquad (3.2)$$

$$= \sum_{i=1}^{m} \left(\left\langle \mathcal{A}_i, \mathcal{U} * \mathcal{U}^\top \right\rangle - y_i\right)^2. \qquad (3.3)$$

We will start with a small random initialization $\mathcal{U}_0 \in \mathbb{R}^{n \times R \times k}$ where each entry is i.i.d. $\mathcal{N}(0, \frac{\alpha^2}{R})$ for some small $\alpha > 0$. Then, the gradient descent iterations are given by

$$\begin{aligned}
\mathcal{U}_{t+1} &= \mathcal{U}_t - \mu \nabla \ell(\mathcal{U}_t) \\
&= \mathcal{U}_t + \mu \mathcal{A}^* \left[\boldsymbol{y} - \mathcal{A}\left(\mathcal{U}_t * \mathcal{U}_t^\top\right)\right] * \mathcal{U}_t \\
&= \left[\mathcal{I} + \mu(\mathcal{A}^* \mathcal{A})\left(\mathcal{X} * \mathcal{X}^\top - \mathcal{U}_t * \mathcal{U}_t^\top\right)\right] * \mathcal{U}_t
\end{aligned}$$
$$(3.4)$$

for some suitably small stepsize $\mu > 0$. Here $\mathcal{A}^* : \mathbb{R}^m \to S^{n \times n \times k}$ denotes the adjoint of $\mathcal{A}$ which is given by $\mathcal{A}^* \boldsymbol{z} = \sum_{i=1}^{m} z_i \mathcal{A}_i$.

Moreover, we say that a measurement operator $\mathcal{A} : S^{n \times n \times k} \to \mathbb{R}^m$ satisfies the Restricted Isometry Property (RIP) of rank-$r$ with constant $\delta > 0$ (abbreviated RIP($r, \delta$)), if we have

$$(1 - \delta)\|\mathcal{Z}\|_F^2 \leq \|\mathcal{A}(\mathcal{Z})\|_2^2 \leq (1 + \delta)\|\mathcal{Z}\|_F^2,$$

for all $\mathcal{Z} \in S^{n \times n \times k}$ with tubal-rank $\leq r$. We note that an RIP condition is a standard condition in the literature, and is used in similar works such as (Li et al., 2018; Stöger & Soltanolkotabi, 2021). This condition is necessary to ensure that there is only one low tubal rank tensor for which the loss function is zero, and that this tensor could be recovered stably in the presence of noise.

**Results** We have analyzed the convergence process of the gradient descent iterates (3.4) in the scenario of small random initialization and overparametrization. Namely, with the ground truth tensor $\mathcal{X} \in \mathbb{R}^{n \times r \times k}$, we assume the initialization $\mathcal{U}_0 \in \mathbb{R}^{n \times R \times k}$ is such that each entry is i.i.d. $\mathcal{N}(0, \frac{\alpha^2}{R})$ with small scaling parameter $\alpha > 0$ and the second dimension $R$ exceeding three timesthe ground truth dimension $r$. Below, we present the direct results of our analysis.

**Theorem 3.1.** *Suppose we have $m$ linear measurements $y = \mathcal{A}(\mathcal{X} * \mathcal{X}^\top)$ of a tubal positive semidefinite tensor $\mathcal{X} * \mathcal{X}^\top \in S_+^{n \times n \times k}$ where $\mathcal{X} \in \mathbb{R}^{n \times r \times k}$ has tubal rank $r \leq n$. We assume $\mathcal{A}$ satisfies $RIP(2r+1, \delta)$ with $\delta \leq c\kappa^{-4} r^{-1/2}$. Suppose we fit a model $\mathcal{X} * \mathcal{X}^\top = \mathcal{U} * \mathcal{U}^\top$ where $\mathcal{U} \in \mathbb{R}^{n \times R \times k}$ with $R \geq 3r$ and obtain $\mathcal{U}$ by running the gradient descent iterations*

$$\mathcal{U}_{t+1} = \left[ \mathcal{I} + \mu(\mathcal{A}^*\mathcal{A})\left( \mathcal{X} * \mathcal{X}^\top - \mathcal{U}_t * \mathcal{U}_t^\top \right) \right] * \mathcal{U}_t$$

*with a stepsize $\mu \leq c\sqrt{k}\kappa^{-4}\|\mathcal{X}\|^2$ starting from the initialization $\mathcal{U}_0 \in \mathbb{R}^{n \times R \times k}$ where each entry is i.i.d. $\mathcal{N}(0, \frac{\alpha^2}{R})$. Then, if the scale of the initialization satisfies*

$$\alpha \lesssim \frac{\sigma_{min}(\mathcal{X})}{\kappa^2 \min\{n, R\}\sqrt{k}} \left( \frac{C_2 \kappa^2 \sqrt{n}}{\sqrt{\min\{n, R\}}} \right)^{-16\kappa^2},$$

*then after*

$$\widehat{t} \lesssim \frac{1}{\mu\sigma_{min}(\mathcal{X})^2} \ln\left( \frac{C_1 \kappa n}{\min\{n, R\}} \min\left\{ 1, \frac{\kappa r}{k(\min\{n,R\}-r)} \right\} \frac{\|\mathcal{X}\|}{k\alpha} \right)$$

*iterations, we have that*

$$\frac{\|\mathcal{U}_{\widehat{t}} * \mathcal{U}_{\widehat{t}}^\top - \mathcal{X} * \mathcal{X}^\top\|_F^2}{\|\mathcal{X}\|^2} \lesssim$$

$$k^{\frac{61}{32}} r^{\frac{1}{8}} \kappa^{\frac{-3}{16}} (\min\{n, R\} - r)^{\frac{3}{8}} \left[ \frac{C_2 \kappa^2 \sqrt{n}}{\sqrt{\min\{n,R\}}} \right]^{21\kappa^2} \left[ \frac{\alpha}{\|\mathcal{X}\|} \right]^{\frac{21}{16}}$$

*holds with probability at least $1 - Cke^{-\tilde{c}R}$. Here, $c, \tilde{c}, C, C_1, C_2 > 0$ are fixed numerical constants.*

Intuitively, this means that if the initialization is sufficiently small, gradient descent will approximately recover the low tubal rank tensor $\mathcal{X} * \mathcal{X}^\top$ after $\widehat{t}$ iterations. Note that the reconstruction error can be made arbitrarily small by making the size of the random initialization $\alpha$ arbitrarily small. This comes at the expense of requiring more iterations. However, this impact is mild as the number of iterations grows only logarithmically with respect to $\alpha$.

Although the above theorem holds for any $R \geq 3r$, it is perhaps most interesting in the case where $R \geq n$ as then every $n \times n \times k$ tubal positive semidefinite tensor can be expressed as $\mathcal{U} * \mathcal{U}^\top$ for some $\mathcal{U} \in \mathbb{R}^{n \times R \times k}$. Hence, the learner model does not assume that the ground truth tensor has low tubal rank, yet gradient descent is able to recover the ground truth tensor instead of any of the infinitely many high tubal rank tensors whose measurements match that of the ground truth tensor.

We note that (Zhang et al., 2019) shows that a random sub-Gaussian measurement operator $\mathcal{A} : \mathbb{R}^{n \times n \times k} \to \mathbb{R}^m$ will satisfy the RIP for tubal rank-$r$ tensors with RIP constant $\delta$ with high probability if $m \geq O(rnk/\delta^2)$. To obtain an RIP

constant of $\delta = O(\kappa^{-4} r^{-1/2})$, one needs $m \geq O(\kappa^8 r^2 nk)$ random sub-Gaussian measurements.

Additionally, we acknowledge that the parameter dependence in Theorem 3.1 may initially seem unfamiliar. However, it aligns well with intuition and prior work: when the tensor is ill-conditioned – i.e., possesses a small tubal singular value – gradient descent without regularization naturally struggles to recover the rank-one component unless the initialization is sufficiently small. While our bound exhibits exponential dependence on the condition number, this is consistent with known results in the matrix setting (e.g., see Lemma 8.6 in (Stöger & Soltanolkotabi, 2021)). Although the necessity of exponential dependence remains an open question, it presents a compelling direction for future research. Moreover, our numerical experiments (see Figure 4) support a polynomial relationship between the test error and the initialization parameter $\alpha$, and while the empirical degree may differ slightly, our theoretical exponent $\frac{21}{16}$ appears to closely approximate the observed behavior.

## 4. Proof Outline

In this section, we turn our attention to giving an overview of the key ideas of the proof.

In our analysis, we demonstrate that the trajectory of gradient descent iterations can be approximately divided into two distinct stages: (I) a spectral stage and (II) a convergence stage described below.

*(I) The spectral stage.* In the spectral stage, where we show that the gradient descent starting from random initialization behaves similarly to spectral initialization, enabling us to prove that by the end of this stage, the column spaces of the tensor iterates $\mathcal{U}_t$ (3.4) and the ground truth matrix $\mathcal{X}$ are sufficiently aligned. Namely, we show that the first few iterations of the gradient descent algorithm $\mathcal{U}_t$ can be approximated by the iteration of the tensor power method modulo normalization (see, e.g.(Gleich et al., 2013)) defined as

$$\widetilde{\mathcal{U}}_t = \left( \mathcal{I} + \mu\mathcal{A}^*\mathcal{A}(\mathcal{X} * \mathcal{X}^\top) \right)^{*t} * \mathcal{U}_0 \in \mathbb{R}^{n \times R \times k}.$$

We call this part of the evolution of the gradient descent iteration the "spectral stage" since, due to its similarity to the power method, at the end of this stage the iterates $\mathcal{U}_t$ will be closely aligned with the classical t-SVD spectral initialization of (Liu et al., 2024).

*(II) The convergence stage.* In the convergence stage, the gradient iterates converge approximately to the underlying low tubal-rank tensor $\mathcal{X} * \mathcal{X}^\top$ at a geometric rate until reaching a certain error floor which is dependent on the initialization scale.

The cornerstone of the analysis of this stage is the de-

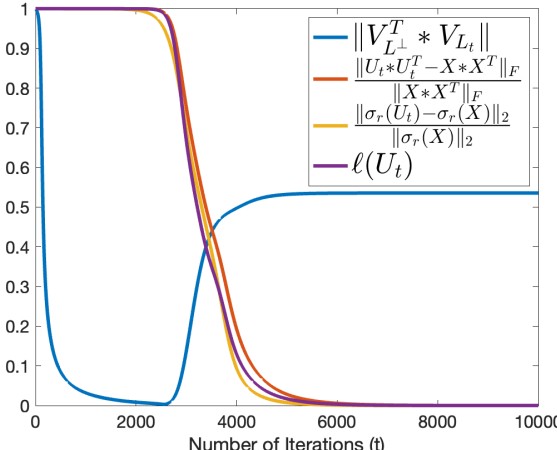

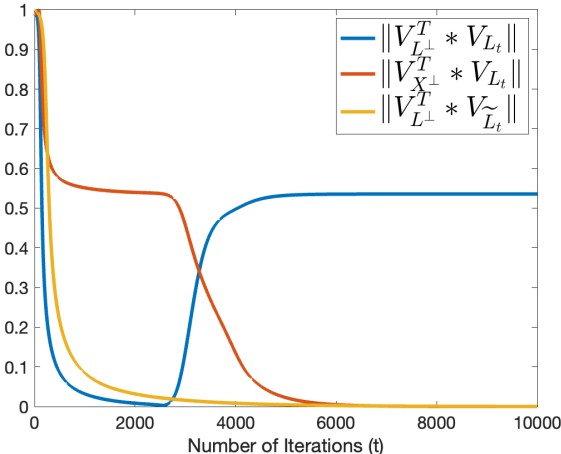

Figure 2: Illustration of (top figure) the two stages of gradient descent algorithm: the spectral alignment stage for $1 \leq t \lesssim 3000$ and the convergence stage $3000 \lesssim t$ and (bottom figure) more details on the alignment phase for the gradient descent progress. In the ground truth tensor $\mathcal{X} \in \mathbb{R}^{n \times r \times k}$, we set $n = 10, k = 4, r = 3$.

composition of the tensor gradient iterates $\mathcal{U}_t$ into two components, the so-called "signal" and "noise" terms. This is done by adapting similar decomposition methods used in recent works analyzing implicit bias phenomenon for gradient descent in the matrix setting (see (Stöger & Soltanolkotabi, 2021; Li et al., 2018)) to our tensor setting. Accordingly, let the tensor-column subspace of the ground truth tensor $\mathcal{X} \in \mathbb{R}^{n \times r \times k}$ be denoted by $\mathcal{V}_{\mathcal{X}}$ with the corresponding basis $\mathcal{V}_{\mathcal{X}} \in \mathbb{R}^{n \times r \times k}$. Consider the tensor $\mathcal{V}_{\mathcal{X}} * \mathcal{U}_t \in \mathbb{R}^{r \times R \times k}$ with its t-SVD decomposition $\mathcal{V}_{\mathcal{X}} * \mathcal{U}_t = \mathcal{V}_t * \Sigma_t * \mathcal{W}_t^\top$. For $\mathcal{W}_t \in \mathbb{R}^{R \times r \times k}$, we

denote by $\mathcal{W}_{t,\perp} \in \mathbb{R}^{R \times (n-r) \times k}$ a tensor whose tensor-column subspace is orthogonal to those of $\mathcal{W}_t$, that is $\|\mathcal{W}_{t,\perp}^\top * \mathcal{W}_t\| = 0$ and its projection operator $\mathcal{P}_{\mathcal{W}_{t,\perp}}$ is defined as $\mathcal{P}_{\mathcal{W}_{t,\perp}} = \mathcal{W}_{t,\perp} * \mathcal{W}_{t,\perp}^\top = \mathcal{I} - \mathcal{W}_t * \mathcal{W}_t^\top$.

We then decompose the gradient descent iterates (3.4) as follows

$$\mathcal{U}_t = \mathcal{U}_t * \mathcal{W}_t * \mathcal{W}_t^\top + \mathcal{U}_t * \mathcal{W}_{t,\perp} * \mathcal{W}_{t,\perp}^\top \quad (4.1)$$

referring to the tensors $\mathcal{U}_t * \mathcal{W}_t * \mathcal{W}_t^\top$ as the signal term of the gradient descent iterates, and to the tensors $\mathcal{U}_t * \mathcal{W}_{t,\perp} * \mathcal{W}_{t,\perp}^\top$ as the noise term. The advantage of such a decomposition is that the tensor-column space of the noise term $\mathcal{U}_t * \mathcal{W}_{t,\perp} * \mathcal{W}_{t,\perp}^\top$ is orthogonal to the tensor-column subspace of the ground truth $\mathcal{X}$ allowing for a rigorous analysis of the convergence process of the two components separately.

At the convergence stage, we show that symmetric tensor $\mathcal{U}_t * \mathcal{W}_t * \mathcal{W}_t^\top * \mathcal{U}_t^\top$ built from the signal term converges towards the ground truth tensor $\mathcal{X} * \mathcal{X}^\top$, whereas the spectral norm of the noise term $\|\mathcal{U}_t * \mathcal{W}_{t,\perp}\|$, stays small.

**Additional challenges in the tensor setting vs. matrix setting** When coming from the matrix case to the tensor setting com, there are several important differences and challenges, which need to be carefully considered and are described below.

- In contrast to the matrix case, the range and kernel of a third-order tubal tensor can include overlapping generator elements (we refrain from using the term basis, in the sense that knowledge of the multirank and complimentary tubal scalar of a tensor must be included to describe the range). Namely, if in the t-SVD (2.1) of a symmetric tensor $\mathcal{X}$ the tensor $\Sigma$ contains $q$ non-invertible tubes – tubes that have zero elements in the Fourier domain –, then there are $q$ common generators for the range and the kernel of $\mathcal{X}$, please see (Kilmer et al., 2013) for more details. With this phenomenon, the decomposition (C.1) of the gradient iterates into signal and noise term is not available for non-invertible tubes, which is why we need to work with a more intricate notion of condition number.

- As stated in (Gleich et al., 2013), running the power method for tubal tensors of dimensions $n \times n \times k$ is equivalent to running in parallel $k$ independent matrix power methods in Fourier domain. However, running gradient descent in the tubal tensor setting is not equivalent to running $k$ gradient descent algorithms independently in Fourier space. This can be easily seen when transforming the measurement operator part of the gradient descent iterates.

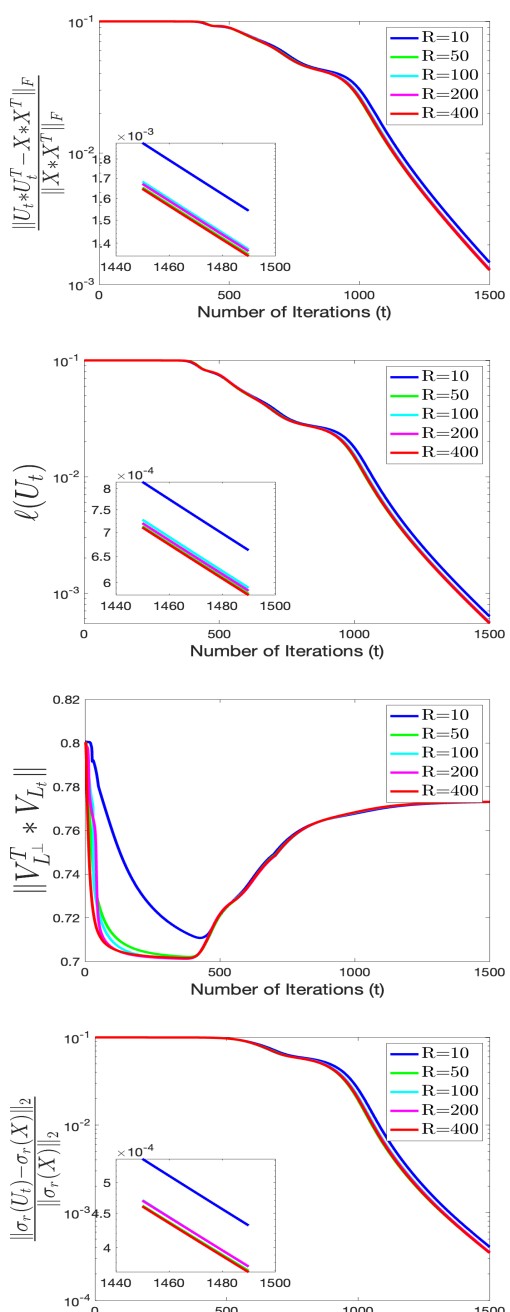

Figure 3: Outcomes of employing gradient descent to minimize the loss function (3.2) with different overparametrization rates. We set $n = 10, k = 4, r = 3$ in the ground truth tensor $\mathcal{X} \in \mathbb{R}^{n \times r \times k}$ and for initialization $\mathcal{U}_0 \in \mathbb{R}^{n \times R \times k}$, we set the over-rank to $R = 10, 50, 100, 200, 400$. For each $R$ we plot the average over twenty experiments. The plots for $\frac{\|\mathcal{U}_t * \mathcal{U}_t^\top - \mathcal{X} * \mathcal{X}^\top\|_F}{\|\mathcal{X} * \mathcal{X}^\top\|_F}$, $\ell(U_t)$ and $\frac{\|\sigma_r(\mathcal{U}_t) - \sigma_r(\mathcal{X})\|_2}{\|\sigma_r(\mathcal{X})\|_2}$ are semi-log plots.

Namely, let as before $y = \mathcal{A}(\mathcal{X} * \mathcal{X}^\top) \in \mathbb{R}^m$ with $y_i = \left\langle \mathcal{A}_i, \mathcal{X} * \mathcal{X}^\top \right\rangle = \left\langle \overline{\mathcal{A}_i}, \overline{\mathcal{X} * \mathcal{X}^\top} \right\rangle = \sum_{q=1}^k \left\langle \overline{A_i}^{(q)}, \overline{X}^{(q)} \overline{X}^{(q)\mathrm{H}} \right\rangle$, $j = 1, \dots m$ then $\mathcal{A}^* \mathcal{A}(\mathcal{X} * \mathcal{X}^\top) = \mathcal{A}^*(y) = \sum_{i=1}^m y_i \mathcal{A}_i \in S^{n \times n \times k}$ and the for $j$-th slice in the Fourier domain, we get $\overline{\mathcal{A}^* \mathcal{A}(\mathcal{X} * \mathcal{X}^\top)}^{(j)} = \sum_{i=1}^m \sum_{j=1}^k \overline{A_i}^{(j)} \left\langle \overline{A_i}^{(q)}, \overline{X}^{(q)} \overline{X}^{(q)\mathrm{H}} \right\rangle$. This means that in each Fourier slice $\overline{\mathcal{U}}_t^{(j)}$ of the gradient descent iterates (3.4) we have the full information about the ground truth tensor $\mathcal{X} * \mathcal{X}^\top$ and not only about its $j$-th slice. In the spectral stage, this fact does not cause significant difficulties. However, in the convergence stage, in order to get the global estimates, it requires a thorough and vigilant analysis of intersections between the slices in the Fourier domain.

In particular, this required nontrivial estimations, such as those presented in Lemmas E.4 and E.5, to control these interactions and provide the respective bounds, which require control of proximity of the auxiliary parameter $\overline{\left(\mathcal{A}^* \mathcal{A}(\mathcal{X} * \mathcal{X}^\top - \mathcal{U}_t * \mathcal{U}_t^\top)\right)}^{(j)}$ to the corresponding $j$th Fourier slice of $\mathcal{X} * \mathcal{X}^\top - \mathcal{U}_t * \mathcal{U}_t^\top$ via the RIP property of the measurement operator $\mathcal{A}$ and aligned matrix subspaces. Another important point is that one need to choose the learning rate $\mu$ and the initialization scale $\alpha$ carefully for the noise term $\mathcal{U}_t * \mathcal{W}_{\perp,t}$ to grow slowly enough in each of the tensor slices in order to not allow overtaking the signal term $\mathcal{U}_t * \mathcal{W}_t$ in the norm, see, e.g., Theorem E.1 and the usage of Lemma E.3 in its proof.

## 5. Numerical Experiments

To verify our theoretical findings, we set multiple numerical tests: from showing two phases of the gradient descent algorithm to demonstrating the advantages of overparametrization. These experimental results showcase not only the implicit regularization for the gradient descent algorithm toward low-tubal-rank tensors but also demonstrate the firmness of our theoretical findings.

Our experiments were conducted on a MacBook Pro equipped with an Apple M1 processor and 16GB of memory, using MATLAB 2023a software. The corresponding code is available in our GitHub repository, https://github.com/AnnaVeselovskaUA/tubal-tensor-implicit-reg-GD.git.

We generate the ground truth tensor $\mathcal{T} \in \mathbb{R}^{n \times n \times k}$ with tubal rank $r$ by $\mathcal{T} = \mathcal{X} * \mathcal{X}^\top$, where the entries of $\mathcal{X} \in \mathbb{R}^{n \times r \times k}$ are i.i.d. sampled from a Gaussian distribution $\mathcal{N}(0, 1)$, and then $\mathcal{X}$ is normalized. The entries of measurement tensor $\mathcal{A}_i$ are i.i.d. sampled from a Gaussian distribution $\mathcal{N}(0, \frac{1}{m})$. In the following, we describe dif-

ferent testing scenarios for recovery of $\mathcal{T}$ via the gradient descent algorithm and their outcome. For all the experiments, we set the dimensions to $n = 10, k = 4, r = 3$, the learning rate $\mu = 10^{-5}$, and the number of measurements $m = 254$.

**Illustration of the two convergence stages.** To illustrate the convergence process of the gradient iterates, for the ground truth tensor $\boldsymbol{\mathcal{X}} * \boldsymbol{\mathcal{X}}^\top \in \mathbb{R}^{n \times n \times k}$ and its counterpart $\boldsymbol{\mathcal{U}}_t * \boldsymbol{\mathcal{U}}_t^\top \in \mathbb{R}^{n \times n \times k}$ being learned by the gradient descent, we consider the training error $\ell(U_t)$, the test error $\frac{\|\boldsymbol{\mathcal{U}}_t * \boldsymbol{\mathcal{U}}_t^\top - \boldsymbol{\mathcal{X}} * \boldsymbol{\mathcal{X}}^\top\|_F}{\|\boldsymbol{\mathcal{X}} * \boldsymbol{\mathcal{X}}^\top\|_F}$, and the test error for their $r$th singular tubes $\sigma_r(\boldsymbol{\mathcal{U}}_t), \sigma_r(\boldsymbol{\mathcal{X}}) \in \mathbb{R}^k$, $\frac{\|\sigma_r(\boldsymbol{\mathcal{U}}_t) - \sigma_r(\boldsymbol{\mathcal{X}})\|_2}{\|\sigma_r(\boldsymbol{\mathcal{X}})\|_2}$. Moreover, we also take into our consideration the tensor subspace $\mathcal{L}$ spanned by the tensor-columns corresponding to the first $r$ singular-tubes of the tensor $\mathcal{A}^* \mathcal{A}(\boldsymbol{\mathcal{X}} * \boldsymbol{\mathcal{X}}^\top)$ and denote by $\mathcal{L}_t$ the tensor-column subspace spanned by the tensor-columns corresponding to the first $r$ singular tubes $\boldsymbol{\mathcal{U}}_t * \boldsymbol{\mathcal{U}}_t^\top$. We note that although Theorem 3.1 bounded a relative error with $\|\boldsymbol{\mathcal{X}}\|^2$ in the denominator, we use $\|\boldsymbol{\mathcal{X}} * \boldsymbol{\mathcal{X}}^\top\|_F$ in the denominator of the relative error for our experiments as it is a more natural relative error to consider. Furthermore, since $\|\boldsymbol{\mathcal{X}} * \boldsymbol{\mathcal{X}}^\top\|_F \geq \|\boldsymbol{\mathcal{X}}\|^2$, and $\|\boldsymbol{\mathcal{X}} * \boldsymbol{\mathcal{X}}^\top\|_F$ could be much larger than $\|\boldsymbol{\mathcal{X}}\|^2$ in cases where the singular values of $\boldsymbol{\mathcal{X}} * \boldsymbol{\mathcal{X}}^\top$ vary drastically, the result of Theorem 3.1 is stronger than if we bounded the more natural Frobenius norm error. Besides, the qualitative behavior in the numerical simulation will be the same for the two error measures as generically they will just differ by a dimensional factor.

Figures 2 demonstrates that the convergence analysis can be divided into two stages: the spectral and the convergence stage. We see that in the first stage ($1 \leq t \lesssim 3000$), the first $r$ tensor-columns of $\boldsymbol{\mathcal{U}}_t * \boldsymbol{\mathcal{U}}_t^\top$ learn the tensor column subspace corresponding to the first $r$ singular-tubes of the tensor $\mathcal{A}^* \mathcal{A}(\boldsymbol{\mathcal{X}} * \boldsymbol{\mathcal{X}}^\top)$, i.e. the principal angle between the tensor column subspaces $\mathcal{L}_t$ and $\mathcal{L}$ becomes small. Namely, as one can observe in Figure 2 (bottom), the principal angle between the two subspaces, $\|\boldsymbol{\mathcal{V}}_{\mathcal{L}^\perp}^\top * \boldsymbol{\mathcal{V}}_{\mathcal{L}_t}\|$, decreases where as the principal angle between $\boldsymbol{\mathcal{X}}$ and $\mathcal{L}_t$ reaches certain plateau, see the behavior of $\|\boldsymbol{\mathcal{V}}_{\boldsymbol{\mathcal{X}}^\perp}^\top * \boldsymbol{\mathcal{V}}_{\mathcal{L}_t}\|$. At the same time, test errors $\frac{\|\boldsymbol{\mathcal{U}}_t * \boldsymbol{\mathcal{U}}_t^\top - \boldsymbol{\mathcal{X}} * \boldsymbol{\mathcal{X}}^\top\|_F}{\|\boldsymbol{\mathcal{X}} * \boldsymbol{\mathcal{X}}^\top\|_F}$ and $\frac{\|\sigma_r(\boldsymbol{\mathcal{U}}_t) - \sigma_r(\boldsymbol{\mathcal{X}})\|_2}{\|\sigma_r(\boldsymbol{\mathcal{X}})\|_2}$ stay large. In the second stage, we see that the test error $\frac{\|\boldsymbol{\mathcal{U}}_t * \boldsymbol{\mathcal{U}}_t^\top - \boldsymbol{\mathcal{X}} * \boldsymbol{\mathcal{X}}^\top\|_F}{\|\boldsymbol{\mathcal{X}} * \boldsymbol{\mathcal{X}}^\top\|_F}$ starts decreasing, meaning that the gradient descent iterates $\boldsymbol{\mathcal{U}}_t * \boldsymbol{\mathcal{U}}_t^\top$ start converging to $\boldsymbol{\mathcal{X}} * \boldsymbol{\mathcal{X}}^\top$ by learning more about the tensor-column subspace of the ground truth tensor. At the same time, the test error over $r$th singular tube $\frac{\|\sigma_r(\boldsymbol{\mathcal{U}}_t) - \sigma_r(\boldsymbol{\mathcal{X}})\|_2}{\|\sigma_r(\boldsymbol{\mathcal{X}})\|_2}$ starts decreasing too and as a result converges to zero. We also see that in this stage the principal angle between $\mathcal{L}_t$ and $\mathcal{L}$ grows, which is also intuitive as the tensor-column subspace $\mathcal{L}$ does not have the full information about the tensor-column subspace of

the ground truth tensor $\boldsymbol{\mathcal{X}} * \boldsymbol{\mathcal{X}}^\top$, and learning more about $\boldsymbol{\mathcal{X}} * \boldsymbol{\mathcal{X}}^\top$ leads to a larger error in terms of principal angles of the two.

**Depiction of the alignment stage.** In this experiment, we illustrate that gradient descent with small initialization behaves similarly to the tensor-power method modulo normalization in the first few iterations, bringing the gradient iterates close to the spectral tubal initialization, used, e.g., in (Liu et al., 2024). Here, as before $\mathcal{L}$ denote the tensor subspace spanned by the tensor-columns corresponding to the first $r$ singular-tubes of tensor $\mathcal{A}^* \mathcal{A}(\boldsymbol{\mathcal{X}} * \boldsymbol{\mathcal{X}}^\top)$ and $\mathcal{L}_t$ is the tensor-column subspace corresponding to the first $r$ singular tubes $\boldsymbol{\mathcal{U}}_t * \boldsymbol{\mathcal{U}}_t^\top$. Additionally, $\widetilde{\mathcal{L}}_t$ denotes the tensor-column subspace spanned by the first $r$ singular-tubes of the tensor $\widetilde{\boldsymbol{\mathcal{U}}}_t * \widetilde{\boldsymbol{\mathcal{U}}}_t^\top$, where $\widetilde{\boldsymbol{\mathcal{U}}}_t^\top = \left(\boldsymbol{\mathcal{I}} + \mathcal{A}^* \mathcal{A}(\boldsymbol{\mathcal{X}} * \boldsymbol{\mathcal{X}}^\top)\right)^{*t} * \boldsymbol{\mathcal{U}}_0$. In Figure 2 (bottom), we see that $\boldsymbol{\mathcal{U}}_t$ and $\widetilde{\boldsymbol{\mathcal{U}}}_t$ learn the subspace $\mathcal{L}$ almost at the same rate in the first iterations, $1 \leq t \lesssim 3000$. In the same figure, we observe that also the angle between $\boldsymbol{\mathcal{V}}_{\boldsymbol{\mathcal{X}}}$ and $\mathcal{L}_t$, respectively $\widetilde{\mathcal{L}}_t$, decreases monotonically in the spectral stage. Then at the beginning of the convergence stage, $3000 \lesssim t$, the angle between $\boldsymbol{\mathcal{V}}_{\boldsymbol{\mathcal{X}}}$ and $\mathcal{L}_t$ starts decreasing gradually and converges to zero, as expected since $\boldsymbol{\mathcal{U}}_t * \boldsymbol{\mathcal{U}}_t^\top$ converges to $\boldsymbol{\mathcal{X}} * \boldsymbol{\mathcal{X}}^\top$. Whereas the principal angle between $\mathcal{L}$ and $\mathcal{L}_t$ growths until it reaches a certain plateau.

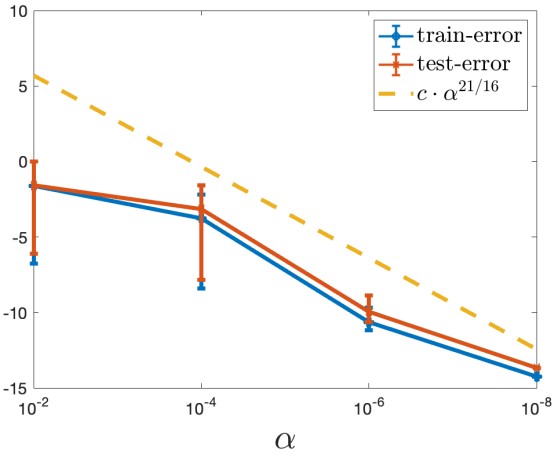

Figure 4: Impact of different initialization scales on the test and the training error. The data are represented in the log-log plot. We set $n = 10, k = 4, r = 3$ in the ground truth tensor $\boldsymbol{\mathcal{X}} \in \mathbb{R}^{n \times r \times k}$ and for initialization $\boldsymbol{\mathcal{U}}_0 = \alpha \boldsymbol{\mathcal{U}} \in \mathbb{R}^{n \times R \times k}$ with $R = 200$ and different scales of $\alpha$. The plot depicts the averaged value for five runs and the bars represent the deviations from the mean value. For illustration, we also depict the theoretical test error bound obtained in Theorem 3.1. As one can see, the numerical error resembles the theoretical behavior of $C_{n,k,r,\kappa} \cdot \alpha^{\frac{21}{16}}$.

**Test and train error under different scales of initialization.** In this experiment, we explore the influence of the initialization scale, denoted by $\alpha$, on the training and the test error. With $R = 200$, we apply gradient descent for various values of $\alpha$, halting the iterations at $t = 3500$ in each run. The results, presented in Figure 4, demonstrate a reduction in test error as $\alpha$ decreases. Notably, the figure indicates that the test error follows an almost polynomial relationship with the initialization scale $\alpha$. This observation is consistent with our theoretical predictions, which also forecast a decrease in test error at a rate of $\alpha$, see Theorem 3.1.

**Impact of different levels of overparameterization on the convergence.** In this numerical analysis, we set $\alpha = 10^{-7}$ and examined the convergence speed of gradient descent to the ground truth tensor for various overparameterization rates $R$. We run the experiment twenty times for each value of $R$ and plot the averaged values per each iteration. The results, shown in Figure 3, reveal that increasing the number of tensor columns $R$, that is, overparameterizing, accelerates the convergence rate, resulting in fewer iterations to reach the desired error level. Additionally, overparameterization reduces the test error and the training error by affecting the spectral stages.

## 6. Conclusion and Outlook

In this paper, we focused on studying the implicit regularization of tubal tensor factorizations via gradient descent by showing that with small random initialization and overparametrization, the gradient descent algorithm is biased towards a low-tubal-rank solution. We have shown that the first iterations of gradient descent with small random initialization behave similarly to the tensor power method, which leads to learning in these first iterations the tensor-column spaces close to the tensor-column space of the ground truth. We also demonstrate that the implicit regularization from small random initialization guides the gradient descent iterations toward low-tubal rank solutions that are not only globally optimal but also generalize well.

## Acknowledgments

AV and FK acknowledge support by the German Science Foundation (DFG) in the context of the collaborative research center TR-109, the Emmy Noether junior research group KR 4512/1-1 and the Bavarian Funding Program for Initiating International Research Cooperation, as well as by the Munich Data Science Institute and Munich Center for Machine Learning. SK acknowledges support by the United States National Science Foundation in the context of the Foundations of Data Science Institute funded by grant NSF DMS 2022205. MI acknowledges support by the United States National Science Foundation grants NSF DMS 2108479 and NSF EDU DGE 2152014.

## Impact Statement

This paper presents work whose goal is to advance the field of Machine Learning, and more specifically, the theoretical understanding of implicit regularization as a tool for structured recovery problems. There are many potential societal consequences of our work, none which we feel must be specifically highlighted here.

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

# Supplementary Material

## A. Outline of Appendices

For ease of organization, we divide the supplementary material into appendices as follows. In Appendix B, we define some additional notation, including the angles between two tensor-column subspaces. In Appendix C, we decompose the gradient descent iterates into a "signal" term and a "noise" term, which will aid us in our analysis. In Appendices D and E, we analyze the spectral and convergence stages, respectively, of the gradient descent iterations. In Appendix F, we prove our main result.

To avoid breaking up the flow of our analysis, we put some technical lemmas in the last few appendices instead of in the previously mentioned appendices. In Appendix G, we prove some properties of measurement operators which satisfy the restricted isometry property. In Appendix H, we prove some properties of matrices and their subspaces. Finally, in Appendix I, we prove some properties of random Gaussian tubal tensors.

## B. Additional Notation

For a tensor $\mathcal{Y} \in \mathbb{R}^{n \times r \times k}$, we denote its t-SVD by $\mathcal{Y} = \mathcal{V}_{\mathcal{Y}} * \Sigma_{\mathcal{Y}} * \mathcal{W}_{\mathcal{Y}}^\top$ with the two orthogonal tensor $\mathcal{V}_{\mathcal{Y}}, \mathcal{W}_{\mathcal{Y}} \in \mathbb{R}^{n \times r \times k}$, and the f-diagonal tensor $\Sigma_{\mathcal{Y}} \in \mathbb{R}^{r \times r \times k}$. We will refer to $\mathcal{V}_{\mathcal{Y}}$ as the tensor-column subspace of $\mathcal{Y}$ and by $\mathcal{V}_{\mathcal{Y}^\perp} \in \mathbb{R}^{n \times (n-r) \times k}$ we denote the tensor-column subspace orthogonal to $\mathcal{V}_{\mathcal{Y}}$ with its projection operator $\mathcal{V}_{\mathcal{Y}^\perp} * \mathcal{V}_{\mathcal{Y}^\perp}^\top = \mathcal{I} - \mathcal{V}_{\mathcal{Y}} * \mathcal{V}_{\mathcal{Y}}^\top$.

We measure the angles between two tensor-column subspaces $\mathcal{Y}_1$ and $\mathcal{Y}_2$ by the tensor-spectral norm $\|\mathcal{V}_{\mathcal{Y}_1^\perp} * \mathcal{V}_{\mathcal{Y}_2}\|$ which according to (Liu et al., 2019; Gleich et al., 2013; Kilmer & Martin, 2011) is equal to

$$\|\mathcal{V}_{\mathcal{Y}_1^\perp}^\top * \mathcal{V}_{\mathcal{Y}_2}\| = \|\overline{\mathcal{V}_{\mathcal{Y}_1^\perp}^\top * \mathcal{V}_{\mathcal{Y}_2}}\| = \|\overline{\mathcal{V}_{\mathcal{Y}_1^\perp}^\top}\,\overline{\mathcal{V}_{\mathcal{Y}_2}}\|.$$

which means that the largest principal angle between $\mathcal{Y}_1$ and $\mathcal{Y}_2$ equals to that of these two subspaces represented in the Fourier domain. In the Fourier domain, since $\overline{\mathcal{V}_{\mathcal{Y}_1^\perp}^\top} \in \mathbb{C}^{(n-r)k \times nk}$ and $\overline{\mathcal{V}_{\mathcal{Y}_2}} \in \mathbb{C}^{nk \times nk}$ are block diagonal matrices, it holds that

$$\|\overline{\mathcal{V}_{\mathcal{Y}_1^\perp}^\top}\,\overline{\mathcal{V}_{\mathcal{Y}_2}}\| = \left\| \begin{pmatrix} \overline{\mathcal{V}_{\mathcal{Y}_1^\perp}^\top}^{(1)} & & & \\ & \overline{\mathcal{V}_{\mathcal{Y}_1^\perp}^\top}^{(2)} & & \\ & & \ddots & \\ & & & \overline{\mathcal{V}_{\mathcal{Y}_1^\perp}^\top}^{(k)} \end{pmatrix} \begin{pmatrix} \overline{\mathcal{V}_{\mathcal{Y}_2}}^{(1)} & & & \\ & \overline{\mathcal{V}_{\mathcal{Y}_2}}^{(2)} & & \\ & & \ddots & \\ & & & \overline{\mathcal{V}_{\mathcal{Y}_2}}^{(k)} \end{pmatrix} \right\| = \max_{1 \leq j \leq k} \left\| \overline{\mathcal{V}_{\mathcal{Y}_1^\perp}^\top}^{(j)} \overline{\mathcal{V}_{\mathcal{Y}_2}}^{(j)} \right\|$$

## C. Signal Decomposition

Recall that the gradient descent iterates are defined in (3.4) as

$$\begin{aligned} \mathcal{U}_{t+1} &= \mathcal{U}_t - \mu \nabla \ell(\mathcal{U}_t) \\ &= \mathcal{U}_t + \mu \mathcal{A}^* \left[ y - \mathcal{A}\left( \mathcal{U}_t * \mathcal{U}_t^\top \right) \right] * \mathcal{U}_t \\ &= \left[ \mathcal{I} + \mu(\mathcal{A}^*\mathcal{A})\left( \mathcal{X} * \mathcal{X}^\top - \mathcal{U}_t * \mathcal{U}_t^\top \right) \right] * \mathcal{U}_t. \end{aligned}$$

For the ground truth tensor $\mathcal{X} \in \mathbb{R}^{n \times r \times k}$, consider its tensor-column subspace $\mathcal{V}_{\mathcal{X}}$ with the corresponding basis $\mathcal{V}_{\mathcal{X}} \in \mathbb{R}^{n \times r \times k}$. Consider the tensor $\mathcal{V}_{\mathcal{X}} * \mathcal{U}_t \in \mathbb{R}^{r \times R \times k}$ with its t-SVD decomposition $\mathcal{V}_{\mathcal{X}} * \mathcal{U}_t = \mathcal{V}_t * \Sigma_t * \mathcal{W}_t^\top$. For $\mathcal{W}_t \in \mathbb{R}^{R \times r \times k}$, we denote by $\mathcal{W}_{t,\perp} \in \mathbb{R}^{R \times (n-r) \times k}$ a tensor whose tensor-column subspace is orthogonal to those of $\mathcal{W}_t$, that is $\|\mathcal{W}_{t,\perp}^\top * \mathcal{W}_t\| = 0$ and its projection operator $\mathcal{P}_{\mathcal{W}_{t,\perp}}$ is defined as $\mathcal{P}_{\mathcal{W}_{t,\perp}} = \mathcal{W}_{t,\perp} * \mathcal{W}_{t,\perp}^\top = \mathcal{I} - \mathcal{W}_t * \mathcal{W}_t^\top$. We then decompose the gradient descent iterates $\mathcal{U}_t$ as follows

$$\mathcal{U}_t = \mathcal{U}_t * \mathcal{W}_t * \mathcal{W}_t^\top + \mathcal{U}_t * \mathcal{W}_{t,\perp} * \mathcal{W}_{t,\perp}^\top \tag{C.1}$$

We will refer to the tensors $\mathcal{U}_t * \mathcal{W}_t * \mathcal{W}_t^\top$ as the signal term of the gradient descent iterates, and the tensors $\mathcal{U}_t * \mathcal{W}_{t,\perp} * \mathcal{W}_{t,\perp}^\top$ will be named as the noise term.

**Lemma C.1.** *The tensor-column space of the noise term $\mathcal{U}_t * \mathcal{W}_{t,\perp} * \mathcal{W}_{t,\perp}^\top$ is orthogonal to the tensor-column subspace of the $\mathcal{X}$, namely $\mathcal{V}_{\mathcal{X}}^\top * \mathcal{U}_t * \mathcal{W}_{t,\perp} * \mathcal{W}_{t,\perp}^\top = 0$. Moreover, if $\mathcal{V}_{\mathcal{X}}^\top * \mathcal{U}_t$ is full tubal-rank with all invertible singular tubes, then the signal term*

$$\mathcal{U}_t * \mathcal{W}_t * \mathcal{W}_t^\top$$

*has tubal-rank $r$ with all invertible singular tubes and the noise term has tubal rank at most $R - r$.*

*Proof.* $\mathcal{V}_{\mathcal{X}}^\top * \mathcal{U}_t * \mathcal{W}_{t,\perp} * \mathcal{W}_{t,\perp}^\top = \mathcal{V}_{\mathcal{X}}^\top * \mathcal{U}_t * (\mathcal{I} - \mathcal{W}_t * \mathcal{W}_t^\top) = \mathcal{V}_{\mathcal{X}}^\top * \mathcal{U}_t - \mathcal{V}_{\mathcal{X}}^\top * \mathcal{U}_t * \mathcal{W}_t * \mathcal{W}_t^\top = 0 \in \mathbb{R}^{r \times R \times k}$. The second part follows fact that if $\mathcal{V}_{\mathcal{X}}^\top * \mathcal{U}_t$ is full tubal rank with all invertible singular tubes then all the slices in the Fourier have full rank. $\square$

# D. Analysis of the Spectral Stage

The goal of this section is to show that the first few iterations of the gradient descent algorithm can be approximated by the iteration of the tensor power method modulo normalization defined as

$$\widetilde{\mathcal{U}}_t = \left(\mathcal{I} + \mu \mathcal{A}^* \mathcal{A}(\mathcal{X} * \mathcal{X}^\top)\right)^{*t} * \mathcal{U}_0 = \mathcal{Z}_t * \mathcal{U}_0 \in \mathbb{R}^{n \times R \times k}.$$

with the tensor power method iteration $\mathcal{Z}_t =: \left(\mathcal{I} + \mu \mathcal{A}^* \mathcal{A}(\mathcal{X} * \mathcal{X}^\top)\right)^{*t} \in \mathbb{R}^{n \times n \times k}$. Moreover, this will result in the feature that after the first few iterations, the tensor-column span of the signal term $\mathcal{U}_t * \mathcal{W}_t * \mathcal{W}_t^\top$ becomes aligned with the tensor-column span of $\mathcal{X}$, and that the noise term $\mathcal{U}_t * \mathcal{W}_{t,\perp}$ is relatively small compared to signal term in terms of the norm, indicating that the signal term dominates the noise term.

For this, let us denote the difference between the power method and the gradient descent iterations by

$$\mathcal{E}_t := \mathcal{U}_t - \widetilde{\mathcal{U}}_t. \tag{D.1}$$

For convenience, throughout this section, we will denote by $\mathcal{M}$ the tensor $\mathcal{M} := \mathcal{A}^* \mathcal{A}(\mathcal{X} * \mathcal{X}^\top) \in \mathbb{R}^{n \times n \times k}$, so that $\widetilde{\mathcal{U}}_t = (\mathcal{I} + \mu \mathcal{M})^{*t} * \mathcal{U}_0$ and $\mathcal{Z}_t = (\mathcal{I} + \mu \mathcal{M})^{*t}$.

In the first result of this section, the following lemma, we show that $\mathcal{E}_t$ can be made small via an appropriate initialization scale.

**Lemma D.1.** *Suppose that $\mathcal{A} : S^{n \times n \times k} \to \mathbb{R}^m$ satisfies $RIP(2, \delta_1)$ and let $t^\star$ be defined as*

$$t^\star = \min \left\{ j \in \mathbb{N} \colon \|\widetilde{\mathcal{U}}_{j-1} - \mathcal{U}_{j-1}\| > \|\widetilde{\mathcal{U}}_{j-1}\| \right\}. \tag{D.2}$$

*Then for all integers $t$ such that $1 \leq t \leq t^\star$ it holds that*

$$\|\mathcal{E}_t\| = \|\mathcal{U}_t - \widetilde{\mathcal{U}}_t\| \leq 8(1 + \delta_1 \sqrt{k}) \sqrt{k \min\{n, R\}} \frac{\alpha^3}{\|\mathcal{M}\|} \|\mathcal{U}\|^3 (1 + \mu \|\mathcal{M}\|)^{3t}. \tag{D.3}$$

*Proof.* Similarly to the matrix case in (Stöger & Soltanolkotabi, 2021), in the tubal tensor case it can be shown that for $t \geq 1$, the difference tensor $\mathcal{E}_t = \mathcal{U}_t - \widetilde{\mathcal{U}}_t$ can be represented as

$$\mathcal{E}_t = \mathcal{U}_t - \widetilde{\mathcal{U}}_t = \sum_{j=1}^t (\mathcal{I} + \mu \mathcal{M})^{*(t-j)} \widehat{\mathcal{E}}_j \tag{D.4}$$

with $\widehat{\mathcal{E}}_j = \mu \mathcal{A}^* \mathcal{A}(\mathcal{U}_{j-1} * \mathcal{U}_{j-1}^\top) * \mathcal{U}_{j-1}$. To estimate $\|\mathcal{E}_t\|$, we will first estimate each summand in (D.4) separately. First, we can proceed with the following simple estimation

$$\|(\mathcal{I} + \mu \mathcal{M})^{*(t-j)} \widehat{\mathcal{E}}_j\| \leq \|(\mathcal{I} + \mu \mathcal{M})\|^{(t-j)} \|\widehat{\mathcal{E}}_j\| \leq \left(1 + \mu \|\mathcal{M}\|\right)^{(t-j)} \|\widehat{\mathcal{E}}_j\|.$$

Now, for $\|\widehat{\mathcal{E}}_j\|$, using the fact that the spectral norm of tubal tensors is sub-multiplicative, we get that

$$\|\widehat{\mathcal{E}}_j\| = \mu \|\mathcal{A}^* \mathcal{A}(\mathcal{U}_{j-1} * \mathcal{U}_{j-1}^\top) * \mathcal{U}_{j-1}\| \leq \mu \|\mathcal{A}^* \mathcal{A}(\mathcal{U}_{j-1} * \mathcal{U}_{j-1}^\top)\| \cdot \|\mathcal{U}_{j-1}\|.$$

Since operator $\mathcal{A}$ satisfies RIP$(2, \delta_1)$, by Lemma G.3, $\mathcal{A}$ also satisfies S2NRIP$(\delta_1\sqrt{k})$, which provides the following estimate

$$\|\mathcal{A}^*\mathcal{A}(\mathcal{U}_{j-1} * \mathcal{U}_{j-1}^\top)\| \leq (1+\delta_1\sqrt{k})\|\mathcal{U}_{j-1} * \mathcal{U}_{j-1}^\top\|_* = (1+\delta_1\sqrt{k})\|\mathcal{U}_{j-1}\|_F^2.$$

All this together leads to

$$\|\mathcal{E}_t\| = \|\mathcal{U}_t - \widetilde{\mathcal{U}}_t\| \leq \mu(1+\delta_1\sqrt{k})\sum_{j=1}^t \left(1 + \mu\|\mathcal{M}\|\right)^{(t-j)}\|\mathcal{U}_{j-1}\|_F^2\|\mathcal{U}_{j-1}\|. \tag{D.5}$$

From here, we want to bound $\|\mathcal{E}_t\|$ in terms of the initialization scale $\alpha$ and the data-related norm $\|\mathcal{M}\|$. For this, we first use the fact that the tensor Frobenius norm above can be bounded as $\|\mathcal{U}_{j-1}\|_F \leq \sqrt{k\min\{n, R\}}\|\mathcal{U}_{j-1}\|$. Then since for all $1 \leq j \leq t^\star$ we have $\|\widetilde{\mathcal{U}}_{j-1} - \mathcal{U}_{j-1}\| \leq \|\widetilde{\mathcal{U}}_{j-1}\|$, the spectral norm of $\mathcal{U}_{j-1}$ can be bounded as

$$\|\mathcal{U}_{j-1}\| \leq \|\widetilde{\mathcal{U}}_{j-1}\| + \|\mathcal{U}_{j-1} - \widetilde{\mathcal{U}}_{j-1}\| \leq 2\|\widetilde{\mathcal{U}}_{j-1}\|.$$

This gives us the following upper bound

$$\|\mathcal{E}_t\| \leq 8\mu(1+\delta_1\sqrt{k})\sqrt{k\min\{n, R\}}\sum_{j=1}^t (1+\mu\|\mathcal{M}\|)^{t-j}\|\widetilde{\mathcal{U}}_{j-1}\|^3. \tag{D.6}$$

As for iterations of the tensor power method, it holds that

$$\|\widetilde{\mathcal{U}}_{j-1}\| = \|(\mathcal{I} + \mu\mathcal{M})^{*(j-1)} * \mathcal{U}_0\| \leq \|(\mathcal{I} + \mu\mathcal{M})^{*(j-1)}\|\|\mathcal{U}_0\| \leq (1+\mu\|\mathcal{M}\|)^{j-1}\|\mathcal{U}_0\| = \alpha(1+\mu\|\mathcal{M}\|)^{j-1}\|\mathcal{U}\|,$$

we can proceed with (D.6) as follows

$$\|\mathcal{E}_t\| \leq 8\mu(1+\delta_1\sqrt{k})\sqrt{k\min\{n, R\}}\alpha^3\|\mathcal{U}\|^3 \sum_{j=1}^t (1+\mu\|\mathcal{M}\|)^{t+2j-3}.$$

Now, the sum on the right-hand side can be estimated as

$$\sum_{j=1}^t (1+\mu\|\mathcal{M}\|)^{t+2j-3} = (1+\mu\|\mathcal{M}\|)^{t-1}\sum_{j=1}^t (1+\mu\|\mathcal{M}\|)^{2j-2} = (1+\mu\|\mathcal{M}\|)^{t-1}\frac{(1+\mu\|\mathcal{M}\|)^{2t} - 1}{(1+\mu\|\mathcal{M}\|)^2 - 1}$$

$$= (1+\mu\|\mathcal{M}\|)^{t-1}\frac{(1+\mu\|\mathcal{M}\|)^{2t} - 1}{\mu\|\mathcal{M}\|(2 + \mu\|\mathcal{M}\|)} \leq \frac{(1+\mu\|\mathcal{M}\|)^{3t}}{\mu\|\mathcal{M}\|},$$

which gives us the final estimation for the norm of $\mathcal{E}_t$ as follows

$$\|\mathcal{E}_t\| \leq 8(1+\delta_1\sqrt{k})\sqrt{k\min\{n, R\}}\frac{\alpha^3}{\|\mathcal{M}\|}\|\mathcal{U}\|^3(1+\mu\|\mathcal{M}\|)^{3t}$$

and finishes the proof. $\qquad\square$

The following lemma provides a lower bound for $t^\star$, indicating the duration for which the approximation in Lemma D.1 remains valid.

**Lemma D.2.** *Consider tensors* $\mathcal{M} := \mathcal{A}^*\mathcal{A}(\mathcal{X} * \mathcal{X}^\top) \in \mathbb{R}^{n \times n \times k}$ *and* $\widetilde{\mathcal{U}}_t := (\mathcal{I} + \mu\mathcal{M})^{*t} * \mathcal{U}_0$. *Let* $\overline{\mathcal{M}} \in \mathbb{C}^{nk \times nk}$ *be the corresponding block diagonal form of the tensor* $\mathcal{M}$ *with the leading eigenvector* $v_1 \in \mathbb{C}^{nk}$, *then*

$$t^\star \geq \left\lfloor \frac{\ln\left(\frac{\|\mathcal{M}\| \cdot \|\overline{\mathcal{U}_0}^\mathrm{H} v_1\|_{\ell_2}}{8(1+\delta_1\sqrt{k})\sqrt{k\min\{n, R\}}\alpha^3\|\mathcal{U}\|^3}\right)}{2\ln(1 + \mu\|\mathcal{M}\|)} \right\rfloor \tag{D.7}$$

*Proof.* Let $\overline{\widetilde{\mathcal{U}}}_t \in \mathbb{C}^{nk \times Rk}$ be the corresponding block diagonal form of tensor $\widetilde{\mathcal{U}}_t$. By the definition of the spectral tensor norm, we have $\|\widetilde{\mathcal{U}}_t\| = \|\overline{\widetilde{\mathcal{U}}}_t\|$ and the definition of the matrix norm gives $\|\overline{\widetilde{\mathcal{U}}}_t\| \geq \left\|\overline{\widetilde{\mathcal{U}}}_t^{\mathrm{H}} v_1\right\|_{\ell_2}$. For the block diagonal version of $\widetilde{\mathcal{U}}_t$, the following properties (see, e.g., (Liu et al., 2019)) holds

$$\overline{\widetilde{\mathcal{U}}}_t = \overline{(\mathcal{I} + \mu\mathcal{M})^{*t} * \mathcal{U}_0} = \overline{(\mathcal{I} + \mu\mathcal{M})^{*t}} \cdot \overline{\mathcal{U}_0} = \overline{(\mathcal{I} + \mu\mathcal{M})}^t \cdot \overline{\mathcal{U}_0}. \tag{D.8}$$

This allows us to proceed as follows

$$\overline{\widetilde{\mathcal{U}}}_t^{\mathrm{H}} v_1 = \left(\overline{(\mathcal{I} + \mu\mathcal{M})}^t \cdot \overline{\mathcal{U}_0}\right)^{\mathrm{H}} v_1 = \overline{\mathcal{U}_0}^{\mathrm{H}} \overline{(\mathcal{I} + \mu\mathcal{M})}^{t\,\mathrm{H}} v_1 = (1 + \mu\|\mathcal{M}\|)^t \overline{\mathcal{U}_0}^{\mathrm{H}} v_1,$$

where for the last equality we used the fact that block-diagonal matrix $\overline{(\mathcal{I} + \mu\mathcal{M})}$ has the same set of eigenvectors as matrix $\overline{\mathcal{M}}$. From here, we get $\|\widetilde{\mathcal{U}}_t\| \geq \left\|\overline{\widetilde{\mathcal{U}}}_t^{\mathrm{H}} v_1\right\|_{\ell_2} = (1 + \mu\|\mathcal{M}\|)^t \left\|\overline{\mathcal{U}_0}^{\mathrm{H}} v_1\right\|_{\ell_2}$. Then, applying Lemma D.1, the relative error in the spectral norm between $\widetilde{\mathcal{U}}_t$ and $\mathcal{U}_t$ can be estimated as

$$\frac{\|\widetilde{\mathcal{U}}_t - \mathcal{U}_t\|}{\|\widetilde{\mathcal{U}}_t\|} \leq 8(1 + \delta_1\sqrt{k}) \frac{\sqrt{k \min\{n, R\}}\alpha^3}{\|\mathcal{M}\| \cdot \left\|\overline{\mathcal{U}_0}^{\mathrm{H}} v_1\right\|_{\ell_2}} \|\mathcal{U}\|^3 (1 + \mu\|\mathcal{M}\|)^{2t}.$$

Setting the bound above to be smaller than 1 and solving for $t$, we get

$$t < \frac{\ln\left(\frac{\|\mathcal{M}\| \cdot \left\|\overline{\mathcal{U}_0}^{\mathrm{H}} v_1\right\|_{\ell_2}}{8(1 + \delta_1\sqrt{k})\sqrt{k \min\{n, R\}}\alpha^3 \|\mathcal{U}\|^3}\right)}{2\ln(1 + \mu\|\mathcal{M}\|)}.$$

Since $t \in \mathbb{N}$ with $t \leq t^\star$ should be such that $\frac{\|\widetilde{\mathcal{U}}_{t-1} - \mathcal{U}_{t-1}\|}{\|\widetilde{\mathcal{U}}_{t-1}\|} < 1$, we can choose $t^\star$ as the floor-value of the right-hand side above. $\qquad\square$

To show that the tensor column subspaces of the tensor power method iterates and the gradient descent iterates are aligned after the alignment phase, we use the largest principal angle between two tensor-column subspaces as the potential function for analysis. Borrowing the idea from (Gleich et al., 2013), we will show that the power method iteration in the tensor domain can be transformed to the classical subspace iteration in the frequency domain.

For this, consider the power method iterates $\widetilde{\mathcal{U}}_t = (\mathcal{I} + \mu\mathcal{M})^{*t} * \mathcal{U}_0$, the iterates $\mathcal{Z}_t = (\mathcal{I} + \mu\mathcal{M})^{*t}$ and the gradient descent iterates $\mathcal{U}_t$ represented as $\mathcal{U}_t = \widetilde{\mathcal{U}}_t + \mathcal{E}_t = \mathcal{Z}_t * \mathcal{U}_0 + \mathcal{E}_t$. All these tensors have their counterparts in the Fourier domain, which we will denote respectively as $\overline{\widetilde{\mathcal{U}}}_t$, $\overline{\mathcal{Z}}_t$ and $\overline{\mathcal{U}}_t$.

As before, consider $\mathcal{M} = \mathcal{A}^*\mathcal{A}(\mathcal{X} * \mathcal{X}^\top) \in \mathbb{R}^{n \times n \times k}$ with its t-SVD $\mathcal{M} = \mathcal{V}_\mathcal{M} * \Sigma_\mathcal{M} * \mathcal{W}_\mathcal{M}^\top$ and its Fourier domain representative $\overline{\mathcal{M}} \in \mathbb{C}^{nk \times nk}$. We denote by $\mathcal{L} \in \mathbb{R}^{n \times r \times k}$ the tensor column subspace spanned by the tensor columns corresponding to the first $r$ singular tubes, that is $\mathcal{L} := \mathcal{V}_\mathcal{M}(:, 1:r, :) \in \mathbb{R}^{n \times r \times k}$. Note that $\mathcal{L}$ is also the subspace spanned by the tensor columns corresponding to the first $r$ singular tubes of the tensor $\mathcal{Z}_t \in \mathbb{R}^{n \times n \times k}$.

By $\mathcal{L}_t \in \mathbb{R}^{n \times n \times k}$ we will donate the tensor-column subspace spanned by the tensor columns corresponding to the first $r$ singular tubes of the gradient descent iterates $\mathcal{U}_t = \mathcal{Z}_t * \mathcal{U}_0 + \mathcal{E}_t$. More concretely, for $\mathcal{U}_t = \sum_{s=1}^{R} \mathcal{V}_{\mathcal{U}_t}(:, s, :) * \Sigma_{\mathcal{U}_t}(s, s, :) * \mathcal{W}_{\mathcal{U}_t}^\top(:, s, :)$ and the corresponding Fourier domain representation $\overline{\mathcal{U}}_t = \mathrm{diag}(\overline{U}_t^{(1)}, \overline{U}_t^{(2)}, \ldots, \overline{U}_t^{(k)})$, where $\overline{U}_t^{(j)} = \sum_\ell \sigma_\ell^{(j)} v_\ell^{(j)} w_\ell^{(j)\mathrm{H}} = U_{U_t}^{(j)} \Sigma_{U_t}^{(j)} W_{U_t}^{(j)\mathrm{H}}$, we define the corresponding new tensors $\mathcal{L}_t := \mathcal{V}_{\mathcal{U}_t}(:, 1:r, :) \in \mathbb{R}^{n \times r \times k}$ and their Fourier domain representations

$$\overline{\mathcal{L}}_t = \mathrm{diag}(\overline{L}_t^{(1)}, \overline{L}_t^{(2)}, \ldots, \overline{L}_t^{(k)}) \tag{D.9}$$

**Lemma D.3.** *Consider the tensor iterates* $\mathcal{Z}_t = (\mathcal{I} + \mu\mathcal{M})^{*t}$ *with its block-matrix representation*

$$\overline{\mathcal{Z}}_t = bdiag(\mathcal{Z}_t) = diag(\overline{Z}_t^{(1)}, \overline{Z}_t^{(2)}, \ldots, \overline{Z}_t^{(k)}). \tag{D.10}$$

*and the tensors*

$$\mathcal{E}_t = \mathcal{U}_t - \widetilde{\mathcal{U}}_t \in \mathbb{R}^{n \times R \times k}$$
$$\mathcal{U}_0 = \alpha\mathcal{U} \in \mathbb{R}^{n \times R \times k}, \quad \alpha > 0.$$

*Assume that for each $1 \leq j \leq k$, it holds that*

$$\sigma_{r+1}(\overline{Z_t}^{(j)})\|\mathcal{U}\| + \frac{\|\mathcal{E}_t\|}{\alpha} < \sigma_r(\overline{Z_t}^{(j)})\sigma_{min}(\overline{\mathcal{V}_{\mathcal{L}}^\top * \mathcal{U}}). \tag{D.11}$$

*Then for each $1 \leq j \leq k$, the following two inequalities hold*

$$\sigma_r\big(\overline{U_t}^{(j)}\big) = \sigma_r\big(\overline{Z_t}^{(j)}\overline{U_0}^{(j)} + \overline{E_t}^{(j)}\big) \geq \alpha\sigma_r(\overline{Z_t}^{(j)})\sigma_{min}(\overline{\mathcal{V}_{\mathcal{L}}^\top * \mathcal{U}}) - \|\mathcal{E}_t\|, \tag{D.12}$$

$$\sigma_{r+1}\big(\overline{U_t}^{(j)}\big) = \sigma_{r+1}\big(\overline{Z_t}^{(j)}\overline{U_0}^{(j)} + \overline{E_t}^{(j)}\big) \leq \alpha\sigma_{r+1}(\overline{Z_t}^{(j)})\|\mathcal{U}\| + \|\mathcal{E}_t\| \tag{D.13}$$

*Moreover, the principal angle between the tensor-column subspaces $\mathcal{L}$ and $\mathcal{L}_t$ is bounded as follows*

$$\|\mathcal{V}_{\mathcal{L}^\perp}^\top * \mathcal{V}_{\mathcal{L}_t}\| \leq \max_{1 \leq j \leq k} \frac{\alpha\sigma_{r+1}(\overline{Z_t}^{(j)})\|\mathcal{U}\| + \|\mathcal{E}_t\|}{\sigma_r(\overline{Z_t}^{(j)})\sigma_{min}(\overline{\mathcal{V}_{\mathcal{L}}^\top * \mathcal{U}}) - \alpha\sigma_{r+1}(\overline{Z_t}^{(j)})\|\mathcal{U}\| - \|\mathcal{E}_t\|} \tag{D.14}$$

*Proof.* For some $t \in \mathbb{N}$, consider tensor $\mathcal{Z}_t = (\mathcal{I} + \mu\mathcal{M})^{*t}$ with its block-matrix representation

$$\overline{\mathcal{Z}_t} = \mathrm{bdiag}(\mathcal{Z}_t) = \mathrm{diag}(\overline{Z_t}^{(1)}, \overline{Z_t}^{(2)}, \ldots, \overline{Z_t}^{(k)}) = \begin{pmatrix} \overline{Z_t}^{(1)} & & & \\ & \overline{Z_t}^{(2)} & & \\ & & \ddots & \\ & & & \overline{Z_t}^{(k)} \end{pmatrix}.$$

As we assume the symmetric tensor case scenario, the block-diagonal matrix representation $\overline{Z_t}$ consists of symmetric matrices $\overline{Z_t}^{(j)} \in \mathbb{C}^{n \times n}$. At the same time, according to (Gleich et al., 2013), the gradient descent tensors $\mathcal{U}_t = \mathcal{Z}_t * \mathcal{U}_0 + \mathcal{E}_t$ have their block-diagonal matrix representation

$$\mathcal{U}_t = \mathcal{Z}_t * \mathcal{U}_0 + \mathcal{E}_t \Leftrightarrow \overline{Z_t}\overline{U_0} + \overline{E_t} = \begin{pmatrix} \overline{Z_t}^{(1)}\overline{U_0}^{(1)} & & & \\ & \overline{Z_t}^{(2)}\overline{U_0}^{(2)} & & \\ & & \ddots & \\ & & & \overline{Z_t}^{(k)}\overline{U_0}^{(k)} \end{pmatrix} + \begin{pmatrix} \overline{E_t}^{(1)} & & & \\ & \overline{E_t}^{(2)} & & \\ & & \ddots & \\ & & & \overline{E_t}^{(k)} \end{pmatrix}. \tag{D.15}$$

Using Weyl's inequality in each block, we have

$$\sigma_r\big(\overline{Z_t}^{(j)}\overline{U_0}^{(j)} + \overline{E_t}^{(j)}\big) \geq \sigma_r\big(\overline{Z_t}^{(j)}\overline{U_0}^{(j)}\big) - \|\overline{E_t}^{(j)}\| \geq \sigma_r\Big((\overline{V_{\mathcal{L}}}^{(j)})^{\mathrm{H}}\overline{Z_t}^{(j)}\overline{U_0}^{(j)}\Big) - \|\overline{E_t}^{(j)}\|.$$

Now, for the singular value above we get the following estimation

$$\begin{aligned}
\sigma_r\Big((\overline{V_{\mathcal{L}}}^{(j)})^{\mathrm{H}}\overline{Z_t}^{(j)}\overline{U_0}^{(j)}\Big) &= \sigma_{min}\Big(\overline{V_{\mathcal{L}}}^{(j)\mathrm{H}}\overline{Z_t}^{(j)}V_{\mathcal{L}}^{(j)}V_{\mathcal{L}}^{(j)\mathrm{H}}\overline{U_0}^{(j)}\Big) \\
&\geq \sigma_{min}\Big(\overline{V_{\mathcal{L}}}^{(j)\mathrm{H}}\overline{Z_t}^{(j)}\overline{V_{\mathcal{L}}}^{(j)}\Big)\sigma_{min}\Big(\overline{V_{\mathcal{L}}}^{(j)\mathrm{H}}\overline{U_0}^{(j)}\Big) \\
&= \sigma_r(\overline{Z_t}^{(j)})\sigma_{min}\Big(\overline{V_{\mathcal{L}}}^{(j)\mathrm{H}}\overline{U_0}^{(j)}\Big) \geq \alpha\sigma_r(\overline{Z_t}^{(j)})\sigma_{min}\Big(\overline{V_{\mathcal{L}}}^{(j)\mathrm{H}}\overline{U}^{(j)}\Big) \\
&= \alpha\sigma_r(\overline{Z_t}^{(j)})\sigma_{min}\Big(\overline{V_{\mathcal{L}}^{\mathrm{H}}}^{(j)}\overline{U}^{(j)}\Big) \geq \alpha\sigma_r(\overline{Z_t}^{(j)})\sigma_{min}\Big(\overline{\mathcal{V}_{\mathcal{L}}^\top * \mathcal{U}}\Big)
\end{aligned}$$

where in the last line we used that for each tensor it holds in the Fourier domain $\overline{V_{\mathcal{L}}}^{(j)\mathrm{H}} = \overline{\mathcal{V}_{\mathcal{L}}^\mathrm{T}}^{(j)}$.

To show inequality (D.13), we can use Weyl's bounds and then the Courant-Fisher theorem, which leads to

$$\begin{aligned}
\sigma_{r+1}\big(\overline{Z_t}^{(j)}\overline{U_0}^{(j)} + \overline{E_t}^{(j)}\big) &\leq \sigma_{r+1}\big(\overline{Z_t}^{(j)}\overline{U_0}^{(j)}\big) + \|\overline{E_t}^{(j)}\| \leq \sigma_{r+1}\big(\overline{Z_t}^{(j)}\overline{U_0}^{(j)}\big) + \|\mathcal{E}_t\| \\
&\leq \sigma_{r+1}\big(\overline{Z_t}^{(j)}\big)\|\overline{U_0}^{(j)}\| + \|\mathcal{E}_t\| \leq \alpha\sigma_{r+1}\big(\overline{Z_t}^{(j)}\big)\|\mathcal{U}\| + \|\mathcal{E}_t\|.
\end{aligned}$$

Now, for estimation of $\|\mathcal{V}_{\mathcal{L}}^\perp * \mathcal{V}_{\mathcal{L}_t}\|$, let us recall that $\mathcal{L}$ is the tensor column subspace spanned by the tensor columns corresponding to the first $r$ singular tubes of tensor $\mathcal{Z}_t = (\mathcal{I} - \mu\mathcal{M})^{*t} \in \mathbb{R}^{n \times n \times k}$, and $\mathcal{L}_t$ is the tensor-column subspace

spanned by the tensor-columns corresponding to the first $r$ singular tubes of the gradient descent iterates $\mathcal{U}_t = \mathcal{Z}_t * \mathcal{U}_0 + \mathcal{E}_t$, and consider Fourier-domain representation (D.15) of $\mathcal{U}_t$. Here, for each $1 \leq j \leq k$, the matrices $\overline{Z_t}^{(j)}\overline{U_0}^{(j)} + \overline{E_t}^{(j)}$ can be represented as

$$\underbrace{\overline{Z_t}^{(j)}\overline{U_0}^{(j)} + \overline{E_t}^{(j)}}_{\widetilde{A}^{(j)}} = \underbrace{\overline{Z_t}^{(j)}\overline{V_{\mathcal{L}}}^{(j)}\overline{V_{\mathcal{L}}}^{(j)\,\mathrm{H}}\overline{U_0}^{(j)}}_{A^{(j)}} + \underbrace{\overline{Z_t}^{(j)}\overline{V_{\mathcal{L}^\perp}}^{(j)}\overline{V_{\mathcal{L}^\perp}}^{(j)\,\mathrm{H}}\overline{U_0}^{(j)} + \overline{E_t}^{(j)}}_{C^{(j)}}. \tag{D.16}$$

As the tensor-column space $\mathcal{V}_{\mathcal{L}}$ is $r$-dimensional, each of matrices $\overline{V_{\mathcal{L}}}^{(j)}$ has rank $r$, see (Gleich et al., 2013). Since the matrices $\overline{Z_t}^{(j)}$ can be decomposed as

$$\overline{Z_t}^{(j)} = \overline{V_{\mathcal{L}}}^{(j)}\Sigma_{\mathcal{L}}^{(j)}\overline{V_{\mathcal{L}}}^{(j)\,\mathrm{H}} + \overline{V_{\mathcal{L}^\perp}}^{(j)}\Sigma_{\mathcal{L}^\perp}^{(j)}\overline{V_{\mathcal{L}^\perp}}^{(j)\,\mathrm{H}}$$

we have that

$$\overline{Z_t}^{(j)}\overline{V_{\mathcal{L}}}^{(j)}\overline{V_{\mathcal{L}}}^{(j)\,\mathrm{H}}\overline{U_0}^{(j)} = \overline{V_{\mathcal{L}}}^{(j)}\Sigma_{\mathcal{L}}^{(j)}\overline{V_{\mathcal{L}}}^{(j)\,\mathrm{H}}\overline{U_0}^{(j)}. \tag{D.17}$$

As $\overline{U_0}^{(j)} \in \mathbb{C}^{r \times R}$ has rank $r$, $\overline{V_{\mathcal{L}}}^{(j)\,\mathrm{H}}\overline{U_0}^{(j)}$ has rank $r$, which means that the product above has rank $r$ too. Due to (D.17), we see that

$$\overline{Z_t}^{(j)}\overline{V_{\mathcal{L}}}^{(j)}\overline{V_{\mathcal{L}}}^{(j)\,\mathrm{H}}\overline{U_0}^{(j)} = \overline{V_{\mathcal{L}}}^{(j)}\overline{V_{\mathcal{L}}}^{(j)\,\mathrm{H}}\overline{Z_t}^{(j)}\overline{V_{\mathcal{L}}}^{(j)}\overline{V_{\mathcal{L}}}^{(j)\,\mathrm{H}}\overline{U_0}^{(j)},$$

which makes $\overline{V_{\mathcal{L}}}^{(j)}$ to the column subspace of $\overline{Z_t}^{(j)}\overline{V_{\mathcal{L}}}^{(j)}\overline{V_{\mathcal{L}}}^{(j)\,\mathrm{H}}\overline{U_0}^{(j)}$. Considering the gap between the singular values of for matrices $A^{(j)}$ and $\widetilde{A}^{(j)}$ in (D.16), namely $\delta^{(j)} = \sigma_r(A^{(j)}) - \sigma_{r+1}(\widetilde{A}^{(j)})$, and using Wedin's $\sin\theta$ theorem (Wedin, 1972), for each $1 \leq j \leq k$ we get

$$\|\overline{V_{\mathcal{L}^\perp}}^{(j)\,\mathrm{H}}\overline{V_{\mathcal{L}_t}}^{(j)}\| \leq \frac{\|C^{(j)}\|}{\delta^{(j)}}.$$

To conduct a further estimation of $\|\overline{V_{\mathcal{L}^\perp}}^{(j)\,\mathrm{H}}\overline{V_{\mathcal{L}_t}}^{(j)}\|$, we analyze lower and upper bounds for the denominator and the numerator above. We start with the denominator first

$$\delta^{(j)} = \sigma_r(A^{(j)}) - \sigma_{r+1}(\widetilde{A}^{(j)})$$
$$= \sigma_r(\overline{Z_t}^{(j)}\overline{V_{\mathcal{L}}}^{(j)}\overline{V_{\mathcal{L}}}^{(j)\,\mathrm{H}}\overline{U_0}^{(j)}) - \sigma_{r+1}(\overline{Z_t}^{(j)}\overline{U_0}^{(j)} + \overline{E_t}^{(j)}).$$

Using properties of singular values of the matrix product for the first term above and Weyl's bound for the second term, we get

$$\delta^{(j)} \geq \sigma_r(\overline{Z_t}^{(j)})\sigma_{min}\left(\overline{V_{\mathcal{L}}}^{(j)\,\mathrm{H}}\overline{U_0}^{(j)}\right) - \sigma_{r+1}\left(\overline{Z_t}^{(j)}\overline{U_0}^{(j)}\right) - \|\overline{E_t}^{(j)})\|$$
$$\geq \sigma_r(\overline{Z_t}^{(j)})\sigma_{min}\left(\overline{\mathcal{V}_{\mathcal{L}}^\top * \mathcal{U}_0}\right) - \sigma_{r+1}\left(\overline{Z_t}^{(j)}\overline{U_0}^{(j)}\right) - \|\mathcal{E}_t\|. \tag{D.18}$$

For the norm of $C^{(j)}$, the following upper bound can be established

$$\|C^{(j)}\| \leq \|\overline{Z_t}^{(j)}\overline{V_{\mathcal{L}^\perp}}^{(j)}\overline{V_{\mathcal{L}^\perp}}^{(j)\,\mathrm{H}}\overline{U_0}^{(j)}\| + \|\overline{E_t}^{(j)}\|$$
$$\leq \|\overline{Z_t}^{(j)}\overline{V_{\mathcal{L}^\perp}}^{(j)}\overline{V_{\mathcal{L}^\perp}}^{(j)\,\mathrm{H}}\|\|\overline{U_0}^{(j)}\| + \|\mathcal{E}_t\|$$
$$\leq \alpha\sigma_{r+1}(\overline{Z_t}^{(j)})\|\mathcal{U}\| + \|\mathcal{E}_t\| \tag{D.19}$$

Now, combining bounds (D.18) and (D.19), one obtains that

$$\|\mathcal{V}_{\mathcal{L}^\perp}^\top * \mathcal{V}_{\mathcal{L}_t}\| = \max_{1 \leq j \leq k}\|\overline{V_{\mathcal{L}^\perp}}^{(j)\,\mathrm{H}}\overline{V_{\mathcal{L}_t}}^{(j)}\| \leq \max_{1 \leq j \leq k}\frac{\alpha\sigma_{r+1}(\overline{Z_t}^{(j)})\|\mathcal{U}\| + \|\mathcal{E}_t\|}{\sigma_r(\overline{Z_t}^{(j)})\sigma_{min}\left(\overline{\mathcal{V}_{\mathcal{L}}^\top * \mathcal{U}}\right) - \sigma_{r+1}\left(\overline{Z_t}^{(j)}\overline{U}^{(j)}\right) - \|\mathcal{E}_t\|} :$$

Using in the denominator the fact that $\sigma_{r+1}\left(\overline{Z_t}^{(j)}\overline{U_0}^{(j)}\right) \leq \alpha\sigma_{r+1}(\overline{Z_t}^{(j)})\|\overline{U}^{(j)}\| \leq \alpha\sigma_{r+1}(\overline{Z_t}^{(j)})\|\mathcal{U}\|$ finishes the proof of this lemma. $\qquad\square$

Further, we consider the gradient descent iterates with its t-SVD

$$\boldsymbol{\mathcal{U}}_t = \sum_{s=1}^{R} \boldsymbol{\mathcal{V}}_{\boldsymbol{\mathcal{U}}_t}(:,s,:) * \Sigma_{\boldsymbol{\mathcal{U}}_t}(s,s,:) * \boldsymbol{\mathcal{W}}_{\boldsymbol{\mathcal{U}}_t}^{\top}(:,s,:)$$

and the corresponding Fourier domain representation $\overline{\boldsymbol{\mathcal{U}}}_t = \mathrm{diag}(\overline{U_t}^{(1)}, \overline{U_t}^{(2)}, \ldots, \overline{U_t}^{(k)})$, where $\overline{U_t}^{(j)} = \sum_{\ell=1}^{R} \sigma_{\ell}^{(j)} v_{\ell}^{(j)} w_{\ell}^{(j)\mathrm{H}} = V_{U_t}^{(j)} \Sigma_{U_t}^{(j)} W_{U_t}^{(j)\mathrm{H}}$ and its signal-noise term decomposition

$$\boldsymbol{\mathcal{U}}_t = \boldsymbol{\mathcal{U}}_t * \boldsymbol{\mathcal{W}}_t * \boldsymbol{\mathcal{W}}_t^{\top} + \boldsymbol{\mathcal{U}}_t * \boldsymbol{\mathcal{W}}_{t,\perp} * \boldsymbol{\mathcal{W}}_{t,\perp}^{\top}.$$

We also define the corresponding new tensors

$$\boldsymbol{\mathcal{L}}_t = \sum_{s=1}^{r} \boldsymbol{\mathcal{V}}_{\boldsymbol{\mathcal{U}}_t}(:,s,:) * \Sigma_{\boldsymbol{\mathcal{U}}_t}(s,s,:) * \boldsymbol{\mathcal{W}}_{\boldsymbol{\mathcal{L}}_t}^{\top}(:,s,:) \tag{D.20}$$

$$\boldsymbol{\mathcal{N}}_t = \sum_{s=r+1}^{R} \boldsymbol{\mathcal{V}}_{\boldsymbol{\mathcal{U}}_t}(:,s,:) * \Sigma_{\boldsymbol{\mathcal{U}}_t}(s,s,:) * \boldsymbol{\mathcal{W}}_{\boldsymbol{\mathcal{U}}_t}^{\top}(:,s,:) \tag{D.21}$$

and their Fourier domain representations

$$\overline{\boldsymbol{\mathcal{L}}}_t = \mathrm{diag}(\overline{L_t}^{(1)}, \overline{L_t}^{(2)}, \ldots, \overline{L_t}^{(k)}), \qquad \overline{L_t}^{(j)} = \sum_{\ell=1}^{r} \sigma_{\ell}^{(j)} v_{\ell}^{(j)} w_{\ell}^{(j)\mathrm{H}} = V_{\boldsymbol{\mathcal{L}}_t}^{(j)} \Sigma_{\boldsymbol{\mathcal{L}}_t}^{(j)} W_{\boldsymbol{\mathcal{L}}_t}^{(j)\mathrm{H}} \tag{D.22}$$

$$\overline{\boldsymbol{\mathcal{N}}}_t = \mathrm{diag}(\overline{N_t}^{(1)}, \overline{N_t}^{(2)}, \ldots, \overline{N_t}^{(k)}), \qquad \overline{N_t}^{(j)} = \sum_{\ell=r+1}^{R} \sigma_{\ell}^{(j)} v_{\ell}^{(j)} w_{\ell}^{(j)\mathrm{H}} = V_{\boldsymbol{\mathcal{N}}_t}^{(j)} \Sigma_{\boldsymbol{\mathcal{N}}_t}^{(j)} W_{\boldsymbol{\mathcal{N}}_t}^{(j)\mathrm{H}} \tag{D.23}$$

**Lemma D.4.** *Assume* $\|\boldsymbol{\mathcal{V}}_{\boldsymbol{\mathcal{X}}^{\perp}}^{\top} * \boldsymbol{\mathcal{V}}_{\boldsymbol{\mathcal{L}}_t}\| \leq \frac{1}{2}$. *Then it holds that*

$$\|\boldsymbol{\mathcal{W}}_{\boldsymbol{\mathcal{L}}_t^{\perp}}^{\top} * \boldsymbol{\mathcal{W}}_t\| \leq 2 \max_{1 \leq j \leq k} \frac{\sigma_{r+1}\left(\overline{U_t}^{(j)}\right)}{\sigma_r\left(\overline{U_t}^{(j)}\right)} \|\boldsymbol{\mathcal{V}}_{\boldsymbol{\mathcal{X}}^{\perp}}^{\top} * \boldsymbol{\mathcal{V}}_{\boldsymbol{\mathcal{L}}_t}\|. \tag{D.24}$$

*Proof.* Consider $\|\boldsymbol{\mathcal{W}}_{\boldsymbol{\mathcal{L}}_t^{\perp}}^{\mathrm{T}} * \boldsymbol{\mathcal{W}}_t\| = \max_{1 \leq j \leq k} \|\overline{W_{\boldsymbol{\mathcal{L}}_t^{\perp}}}^{(j)\mathrm{H}} \overline{W_t}^{(j)}\|$. For each $1 \leq j \leq k$, we can now exploit the results of Lemma A.1 in (Stöger & Soltanolkotabi, 2021), to get that

$$\|(\overline{W_{\boldsymbol{\mathcal{L}}_t^{\perp}}^{\top}})^{(j)} \overline{W_t}^{(j)}\| \leq \frac{\|\Sigma_{\boldsymbol{\mathcal{N}}_t}^{(j)}\| \|\overline{V_{\boldsymbol{\mathcal{N}}_t}^{\mathrm{H}}}^{(j)} \overline{V_{\boldsymbol{\mathcal{X}}}}^{(j)}\|}{\sigma_{min}\left(\overline{V_{\boldsymbol{\mathcal{X}}}}^{(j)} \overline{U_t}^{(j)}\right)} \quad \text{and} \quad \sigma_{min}(\overline{V_{\boldsymbol{\mathcal{X}}}}^{(j)} \overline{U_t}^{(j)}) \geq \frac{\sigma_{min}(\overline{L_t}^{(j)})}{2}.$$

From here, we can proceed as follows

$$\begin{aligned} \|\boldsymbol{\mathcal{W}}_{\boldsymbol{\mathcal{L}}_t^{\perp}}^{\top} * \boldsymbol{\mathcal{W}}_t\| &= \max_{1 \leq j \leq k} \|\overline{W_{\boldsymbol{\mathcal{L}}_t^{\perp}}^{\mathrm{H}}}^{(j)} \overline{W_t}^{(j)}\| \leq 2 \max_{1 \leq j \leq k} \frac{\|\Sigma_{\boldsymbol{\mathcal{N}}_t}^{(j)}\| \|\overline{V_{\boldsymbol{\mathcal{N}}_t}^{\mathrm{H}}}^{(j)} \overline{V_{\boldsymbol{\mathcal{X}}}}^{(j)}\|}{\sigma_{min}(\overline{L_t}^{(j)})} \\ &= 2 \max_{1 \leq j \leq k} \frac{\sigma_{r+1}(\overline{U_t}^{(j)}) \|\overline{V_{\boldsymbol{\mathcal{N}}_t}^{\mathrm{H}}}^{(j)} \overline{V_{\boldsymbol{\mathcal{X}}}}^{(j)}\|}{\sigma_r(\overline{U_t}^{(j)})} \leq 2 \max_{1 \leq j \leq k} \frac{\sigma_{r+1}(\overline{U_t}^{(j)})}{\sigma_r(\overline{U_t}^{(j)})} \|\boldsymbol{\mathcal{V}}_{\boldsymbol{\mathcal{L}}_t^{\perp}}^{\top} * \boldsymbol{\mathcal{V}}_{\boldsymbol{\mathcal{X}}}\| \\ &= 2 \max_{1 \leq j \leq k} \frac{\sigma_{r+1}(\overline{U_t}^{(j)})}{\sigma_r(\overline{U_t}^{(j)})} \|\boldsymbol{\mathcal{V}}_{\boldsymbol{\mathcal{X}}^{\perp}}^{\top} * \boldsymbol{\mathcal{V}}_{\boldsymbol{\mathcal{L}}_t}\|, \end{aligned}$$

which concludes the proof. $\square$

**Lemma D.5.** *Assume that* $\|\boldsymbol{\mathcal{V}}_{\boldsymbol{\mathcal{X}}^{\perp}}^{\top} * \boldsymbol{\mathcal{V}}_{\boldsymbol{\mathcal{L}}_t}\| \leq \frac{1}{8}$ *for some* $t \geq 1, t \in \mathbb{N}$. *Then for each* $1 \leq j \leq k$, *it holds that*

$$\sigma_r\left(\overline{\boldsymbol{\mathcal{U}}_t * \boldsymbol{\mathcal{W}}_t}^{(j)}\right) \geq \frac{1}{2}\sigma_r\left(\overline{\boldsymbol{\mathcal{U}}_t}^{(j)}\right) \tag{D.25}$$

$$\sigma_1(\overline{\boldsymbol{\mathcal{U}}_t * \boldsymbol{\mathcal{W}}_{t,\perp}}^{(j)}) \leq 2\sigma_{r+1}(\overline{U_t}^{(j)}). \tag{D.26}$$

*Moreover, the principal angles between the tensor-column subspaces spanned by $\mathcal{X}$ and $\mathcal{U}_t \mathcal{W}_t$ can be estimated as follows*

$$\|\mathcal{V}_{\mathcal{X}^\perp} * \mathcal{V}_{\mathcal{U}_t \mathcal{W}_t}\| \leq 7\|\mathcal{V}_{\mathcal{X}^\perp}^\top * \mathcal{V}_{\mathcal{L}_t}\| \tag{D.27}$$

$$\|\mathcal{U}_t * \mathcal{W}_{t,\perp}\| \leq 2 \max_{1 \leq j \leq k} \sigma_{r+1}(\overline{U}_t^{(j)}). \tag{D.28}$$

*Proof.* We assume that $\|\mathcal{V}_{\mathcal{X}^\perp}^\top * \mathcal{V}_{\mathcal{L}_t}\| \leq \frac{1}{8}$, then due to Lemma D.4, we obtain that

$$\|\mathcal{W}_{\mathcal{L}_t^\perp}^\top * \mathcal{W}_t\| \leq 2 \max_{1 \leq j \leq k} \frac{\sigma_{r+1}\left(\overline{U}_j^{(j)}\right)}{\sigma_r\left(\overline{U}_j^{(j)}\right)} \|\mathcal{V}_{\mathcal{X}^\perp}^\top * \mathcal{V}_{\mathcal{L}_t}\| \leq \frac{1}{4}. \tag{D.29}$$

Now, to estimate $\sigma_r\left(\overline{\mathcal{U}_t * \mathcal{W}_t}^{(j)}\right)$, we see that for each $1 \leq j \leq k$, it holds that

$$\sigma_r\left(\overline{\mathcal{U}_t * \mathcal{W}_t}^{(j)}\right)^2 = \sigma_r\left(\left(\overline{\mathcal{U}_t * \mathcal{W}_t}^{(j)}\right)^{\mathrm{H}} \overline{\mathcal{U}_t * \mathcal{W}_t}^{(j)}\right) = \sigma_r\left(\overline{W}_t^{(j)\mathrm{H}} \overline{U}_t^{(j)\mathrm{H}} \overline{U}_t^{(j)} \overline{W}_t^{(j)}\right) \tag{D.30}$$

Since $\overline{U}_t^{(j)\mathrm{H}} \overline{U}_t^{(j)} = \overline{L}_t^{(j)\mathrm{H}} \overline{L}_t^{(j)} + \overline{N}_t^{(j)\mathrm{H}} \overline{N}_t^{(j)}$, we get that

$$\sigma_r\left(\overline{\mathcal{U}_t * \mathcal{W}_t}^{(j)}\right)^2 \geq \sigma_r\left(\overline{W}_t^{(j)\mathrm{H}} \overline{L}_t^{(j)\mathrm{H}} \overline{L}_t^{(j)} \overline{W}_t^{(j)}\right) = \sigma_r\left(\overline{W}_t^{(j)\mathrm{H}} \overline{L}_t^{(j)}\right)^2$$

$$\geq \sigma_r\left(\overline{W}_t^{(j)\mathrm{H}} W_{\overline{L}_t^{(j)}}\right)^2 \sigma_r\left(\overline{L}_t^{(j)}\right)^2 \geq (1 - \|\mathcal{W}_{\mathcal{L}_t^\perp} * \mathcal{W}_t^T\|^2)\sigma_r\left(\overline{U}_t^{(j)}\right)^2,$$

where in the last line we used the definition of the principal angle between tensor column subspaces and the corresponding properties in their Fourier domain slices, namely

$$\sigma_r\left(\overline{W}_t^{(j)\mathrm{H}} W_{\overline{L}_t^{(j)}}\right)^2 = 1 - \|\overline{W}_t^{(j)\mathrm{H}} W_{\overline{L}_t^{(j)}}^\perp\|^2 \geq 1 - \max_{1 \leq j \leq k} \|\overline{W}_t^{(j)\mathrm{H}} W_{\overline{L}_t^{(j)}}^\perp\|^2 = 1 - \|\mathcal{W}_{\mathcal{L}_t^\perp} * \mathcal{W}_t^T\|^2.$$

Due to our assumption $\|\mathcal{V}_{\mathcal{X}^\perp}^\top * \mathcal{V}_{\mathcal{L}_t}\| \leq \frac{1}{8}$, we can see that in the Fourier domain, the subspaces spanned by $\overline{V}_{\mathcal{X}_t^\perp}^{(j)}$ and $\overline{V}_{\mathcal{L}_t}^{(j)} = V_{\overline{L}_t^{(j)}}$ are close enough. Then, decomposing $\overline{U}_t^{(j)}$ into two different ways, namely as

$$\overline{U}_t^{(j)} = \sum_{\ell=1}^R \sigma_\ell^{(j)} v_\ell^{(j)} w_\ell^{(j)\mathrm{H}} = \overline{L}_t^{(j)} + \overline{N}_t^{(j)}$$

and as

$$\overline{U}_t^{(j)} = \overline{U}_t^{(j)} \overline{W}_t^{(j)} \overline{W}_t^{(j)\mathrm{H}} + \overline{U}_t^{(j)} \overline{W}_{t,\perp}^{(j)} \overline{W}_{t,\perp}^{(j)\mathrm{H}},$$

according to Lemma H.1, one obtains for each $1 \leq j \leq k$ that

$$\|\overline{V}_{\mathcal{X}_t^\perp}^{(j)\mathrm{H}} V_{\overline{U}_t^{(j)} \overline{W}_t^{(j)}}\| \leq 7\|\overline{V}_{\mathcal{X}_t^\perp}^{(j)\mathrm{H}} \overline{V}_{\mathcal{L}_t}^{(j)}\|$$

$$\|\overline{U}_t^{(j)} \overline{W}_{t,\perp}^{(j)}\| \leq 2\sigma_{r+1}(\overline{U}_t^{(j)}),$$

where the last inequality is equivalent to $\sigma_1(\overline{\mathcal{U}_t * \mathcal{W}_{t,\perp}}^{(j)}) \leq 2\sigma_{r+1}(\overline{U}_t^{(j)})$. According to the definition of principal angles between tensor subspaces, this implies that

$$\|\mathcal{V}_{\mathcal{X}^\perp}^\top * \mathcal{V}_{\mathcal{U}_t * \mathcal{W}_t}\| = \max_j \|\overline{V}_{\mathcal{X}_t^\perp}^{(j)\mathrm{H}} V_{\overline{U}_t^{(j)} \overline{W}_t^{(j)}}\| \leq 7 \max_j \|\overline{V}_{\mathcal{X}_t^\perp}^{(j)\mathrm{H}} \overline{V}_{\mathcal{L}_t}^{(j)}\| = 7\|\mathcal{V}_{\mathcal{X}^\perp}^\top * \mathcal{V}_{\mathcal{L}_t}\|.$$

In the same way, $\|\mathcal{U}_t * \mathcal{W}_{t,\perp}\| = \max_j \|\overline{U}_t^{(j)} \overline{W}_{t,\perp}^{(j)}\| \leq 2 \max_j \sigma_{r+1}(\overline{U}_t^{(j)})$, which finishes the proof. $\square$

**Lemma D.6.** *Consider a tensor $\mathcal{T} := \mathcal{X} * \mathcal{X}^\top \in S_+^{n \times n \times k}$ with tubal rank $r \leq n$. Assume that measurement operator $\mathcal{A}$ is such that*

$$\mathcal{M} = \mathcal{A}^* \mathcal{A}(\mathcal{T}) = \mathcal{T} + \mathcal{E} \quad \in S_+^{n \times n \times k}$$

*and for for each $1 \leq j \leq k$ one has $\|\overline{E}^{(j)}\| \leq \delta \lambda_r(\overline{\mathcal{T}}^{(j)})$ with $\delta \leq \frac{1}{4}$. For the same $\mathcal{M}$ with its t-SVD $\mathcal{M} = \mathcal{V}_{\mathcal{M}} * \Sigma_{\mathcal{M}} * \mathcal{W}_{\mathcal{M}}^{\top}$, let $\mathcal{L} \in \mathbb{R}^{n \times r \times k}$ denote the tensor column subspace spanned by the tensor-columns corresponding to the first $r$ singular tubes, that is $\mathcal{L} := \mathcal{V}_{\mathcal{M}}(:, 1:r, :) \in \mathbb{R}^{n \times r \times k}$.*

*Then, in each Fourier slice $j$, $1 \leq j \leq k$, it holds that*

$$(1 - \delta)\lambda_1(\overline{T}^{(j)}) \leq \lambda_1(\overline{M}^{(j)}) \leq (1 + \delta)\lambda_1(\overline{T}^{(j)}) \tag{D.31}$$

$$\lambda_{r+1}(\overline{M}^{(j)}) \leq \delta \lambda_r(\overline{T}^{(j)}) \tag{D.32}$$

$$\lambda_r(\overline{M}^{(j)}) \geq (1 - \delta)\lambda_r(\overline{T}^{(j)}), \tag{D.33}$$

*and*

$$(1 - \delta)\|\mathcal{T}\| \leq \|\mathcal{M}\| \leq (1 + \delta)\|\mathcal{T}\| \tag{D.34}$$

*Moreover, the tensor-column subspaces of $\mathcal{X}$ and $\mathcal{L}$ are aligned, namely*

$$\|\mathcal{V}_{\mathcal{X}^{\perp}}^{\top} * \mathcal{V}_{\mathcal{L}}\| \leq 2\delta \tag{D.35}$$

*Proof.* Consider tensor $\mathcal{T} := \mathcal{X} * \mathcal{X}^{\top} \in S_{+}^{n \times n \times k}$. Due to the definition of tensor transpose and conjugate symmetry of Fourier coefficients (Kilmer & Martin, 2011), the Fourier slices of $\mathcal{T}$ are defined as $\overline{T}^{(j)} = \overline{X}^{(j)}\overline{X}^{(j)H}$. That is, each face of $\mathcal{T}$ is Hermitian and at least positive semidefinite. As we assume that for each $j$, $1 \leq j \leq k$, one has $\|\overline{E_t}^{(j)}\| \leq \delta \lambda_r(\overline{\mathcal{T}}^{(j)})$ using Weyl's inequality in each of the Fourier slices, we obtain the first three inequalities.

To show that the tensor subspace $\mathcal{V}_{\mathcal{X}}$ and $\mathcal{V}_{\mathcal{L}}$ are aligned, we use first the definition

$$\|\mathcal{V}_{\mathcal{X}^{\perp}}^{\top} * \mathcal{V}_{\mathcal{L}}\| = \max_{1 \leq j \leq k} \|\overline{V_{\mathcal{X}^{\perp}}^{(j)}}^{H} \overline{V_{\mathcal{L}}^{(j)}}\| \tag{D.36}$$

For the estimation of $\|\overline{V_{\mathcal{X}^{\perp}}^{(j)}}^{H} \overline{V_{\mathcal{L}}^{(j)}}\|$ in each of the Fourier slices, we apply Wedin's $\sin \Theta$ theorem. For this, denote $\mathcal{L} := \mathcal{V}_{\mathcal{M}}(:, 1:r, :) \in \mathbb{R}^{n \times r \times k}$ and let $\overline{V}_{\mathcal{L}}^{(j)}$ denote the corresponding Fourier slices of $\mathcal{L} \in \mathbb{R}^{n \times r \times k}$. Since in the Fourier space, it holds that $\overline{M}^{(j)} = \overline{T}^{(j)} + \overline{E}^{(j)}$ and $\overline{V}_{\mathcal{L}}^{(j)}$ encompasses the first $r$ eigenvectors of $\overline{M}^{(j)}$, from Wedin's $\sin \Theta$ theorem, we obtain

$$\|\overline{V_{\mathcal{X}^{\perp}}^{(j)}}^{H} \overline{V_{\mathcal{L}}^{(j)}}\| \leq \frac{\|\overline{E}^{(j)}\|}{\xi^{(j)}},$$

with $\xi^{(j)} := \lambda_r(\overline{T}^{(j)}) - \lambda_{r+1}(\overline{M}^{(j)})$. Using estimate (D.32), $\xi^{(j)}$ can be lower-bounded as

$$\xi^{(j)} := \lambda_r(\overline{T}^{(j)}) - \lambda_{r+1}(\overline{M}^{(j)}) \geq \lambda_r(\overline{T}^{(j)}) - \delta \lambda_r(\overline{T}^{(j)}) = (1 - \delta)\lambda_r(\overline{T}^{(j)}).$$

Using the bound the the assumptions that $\|\overline{E_t}^{(j)}\| \leq \delta \lambda_r(\overline{\mathcal{T}}^{(j)})$ and $\delta \leq \frac{1}{2}$, we get

$$\|\overline{V_{\mathcal{X}^{\perp}}^{(j)}}^{H} \overline{V_{\mathcal{L}}^{(j)}}\| \leq \frac{\delta}{1 - \delta} \leq 2\delta.$$

Coming back to equality (D.36), we obtain the stated bound for the principal angle between the two tensor column subspaces. $\qquad\square$

**Lemma D.7.** *Consider a tensor $\mathcal{X} * \mathcal{X}^{\top} \in S_{+}^{n \times n \times k}$ with tubal rank $r \leq n$. Assume that measurement operator $\mathcal{A}$ is such that*

$$\mathcal{M} = \mathcal{A}^{*}\mathcal{A}(\mathcal{X} * \mathcal{X}^{\top}) = \mathcal{X} * \mathcal{X}^{\top} + \mathcal{E}$$

*and for each, $j$, $1 \leq j \leq k$, one has $\|\overline{E}^{(j)}\| \leq \delta \lambda_r(\overline{X}^{(j)}\overline{X}^{(j)H})$ with $\delta \leq c_1$. Moreover, assume that for difference tensor $\mathcal{E}_t = \mathcal{U}_t - \widetilde{\mathcal{U}}_t$ it holds that*

$$\gamma := \frac{\alpha \max_{1 \leq j \leq k} \sigma_{r+1}(\overline{Z}_t^{(j)})\|\mathcal{U}\| + \|\mathcal{E}_t\|}{\min_{1 \leq j \leq k} \sigma_r(\overline{Z}_t^{(j)})} \frac{1}{\alpha \sigma_{min}(\mathcal{V}_{\mathcal{L}}^{\top} * \mathcal{U})} \leq c_2 \kappa^{-2}, \tag{D.37}$$

*where $c_1, c_2 > 0$ are sufficiently small absolute constants. Then for the signal and noise term of the gradient descent* (C.1), *we have*

$$\|\mathcal{V}_{\mathcal{X}^\perp}^\top * \mathcal{V}_{\mathcal{U}_t * \mathcal{W}_t}\| \leq 14(\delta + \gamma) \tag{D.38}$$

$$\|\mathcal{U}_t * \mathcal{W}_{t,\perp}\| \leq \frac{\kappa^{-2}}{8} \alpha \min_{1 \leq j \leq k} \sigma_r(\overline{Z}_t^{(j)}) \sigma_{min}(\overline{\mathcal{V}_{\mathcal{L}}^\top * \mathcal{U}}) \tag{D.39}$$

*and for each $j$, $1 \leq j \leq k$, it holds that*

$$\sigma_{min}(\overline{\mathcal{U}_t * \mathcal{W}_t}^{(j)}) \geq \frac{1}{4} \alpha \min_{1 \leq j \leq k} \sigma_r(\overline{Z}_t^{(j)}) \sigma_{min}(\overline{\mathcal{V}_{\mathcal{L}}^\top * \mathcal{U}}) \tag{D.40}$$

$$\sigma_1(\overline{\mathcal{U}_t * \mathcal{W}_{t,\perp}}^{(j)}) \leq \frac{\kappa^{-2}}{8} \alpha \min_{1 \leq j \leq k} \sigma_r(\overline{Z}_t^{(j)}) \sigma_{min}(\overline{\mathcal{V}_{\mathcal{L}}^\top * \mathcal{U}}) \tag{D.41}$$

*Proof.* To prove the above-stated properties, we will use Lemma D.3. Therefore, we start by checking the conditions of this lemma. Sufficiently small $c_2$ and the assumption $\gamma \leq c_2 \kappa^{-2}$ allows for $\gamma \leq \frac{1}{2}$. This means that

$$\frac{\alpha \max_{1 \leq j \leq k} \sigma_{r+1}(\overline{Z}_t^{(j)}) \|\mathcal{U}\| + \|\mathcal{E}_t\|}{\min_{1 \leq j \leq k} \sigma_r(\overline{Z}_t^{(j)})} \frac{1}{\alpha \sigma_{min}(\overline{\mathcal{V}_{\mathcal{L}}^\top * \mathcal{U}})} \leq \frac{1}{2}$$

and in each of the Fourier slices we have

$$\sigma_{r+1}(\overline{Z}_t^{(j)}) \|\mathcal{U}\| + \frac{\|\mathcal{E}_t\|}{\alpha} \leq \frac{1}{2} \sigma_r(\overline{Z}_t^{(j)}) \sigma_{min}(\overline{\mathcal{V}_{\mathcal{L}}^\top * \mathcal{U}}),$$

fulfilling the assumption of Lemma D.3. Hence, from Lemma D.3, we conclude that

$$\|\mathcal{V}_{\mathcal{L}^\perp}^\top * \mathcal{V}_{\mathcal{L}_t}\| \leq \max_{1 \leq j \leq k} \frac{\alpha \sigma_{r+1}(\overline{Z}_t^{(j)}) \|\mathcal{U}\| + \|\mathcal{E}_t\|}{\alpha \sigma_r(\overline{Z}_t^{(j)}) \sigma_{min}(\overline{\mathcal{V}_{\mathcal{L}}^\top * \mathcal{U}}) - \alpha \sigma_{r+1}(\overline{Z}_t^{(j)}) \|\mathcal{U}\| - \|\mathcal{E}_t\|} \tag{D.42}$$

$$\leq \frac{\alpha \max_{1 \leq j \leq k} \sigma_{r+1}(\overline{Z}_t^{(j)}) \|\mathcal{U}\| + \|\mathcal{E}_t\|}{\alpha \min_{1 \leq j \leq k} \sigma_r(\overline{Z}_t^{(j)}) \sigma_{min}(\overline{\mathcal{V}_{\mathcal{L}}^\top * \mathcal{U}}) - \alpha \max_{1 \leq j \leq k} \sigma_{r+1}(\overline{Z}_t^{(j)}) \|\mathcal{U}\| - \|\mathcal{E}_t\|}, \tag{D.43}$$

and, moreover, together with Lemma D.5 and the assumption $\gamma \leq \frac{1}{2}$ we get

$$\min_{1 \leq j \leq k} \sigma_r(\overline{U}_t^{(j)}) \geq \alpha \min_{1 \leq j \leq k} \sigma_r(\overline{Z}_t^{(j)}) \sigma_{min}(\overline{\mathcal{V}_{\mathcal{L}}^\top * \mathcal{U}}) - \|\mathcal{E}_t\| \geq \frac{\alpha}{2} \min_{1 \leq j \leq k} \sigma_r(\overline{Z}_t^{(j)}) \sigma_{min}(\overline{\mathcal{V}_{\mathcal{L}}^\top * \mathcal{U}}) \tag{D.44}$$

$$\max_{1 \leq j \leq k} \sigma_{r+1}(\overline{U}_t^{(j)}) \leq \alpha \min_{1 \leq j \leq k} \sigma_r \sigma_r(\overline{Z}_t^{(j)}) \|\mathcal{U}\| + \|\mathcal{E}_t\| \leq \alpha \gamma \min_{1 \leq j \leq k} \sigma_r(\overline{Z}_t^{(j)}) \sigma_{min}(\overline{\mathcal{V}_{\mathcal{L}}^\top * \mathcal{U}}) \tag{D.45}$$

The last two inequalities, allow extend bound (D.42) as follows

$$\|\mathcal{V}_{\mathcal{L}^\perp}^\top * \mathcal{V}_{\mathcal{L}_t}\| \leq \frac{\gamma}{1 - \gamma} \tag{D.46}$$

Now, consider the principal angle between $\mathcal{X}$ and $\mathcal{L}_t$ using its definition

$$\|\mathcal{V}_{\mathcal{X}^\perp}^\top * \mathcal{V}_{\mathcal{L}_t}\| = \max_{1 \leq j \leq k} \|\overline{V}_{\mathcal{X}^\perp}^{(j)}{}^{\mathrm{H}} \overline{V}_{\mathcal{L}_t}^{(j)}\| = \max_{1 \leq j \leq k} \|\overline{V}_{\mathcal{X}^\perp}^{(j)} \overline{V}_{\mathcal{X}^\perp}^{(j)\mathrm{H}} - \overline{V}_{\mathcal{L}_t}^{(j)} \overline{V}_{\mathcal{L}_t}^{(j)\mathrm{H}}\|$$

$$\leq \max_{1 \leq j \leq k} \|\overline{V}_{\mathcal{X}^\perp}^{(j)} \overline{V}_{\mathcal{X}^\perp}^{(j)\mathrm{H}} - \overline{V}_{\mathcal{L}_t}^{(j)} \overline{V}_{\mathcal{L}_t}^{(j)\mathrm{H}}\| \leq \max_{1 \leq j \leq k} \|\overline{V}_{\mathcal{X}^\perp}^{(j)} \overline{V}_{\mathcal{X}^\perp}^{(j)\mathrm{H}} - \overline{V}_{\mathcal{L}}^{(j)} \overline{V}_{\mathcal{L}}^{(j)\mathrm{H}}\| + \|\overline{V}_{\mathcal{L}}^{(j)} \overline{V}_{\mathcal{L}}^{(j)\mathrm{H}} - \overline{V}_{\mathcal{L}_t}^{(j)} \overline{V}_{\mathcal{L}_t}^{(j)\mathrm{H}}\|$$

$$\leq \max_{1 \leq j \leq k} \|\overline{V}_{\mathcal{X}^\perp}^{(j)} \overline{V}_{\mathcal{X}^\perp}^{(j)\mathrm{H}} - \overline{V}_{\mathcal{L}}^{(j)} \overline{V}_{\mathcal{L}}^{(j)\mathrm{H}}\| + \max_{1 \leq j \leq k} \|\overline{V}_{\mathcal{L}}^{(j)} \overline{V}_{\mathcal{L}}^{(j)\mathrm{H}} - \overline{V}_{\mathcal{L}_t}^{(j)} \overline{V}_{\mathcal{L}_t}^{(j)\mathrm{H}}\|$$

$$= \|\mathcal{V}_{\mathcal{X}^\perp}^\top * \mathcal{V}_{\mathcal{L}}\| + \|\mathcal{V}_{\mathcal{L}^\perp}^\top * \mathcal{V}_{\mathcal{L}_t}\|$$

Using the last line above, and inequalities (D.35) and (D.46), we obtain

$$\|\mathcal{V}_{\mathcal{X}^{\perp}}^{\top} * \mathcal{V}_{\mathcal{L}_t}\| \leq 2(\delta + \gamma).$$

From here, allowing $\delta$ and $\gamma$ to be such that $\|\mathcal{V}_{\mathcal{X}^{\perp}}^{\top} * \mathcal{V}_{\mathcal{L}_t}\| \leq \frac{1}{8}$, we can use Lemma D.5 to get

$$\|\mathcal{V}_{\mathcal{X}^{\perp}} * \mathcal{V}_{\mathcal{U}_t \mathcal{W}_t}\| \leq 7\|\mathcal{V}_{\mathcal{X}^{\perp}}^{\top} * \mathcal{V}_{\mathcal{L}_t}\| \leq 14(\delta + \gamma).$$

Furthermore, Lemma D.5 together with inequality (D.45) also results in

$$
\begin{aligned}
\sigma_1(\overline{\mathcal{U}_t * \mathcal{W}_{t,\perp}}^{(j)}) &\leq 2\sigma_{r+1}(\overline{U}_t^{(j)}) \\
&\leq 2\max_{1 \leq j \leq k} \sigma_{r+1}(\overline{U}_t^{(j)}) \\
&\leq 2\gamma\alpha \min_{1 \leq j \leq k} \sigma_r(\overline{Z}_t^{(j)})\sigma_{\min}(\overline{\mathcal{V}_{\mathcal{L}}^{\top} * \mathcal{U}}) \\
&\leq \frac{\kappa^{-2}}{8}\alpha \min_{1 \leq j \leq k} \sigma_r(\overline{Z}_t^{(j)})\sigma_{\min}(\overline{\mathcal{V}_{\mathcal{L}}^{\top} * \mathcal{U}})
\end{aligned}
$$

and for the spectral norm of $\mathcal{U}_t * \mathcal{W}_{t,\perp}$ we get

$$\|\mathcal{U}_t * \mathcal{W}_{t,\perp}\| \leq 2\max_{1 \leq j \leq k}\sigma_{r+1}(\overline{U}_t^{(j)}) \leq \frac{\kappa^{-2}}{8}\alpha \min_{1 \leq j \leq k} \sigma_r(\overline{Z}_t^{(j)})\sigma_{\min}(\overline{\mathcal{V}_{\mathcal{L}}^{\top} * \mathcal{U}}).$$

To conclude the proof, we see that Lemma D.5 together with inequality (D.44) provides for each $j$, $1 \leq j \leq k$, the following lower bound

$$\sigma_r\left(\overline{\mathcal{U}_t * \mathcal{W}_t}^{(j)}\right) \geq \frac{1}{2}\sigma_r\left(\overline{\mathcal{U}}_t^{(j)}\right) \geq \frac{\alpha}{4}\sigma_r(\overline{Z}_t^{(j)})\sigma_{min}(\overline{\mathcal{V}_{\mathcal{L}}^{\top} * \mathcal{U}}) \geq \frac{\alpha}{4}\min_{1 \leq j \leq k}\sigma_r(\overline{Z}_t^{(j)})\sigma_{min}(\overline{\mathcal{V}_{\mathcal{L}}^{\top} * \mathcal{U}}).$$

$\square$

The following lemma shows that for an appropriately chosen initialization, in the first new iteration, the tensor column subspaces between the signal term $\mathcal{U}_t * \mathcal{W}_t$ and the ground truth tensor $\mathcal{X}$ become aligned. Moreover, for each $1 \leq j \leq k$ there is a solid gap between the smallest singular values of the signal term and the largest singular values of the noise term.

**Lemma D.8.** *Assume $\mathcal{A} : S^{n \times n \times k} \to \mathbb{R}^m$ satisfies the S2NRIP($\delta_1$) for some constant $\delta_1 > 0$. Also, assume that*

$$\mathcal{M} := \mathcal{A}^* \mathcal{A}(\mathcal{X} * \mathcal{X}^{\top}) = \mathcal{X} * \mathcal{X}^{\top} + \mathcal{E}$$

*with $\|\overline{E}^{(j)}\| \leq \delta\lambda_r(\overline{X}^{(j)}\overline{X}^{(j)\mathrm{H}})$ for each $1 \leq j \leq k$ and $\delta \leq c_1\kappa^{-2}$.*

*Denote by $\mathcal{L}$ the tensor-columns corresponding to the first $r$ singular tubes in the t-SVD of $\mathcal{M}$, that is, $\mathcal{L} := \mathcal{V}_{\mathcal{M}}(:, 1:r, :) \in \mathbb{R}^{n \times r \times k}$, and define the initialization $\mathcal{U}_0 = \alpha\mathcal{U}$ with the coefficient $\alpha$ such that*

$$\alpha^2 \leq \frac{c\|\mathcal{X}\|^2}{12k\sqrt{\min\{n, R\}}\kappa^2\|\mathcal{U}\|^3}\left(\frac{2\kappa^2\|\mathcal{U}\|^3}{c_3\sigma_{min}(\overline{\mathcal{V}_{\mathcal{L}}^{\top} * \mathcal{U}})}\right)^{-48\kappa^2}\min\left\{\sigma_{min}(\overline{\mathcal{V}_{\mathcal{L}}^{\top} * \mathcal{U}}), \|\overline{\mathcal{U}}_0^{\mathrm{H}} v_1\|_{\ell_2}\right\} \tag{D.47}$$

*where $v_1 \in \mathbb{C}^{nk}$ is the leading eigenvector of matrix $\overline{\mathcal{M}} \in \mathbb{C}^{nk \times nk}$.*

*Assume that learning rate $\mu$ fulfils $\mu \leq c_3\kappa^{-2}\|\mathcal{X}\|^{-2}$, then after $t_{\star}$ iterations with*

$$t_{\star} \asymp \frac{1}{\mu \min_{1 \leq j \leq k} \sigma_r(\overline{X}^{(j)})^2}\ln\left(\frac{2\kappa^2\|\mathcal{U}\|}{c_3\sigma_{min}(\overline{\mathcal{V}_{\mathcal{L}}^{\top} * \mathcal{U})}}\right) \tag{D.48}$$

*it holds that*

$$\|\mathcal{U}_{t_{\star}}\| \leq 3\|\mathcal{X}\| \tag{D.49}$$

$$\|\mathcal{V}_{\mathcal{X}^{\perp}} * \mathcal{V}_{\mathcal{U}_{t_{\star}} * \mathcal{W}_{t_{\star}}}\| \leq c\kappa^{-2}. \tag{D.50}$$

*and for each $1 \leq j \leq k$, we have*

$$\sigma_r\left(\overline{\boldsymbol{\mathcal{U}}_{t_\star} * \boldsymbol{\mathcal{W}}_{t_\star}}^{(j)}\right) \geq \frac{1}{4}\alpha\beta \tag{D.51}$$

$$\sigma_1\left(\overline{\boldsymbol{\mathcal{U}}_{t_\star} * \boldsymbol{\mathcal{W}}_{t_\star,\perp}}^{(j)}\right) \leq \frac{\kappa^{-2}}{8}\alpha\beta \tag{D.52}$$

$$\tag{D.53}$$

*where $\beta$ satisfies $\sigma_{min}(\overline{\boldsymbol{\mathcal{V}}_{\boldsymbol{\mathcal{L}}}^\top * \boldsymbol{\mathcal{U}}}) \leq \beta \leq \sigma_{min}(\overline{\boldsymbol{\mathcal{V}}_{\boldsymbol{\mathcal{L}}}^\top * \boldsymbol{\mathcal{U}}}) \left(\frac{2\kappa^2\|\boldsymbol{\mathcal{U}}\|}{c_3\sigma_{min}(\overline{\boldsymbol{\mathcal{V}}_{\boldsymbol{\mathcal{L}}}^\top * \boldsymbol{\mathcal{U}}})}\right)^{16\kappa^2}$.*

*Proof.* For the proof of this lemma, we want to apply Lemma D.7. The first condition of Lemma D.7 is the following

$$\gamma := \frac{\alpha \max_{1 \leq j \leq k} \sigma_{r+1}(\overline{Z}_t^{(j)})\|\boldsymbol{\mathcal{U}}\| + \|\boldsymbol{\mathcal{E}}_t\|}{\min_{1 \leq j \leq k} \sigma_r(\overline{Z}_t^{(j)})} \frac{1}{\alpha\sigma_{min}(\overline{\boldsymbol{\mathcal{V}}_{\boldsymbol{\mathcal{L}}}^\top * \boldsymbol{\mathcal{U}}})} \leq c_2\kappa^{-2},$$

By the definition of $\gamma$, it is sufficient to show that

$$\max_{1 \leq j \leq k} \sigma_{r+1}(\overline{Z}_t^{(j)})\|\boldsymbol{\mathcal{U}}\| \leq \frac{c_3}{2\kappa^2} \min_{1 \leq j \leq k} \sigma_r(\overline{Z}_t^{(j)})\sigma_{min}(\overline{\boldsymbol{\mathcal{V}}_{\boldsymbol{\mathcal{L}}}^\top * \boldsymbol{\mathcal{U}}}) \tag{D.54}$$

and

$$\|\boldsymbol{\mathcal{E}}_t\| \leq \frac{c_3}{2\kappa^2}\alpha \min_{1 \leq j \leq k} \sigma_r(\overline{Z}_t^{(j)})\sigma_{min}(\overline{\boldsymbol{\mathcal{V}}_{\boldsymbol{\mathcal{L}}}^\top * \boldsymbol{\mathcal{U}}}). \tag{D.55}$$

Since for $\boldsymbol{\mathcal{Z}}_t = (\boldsymbol{\mathcal{I}} + \mu\boldsymbol{\mathcal{M}})^{*t}$ the transformation in the Fourier domain leads to the blocks

$$\overline{Z}_t^{(j)} = (\mathrm{Id} + \mu\overline{M}^{(j)})^t,$$

this means that inequality (D.54) is equivalent to

$$\frac{2\kappa^2\|\boldsymbol{\mathcal{U}}\|}{c_3\sigma_{min}(\overline{\boldsymbol{\mathcal{V}}_{\boldsymbol{\mathcal{L}}}^\top * \boldsymbol{\mathcal{U}}})} \leq \left(\frac{1 + \mu \min_{1 \leq j \leq k} \sigma_r(\overline{M}^{(j)})}{1 + \mu \max_{1 \leq j \leq k} \sigma_{r+1}(\overline{M}^{(j)})}\right)^t,$$

which can be further modified as

$$\ln\left(\frac{2\kappa^2\|\boldsymbol{\mathcal{U}}\|}{\sigma_{min}(\overline{\boldsymbol{\mathcal{V}}_{\boldsymbol{\mathcal{L}}}^\top * \boldsymbol{\mathcal{U}}})}\right) \leq t\ln\left(\frac{1 + \mu \min_{1 \leq j \leq k} \sigma_r(\overline{M}^{(j)})}{1 + \mu \max_{1 \leq j \leq k} \sigma_{r+1}(\overline{M}^{(j)})}\right).$$

Hence, if we take $t_\star$ as follows

$$t_\star := \left\lceil \ln\left(\frac{2\kappa^2\|\boldsymbol{\mathcal{U}}\|}{\sigma_{min}(\overline{\boldsymbol{\mathcal{V}}_{\boldsymbol{\mathcal{L}}}^\top * \boldsymbol{\mathcal{U}}})}\right) \Big/ \ln\left(\frac{1 + \mu \min_{1 \leq j \leq k} \sigma_r(\overline{M}^{(j)})}{1 + \mu \max_{1 \leq j \leq k} \sigma_{r+1}(\overline{M}^{(j)})}\right) \right\rceil \tag{D.56}$$

then condition (D.54) will be satisfied in each block in the Fourier domain. For convenience, we will further denote

$$\psi := \ln\left(\frac{2\kappa^2\|\boldsymbol{\mathcal{U}}\|}{\sigma_{min}(\overline{\boldsymbol{\mathcal{V}}_{\boldsymbol{\mathcal{L}}}^\top * \boldsymbol{\mathcal{U}}})}\right). \tag{D.57}$$

For the second part of Lemma D.7's condition, inequality (D.55), we will use Lemma D.1. To apply this Lemma, the condition $t_\star \leq t^\star$ needs to be satisfied. According to Lemma D.2

$$t^\star \geq \left\lceil \frac{\ln\left(\frac{\|\boldsymbol{\mathcal{M}}\| \cdot \|\overline{\boldsymbol{\mathcal{U}}}_0^{\mathrm{H}} v_1\|_{\ell_2}}{8(1+\delta_1\sqrt{k})\sqrt{k\min\{n,R\}}\alpha^3\|\boldsymbol{\mathcal{U}}\|^3}\right)}{2\ln(1 + \mu\|\boldsymbol{\mathcal{M}}\|)} \right\rceil \tag{D.58}$$

For $t_\star \leq t^\star$ to hold, it will be sufficient to check, e.g., the following condition

$$\frac{\psi}{\ln\left(\frac{1+\mu\min_{1\leq j\leq k}\sigma_r(\overline{M}^{(j)})}{1+\mu\max_{1\leq j\leq k}\sigma_{r+1}(\overline{M}^{(j)})}\right)} \leq \frac{1}{2} \cdot \frac{\ln\left(\frac{\|\boldsymbol{\mathcal{M}}\|\cdot\|\overline{\boldsymbol{\mathcal{U}}}_0^{\mathrm{H}}v_1\|_{\ell_2}}{8(1+\delta_1\sqrt{k})\sqrt{k\min\{n,R\}}\alpha^3\|\boldsymbol{\mathcal{U}}\|^3}\right)}{2\ln\left(1+\mu\|\boldsymbol{\mathcal{M}}\|\right)}.$$

To check this condition let us first analyze the expression $\ln\left(1+\mu\|\boldsymbol{\mathcal{M}}\|\right)/\ln\left(\frac{1+\mu\min_{1\leq j\leq k}\sigma_r(\overline{M}^{(j)})}{1+\mu\max_{1\leq j\leq k}\sigma_{r+1}(\overline{M}^{(j)})}\right)$ first. Using $\frac{x}{1+x} \leq \ln(1+x) \leq x$, we can upper bound the above expression as

$$\frac{\ln\left(1+\mu\|\boldsymbol{\mathcal{M}}\|\right)}{\ln\left(\frac{1+\mu\min_{1\leq j\leq k}\sigma_r(\overline{M}^{(j)})}{1+\mu\max_{1\leq j\leq k}\sigma_{r+1}(\overline{M}^{(j)})}\right)} \leq \frac{\|\boldsymbol{\mathcal{M}}\|(1+\mu\min_{1\leq j\leq k}\sigma_r(\overline{M}^{(j)}))}{\min_{1\leq j\leq k}\sigma_r(\overline{M}^{(j)}) - \max_{1\leq j\leq k}\sigma_{r+1}(\overline{M}^{(j)})} \tag{D.59}$$

From here, applying the PSD of the tensor representatives in the Fourier domain and the assumptions $\delta \leq \frac{1}{3}$ and $\mu \leq c_3\kappa^{-2}\|\boldsymbol{\mathcal{X}}\|^{-2}$ and Lemma D.6, we get

$$\frac{\|\boldsymbol{\mathcal{M}}\|(1+\min_{1\leq j\leq k}\sigma_r(\overline{M}^{(j)}))}{\min_{1\leq j\leq k}\sigma_r(\overline{M}^{(j)}) - \max_{1\leq j\leq k}\sigma_{r+1}(\overline{M}^{(j)})} \leq \frac{(1+\delta)\|\boldsymbol{\mathcal{T}}\|}{(1-2\delta)\lambda_r(\overline{T}^{(j)})}\left(1+c_3(1+\delta)\left(\frac{\lambda_1(\overline{X}^{(j)})}{\kappa\|\boldsymbol{\mathcal{X}}\|}\right)^2\right)$$

$$\leq \kappa^2\frac{(1+\delta)}{(1-2\delta)}(1+c_3(1+\delta)) \leq 8\kappa^2,$$

in the last line, we used the bound on $\delta$ and that $c_3$ can be taken small enough. This means

$$\frac{\ln\left(1+\mu\|\boldsymbol{\mathcal{M}}\|\right)}{\ln\left(\frac{1+\mu\min_{1\leq j\leq k}\sigma_r(\overline{M}^{(j)})}{1+\mu\max_{1\leq j\leq k}\sigma_{r+1}(\overline{M}^{(j)})}\right)} \leq 8\kappa^2. \tag{D.60}$$

Thus, to show that $t_\star \leq t^\star$, it is sufficient to tune the initialization factor $\alpha$ so that

$$\psi \cdot 32\kappa^2 \leq \ln\left(\frac{\|\boldsymbol{\mathcal{M}}\| \cdot \|\overline{\boldsymbol{\mathcal{U}}}_0^{\mathrm{H}}v_1\|_{\ell_2}}{8(1+\delta_1\sqrt{k})\sqrt{k\min\{n,R\}}\alpha^3\|\boldsymbol{\mathcal{U}}\|^3}\right).$$

or using the notation for $\phi$, this is equivalent to

$$\left(\frac{2\kappa^2\|\boldsymbol{\mathcal{U}}\|}{\sigma_{min}(\overline{\boldsymbol{\mathcal{V}}_{\mathcal{L}}^\top * \boldsymbol{\mathcal{U}})}\right)^{32\kappa^2} \leq \frac{\|\boldsymbol{\mathcal{M}}\| \cdot \|\overline{\boldsymbol{\mathcal{U}}}_0^{\mathrm{H}}v_1\|_{\ell_2}}{8(1+\delta_1\sqrt{k})\sqrt{k\min\{n,R\}}\alpha^3\|\boldsymbol{\mathcal{U}}\|^3}$$

Since $\|\overline{\boldsymbol{\mathcal{U}}}_0^{\mathrm{H}}v_1\|_{\ell_2}/\alpha = \|\overline{\boldsymbol{\mathcal{U}}}^{\mathrm{H}}v_1\|_{\ell_2}$, The last inequality is implied if

$$\alpha^2 \leq \left(\frac{2\kappa^2\|\boldsymbol{\mathcal{U}}\|}{\sigma_{min}(\overline{\boldsymbol{\mathcal{V}}_{\mathcal{L}}^\top * \boldsymbol{\mathcal{U}})}\right)^{-32\kappa^2} \frac{\|\boldsymbol{\mathcal{M}}\| \cdot \|\overline{\boldsymbol{\mathcal{U}}}^{\mathrm{H}}v_1\|_{\ell_2}}{8(1+\delta_1\sqrt{k})\sqrt{k\min\{n,R\}}\|\boldsymbol{\mathcal{U}}\|^3},$$

or if we set $\alpha$ even smaller using the fact that $(1+\delta_1\sqrt{k})\sqrt{k} \leq (1+\sqrt{k})\sqrt{k} \leq 2k$ and $\|\boldsymbol{\mathcal{M}}\| \geq \frac{2}{3}\|\boldsymbol{\mathcal{X}}\|^2$ and set the parameter $\alpha$ so that

$$\alpha^2 \leq \left(\frac{2\kappa^2\|\boldsymbol{\mathcal{U}}\|}{\sigma_{min}(\overline{\boldsymbol{\mathcal{V}}_{\mathcal{L}}^\top * \boldsymbol{\mathcal{U}})}\right)^{-32\kappa^2} \frac{\|\boldsymbol{\mathcal{X}}\|^2 \cdot \|\overline{\boldsymbol{\mathcal{U}}}^{\mathrm{H}}v_1\|_{\ell_2}}{24k\sqrt{\min\{n,R\}}\|\boldsymbol{\mathcal{U}}\|^3}.$$

Hence $t_\star \leq t^\star$ is satisfied and applying Lemma D.7, we get

$$\|\boldsymbol{\mathcal{E}}_{t_\star}\| \leq 8(1+\delta_1\sqrt{k})\sqrt{k\min\{n,R\}}\frac{\alpha^3}{\|\boldsymbol{\mathcal{M}}\|}\|\boldsymbol{\mathcal{U}}\|^3(1+\mu\|\boldsymbol{\mathcal{M}}\|)^{3t_\star} \tag{D.61}$$

Moreover, using $\|\mathcal{M}\| \geq \frac{2}{3}\|\mathcal{X}\|^2$ from Lemma D.6 with $\delta \leq 1/3$ and $(1 + \delta_1\sqrt{k})\sqrt{k} \leq 2k$ , we get

$$\|\mathcal{E}_{t_\star}\| \leq 12k\sqrt{\min\{n, R\}}\frac{\alpha^3}{\|\mathcal{X}\|^2}\|\mathcal{U}\|^3(1 + \mu\|\mathcal{M}\|)^{3t_\star}$$

Hence, using that $\overline{Z_t}^{(j)} = (\mathrm{Id} + \mu\overline{M}^{(j)})^t$ inequality (D.55) will be implied if

$$12k\sqrt{\min\{n, R\}}\frac{\alpha^3}{\|\mathcal{X}\|^2}\|\mathcal{U}\|^3(1 + \mu\|\mathcal{M}\|)^{3t_\star} \leq \frac{c_3}{2\kappa^2}\alpha \min_{1 \leq j \leq k} \sigma_r\left((\mathrm{Id} + \mu\overline{M}^{(j)})^{t_\star}\right)\sigma_{min}(\overline{\mathcal{V}_{\mathcal{L}}^\top * \mathcal{U}}),$$

which is equivalent to

$$\alpha^2 \leq c_3\frac{\|\mathcal{X}\|^2\sigma_{min}(\overline{\mathcal{V}_{\mathcal{L}}^\top * \mathcal{U}})}{12k\sqrt{\min\{n, R\}}\kappa^2\|\mathcal{U}\|^3}\frac{(1 + \mu\lambda_r(\overline{M}^{(j)}))^{t_\star}}{(1 + \mu\|\mathcal{M}\|)^{3t_\star}}, \tag{D.62}$$

for all $j$. To proceed further, let us analyze the last factor from above using the definition of $t_\star$. Note that

$$\frac{(1 + \mu\lambda_r(\overline{M}^{(j)}))^{t_\star}}{(1 + \mu\|\mathcal{M}\|)^{3t_\star}} = \exp\left(t_\star \ln\left(\frac{1 + \mu\lambda_r(\overline{M}^{(j)})}{(1 + \mu\|\mathcal{M}\|)^3}\right)\right) \geq \exp\left(-3t_\star \ln\left((1 + \mu\|\mathcal{M}\|)^3\right)\right)$$

Now, using the definition of $t_\star$, that is $t_\star = \left\lceil \psi / \ln\left(\frac{1 + \mu\min_{1 \leq j \leq k}\sigma_r(\overline{M}^{(j)}}{1 + \mu\max_{1 \leq j \leq k}\sigma_{r+1}(\overline{M}^{(j)})}\right)\right\rceil$ and inequality (D.60), we get

$$\exp\left(-3t_\star \ln\left((1 + \mu\|\mathcal{M}\|)^3\right)\right) \geq \exp\left(-48\psi\kappa^2\right) = \left(\frac{2\kappa^2\|\mathcal{U}\|}{c_3\sigma_{min}(\overline{\mathcal{V}_{\mathcal{L}}^\top * \mathcal{U}})}\right)^{-48\kappa^2} \tag{D.63}$$

Inserting this into inequality (D.62), we get

$$\alpha^2 \leq c_3\frac{\|\mathcal{X}\|^2\sigma_{min}(\overline{\mathcal{V}_{\mathcal{L}}^\top * \mathcal{U}})}{12\,k\sqrt{\min\{n, R\}}\kappa^2\|\mathcal{U}\|^3}\left(\frac{2\kappa^2\|\mathcal{U}\|}{c_3\sigma_{min}(\overline{\mathcal{V}_{\mathcal{L}}^\top * \mathcal{U}})}\right)^{-48\kappa^2}. \tag{D.64}$$

For such $\alpha$, we have shown that inequality (D.55) holds, and the condition of Lemma D.7 is fulfilled, which gives us

$$\|\mathcal{V}_{\mathcal{X}^\perp}^\top * \mathcal{V}_{\mathcal{U}_t * \mathcal{W}_t}\| \leq 14(\delta + \gamma) \leq c\kappa^{-2}, \tag{D.65}$$

where the last inequality follows from our assumption that $\delta \leq c_1\kappa^{-2}$ and $\mu \leq c_3\kappa^{-2}\|\mathcal{X}\|^{-2}$ and from setting the constants $c_1$ and $c_3$ small enough.

Moreover, for each $1 \leq j \leq k$, from Lemma D.7 it follows that

$$\sigma_{min}(\overline{\mathcal{U}_t * \mathcal{W}_t}^{(j)}) \geq \frac{1}{4}\alpha\beta, \tag{D.66}$$

$$\sigma_1(\overline{\mathcal{U}_t * \mathcal{W}_{t,\perp}}^{(j)}) \leq \frac{\kappa^{-2}}{8}\alpha\beta. \tag{D.67}$$

where $\beta := \min_{1 \leq j \leq k} \sigma_r(\overline{Z_t}^{(j)})\sigma_{min}(\overline{\mathcal{V}_{\mathcal{L}}^\top * \mathcal{U}})$.

In the remaining part, we will show that $t_\star$, $\beta$ and $\|\mathcal{U}_{t_\star}\|$ have the properties stated in the lemma.

Let us start with $t_\star$. Using the same inequalities for $\ln(1 + x)$ as above and Lemma D.6, one can show

$$\ln\left(\frac{1 + \mu\min_{1 \leq j \leq k}\sigma_r(\overline{M}^{(j)})}{1 + \mu\max_{1 \leq j \leq k}\sigma_{r+1}(\overline{M}^{(j)})}\right) \geq \frac{\mu\min_{1 \leq j \leq k}\sigma_r(\overline{M}^{(j)})}{1 + \mu\min_{1 \leq j \leq k}\sigma_r(\overline{M}^{(j)})} - \mu\max_{1 \leq j \leq k}\sigma_{r+1}(\overline{M}^{(j)}) \geq \frac{2}{3}\mu\min_{1 \leq j \leq k}\sigma_r(\overline{X}^{(j)})^2$$

and at the same time

$$\ln\left(\frac{1 + \mu\min_{1 \leq j \leq k}\sigma_r(\overline{M}^{(j)})}{1 + \mu\max_{1 \leq j \leq k}\sigma_{r+1}(\overline{M}^{(j)})}\right) \leq \ln\left(1 + \mu\min_{1 \leq j \leq k}\sigma_r(\overline{M}^{(j)})\right) \leq \mu\min_{1 \leq j \leq k}\sigma_r(\overline{M}^{(j)})$$

$$\leq \mu(1+\delta) \min_{1\leq j\leq k} \sigma_r(\overline{X}^{(j)})^2 \leq 4/3\mu \min_{1\leq j\leq k} \sigma_r(\overline{X}^{(j)})^2$$

which shows that, on the one hand,

$$\frac{1}{\ln\left(\frac{1+\mu \min_{1\leq j\leq k}\sigma_r(\overline{M}^{(j)})}{1+\mu \max_{1\leq j\leq k}\sigma_{r+1}(\overline{M}^{(j)})}\right)} \leq \frac{2}{3\mu} \max_{1\leq j\leq k} \frac{1}{\sigma_r(\overline{X}^{(j)})^2} = \frac{2}{3\mu \min_{1\leq j\leq k}\sigma_r(\overline{X}^{(j)})^2}$$

and on the other hand

$$\frac{1}{\ln\left(\frac{1+\mu \min_{1\leq j\leq k}\sigma_r(\overline{M}^{(j)})}{1+\mu \max_{1\leq j\leq k}\sigma_{r+1}(\overline{M}^{(j)})}\right)} \geq \frac{3}{4\mu \min_{1\leq j\leq k}\sigma_r(\overline{X}^{(j)})^2},$$

which shows the desired properties of $t_\star$.

Now, we consider $\beta := \min_{1\leq j\leq k} \sigma_r(\overline{Z_{t_\star}}^{(j)})\sigma_{min}(\overline{\mathcal{V}_{\mathcal{L}}^\top * \mathcal{U}})$. By the definition of $\overline{Z_t}^{(j)}$ and inequality (D.60), we get

$$\left(1+\mu\sigma_r(\overline{M}^{(j)})\right)^{t_\star} = \exp\left(t_\star \ln(1+\mu\sigma_r(\overline{M}^{(j)}))\right) \leq \exp\left(t_\star \ln(1+\mu\|\mathcal{M}\|)\right)$$

$$\leq \exp\left(2\psi \max_{1\leq j\leq k} \frac{\ln(1+\mu\|\mathcal{M}\|)}{\ln\left(\frac{1+\mu\sigma_r(\overline{M}^{(j)})}{1+\mu\sigma_{r+1}(\overline{M}^{(j)})}\right)}\right) \leq \exp(16\psi\kappa^2) = \left(\frac{2\kappa^2\|\mathcal{U}\|}{c_3\sigma_{min}(\overline{\mathcal{V}_{\mathcal{L}}^\top * \mathcal{U}})}\right)^{16\kappa^2}. \quad \text{(D.68)}$$

Since this holds for all $j$, we have

$$\beta \leq \sigma_{min}(\overline{\mathcal{V}_{\mathcal{L}}^\top * \mathcal{U}})\left(\frac{2\kappa^2\|\mathcal{U}\|}{c_3\sigma_{min}(\overline{\mathcal{V}_{\mathcal{L}}^\top * \mathcal{U}})}\right)^{16\kappa^2}.$$

Finally, we come to the properties of $\mathcal{U}_{t^\star}$. By the representation $\mathcal{U}_{t_\star} = \mathcal{Z}_{t_\star} * \mathcal{U}_0 + \mathcal{E}_{t_\star}$, we get

$$\|\mathcal{U}_{t^\star}\| \leq \alpha\|\mathcal{Z}_{t_\star}\|\|\mathcal{U}\| + \|\mathcal{E}_{t_\star}\|.$$

From (D.55), we get

$$\|\mathcal{E}_t\| \leq \frac{c_3}{2\kappa^2}\alpha\|\mathcal{Z}_t\|\sigma_{min}(\overline{\mathcal{V}_{\mathcal{L}}}^{\mathrm{H}}\overline{\mathcal{U}}) \leq \frac{c_3}{2\kappa^2}\alpha\|\mathcal{Z}_t\|\sigma_{min}(\overline{\mathcal{V}_{\mathcal{L}}}^{\mathrm{H}})\sigma_{max}(\overline{\mathcal{U}}) \leq \alpha\|\mathcal{Z}_t\|\|\mathcal{U}\|,$$

which allows us to proceed as follows

$$\|\mathcal{U}_{t^\star}\| \leq 2\alpha\|\mathcal{Z}_{t_\star}\|\|\mathcal{U}\| \leq 2\alpha(1+\mu\|\mathcal{M}\|)^{t_\star}\|\mathcal{U}\|,$$

$$= 2\alpha\ln\left(t_\star(1+\mu\|\mathcal{M}\|)\right)\|\mathcal{U}\| \leq 2\alpha\|\mathcal{U}\|\left(\frac{2\kappa^2\|\mathcal{U}\|}{c_3\sigma_{min}(\overline{\mathcal{V}_{\mathcal{L}}^\top * \mathcal{U}})}\right)^{16\kappa^2}$$

$$\leq 2\|\mathcal{X}\|\sqrt{\frac{c_3\sigma_{min}(\overline{\mathcal{V}_{\mathcal{L}}^\top * \mathcal{U}})}{12\,k\sqrt{\min\{n,R\}}\kappa^2\|\mathcal{U}\|}}\left(\frac{2\kappa^2\|\mathcal{U}\|}{c_3\sigma_{min}(\overline{\mathcal{V}_{\mathcal{L}}^\top * \mathcal{U}})}\right)^{-8\kappa^2} \leq 3\|\mathcal{X}\|,$$

where for the second inequality above we used (D.68) and in the last one an upper bound on $\alpha$ from (D.64) has been applied.

$\square$

The results in Lemma D.8 hold for any initialization $\mathcal{U}$. Below, we will use the fact that $\mathcal{U}$ is a tensor with Gaussian entries. This yields the following lemma, which shows that with initialization scale $\alpha > 0$ chosen sufficiently small, the properties stated in Lemma D.8 hold with high probability.

**Lemma D.9.** *Fix a sufficiently small constant $c > 0$. Let $\mathcal{U} \in \mathbb{R}^{n\times R\times k}$ be a random tubal tensor with i.i.d. $\mathcal{N}(0, \frac{1}{R})$ entries, and let $\epsilon \in (0,1)$. Assume that $\mathcal{A} : S^{n\times n\times k} \to \mathbb{R}^m$ satisfies the S2NRIP($\delta_1$) for some constant $\delta_1 > 0$. Also, assume that*

$$\mathcal{M} := \mathcal{A}^*\mathcal{A}(\mathcal{X} * \mathcal{X}^\top) = \mathcal{X} * \mathcal{X}^\top + \mathcal{E}$$

*with $\|\overline{E}^{(j)}\| \leq \delta\lambda_r(\overline{X}^{(j)}\overline{X}^{(j)\,\mathrm{H}})$ for each $1 \leq j \leq k$, where $\delta \leq c_1\kappa^{-2}$. Let $\mathcal{U}_0 = \alpha\mathcal{U}$ where*

$$\alpha^2 \lesssim \begin{cases} \dfrac{\epsilon\min\{n,R\}\|\boldsymbol{\mathcal{X}}\|^2}{k^2 n^{3/2}\kappa^2}\left(\dfrac{2\kappa^2 kn^{3/2}}{c_3\min\{n,R\}^{3/2}\epsilon}\right)^{-24\kappa^2} & \text{if } R \geq 3r \\[4mm] \dfrac{\epsilon\|\boldsymbol{\mathcal{X}}\|^2}{k^2 n^{3/2}\kappa^2}\left(\dfrac{2\kappa^2 kn^{3/2}}{c_3 r^{1/2}\epsilon}\right)^{-24\kappa^2} & \text{if } R < 3r \end{cases}.$$

*Assume the step size satisfies $\mu \leq c_2\kappa^{-2}\|\boldsymbol{\mathcal{X}}\|^2$. Then, with probability at least $1-p$ where*

$$p = \begin{cases} k(\tilde{C}\epsilon)^{R-r+1} + ke^{-\tilde{c}R} & \text{if } R \geq 2r \\ k\epsilon^2 + ke^{-\tilde{c}R} & \text{if } R < 2r \end{cases}$$

*the following statement holds. After*

$$t_\star \lesssim \begin{cases} \dfrac{1}{\mu\min_{1\leq j\leq k}\sigma_r(\overline{X}^{(j)})^2}\ln\left(\dfrac{2\kappa^2\sqrt{n}}{c_3\epsilon\sqrt{\min\{n;R\}}}\right) & \text{if } R \geq 3r \\[4mm] \dfrac{1}{\mu\min_{1\leq j\leq k}\sigma_r(\overline{X}^{(j)})^2}\ln\left(\dfrac{2\kappa^2\sqrt{rn}}{c_3\epsilon}\right) & \text{if } R < 3r \end{cases}$$

*iterations, it holds that*

$$\|\boldsymbol{\mathcal{U}}_{t_\star}\| \leq 3\|\boldsymbol{\mathcal{X}}\| \tag{D.69}$$

$$\|\boldsymbol{\mathcal{V}}_{\boldsymbol{\mathcal{X}}^\perp} * \boldsymbol{\mathcal{V}}_{\boldsymbol{\mathcal{U}}_{t_\star}*\boldsymbol{\mathcal{W}}_{t_\star}}\| \leq c\kappa^{-2}. \tag{D.70}$$

*and for each $1 \leq j \leq k$, we have*

$$\sigma_r\left(\overline{\boldsymbol{\mathcal{U}}_{t_\star}*\boldsymbol{\mathcal{W}}_{t_\star}}^{(j)}\right) \geq \frac{1}{4}\alpha\beta \tag{D.71}$$

$$\sigma_1\left(\overline{\boldsymbol{\mathcal{U}}_{t_\star}*\boldsymbol{\mathcal{W}}_{t_\star,\perp}}^{(j)}\right) \leq \frac{\kappa^{-2}}{8}\alpha\beta \tag{D.72}$$

$$\tag{D.73}$$

*where*

$$\beta \lesssim \begin{cases} \epsilon\sqrt{k}\left(\dfrac{2\kappa^2\sqrt{n}}{c_3\epsilon\sqrt{\min\{n;R\}}}\right)^{16\kappa^2} & \text{if } R \geq 3r \\[4mm] \dfrac{\epsilon\sqrt{k}}{r}\left(\dfrac{2\kappa^2\sqrt{rn}}{c_3\epsilon}\right)^{16\kappa^2} & \text{if } R < 3r \end{cases}$$

*and*

$$\beta \gtrsim \begin{cases} \epsilon\sqrt{k} & \text{if } R \geq 3r \\[3mm] \dfrac{\epsilon\sqrt{k}}{r} & \text{if } R < 3r \end{cases}.$$

*Proof.* By Lemma I.3, we have that $\|\boldsymbol{\mathcal{U}}\| \lesssim \sqrt{\dfrac{k\max\{n,R\}}{R}} = \sqrt{\dfrac{kn}{\min\{n;R\}}}$ with probability at least $1 - O(ke^{-c\max\{n,R\}})$. Also, by Lemma I.4, we have that $\|\overline{\boldsymbol{\mathcal{U}}}^H v_1\|_{\ell_2} = \|\boldsymbol{\mathcal{U}}^\top * \boldsymbol{\mathcal{V}}_1\|_F \asymp \sqrt{k}$ with probability at least $1 - O(ke^{-cR})$. Since $\boldsymbol{\mathcal{U}} \in \mathbb{R}^{n\times R\times k}$ has i.i.d. $\mathcal{N}(0,\frac{1}{R})$ entries and $\boldsymbol{\mathcal{V}}_{\mathcal{L}} * \boldsymbol{\mathcal{V}}_{\mathcal{L}} = \boldsymbol{\mathcal{I}}$, by rotational invariance, $\boldsymbol{\mathcal{V}}_{\mathcal{L}}^\top * \boldsymbol{\mathcal{U}} \in \mathbb{R}^{r\times R\times k}$ also has i.i.d. $\mathcal{N}(0,\frac{1}{R})$ entries. Hence, the lower bound on $\sigma_{\min}(\boldsymbol{\mathcal{V}}_{\mathcal{L}}^\top * \boldsymbol{\mathcal{U}})$ in Lemma I.2 applies. If $r \leq R \leq 2r$, we have

$$\sigma_{\min}(\boldsymbol{\mathcal{V}}_{\mathcal{L}}^\top * \boldsymbol{\mathcal{U}}) \geq \frac{\epsilon\sqrt{k}}{\sqrt{rR}} \gtrsim \frac{\epsilon\sqrt{k}}{r}$$

with probability at least $1 - k\epsilon^2$. If $2r < R < 3r$, we have

$$\sigma_{\min}(\mathcal{V}_{\mathcal{L}}^\top * \mathcal{U}) \geq \frac{\epsilon\sqrt{k}(\sqrt{R} - \sqrt{2r-1})}{\sqrt{R}} \geq \frac{\epsilon\sqrt{k}(R - (2r-1))}{\sqrt{r}(\sqrt{R} + \sqrt{2r-1})} \gtrsim \frac{\epsilon\sqrt{k}}{r}$$

with probability at least $1 - k(C\epsilon)^{R-2r+1} - ke^{-cR}$. If $R \geq 3r$, we have

$$\sigma_{\min}(\mathcal{V}_{\mathcal{L}}^\top * \mathcal{U}) \geq \frac{\epsilon\sqrt{k}(\sqrt{R} - \sqrt{2r-1})}{\sqrt{R}} = \epsilon\sqrt{k}\left(1 - \sqrt{\frac{2r-1}{R}}\right) \gtrsim \epsilon\sqrt{k}$$

with probability at least $1 - k(C\epsilon)^{R-2r+1} - ke^{-cR}$.

Therefore, the above bounds on $\|\mathcal{U}\|$, $\|\overline{\mathcal{U}}^H v_1\|_{\ell_2}$, and $\sigma_{\min}(\mathcal{V}_{\mathcal{L}}^\top * \mathcal{U})$ all hold simultaneously with probability at least $1 - p$ where

$$p = \begin{cases} k(\tilde{C}\epsilon)^{R-r+1} + ke^{-\tilde{c}R} & \text{if } R \geq 2r \\ k\epsilon^2 + ke^{-\tilde{c}R} & \text{if } R < 2r \end{cases}.$$

Provided that all three of these bounds hold, one can substitute these into Lemma D.8 to obtain the desired result. $\qquad\square$

## E. Analysis of Convergence Stage

In this section, we will prove that after passing the spectral stage, $\mathcal{U}_t * \mathcal{U}_t^\top$ goes into the convergence process towards the ground truth tensor $\mathcal{X} * \mathcal{X}^\top$ in the Frobenius norm. For this, we will first show that in each of the tensor slices $\sigma_{\min}(\mathcal{V}_{\mathcal{X}}^\top * \mathcal{U}_{t+1}{}^{(j)})$ grows exponentially, see Lemma E.1, whereas the noise terms $\|\overline{\mathcal{U}_{t+1} * \mathcal{W}_{t+1,\perp}}{}^{(j)}\|$, $1 \leq j \leq k$, grow slower, see Lemma E.3. Moreover, in Lemma E.5, we show that the tensor column spaces of the signal term $\mathcal{U}_t * \mathcal{W}_t$ and the ground truth $\mathcal{X}$ stay aligned. With this, and several auxiliary lemmas in place, we show that

**Lemma E.1.** *Assume that the following conditions hold*

$$\mu \leq c\|\mathcal{X}\|^{-2}\kappa^{-2}$$
$$\|\mathcal{U}_t\| \leq 3\|\mathcal{X}\|$$
$$\|\mathcal{V}_{\mathcal{X}^\perp}^\top * \mathcal{V}_{\mathcal{U}_t * \mathcal{W}_t}\| \leq c\kappa^{-1}$$

*and*

$$\|(\mathcal{A}^*\mathcal{A} - \mathcal{I})(\mathcal{X} * \mathcal{X}^\top - \mathcal{U}_t * \mathcal{U}_t^\top)\| \leq c\sigma_{min}^2(\overline{\mathcal{X}}). \tag{E.1}$$

*Moreover, assume that $\mathcal{V}_{\mathcal{X}}^\top * \mathcal{U}_t$ has full tubal rank with all invertible t-SVD-singular tubes. Then, for each $j$, $1 \leq j \leq k$, it holds that*

$$\sigma_{min}(\overline{\mathcal{V}_{\mathcal{X}}^\top * \mathcal{U}_{t+1}}{}^{(j)}) \geq \sigma_{min}(\overline{\mathcal{V}_{\mathcal{X}}^\top * \mathcal{U}_{t+1} * \mathcal{W}_t}{}^{(j)}) \geq \sigma_{min}(\overline{\mathcal{V}_{\mathcal{X}}^\top * \mathcal{U}_t}{}^{(j)})\left(1 + \frac{1}{4}\mu\sigma_{min}^2(\overline{\mathcal{X}}) - \mu\sigma_{min}^2(\overline{\mathcal{V}_{\mathcal{X}}^\top * \mathcal{U}_t}{}^{(j)})\right).$$

*Proof.* Consider the tensor $\mathcal{V}_{\mathcal{X}}^\top * \mathcal{U}_{t+1} * \mathcal{W}_t$. Using the definition of $\mathcal{U}_{t+1}$ in terms of $\mathcal{U}_t$, we can rewrite it as

$$\mathcal{V}_{\mathcal{X}}^\top * \mathcal{U}_{t+1} * \mathcal{W}_t = \mathcal{V}_{\mathcal{X}}^\top * \left(\mathcal{I} + \mu\mathcal{A}^*\mathcal{A}(\mathcal{X} * \mathcal{X}^\top - \mathcal{U}_t * \mathcal{U}_t^\top)\right) * \mathcal{U}_t * \mathcal{W}_t.$$

This representation leads to the following representation of the RHS above in the Fourier domain

$$\overline{V}_{\mathcal{X}}^{(j)\,H}(\mathrm{Id} + \mu\overline{(\mathcal{A}^*\mathcal{A}(\mathcal{X} * \mathcal{X}^\top - \mathcal{U}_t * \mathcal{U}_t^\top))}{}^{(j)})\overline{U}_t^{(j)}\overline{W}_t^{(j)} := H^{(j)}.$$

Note that here $\overline{(\mathcal{A}^*\mathcal{A}(\mathcal{X} * \mathcal{X}^\top - \mathcal{U}_t * \mathcal{U}_t^\top))}{}^{(j)}$ can not be represented as an independent slice of measurements of $\overline{X}^{(j)}\overline{X}^{(j)H} - \overline{U}_t^{(j)}\overline{U}_t^{(j)H}$ as it involved the information about all the slices $1 \leq j \leq k$.

Due to our assumptions on $\|\mathcal{U}_t\|$ and the tensor spectral norm property, we get

$$\|\overline{V}_{\mathcal{X}}^{(j)\,H}\overline{U}_t^{(j)}\| \leq \|\overline{U}_t^{(j)}\| \leq \|\mathcal{U}_t\| \leq 3\|\mathcal{X}\|.$$

This in turn is leading to

$$\mu \leq c\|\boldsymbol{\mathcal{X}}\|^{-2}\kappa^{-2} \leq \tilde{c}\|\overline{V}_{\boldsymbol{\mathcal{X}}}^{(j)\,\mathrm{H}}\overline{U}_t^{(j)}\|^{-2}.$$

This property of $\mu$ together with the nature of $\overline{W}_t^{(j)}$ and $\overline{V}_{\boldsymbol{\mathcal{X}}}^{(j)}$ coming along from the signal-noise-term decomposition (C.1) leads to the fulfilled conditions of Lemma H.2. Applying Lemma H.2 to the matrix $H^{(j)}$, the smallest singular value of matrix $H^{(j)}$ can be estimated as

$$\sigma_{min}(H^{(j)}) \geq \left(1+\mu\sigma_{min}^2(\overline{X}^{(j)})-\mu\|P_1^{(j)}\|-\mu\|P_2^{(j)}\|-\mu^2\|P_3^{(j)}\|\right)\sigma_{min}(\overline{V}_{\boldsymbol{\mathcal{X}}}^{(j)\,\mathrm{H}}\overline{U}_t^{(j)})\left(1-\mu\sigma_{min}^2(\overline{V}_{\boldsymbol{\mathcal{X}}}^{(j)\,\mathrm{H}}\overline{U}_t^{(j)})\right). \quad \text{(E.2)}$$

with

$$\|P_1^{(j)}\| \leq 4\|\overline{U}_t^{(j)}\overline{W}_t^{(j)}\|^2\|\overline{V}_{\boldsymbol{\mathcal{X}}\perp}^{(j)}V_{\overline{U}_t^{(j)}\overline{W}_t^{(j)}}\|^2$$

$$\|P_2^{(j)}\| \leq 4\left\|\overline{(\mathcal{A}^*\mathcal{A}(\boldsymbol{\mathcal{X}}*\boldsymbol{\mathcal{X}}^\top-\boldsymbol{\mathcal{U}}_t*\boldsymbol{\mathcal{U}}_t^\top))}^{(j)}-\overline{X}^{(j)}\overline{X}^{(j)\mathrm{H}}+\overline{U}_t^{(j)}\overline{U}_t^{(j)\mathrm{H}}\right\|$$

$$\|P_3^{(j)}\| \leq 2\|\overline{X}^{(j)}\|^2\|\overline{U}_t^{(j)}\overline{W}_t^{(j)}\|^2.$$

Further, we will make the above bounds for $\|P_i^{(j)}\|$, $i \in \{1,2,3\}$, more precise using information about the tensor setting.

First of all since $\|\overline{U}_t^{(j)}\overline{W}_t^{(j)}\| \leq \|\overline{U}_t^{(j)}\| \leq \|\boldsymbol{\mathcal{U}}_t\| \leq 3\|\boldsymbol{\mathcal{X}}\|$, we get $\|P_1^{(j)}\| \leq 36\|\boldsymbol{\mathcal{X}}\|^2\|\overline{V}_{\boldsymbol{\mathcal{X}}\perp}^{(j)}V_{\overline{U}_t^{(j)}\overline{W}_t^{(j)}}\|^2$. Moreover, since $\overline{V}_{\boldsymbol{\mathcal{X}}\perp}^{(j)}V_{\overline{U}_t^{(j)}\overline{W}_t^{(j)}} = \overline{\boldsymbol{\mathcal{V}}_{\boldsymbol{\mathcal{X}}}^\top * \boldsymbol{\mathcal{V}}_{\boldsymbol{\mathcal{U}}_t*\boldsymbol{\mathcal{W}}_t}}^{(j)}$ and $\|\boldsymbol{\mathcal{V}}_{\boldsymbol{\mathcal{X}}\perp} * \boldsymbol{\mathcal{V}}_{\boldsymbol{\mathcal{U}}_t*\boldsymbol{\mathcal{W}}_t}\| \leq c\kappa^{-1}$ due to the assumption, it follows that for each $j$, $1 \leq j \leq k$, it holds that $\|\overline{V}_{\boldsymbol{\mathcal{X}}\perp}^{(j)}V_{\overline{U}_t^{(j)}\overline{W}_t^{(j)}}\| \leq c\kappa^{-1}$. This allows for the following estimation

$$\|P_1^{(j)}\| \leq 36\|\boldsymbol{\mathcal{X}}\|^2 c\kappa^{-1} \leq \frac{1}{4}\sigma_{min}^2(\overline{\boldsymbol{\mathcal{X}}}),$$

where the last inequality follows from the fact that $c > 0$ is small enough.

Before proceeding with $\|P_2^{(j)}\|$, consider

$$(\mathcal{A}^*\mathcal{A}-\mathcal{I})(\boldsymbol{\mathcal{X}}*\boldsymbol{\mathcal{X}}^\top-\boldsymbol{\mathcal{U}}_t*\boldsymbol{\mathcal{U}}_t^\top) = (\mathcal{A}^*\mathcal{A})(\boldsymbol{\mathcal{X}}*\boldsymbol{\mathcal{X}}^\top-\boldsymbol{\mathcal{U}}_t*\boldsymbol{\mathcal{U}}_t^\top)-(\boldsymbol{\mathcal{X}}*\boldsymbol{\mathcal{X}}^\top-\boldsymbol{\mathcal{U}}_t*\boldsymbol{\mathcal{U}}_t^\top).$$

The RHS from above has the following slices in the Fourier domain

$$\overline{(\mathcal{A}^*\mathcal{A})(\boldsymbol{\mathcal{X}}*\boldsymbol{\mathcal{X}}^\top-\boldsymbol{\mathcal{U}}_t*\boldsymbol{\mathcal{U}}_t^\top)}^{(j)}-\left(\overline{X}^{(j)}\overline{X}^{(j)\mathrm{H}}-\overline{U}_t^{(j)}\overline{U}_t^{(j)\mathrm{H}}\right),$$

the norm of which (due to assumption (E.1) and the definition of the tensor spectral norm) can be bounded as

$$\|\overline{(\mathcal{A}^*\mathcal{A})(\boldsymbol{\mathcal{X}}*\boldsymbol{\mathcal{X}}^\top-\boldsymbol{\mathcal{U}}_t*\boldsymbol{\mathcal{U}}_t^\top)}^{(j)}-\left(\overline{X}^{(j)}\overline{X}^{(j)\mathrm{H}}-\overline{U}_t^{(j)}\overline{U}_t^{(j)\mathrm{H}}\right)\| \leq \|(\mathcal{A}^*\mathcal{A}-\mathcal{I})(\boldsymbol{\mathcal{X}}*\boldsymbol{\mathcal{X}}^\top-\boldsymbol{\mathcal{U}}_t*\boldsymbol{\mathcal{U}}_t^\top)\| \leq c\sigma_{min}^2(\overline{\boldsymbol{\mathcal{X}}}).$$

This leads to the following estimation

$$\|P_2^{(j)}\| \leq 4c\sigma_{min}^2(\overline{\boldsymbol{\mathcal{X}}})$$

To further assess $\|P_3^{(j)}\|$, we take into account that matrix $\overline{W}_t^{(j)}$ is an orthogonal matrix and the assumption $\|\boldsymbol{\mathcal{U}}_t\| \leq 3\|\boldsymbol{\mathcal{X}}\|$, which allows for the next bound

$$\|P_3^{(j)}\| \leq 2\|\overline{X}^{(j)}\|^2\|\overline{U}_t^{(j)}\overline{W}_t^{(j)}\|^2 \leq 2\|\boldsymbol{\mathcal{X}}\|^2\|\overline{U}_t^{(j)}\|^2 \leq 2\|\boldsymbol{\mathcal{X}}\|^2\|\boldsymbol{\mathcal{U}}_t\|^2 \leq 18\|\boldsymbol{\mathcal{X}}\|^4.$$

Inserting the newly obtained estimates for $\|P_i^{(j)}\|$, $i \in \{1,2,3\}$, into (E.2), we get

$$\sigma_{min}(H^{(j)}) \geq (1+\mu\sigma_{min}^2(\overline{X}^{(j)})-\frac{\mu}{4}\sigma_{min}^2(\overline{\boldsymbol{\mathcal{X}}})-4\mu c\sigma_{min}^2(\overline{\boldsymbol{\mathcal{X}}})-18\mu^2\|\boldsymbol{\mathcal{X}}\|^4)\cdot$$

$$\cdot\sigma_{min}(\overline{V}_{\boldsymbol{\mathcal{X}}}^{(j)\,\mathrm{H}}\overline{U}_t^{(j)})\left(1-\mu\sigma_{min}^2(\overline{V}_{\boldsymbol{\mathcal{X}}}^{(j)\,\mathrm{H}}\overline{U}_t^{(j)})\right)$$

$$\geq (1+\mu\sigma_{min}^2(\overline{\boldsymbol{\mathcal{X}}})-\frac{\mu}{4}\sigma_{min}^2(\overline{\boldsymbol{\mathcal{X}}})-4\mu c\sigma_{min}^2(\overline{\boldsymbol{\mathcal{X}}})-18\mu^2\|\boldsymbol{\mathcal{X}}\|^4)\sigma_{min}(\overline{V}_{\boldsymbol{\mathcal{X}}}^{(j)\,\mathrm{H}}\overline{U}_t^{(j)})\left(1-\mu\sigma_{min}^2(\overline{V}_{\boldsymbol{\mathcal{X}}}^{(j)\,\mathrm{H}}\overline{U}_t^{(j)})\right).$$

Now, according to the assumption on $\mu$, we get

$$\mu^2\|\boldsymbol{\mathcal{X}}\|^4 \leq \mu c\kappa^{-2}\|\boldsymbol{\mathcal{X}}\|^{-2}\|\boldsymbol{\mathcal{X}}\|^4 = \mu c\frac{\sigma_{min}^2(\overline{\boldsymbol{\mathcal{X}}})}{\|\boldsymbol{\mathcal{X}}\|^2}\|\boldsymbol{\mathcal{X}}\|^{-2}\|\boldsymbol{\mathcal{X}}\|^4 = c\mu\sigma_{min}^2(\overline{\boldsymbol{\mathcal{X}}})$$

Taking $c$ small enough allows for the following estimation

$$\sigma_{min}(H^{(j)}) \geq \sigma_{min}(\overline{V}_{\boldsymbol{\mathcal{X}}}^{(j)\,\mathrm{H}}\overline{U}_t^{(j)})\big(1 + \frac{1}{2}\mu\sigma_{min}^2(\overline{\boldsymbol{\mathcal{X}}})\big)\big(1 - \mu\sigma_{min}^2(\overline{V}_{\boldsymbol{\mathcal{X}}}^{(j)\,\mathrm{H}}\overline{U}_t^{(j)})\big)$$

$$= \sigma_{min}(\overline{V}_{\boldsymbol{\mathcal{X}}}^{(j)\,\mathrm{H}}\overline{U}_t^{(j)})\Big(1 + \frac{1}{2}\mu\sigma_{min}^2(\overline{\boldsymbol{\mathcal{X}}})\big(1 - \mu\sigma_{min}^2(\overline{V}_{\boldsymbol{\mathcal{X}}}^{(j)\,\mathrm{H}}\overline{U}_t^{(j)})\big) - \mu\sigma_{min}^2(\overline{V}_{\boldsymbol{\mathcal{X}}}^{(j)\,\mathrm{H}}\overline{U}_t^{(j)})\Big)$$

Now, since $\sigma_{min}(\overline{V}_{\boldsymbol{\mathcal{X}}}^{(j)\,\mathrm{H}}\overline{U}_t^{(j)}) \leq \sigma_{min}(\overline{U}_t^{(j)}) \leq \|\boldsymbol{\mathcal{U}}_t\| \leq 3\|\boldsymbol{\mathcal{X}}\|$, we have that

$$\mu\sigma_{min}^2(\overline{V}_{\boldsymbol{\mathcal{X}}}^{(j)\,\mathrm{H}}\overline{U}_t^{(j)}) \leq \mu 9\|\boldsymbol{\mathcal{X}}\|^2 \leq 9c\kappa^{-2} \leq \frac{1}{2}$$

due to the fact that $c > 0$ can be chosen small enough. The last part of Lemma's proof follows from $\sigma_{min}(\boldsymbol{\mathcal{V}}_{\boldsymbol{\mathcal{X}}}^{\top} * \boldsymbol{\mathcal{U}}_{t+1}^{(j)}) \geq \sigma_{min}(\boldsymbol{\mathcal{V}}_{\boldsymbol{\mathcal{X}}}^{\top} * \boldsymbol{\mathcal{U}}_{t+1} * \boldsymbol{\mathcal{W}}_t^{(j)})$ and $\sigma_{min}(\boldsymbol{\mathcal{V}}_{\boldsymbol{\mathcal{X}}}^{\top} * \boldsymbol{\mathcal{U}}_{t+1} * \boldsymbol{\mathcal{W}}_t^{(j)}) = \sigma_{min}(H^{(j)})$, which completes the argument. $\qquad\square$

The next two lemmas will allow us to show that in each of the Fourier slices the noise term part of the gradient descent iterates is growing slower than its signal term part.

**Lemma E.2.** *Assume that $\mu \leq c\min\big\{\frac{1}{10}\|\boldsymbol{\mathcal{X}}\|^{-2}, \|(\mathcal{A}^*\mathcal{A} - \mathcal{I})(\boldsymbol{\mathcal{X}} * \boldsymbol{\mathcal{X}}^{\top} - \boldsymbol{\mathcal{U}}_t * \boldsymbol{\mathcal{U}}_t^{\top})\|^{-1}\big\}$ and $\|\boldsymbol{\mathcal{U}}_t\| \leq 3\|\boldsymbol{\mathcal{X}}\|$. Moreover, suppose that $\boldsymbol{\mathcal{V}}_{\boldsymbol{\mathcal{X}}}^{\top} * \boldsymbol{\mathcal{U}}_t$ has full tubal rank with all invertible t-SVD-tubes and $\|\boldsymbol{\mathcal{V}}_{\boldsymbol{\mathcal{X}}^{\perp}}^{\top} * \boldsymbol{\mathcal{V}}_{\boldsymbol{\mathcal{U}}_t * \boldsymbol{\mathcal{W}}_t}\| \leq c\kappa^{-1}$ with a sufficiently small contact $c > 0$. Then, the principal angle between $\boldsymbol{\mathcal{V}}_{\boldsymbol{\mathcal{X}}^{\perp}}$ and $\boldsymbol{\mathcal{V}}_{\boldsymbol{\mathcal{U}}_{t+1} * \boldsymbol{\mathcal{W}}_t}$ can be bounded as follows*

$$\|\boldsymbol{\mathcal{V}}_{\boldsymbol{\mathcal{X}}^{\perp}}^{\top} * \boldsymbol{\mathcal{V}}_{\boldsymbol{\mathcal{U}}_{t+1} * \boldsymbol{\mathcal{W}}_t}\| \leq 2\|\boldsymbol{\mathcal{V}}_{\boldsymbol{\mathcal{X}}^{\perp}}^{\top} * \boldsymbol{\mathcal{V}}_{\boldsymbol{\mathcal{U}}_t * \boldsymbol{\mathcal{W}}_t}\| + 2\mu\|(\mathcal{A}^*\mathcal{A})(\boldsymbol{\mathcal{X}} * \boldsymbol{\mathcal{X}}^{\top} - \boldsymbol{\mathcal{U}}_t * \boldsymbol{\mathcal{U}}_t^{\top})\|.$$

*In particular, it holds that $\|\boldsymbol{\mathcal{V}}_{\boldsymbol{\mathcal{X}}^{\perp}}^{\top} * \boldsymbol{\mathcal{V}}_{\boldsymbol{\mathcal{U}}_{t+1} * \boldsymbol{\mathcal{W}}_t}\| \leq 1/50$.*

*Proof.* By the definition of $\boldsymbol{\mathcal{U}}_{t+1}$, we have

$$\boldsymbol{\mathcal{U}}_{t+1} * \boldsymbol{\mathcal{W}}_t = \big(\mathcal{I} + \mu\mathcal{A}^*\mathcal{A}(\boldsymbol{\mathcal{X}} * \boldsymbol{\mathcal{X}}^{\top} - \boldsymbol{\mathcal{U}}_t * \boldsymbol{\mathcal{U}}_t^{\top})\big) * \boldsymbol{\mathcal{U}}_t * \boldsymbol{\mathcal{W}}_t \qquad \in \mathbb{R}^{n \times r \times k},$$

which allows for the following representation in the Fourier domain

$$\overline{\boldsymbol{\mathcal{U}}_{t+1} * \boldsymbol{\mathcal{W}}_t}^{(j)} = \Big(\mathrm{Id} + \mu\overline{\mathcal{A}^*\mathcal{A}(\boldsymbol{\mathcal{X}} * \boldsymbol{\mathcal{X}}^{\top} - \boldsymbol{\mathcal{U}}_t * \boldsymbol{\mathcal{U}}_t^{\top})}^{(j)}\Big)\overline{\boldsymbol{\mathcal{U}}_t * \boldsymbol{\mathcal{W}}_t}^{(j)} \quad \in \mathbb{C}^{n \times r}, \quad 1 \leq j \leq k.$$

Consider the SVD decomposition $\overline{\boldsymbol{\mathcal{U}}_t * \boldsymbol{\mathcal{W}}_t}^{(j)} = V_{\overline{\boldsymbol{\mathcal{U}}_t * \boldsymbol{\mathcal{W}}_t}^{(j)}} \Sigma_{\overline{\boldsymbol{\mathcal{U}}_t * \boldsymbol{\mathcal{W}}_t}^{(j)}} W_{\overline{\boldsymbol{\mathcal{U}}_t * \boldsymbol{\mathcal{W}}_t}^{(j)}}^{\mathrm{H}}$ and denote by $Z^{(j)}$ the matrix

$$Z^{(j)} := \Big(\mathrm{Id} + \mu\overline{\mathcal{A}^*\mathcal{A}(\boldsymbol{\mathcal{X}} * \boldsymbol{\mathcal{X}}^{\top} - \boldsymbol{\mathcal{U}}_t * \boldsymbol{\mathcal{U}}_t^{\top})}^{(j)}\Big) V_{\overline{\boldsymbol{\mathcal{U}}_t * \boldsymbol{\mathcal{W}}_t}^{(j)}} \qquad \in \mathbb{C}^{n \times r}.$$

Since by assumption $\overline{\boldsymbol{\mathcal{U}}_t * \boldsymbol{\mathcal{W}}_t}^{(j)}$ has full rank (due to full-rankness of $\boldsymbol{\mathcal{V}}_{\boldsymbol{\mathcal{X}}}^{\top} * \boldsymbol{\mathcal{U}}_t$, see Lemma C.1), matrix $Z^{(j)}$ has the same column space as $\overline{\boldsymbol{\mathcal{U}}_{t+1} * \boldsymbol{\mathcal{W}}_t}^{(j)}$ and the principal angle between tensor subspaces $\boldsymbol{\mathcal{V}}_{\boldsymbol{\mathcal{X}}^{\perp}}$ and $\boldsymbol{\mathcal{V}}_{\boldsymbol{\mathcal{U}}_{t+1} * \boldsymbol{\mathcal{W}}_t}$ can be computed via $Z^{(j)}$ as

$$\|\boldsymbol{\mathcal{V}}_{\boldsymbol{\mathcal{X}}^{\perp}}^{\top} * \boldsymbol{\mathcal{V}}_{\boldsymbol{\mathcal{U}}_{t+1} * \boldsymbol{\mathcal{W}}_t}\| = \max_{1 \leq j \leq k}\|\overline{V}_{\boldsymbol{\mathcal{X}}^{\perp}}^{(j)\mathrm{H}}\overline{V}_{\boldsymbol{\mathcal{U}}_{t+1} * \boldsymbol{\mathcal{W}}_t}^{(j)}\| = \max_{1 \leq j \leq k}\|\overline{V}_{\boldsymbol{\mathcal{X}}^{\perp}}^{(j)\mathrm{H}}\overline{V}_{\overline{\boldsymbol{\mathcal{U}}_t * \boldsymbol{\mathcal{W}}_t}^{(j)}}\| = \max_{1 \leq j \leq k}\|\overline{V}_{\boldsymbol{\mathcal{X}}^{\perp}}^{(j)\mathrm{H}}V_{Z^{(j)}}\|.$$

Now, we will consider each of the terms $\|\overline{V}_{\boldsymbol{\mathcal{X}}^{\perp}}^{(j)\mathrm{H}}V_{Z^{(j)}}\|$ separately and bound them as follows

$$\|\overline{V}_{\boldsymbol{\mathcal{X}}^{\perp}}^{(j)\mathrm{H}}V_{Z^{(j)}}\| \leq \|\overline{V}_{\boldsymbol{\mathcal{X}}^{\perp}}^{(j)\mathrm{H}}V_{Z^{(j)}}\Sigma_{Z^{(j)}}W_{Z^{(j)}}^{\mathrm{H}}\|\|(\Sigma_{Z^{(j)}}W_{Z^{(j)}}^{\mathrm{H}})^{-1}\| = \frac{\|\overline{V}_{\boldsymbol{\mathcal{X}}^{\perp}}^{(j)\mathrm{H}}Z^{(j)}\|}{\sigma_{\min}(Z^{(j)})}. \qquad \text{(E.3)}$$

Using the definition of $Z^{(j)}$, the norm in the numerator above can be estimated as

$$\|\overline{V}_{\boldsymbol{\mathcal{X}}^{\perp}}^{(j)\mathrm{H}}Z^{(j)}\| \leq \|\overline{V}_{\boldsymbol{\mathcal{X}}^{\perp}}^{(j)\mathrm{H}}V_{\overline{\boldsymbol{\mathcal{U}}_t * \boldsymbol{\mathcal{W}}_t}^{(j)}}\| + \mu\|\overline{V}_{\boldsymbol{\mathcal{X}}^{\perp}}^{(j)\mathrm{H}}\overline{\mathcal{A}^*\mathcal{A}(\boldsymbol{\mathcal{X}} * \boldsymbol{\mathcal{X}}^{\top} - \boldsymbol{\mathcal{U}}_t * \boldsymbol{\mathcal{U}}_t^{\top})}^{(j)}\|$$

$$\leq \|\overline{V}_{\boldsymbol{\mathcal{X}}^{\perp}}^{(j)\mathrm{H}}\overline{V}_{\boldsymbol{\mathcal{U}}_t * \boldsymbol{\mathcal{W}}_t}^{(j)}\| + \mu\|\overline{\mathcal{A}^*\mathcal{A}(\boldsymbol{\mathcal{X}} * \boldsymbol{\mathcal{X}}^{\top} - \boldsymbol{\mathcal{U}}_t * \boldsymbol{\mathcal{U}}_t^{\top})}^{(j)}\|$$

$$\leq \|\boldsymbol{\mathcal{V}}_{\boldsymbol{\mathcal{X}}^{\perp}}^{\top} * \boldsymbol{\mathcal{V}}_{\boldsymbol{\mathcal{U}}_t * \boldsymbol{\mathcal{W}}_t}\| + \mu\|\mathcal{A}^*\mathcal{A}(\boldsymbol{\mathcal{X}} * \boldsymbol{\mathcal{X}}^{\top} - \boldsymbol{\mathcal{U}}_t * \boldsymbol{\mathcal{U}}_t^{\top})\|.$$

Using again the definition of $Z^{(j)}$ and Weyl's inequality, the denominator in (E.3) can be estimated from below as follows

$$
\begin{aligned}
\sigma_{\min}(Z^{(j)}) &\geq \sigma_{\min}(V_{\overline{\mathcal{U}_t * \mathcal{W}_t}^{(j)}}) - \mu \| \left( \overline{\mathcal{A}^* \mathcal{A}(\mathcal{X} * \mathcal{X}^\top - \mathcal{U}_t * \mathcal{U}_t^\top)^{(j)}} \right) V_{\overline{\mathcal{U}_t * \mathcal{W}_t}^{(j)}} \| \\
&\geq 1 - \mu \| \overline{\mathcal{A}^* \mathcal{A}(\mathcal{X} * \mathcal{X}^\top - \mathcal{U}_t * \mathcal{U}_t^\top)^{(j)}} \| \geq 1 - \mu \| \mathcal{A}^* \mathcal{A}(\mathcal{X} * \mathcal{X}^\top - \mathcal{U}_t * \mathcal{U}_t^\top) \| \\
&\geq 1 - \mu(\|(\mathcal{A}^* \mathcal{A} - \mathcal{I})(\mathcal{X} * \mathcal{X}^\top - \mathcal{U}_t * \mathcal{U}_t^\top)\| + \|(\mathcal{X} * \mathcal{X}^\top - \mathcal{U}_t * \mathcal{U}_t^\top)\|) \\
&\geq 1 - \mu \left( \|(\mathcal{A}^* \mathcal{A} - \mathcal{I})(\mathcal{X} * \mathcal{X}^\top - \mathcal{U}_t * \mathcal{U}_t^\top)\| + \|\mathcal{X}\|^2 + \|\mathcal{U}_t\|^2 \right) \\
&\geq 1 - \mu \left( \|(\mathcal{A}^* \mathcal{A} - \mathcal{I})(\mathcal{X} * \mathcal{X}^\top - \mathcal{U}_t * \mathcal{U}_t^\top)\| + 10\|\mathcal{X}\|^2 \right) \geq \frac{1}{2},
\end{aligned}
$$

where the last inequality follows from the assumption on $\mu$. Now, we can come back to the estimation of $\|\mathcal{V}_{\mathcal{X}^\perp}^\top * \mathcal{V}_{\mathcal{U}_{t+1} * \mathcal{W}_t}\|$, which due to the combination of the above-carried estimated reads as

$$
\|\mathcal{V}_{\mathcal{X}^\perp}^\top * \mathcal{V}_{\mathcal{U}_{t+1} * \mathcal{W}_t}\| \leq 2\|\mathcal{V}_{\mathcal{X}^\perp}^\top * \mathcal{V}_{\mathcal{U}_t * \mathcal{W}_t}\| + 2\mu \|\mathcal{A}^* \mathcal{A}(\mathcal{X} * \mathcal{X}^\top - \mathcal{U}_t * \mathcal{U}_t^\top)\|
$$

providing the first result from the Lemma. The second bound stated in the Lemma follows from our assumption on $\|\mathcal{V}_{\mathcal{X}^\perp}^\top * \mathcal{V}_{\mathcal{U}_t * \mathcal{W}_t}\|$ and $\mu$ and the fact that the constant $c$ is chosen small enough to make $\|\mathcal{V}_{\mathcal{X}^\perp}^\top * \mathcal{V}_{\mathcal{U}_{t+1} * \mathcal{W}_t}\| \leq \frac{1}{50}$.

$\square$

**Lemma E.3.** *Assume that $\mu \leq c_1 \min \left\{ \frac{1}{10} \|\mathcal{X}\|^{-2}, \|(\mathcal{A}^* \mathcal{A} - \mathcal{I})(\mathcal{X} * \mathcal{X}^\top - \mathcal{U}_t * \mathcal{U}_t^\top)\|^{-1} \right\}$ and $\|\mathcal{U}_t\| \leq 3\|\mathcal{X}\|$. Moreover, suppose that tensor $\mathcal{V}_{\mathcal{X}}^\top * \mathcal{U}_{t+1} * \mathcal{W}_t$ has all invertible t-SVD-tubes and that $\|\mathcal{V}_{\mathcal{X}^\perp}^\top * \mathcal{V}_{\mathcal{U}_t * \mathcal{W}_t}\| \leq c_1 \kappa^{-1}$, with absolute constant $c_1 > 0$ chosen small enough. Then, it holds that*

$$
\begin{aligned}
\|\overline{\mathcal{U}_{t+1} * \mathcal{W}_{t+1,\perp}}^{(j)}\| \leq &\left( 1 - \frac{\mu}{2} \|\overline{\mathcal{U}_t * \mathcal{W}_{t,\perp}}^{(j)}\|^2 + 9\mu \|\overline{\mathcal{V}_{\mathcal{X}^\perp}^\top * \mathcal{V}_{\mathcal{U}_t * \mathcal{W}_t}}^{(j)}\| \|\mathcal{X}\|^2 \right. \\
&\left. + 2\mu \|(\mathcal{A}^* \mathcal{A} - \mathcal{I})(\mathcal{X} * \mathcal{X}^\top - \mathcal{U}_t * \mathcal{U}_t^\top)\| \right) \|\overline{\mathcal{U}_t * \mathcal{W}_{t,\perp}}^{(j)}\|
\end{aligned}
$$

*for each $j$, with $1 \leq j \leq k$.*

*Proof.* First, we will consider tensor $\mathcal{U}_{t+1} * \mathcal{W}_{t+1,\perp}$ splitting it into two different parts, and then will conduct the corresponding norm estimations of each Fourier slices.

To begin with, note that for the tensor-column space of $\mathcal{X}$, that is $\mathcal{V}_{\mathcal{X}}$, it holds that $\mathcal{V}_{\mathcal{X}} * \mathcal{V}_{\mathcal{X}}^\top + \mathcal{V}_{\mathcal{X}^\perp} * \mathcal{V}_{\mathcal{X}^\perp}^\top = \mathcal{I}$ (see, for example, (Liu et al., 2019)). Using this, we can represent $\mathcal{U}_{t+1} * \mathcal{W}_{t+1,\perp}$ as follows

$$
\mathcal{U}_{t+1} * \mathcal{W}_{t+1,\perp} = \mathcal{V}_{\mathcal{X}} * \mathcal{V}_{\mathcal{X}}^\top * \mathcal{U}_{t+1} * \mathcal{W}_{t+1,\perp} + \mathcal{V}_{\mathcal{X}^\perp} * \mathcal{V}_{\mathcal{X}^\perp}^\top * \mathcal{U}_{t+1} * \mathcal{W}_{t+1,\perp} = \mathcal{V}_{\mathcal{X}^\perp} * \mathcal{V}_{\mathcal{X}^\perp}^\top * \mathcal{U}_{t+1} * \mathcal{W}_{t+1,\perp} \quad \text{(E.4)}
$$

where the last equality follows from Lemma C.1 due to the property $\mathcal{V}_{\mathcal{X}}^\top * \mathcal{U}_{t+1} * \mathcal{W}_{t+1,\perp} = 0$.

Now, we split the term $\mathcal{V}_{\mathcal{X}^\perp} * \mathcal{V}_{\mathcal{X}^\perp}^\top * \mathcal{U}_{t+1} * \mathcal{W}_{t+1,\perp}$ into two parts using $\mathcal{W}_t * \mathcal{W}_t^\top + \mathcal{W}_{t,\perp} * \mathcal{W}_{t,\perp}^\top = \mathcal{I}$, which leads to

$$
\mathcal{V}_{\mathcal{X}^\perp} * \mathcal{V}_{\mathcal{X}^\perp}^\top * \mathcal{U}_{t+1} * \mathcal{W}_{t+1,\perp} = \mathcal{V}_{\mathcal{X}^\perp} * \mathcal{V}_{\mathcal{X}^\perp}^\top * \mathcal{U}_{t+1} * \mathcal{W}_t * \mathcal{W}_t^\top * \mathcal{W}_{t+1,\perp} + \mathcal{V}_{\mathcal{X}^\perp} * \mathcal{V}_{\mathcal{X}^\perp}^\top * \mathcal{U}_{t+1} * \mathcal{W}_{t,\perp} * \mathcal{W}_{t,\perp}^\top * \mathcal{W}_{t+1,\perp}
$$
(E.5)

To estimate the norm of $\mathcal{V}_{\mathcal{X}^\perp} * \mathcal{V}_{\mathcal{X}^\perp}^\top * \mathcal{U}_t * \mathcal{W}_{t+1,\perp}$ in each slice in the Fourier domain, we will use the above-given representation and estimate each of the summands individually. Let us start with the second one. Its $j$th slice in the Fourier domain reads as

$$
\overline{(\mathcal{V}_{\mathcal{X}^\perp} * \mathcal{V}_{\mathcal{X}^\perp}^\top * \mathcal{U}_{t+1} * \mathcal{W}_{t,\perp} * \mathcal{W}_{t,\perp}^\top * \mathcal{W}_{t+1,\perp})}^{(j)} = \overline{V}_{\mathcal{X}^\perp}^{(j)} \overline{V}_{\mathcal{X}^\perp}^{(j)\mathrm{H}} \overline{U}_{t+1}^{(j)} \overline{W}_{t,\perp}^{(j)} \overline{W}_{t,\perp}^{(j)\mathrm{H}} \overline{W}_{t+1,\perp}^{(j)}.
$$

Due to the orthogonality of the columns of $\overline{V}_{\mathcal{X}^\perp}^{(j)}$, it holds that $\|\overline{V}_{\mathcal{X}^\perp}^{(j)} \overline{V}_{\mathcal{X}^\perp}^{(j)\mathrm{H}} \overline{U}_{t+1}^{(j)} \overline{W}_{t,\perp}^{(j)} \overline{W}_{t,\perp}^{(j)\mathrm{H}} \overline{W}_{t+1,\perp}^{(j)}\| = \|\overline{V}_{\mathcal{X}^\perp}^{(j)\mathrm{H}} \overline{U}_{t+1}^{(j)} \overline{W}_{t,\perp}^{(j)} \overline{W}_{t,\perp}^{(j)\mathrm{H}} \overline{W}_{t+1,\perp}^{(j)}\|$. In the Fourier domain, this allows us to focus on $j$th slices of the last one

$$
\overline{V}_{\mathcal{X}^\perp}^{(j)\mathrm{H}} \overline{U}_{t+1}^{(j)} \overline{W}_{t,\perp}^{(j)} \overline{W}_{t,\perp}^{(j)\mathrm{H}} \overline{W}_{t+1,\perp}^{(j)} := G_2^{(j)}.
$$

Due to the definition of the gradient descent iterates $\mathcal{U}_{t+1}$, we have the following representation for its blocks $\overline{U}_{t+1}^{(j)}$ in the Fourier domain

$$\overline{U}_{t+1}^{(j)} = \left(\mathrm{Id} + \mu\overline{(\mathcal{A}^*\mathcal{A}(\boldsymbol{\mathcal{X}} * \boldsymbol{\mathcal{X}}^\top - \mathcal{U}_t * \mathcal{U}_t^\top))}^{(j)}\right)\overline{U}_t^{(j)}$$

To upper bound the norm of $G_2^{(j)}$, we want to apply Lemma H.3. Due to the assumptions in this lemma that $\mathcal{V}_{\boldsymbol{\mathcal{X}}}^\top * \mathcal{U}_{t+1} * \mathcal{W}_t$ has full tubal rank with all invertible t-SVD-tubes and $\|\mathcal{V}_{\boldsymbol{\mathcal{X}}^\perp}^\top * \mathcal{V}_{\mathcal{U}_t * \mathcal{W}_t}\| \le c\kappa^{-1}$ in addition to the conditions on $\mu$ and the decomposition of gradient descent iterates into the signal and noise term, the conditions of Lemma H.3 are satisfied for the choice $Y_1 = \overline{U}_{t+1}^{(j)}$ and $Y = \overline{U}_t^{(j)}$ and $Z$ as $Z = \overline{(\mathcal{A}^*\mathcal{A}(\boldsymbol{\mathcal{X}} * \boldsymbol{\mathcal{X}}^\top - \mathcal{U}_t * \mathcal{U}_t^\top))}^{(j)}$. This allows to upper-bound the norm of $G_2^{(j)}$ as follows

$$\|G_2^{(j)}\| \le \|\overline{U}_t^{(j)}\overline{W}_{t,\perp}^{(j)}\|\left(1 - \mu\|\overline{U}_t^{(j)}\overline{W}_{t,\perp}^{(j)}\|^2 + \mu\|\overline{(\mathcal{A}^*\mathcal{A}(\boldsymbol{\mathcal{X}} * \boldsymbol{\mathcal{X}}^\top - \mathcal{U}_t * \mathcal{U}_t^\top))}^{(j)} - (\overline{X}^{(j)}\overline{X}^{(j)\mathrm{H}} - \overline{U}_t^{(j)}\overline{U}_t^{(j)\mathrm{H}})\|\right)$$
$$+ \mu^2\left(\|\overline{U}_t^{(j)}\overline{W}_t^{(j)}\|^2 + \|\overline{(\mathcal{A}^*\mathcal{A}(\boldsymbol{\mathcal{X}} * \boldsymbol{\mathcal{X}}^\top - \mathcal{U}_t * \mathcal{U}_t^\top))}^{(j)} - (\overline{X}^{(j)}\overline{X}^{(j)\mathrm{H}} - \overline{U}_t^{(j)}\overline{U}_t^{(j)\mathrm{H}})\|\right)\|\overline{U}_t^{(j)}\overline{W}_{t,\perp}^{(j)}\|^3$$

Using now the fact that for each $j$ it holds that

$$\|\overline{(\mathcal{A}^*\mathcal{A}(\boldsymbol{\mathcal{X}} * \boldsymbol{\mathcal{X}}^\top - \mathcal{U}_t * \mathcal{U}_t^\top))}^{(j)} - (\overline{X}^{(j)}\overline{X}^{(j)\mathrm{H}} - \overline{U}_t^{(j)}\overline{U}_t^{(j)\mathrm{H}})\| \le \|(\mathcal{A}^*\mathcal{A} - \mathcal{I})(\boldsymbol{\mathcal{X}} * \boldsymbol{\mathcal{X}}^\top - \mathcal{U}_t * \mathcal{U}_t^\top)\|$$

and that $\|\overline{U}_t^{(j)}\| \le \|\mathcal{U}_t\| \le 3\|\boldsymbol{\mathcal{X}}\|$, we can proceed with the bound for the norm of $G_2^{(j)}$ as below

$$\|G_2^{(j)}\| \le \|\overline{U}_t^{(j)}\overline{W}_{t,\perp}^{(j)}\|\left(1 - \mu\|\overline{U}_t^{(j)}\overline{W}_{t,\perp}^{(j)}\|^2 + \mu\|(\mathcal{A}^*\mathcal{A} - \mathcal{I})(\boldsymbol{\mathcal{X}} * \boldsymbol{\mathcal{X}}^\top - \mathcal{U}_t * \mathcal{U}_t^\top)\|\right)$$
$$+ \mu^2\left(9\|\boldsymbol{\mathcal{X}}\|^2 + \|(\mathcal{A}^*\mathcal{A} - \mathcal{I})(\boldsymbol{\mathcal{X}} * \boldsymbol{\mathcal{X}}^\top - \mathcal{U}_t * \mathcal{U}_t^\top)\|\right)\|\overline{U}_t^{(j)}\overline{W}_{t,\perp}^{(j)}\|^3$$

Further, using the assumption $\mu \le c_1 \min\left\{\frac{1}{10}\|\boldsymbol{\mathcal{X}}\|^{-2}, \|(\mathcal{A}^*\mathcal{A} - \mathcal{I})(\boldsymbol{\mathcal{X}} * \boldsymbol{\mathcal{X}}^\top - \mathcal{U}_t * \mathcal{U}_t^\top)\|^{-1}\right\}$, we get

$$\|G_2^{(j)}\| \le \|\overline{U}_t^{(j)}\overline{W}_{t,\perp}^{(j)}\|\left(1 - \mu\|\overline{U}_t^{(j)}\overline{W}_{t,\perp}^{(j)}\|^2 + \mu\|(\mathcal{A}^*\mathcal{A} - \mathcal{I})(\boldsymbol{\mathcal{X}} * \boldsymbol{\mathcal{X}}^\top - \mathcal{U}_t * \mathcal{U}_t^\top)\|\right) + \frac{\mu}{2}\|\overline{U}_t^{(j)}\overline{W}_{t,\perp}^{(j)}\|^3$$
$$= \|\overline{U}_t^{(j)}\overline{W}_{t,\perp}^{(j)}\|\left(1 - \frac{\mu}{2}\|\overline{U}_t^{(j)}\overline{W}_{t,\perp}^{(j)}\|^2 + \mu\|(\mathcal{A}^*\mathcal{A} - \mathcal{I})(\boldsymbol{\mathcal{X}} * \boldsymbol{\mathcal{X}}^\top - \mathcal{U}_t * \mathcal{U}_t^\top)\|\right).$$

Now, let us return to the first summand in (E.5), that is $\mathcal{V}_{\boldsymbol{\mathcal{X}}}^\top * \mathcal{U}_{t+1} * \mathcal{W}_t * \mathcal{W}_t^\top * \mathcal{W}_{t+1,\perp}$. Using again the fact that $\mathcal{V}_{\boldsymbol{\mathcal{X}}} * \mathcal{U}_{t+1} * \mathcal{W}_{t+1,\perp} = 0$ allows us to rewrite it as

$$\mathcal{V}_{\boldsymbol{\mathcal{X}}}^\top * \mathcal{U}_{t+1} * \mathcal{W}_t * \mathcal{W}_t^\top * \mathcal{W}_{t+1,\perp} = -\mathcal{V}_{\boldsymbol{\mathcal{X}}}^\top * \mathcal{U}_{t+1} * \mathcal{W}_{t,\perp} * \mathcal{W}_{t,\perp}^\top * \mathcal{W}_{t+1,\perp} \tag{E.6}$$

Moreover, for the same summand, the corresponding $j$th slice in the Fourier domain reads as

$$\overline{V}_{\boldsymbol{\mathcal{X}}^\perp}^{(j)\mathrm{H}}\overline{U}_{t+1}^{(j)}\overline{W}_t^{(j)}\overline{W}_t^{(j)\mathrm{H}}\overline{W}_{t+1,\perp}^{(j)} := G_1^{(j)}.$$

Due to relation (E.6) in the tensor domain, in the Fourier domain it holds that

$$\overline{V}_{\boldsymbol{\mathcal{X}}}^{(j)\mathrm{H}}\overline{U}_{t+1}^{(j)}\overline{W}_t^{(j)}\overline{W}_t^{(j)\mathrm{H}}\overline{W}_{t+1,\perp}^{(j)} = -\overline{V}_{\boldsymbol{\mathcal{X}}}^{(j)\mathrm{H}}\overline{U}_{t+1}^{(j)}\overline{W}_{t,\perp}^{(j)}\overline{W}_{t,\perp}^{(j)\mathrm{H}}\overline{W}_{t+1,\perp}^{(j)},$$

which allows to represent $\overline{W}_t^{(j)\mathrm{H}}\overline{W}_{t+1,\perp}^{(j)}$ as

$$\overline{W}_t^{(j)\mathrm{H}}\overline{W}_{t+1,\perp}^{(j)} = -\left(\overline{V}_{\boldsymbol{\mathcal{X}}}^{(j)\mathrm{H}}\overline{U}_{t+1}^{(j)}\overline{W}_t^{(j)}\right)^{-1}\overline{V}_{\boldsymbol{\mathcal{X}}}^{(j)\mathrm{H}}\overline{U}_{t+1}^{(j)}\overline{W}_{t,\perp}^{(j)}\overline{W}_{t,\perp}^{(j)\mathrm{H}}\overline{W}_{t+1,\perp}^{(j)}.$$

Note that the matrix on the RHS above is invertible due to the assumption that $\mathcal{V}_{\boldsymbol{\mathcal{X}}}^\top * \mathcal{U}_{t+1} * \mathcal{W}_t$ has full tubal rank with all invertible t-SVD-tubes. From here, $G_1^{(j)}$ can be represented as

$$G_1^{(j)} = \overline{V}_{\boldsymbol{\mathcal{X}}^\perp}^{(j)\mathrm{H}}\overline{U}_{t+1}^{(j)}\overline{W}_t^{(j)}\left(\overline{V}_{\boldsymbol{\mathcal{X}}}^{(j)\mathrm{H}}\overline{U}_{t+1}^{(j)}\overline{W}_t^{(j)}\right)^{-1}\overline{V}_{\boldsymbol{\mathcal{X}}}^{(j)\mathrm{H}}\overline{U}_{t+1}^{(j)}\overline{W}_{t,\perp}^{(j)}\overline{W}_{t,\perp}^{(j)\mathrm{H}}\overline{W}_{t+1,\perp}^{(j)}.$$

According to Lemma H.3, the norm of $G_1^{(j)}$ can be bounded from above as

$$\|G_1^{(j)}\| \le 2\mu\Big(\|\overline{V}_{\boldsymbol{\mathcal{X}}^\perp}^{(j)\mathrm{H}} V_{\overline{U}_t^{(j)}\overline{W}_t^{(j)}}\|\|\overline{U}_t^{(j)}\overline{W}_t^{(j)}\|^2 + \|\overline{\big(\mathcal{A}^*\mathcal{A}(\boldsymbol{\mathcal{X}}*\boldsymbol{\mathcal{X}}^\top - \boldsymbol{\mathcal{U}}_t*\boldsymbol{\mathcal{U}}_t^\top)\big)}^{(j)} - (\overline{X}^{(j)}\overline{X}^{(j)\mathrm{H}} - \overline{U}_t^{(j)}\overline{U}_t^{(j)\mathrm{H}})\|\Big)\cdot$$
$$\cdot\,\|\overline{V}_{\boldsymbol{\mathcal{X}}^\perp}^{(j)\mathrm{H}} V_{\overline{U}_{t+1}^{(j)}\overline{W}_t^{(j)}}\|\|\overline{U}_t^{(j)}\overline{W}_{t,\perp}^{(j)}\|$$
$$\le 2\mu\Big(\|\overline{V}_{\boldsymbol{\mathcal{X}}^\perp}^{(j)\mathrm{H}} V_{\overline{U}_t^{(j)}\overline{W}_t^{(j)}}\|\|\overline{U}_t^{(j)}\|^2 + \|(\mathcal{A}^*\mathcal{A} - \mathcal{I})(\boldsymbol{\mathcal{X}}*\boldsymbol{\mathcal{X}}^\top - \boldsymbol{\mathcal{U}}_t*\boldsymbol{\mathcal{U}}_t^\top)\|\Big)\cdot\|\overline{V}_{\boldsymbol{\mathcal{X}}^\perp}^{(j)\mathrm{H}} V_{\overline{U}_{t+1}^{(j)}\overline{W}_t^{(j)}}\|\|\overline{U}_t^{(j)}\overline{W}_{t,\perp}^{(j)}\|$$
$$\le 2\mu\Big(\|\overline{V}_{\boldsymbol{\mathcal{X}}^\perp}^{(j)\mathrm{H}} V_{\overline{U}_t^{(j)}\overline{W}_t^{(j)}}\|\|\overline{U}_t^{(j)}\|^2 + \|(\mathcal{A}^*\mathcal{A} - \mathcal{I})(\boldsymbol{\mathcal{X}}*\boldsymbol{\mathcal{X}}^\top - \boldsymbol{\mathcal{U}}_t*\boldsymbol{\mathcal{U}}_t^\top)\|\Big)\cdot\|\boldsymbol{\mathcal{V}}_{\boldsymbol{\mathcal{X}}^\perp}^\top * \boldsymbol{\mathcal{V}}_{\boldsymbol{\mathcal{U}}_{t+1}*\boldsymbol{\mathcal{W}}_t}\|\|\overline{U}_t^{(j)}\overline{W}_{t,\perp}^{(j)}\|$$

Due to $\|\boldsymbol{\mathcal{V}}_{\boldsymbol{\mathcal{X}}^\perp}^\top * \boldsymbol{\mathcal{V}}_{\boldsymbol{\mathcal{U}}_{t+1}*\boldsymbol{\mathcal{W}}_t}\| \le \frac{1}{50}$ from Lemma E.2, the fact that $\|\overline{U}_t^{(j)}\| \le \|\boldsymbol{\mathcal{U}}_t\|$, and our assumption that $\|\boldsymbol{\mathcal{U}}_t\| \le 3\|\boldsymbol{\mathcal{X}}\|$, the norm of $G_1^{(j)}$ can be further bounded as

$$\|G_1^{(j)}\| \le \mu\Big(9\|\overline{V}_{\boldsymbol{\mathcal{X}}^\perp}^{(j)\mathrm{H}} V_{\overline{U}_t^{(j)}\overline{W}_t^{(j)}}\|\|\boldsymbol{\mathcal{X}}\|^2 + \|(\mathcal{A}^*\mathcal{A} - \mathcal{I})(\boldsymbol{\mathcal{X}}*\boldsymbol{\mathcal{X}}^\top - \boldsymbol{\mathcal{U}}_t*\boldsymbol{\mathcal{U}}_t^\top)\|\Big)\|\overline{U}_t^{(j)}\overline{W}_{t,\perp}^{(j)}\|$$
$$= \mu\Big(9\|\overline{(\boldsymbol{\mathcal{V}}_{\boldsymbol{\mathcal{X}}^\perp}^\top * \boldsymbol{\mathcal{V}}_{\boldsymbol{\mathcal{U}}_t*\boldsymbol{\mathcal{W}}_t})}^{(j)}\|\|\boldsymbol{\mathcal{X}}\|^2 + \|(\mathcal{A}^*\mathcal{A} - \mathcal{I})(\boldsymbol{\mathcal{X}}*\boldsymbol{\mathcal{X}}^\top - \boldsymbol{\mathcal{U}}_t*\boldsymbol{\mathcal{U}}_t^\top)\|\Big)\|\overline{U}_t^{(j)}\overline{W}_{t,\perp}^{(j)}\|.$$

Since due to representation (E.4), it holds that $\|\overline{(\boldsymbol{\mathcal{U}}_{t+1} * \boldsymbol{\mathcal{W}}_{t+1,\perp})}^{(j)}\| = \|\overline{(\boldsymbol{\mathcal{V}}_{\boldsymbol{\mathcal{X}}^\perp} * \boldsymbol{\mathcal{U}}_{t+1} * \boldsymbol{\mathcal{W}}_{t+1,\perp})}^{(j)}\|$, combining the inequalities for $\|G_1^{(j)}\|$ and $\|G_2^{(j)}\|$ together with $\overline{U}_t^{(j)}\overline{W}_{t,\perp}^{(j)} = \overline{(\boldsymbol{\mathcal{U}}_t * \boldsymbol{\mathcal{W}}_{t,\perp})}^{(j)}$ leads to the final result

$$\|\overline{(\boldsymbol{\mathcal{U}}_{t+1} * \boldsymbol{\mathcal{W}}_{t+1,\perp})}^{(j)}\| \le \Big(1 - \frac{\mu}{2}\|\overline{(\boldsymbol{\mathcal{U}}_t * \boldsymbol{\mathcal{W}}_{t,\perp})}^{(j)}\|^2 + 9\mu\|\overline{(\boldsymbol{\mathcal{V}}_{\boldsymbol{\mathcal{X}}^\perp}^\top * \boldsymbol{\mathcal{V}}_{\boldsymbol{\mathcal{U}}_t*\boldsymbol{\mathcal{W}}_t})}^{(j)}\|\|\boldsymbol{\mathcal{X}}\|^2$$
$$+ 2\mu\|(\mathcal{A}^*\mathcal{A} - \mathcal{I})(\boldsymbol{\mathcal{X}}*\boldsymbol{\mathcal{X}}^\top - \boldsymbol{\mathcal{U}}_t*\boldsymbol{\mathcal{U}}_t^\top)\|\Big)\|\overline{(\boldsymbol{\mathcal{U}}_t * \boldsymbol{\mathcal{W}}_{t,\perp})}^{(j)}\|.$$

$\square$

The next lemma shows that the tensors $\boldsymbol{\mathcal{W}}_t$ and $\boldsymbol{\mathcal{W}}_{t+1}$ span approximately the same tensor column space.

**Lemma E.4.** *Assume that the following conditions hold*

$$\|\boldsymbol{\mathcal{U}}_t\| \le 3\|\boldsymbol{\mathcal{X}}\|, \tag{E.7}$$
$$\mu \le c\|\boldsymbol{\mathcal{X}}\|^{-2}\kappa^{-2} \tag{E.8}$$
$$\|\boldsymbol{\mathcal{V}}_{\boldsymbol{\mathcal{X}}^\perp}^\top * \boldsymbol{\mathcal{V}}_{\boldsymbol{\mathcal{U}}_t*\boldsymbol{\mathcal{W}}_t}\| \le c\kappa^{-1} \tag{E.9}$$
$$\|\overline{\boldsymbol{\mathcal{U}}_t * \boldsymbol{\mathcal{W}}_{t,\perp}}^{(j)}\| \le 2\sigma_{min}(\overline{\boldsymbol{\mathcal{U}}_t * \boldsymbol{\mathcal{W}}_t}^{(j)}), \tag{E.10}$$
$$\|(\mathcal{A}^*\mathcal{A} - \mathcal{I})(\boldsymbol{\mathcal{X}}*\boldsymbol{\mathcal{X}}^\top - \boldsymbol{\mathcal{U}}_t*\boldsymbol{\mathcal{U}}_t^\top)\| \le c\sigma_{min}^2(\overline{\boldsymbol{\mathcal{X}}}). \tag{E.11}$$

*Then it holds that*

$$\|\boldsymbol{\mathcal{W}}_{t,\perp}^\top * \boldsymbol{\mathcal{W}}_{t+1}\| \le \mu\Big(\tfrac{1}{4800}\sigma_{min}^2(\overline{\boldsymbol{\mathcal{X}}}) + \|\boldsymbol{\mathcal{U}}_t*\boldsymbol{\mathcal{W}}_t\|\|\boldsymbol{\mathcal{U}}_t*\boldsymbol{\mathcal{W}}_{t,\perp}\|\Big)\|\boldsymbol{\mathcal{V}}_{\boldsymbol{\mathcal{X}}^\perp}^\top * \boldsymbol{\mathcal{V}}_{\boldsymbol{\mathcal{U}}_t*\boldsymbol{\mathcal{W}}_t}\| + 4\mu\|(\mathcal{A}^*\mathcal{A} - \mathcal{I})(\boldsymbol{\mathcal{X}}*\boldsymbol{\mathcal{X}}^\top - \boldsymbol{\mathcal{U}}_t*\boldsymbol{\mathcal{U}}_t^\top)\|$$

*and* $\sigma_{min}(\overline{\boldsymbol{\mathcal{W}}_t^\top * \boldsymbol{\mathcal{W}}_{t+1}}^{(j)}) \ge \frac{1}{2}$, $1 \le j \le k$.

*Proof.* To bound the norm of $\boldsymbol{\mathcal{W}}_{t,\perp}^\top * \boldsymbol{\mathcal{W}}_{t+1}$, we will rewrite $\boldsymbol{\mathcal{W}}_{t,\perp}^\top * \boldsymbol{\mathcal{W}}_{t+1}$ in the Fourier domain with the help of Fourier slices of $\boldsymbol{\mathcal{V}}_{\boldsymbol{\mathcal{X}}}^\top * \boldsymbol{\mathcal{U}}_t$. First, note that due to the decomposition of the gradient iterates into the noise and signal term, it holds $\boldsymbol{\mathcal{V}}_{\boldsymbol{\mathcal{X}}}^\top * \boldsymbol{\mathcal{U}}_{t+1} = \boldsymbol{\mathcal{V}}_{\boldsymbol{\mathcal{X}}}^\top * \boldsymbol{\mathcal{U}}_{t+1} * \boldsymbol{\mathcal{W}}_{t+1} * \boldsymbol{\mathcal{W}}_{t+1}^\top$. This allows us to represent the corresponding $j$th Fourier slices of $\boldsymbol{\mathcal{V}}_{\boldsymbol{\mathcal{X}}}^\top * \boldsymbol{\mathcal{U}}_{t+1}$ as $\overline{V}_{\boldsymbol{\mathcal{X}}}^{(j)\mathrm{H}}\overline{U}_{t+1}^{(j)} = \overline{V}_{\boldsymbol{\mathcal{X}}}^{(j)\mathrm{H}}\overline{U}_{t+1}^{(j)}\overline{W}_{t+1}^{(j)}\overline{W}_{t+1}^{(j)\mathrm{H}}$, which means that for each $j$, the matrices $\overline{V}_{\boldsymbol{\mathcal{X}}}^{(j)\mathrm{H}}\overline{U}_{t+1}^{(j)}$ and $\overline{V}_{\boldsymbol{\mathcal{X}}}^{(j)\mathrm{H}}\overline{U}_{t+1}^{(j)}\overline{W}_{t+1}^{(j)}\overline{W}_{t+1}^{(j)\mathrm{H}}$ have the same kernel, and therefore $\overline{U}_{t+1}^{(j)\mathrm{H}}\overline{V}_{\boldsymbol{\mathcal{X}}}^{(j)}$ spans the same subspace as $\overline{W}_{t+1}^{(j)}\overline{W}_{t+1}^{(j)\mathrm{H}}\overline{U}_{t+1}^{(j)\mathrm{H}}\overline{V}_{\boldsymbol{\mathcal{X}}}^{(j)}$. Due to this and the following representation of the matrices

$$\overline{U}_t^{(j)} = \overline{U}_t^{(j)}\overline{W}_t^{(j)}\overline{W}_t^{(j)\mathrm{H}} + \overline{U}_t^{(j)}\overline{W}_t^{(j)}\overline{W}_t^{(j)\mathrm{H}} \tag{E.12}$$
$$\overline{U}_{t+1}^{(j)} = \overline{U}_{t+1}^{(j)}\overline{W}_{t+1}^{(j)}\overline{W}_{t+1}^{(j)\mathrm{H}} + \overline{U}_{t+1}^{(j)}\overline{W}_{t+1}^{(j)}\overline{W}_{t+1}^{(j)\mathrm{H}}, \tag{E.13}$$

we can apply Lemma H.4 to estimate the norm of $\overline{W}_{t,\perp}^{\mathrm{H}}\overline{W}_{t+1}^{(j)}$ taking $Y_1 = \overline{U}_{t+1}^{(j)}$ and $Y = \overline{U}_t^{(j)}$ and $Z$ as

$$Z^{(j)} := \overline{\left(\mathcal{A}^*\mathcal{A}(\boldsymbol{\mathcal{X}}*\boldsymbol{\mathcal{X}}^\top - \boldsymbol{\mathcal{U}}_t*\boldsymbol{\mathcal{U}}_t^\top)\right)}^{(j)}.$$

This gives us the following estimate

$$\|\overline{W}_{t,\perp}^{\mathrm{H}}\overline{W}_{t+1}^{(j)}\| \le \mu\left(1 + \mu\frac{\|Z^{(j)}\|\|\overline{U}_t^{(j)}\overline{W}_t^{(j)}\|}{\sigma_{\min}(\overline{V}_{\boldsymbol{\mathcal{X}}}^{(j)\mathrm{H}}\overline{U}_{t+1}^{(j)})}\right)\|\overline{U}_t^{(j)}\overline{W}_t^{(j)}\|\|\overline{U}_t^{(j)}\overline{W}_{t,\perp}^{(j)}\|\|\overline{V}_{\boldsymbol{\mathcal{X}}}^{(j)\mathrm{H}}V_{\overline{U}_t^{(j)}\overline{W}_t^{(j)}}\| \qquad \text{(E.14)}$$

$$+ \mu\frac{\|Z^{(j)} - (\overline{X}^{(j)}\overline{X}^{(j)\mathrm{H}} - \overline{U}_t^{(j)}\overline{U}_t^{(j)\mathrm{H}})\|}{\sigma_{\min}(\overline{V}_{\boldsymbol{\mathcal{X}}}^{(j)\mathrm{H}}\overline{U}_{t+1}^{(j)})}\|\overline{U}_t^{(j)}\overline{W}_{t,\perp}^{(j)}\|.$$

To proceed further with the upper bound above, we will first show that in each Fourier slice it holds that

$$\sigma_{\min}(\overline{V}_{\boldsymbol{\mathcal{X}}}^{(j)\mathrm{H}}\overline{U}_{t+1}^{(j)}) \ge \frac{1}{2}\sigma_{\min}(\overline{U}_t^{(j)}\overline{W}_t^{(j)}), \quad 1 \le j \le k. \qquad \text{(E.15)}$$

First, note that

$$\sigma_{\min}\left(\overline{V}_{\boldsymbol{\mathcal{X}}}^{(j)\mathrm{H}}\overline{U}_{t+1}^{(j)}\right) \ge \sigma_{\min}\left(\overline{V}_{\boldsymbol{\mathcal{X}}}^{(j)\mathrm{H}}\overline{U}_{t+1}^{(j)}\overline{W}_{t+1}^{(j)}\right) = \sigma_{\min}\left(\overline{V}_{\boldsymbol{\mathcal{X}}}^{(j)\mathrm{H}}(\mathrm{Id} + \mu Z^{(j)})\overline{U}_t^{(j)}\overline{W}_{t+1}^{(j)}\right)$$

$$= \sigma_{\min}\left(\overline{V}_{\boldsymbol{\mathcal{X}}}^{(j)\mathrm{H}}(\mathrm{Id} + \mu Z^{(j)})V_{\overline{U}_t^{(j)}\overline{W}_{t+1}^{(j)}}V_{\overline{U}_t^{(j)}\overline{W}_{t+1}^{(j)}}^{\mathrm{H}}\overline{U}_t^{(j)}\overline{W}_{t+1}^{(j)}\right)$$

$$\ge \sigma_{\min}\left(\overline{V}_{\boldsymbol{\mathcal{X}}}^{(j)\mathrm{H}}(\mathrm{Id} + \mu Z^{(j)})V_{\overline{U}_t^{(j)}\overline{W}_{t+1}^{(j)}}\right) \cdot \sigma_{\min}\left(V_{\overline{U}_t^{(j)}\overline{W}_{t+1}^{(j)}}^{\mathrm{H}}\overline{U}_t^{(j)}\overline{W}_{t+1}^{(j)}\right)$$

$$\ge \left(\sigma_{\min}\left(\overline{V}_{\boldsymbol{\mathcal{X}}}^{(j)\mathrm{H}}V_{\overline{U}_t^{(j)}\overline{W}_{t+1}^{(j)}}\right) - \mu\|\overline{V}_{\boldsymbol{\mathcal{X}}}^{(j)\mathrm{H}}Z^{(j)}V_{\overline{U}_t^{(j)}\overline{W}_{t+1}^{(j)}}\|\right) \cdot \sigma_{\min}\left(V_{\overline{U}_t^{(j)}\overline{W}_{t+1}^{(j)}}^{\mathrm{H}}\overline{U}_t^{(j)}\overline{W}_{t+1}^{(j)}\right).$$

Due to our assumption (E.9) on the principal angle $\|\boldsymbol{\mathcal{V}}_{\boldsymbol{\mathcal{X}}^\perp}^\top * \boldsymbol{\mathcal{V}}_{\boldsymbol{\mathcal{U}}_t*\boldsymbol{\mathcal{W}}_t}\|$ and the properties of the tensor slices, we have that

$$\sigma_{\min}\left(\overline{V}_{\boldsymbol{\mathcal{X}}}^{(j)\mathrm{H}}V_{\overline{U}_t^{(j)}\overline{W}_{t+1}^{(j)}}\right) \ge \sigma_{\min}\left(\overline{\boldsymbol{\mathcal{V}}_{\boldsymbol{\mathcal{X}}}^\top * \boldsymbol{\mathcal{V}}_{\boldsymbol{\mathcal{U}}_t*\boldsymbol{\mathcal{W}}_{t+1}}}\right) = \sqrt{1 - \left\|\overline{\boldsymbol{\mathcal{V}}_{\boldsymbol{\mathcal{X}}}^\top * \boldsymbol{\mathcal{V}}_{\boldsymbol{\mathcal{U}}_t*\boldsymbol{\mathcal{W}}_{t+1}}}\right\|^2} \ge \frac{3}{4},$$

where that last inequality can be guaranteed by choosing $c > 0$ small enough. Thus, to show that relation (E.15) holds we need to demonstrate that $\mu\|\overline{V}_{\boldsymbol{\mathcal{X}}}^{(j)\mathrm{H}}Z^{(j)}V_{\overline{U}_t^{(j)}\overline{W}_{t+1}^{(j)}}\|$ be bounded from above by $\frac{1}{4}$. For this, we will proceed as follows

$$\mu\|\overline{V}_{\boldsymbol{\mathcal{X}}}^{(j)\mathrm{H}}Z^{(j)}V_{\overline{U}_t^{(j)}\overline{W}_{t+1}^{(j)}}\| \le \mu\|Z^{(j)}\| \le \mu\|Z^{(j)} - (\overline{X}^{(j)}\overline{X}^{(j)\mathrm{H}} - \overline{U}_t^{(j)}\overline{U}_t^{(j)\mathrm{H}})\| + \mu\|\overline{X}^{(j)}\overline{X}^{(j)\mathrm{H}} - \overline{U}_t^{(j)}\overline{U}_t^{(j)\mathrm{H}}\|. \quad \text{(E.16)}$$

By the definition of $Z^{(j)}$, for the first summand from above we have

$$\left\|Z^{(j)} - (\overline{X}^{(j)}\overline{X}^{(j)\mathrm{H}} - \overline{U}_t^{(j)}\overline{U}_t^{(j)\mathrm{H}})\right\| = \left\|\overline{\left(\mathcal{A}^*\mathcal{A}(\boldsymbol{\mathcal{X}}*\boldsymbol{\mathcal{X}}^\top - \boldsymbol{\mathcal{U}}_t*\boldsymbol{\mathcal{U}}_t^\top)\right)}^{(j)} - (\overline{X}^{(j)}\overline{X}^{(j)\mathrm{H}} - \overline{U}_t^{(j)}\overline{U}_t^{(j)\mathrm{H}})\right\|$$

$$= \left\|\overline{\left((\mathcal{I} - \mathcal{A}^*\mathcal{A})(\boldsymbol{\mathcal{X}}*\boldsymbol{\mathcal{X}}^\top - \boldsymbol{\mathcal{U}}_t*\boldsymbol{\mathcal{U}}_t^\top)\right)}^{(j)}\right\|$$

$$\le \left\|(\mathcal{I} - \mathcal{A}^*\mathcal{A})(\boldsymbol{\mathcal{X}}*\boldsymbol{\mathcal{X}}^\top - \boldsymbol{\mathcal{U}}_t*\boldsymbol{\mathcal{U}}_t^\top)\right\|$$

and for the second summand, it holds that

$$\|\overline{X}^{(j)}\overline{X}^{(j)\mathrm{H}} - \overline{U}_t^{(j)}\overline{U}_t^{(j)\mathrm{H}}\| \le \|\boldsymbol{\mathcal{X}}*\boldsymbol{\mathcal{X}}^\top - \boldsymbol{\mathcal{U}}_t*\boldsymbol{\mathcal{U}}_t^\top\| \le \|\boldsymbol{\mathcal{X}}\|^2 + \|\boldsymbol{\mathcal{U}}_t\|^2.$$

This allows us to proceed with inequality (E.16) as

$$\mu\|\overline{V}_{\boldsymbol{\mathcal{X}}}^{(j)\mathrm{H}}Z^{(j)}V_{\overline{U}_t^{(j)}\overline{W}_{t+1}^{(j)}}\| \le \mu\left\|(\mathcal{I} - \mathcal{A}^*\mathcal{A})(\boldsymbol{\mathcal{X}}*\boldsymbol{\mathcal{X}}^\top - \boldsymbol{\mathcal{U}}_t*\boldsymbol{\mathcal{U}}_t^\top)\right\| + \mu(\|\boldsymbol{\mathcal{X}}\|^2 + \|\boldsymbol{\mathcal{U}}_t\|^2)$$

$$\le \mu\left\|(\mathcal{I} - \mathcal{A}^*\mathcal{A})(\boldsymbol{\mathcal{X}}*\boldsymbol{\mathcal{X}}^\top - \boldsymbol{\mathcal{U}}_t*\boldsymbol{\mathcal{U}}_t^\top)\right\| + 10\mu\|\boldsymbol{\mathcal{X}}\|^2) \le \mu c\sigma_{\min}^2(\overline{\boldsymbol{\mathcal{X}}}) + 11\mu\|\boldsymbol{\mathcal{X}}\|^2 \le \frac{1}{2},$$

where in the first line we used assumption (E.7), and in the second assumption(E.11). The third inequality above follows from our assumption on $\mu$ and sufficiently small constant $c > 0$. This, in turn, shows that relation (E.15) holds and we can proceed with (E.14) in the following manner

$$\|\overline{W}_{t,\perp}^{\mathrm{H}} \overline{W}_{t+1}^{(j)}\| \leq \mu \left( 1 + 2\mu \frac{\|Z^{(j)}\| \|\overline{U}_t^{(j)} \overline{W}_t^{(j)}\|}{\sigma_{\min}(\overline{U}_t^{(j)} \overline{W}_t^{(j)})} \right) \|\overline{U}_t^{(j)} \overline{W}_t^{(j)}\| \|\overline{U}_t^{(j)} \overline{W}_{t,\perp}^{(j)}\| \|\overline{V}_{\boldsymbol{\mathcal{X}}}^{(j)\mathrm{H}} V_{\overline{U}_t^{(j)} \overline{W}_t^{(j)}}\|$$
$$+ 2\mu \frac{\|Z^{(j)} - (\overline{X}^{(j)} \overline{X}^{(j)\mathrm{H}} - \overline{U}_t^{(j)} \overline{U}_t^{(j)\mathrm{H}})\|}{\sigma_{\min}(\overline{U}_t^{(j)} \overline{W}_t^{(j)})} \|\overline{U}_t^{(j)} \overline{W}_{t,\perp}^{(j)}\|.$$

Now, using assumption (E.10) and the definition of $Z^{(j)}$, we have

$$\|\overline{W}_{t,\perp}^{\mathrm{H}} \overline{W}_{t+1}^{(j)}\| \leq \mu \|\overline{V}_{\boldsymbol{\mathcal{X}}^\perp}^{(j)\mathrm{H}} V_{\overline{U}_t^{(j)} \overline{W}_t^{(j)}}\| \|\overline{U}_t^{(j)} \overline{W}_t^{(j)}\| \|\overline{U}_t^{(j)} \overline{W}_{t,\perp}^{(j)}\|$$
$$+ 4\mu \|\overline{\left(\mathcal{A}^* \mathcal{A}(\boldsymbol{\mathcal{X}} * \boldsymbol{\mathcal{X}}^\top - \boldsymbol{\mathcal{U}}_t * \boldsymbol{\mathcal{U}}_t^\top)\right)}^{(j)} - (\overline{X}^{(j)} \overline{X}^{(j)\mathrm{H}} - \overline{U}_t^{(j)} \overline{U}_t^{(j)\mathrm{H}})\|$$
$$+ 4\mu^2 \|\overline{\left(\mathcal{A}^* \mathcal{A}(\boldsymbol{\mathcal{X}} * \boldsymbol{\mathcal{X}}^\top - \boldsymbol{\mathcal{U}}_t * \boldsymbol{\mathcal{U}}_t^\top)\right)}^{(j)}\| \|\overline{U}_t^{(j)} \overline{W}_t^{(j)}\|^2 \|\overline{V}_{\boldsymbol{\mathcal{X}}^\perp}^{(j)\mathrm{H}} V_{\overline{U}_t^{(j)} \overline{W}_t^{(j)}}\|$$
$$\leq \mu \|\overline{V}_{\boldsymbol{\mathcal{X}}^\perp}^{(j)\mathrm{H}} V_{\overline{U}_t^{(j)} \overline{W}_t^{(j)}}\| \|\overline{U}_t^{(j)} \overline{W}_t^{(j)}\| \|\overline{U}_t^{(j)} \overline{W}_{t,\perp}^{(j)}\|$$
$$+ 4\mu \|(\mathcal{A}^* \mathcal{A} - \mathcal{I})(\boldsymbol{\mathcal{X}} * \boldsymbol{\mathcal{X}}^\top - \boldsymbol{\mathcal{U}}_t * \boldsymbol{\mathcal{U}}_t^\top)\|$$
$$+ 4\mu^2 \|\mathcal{A}^* \mathcal{A}(\boldsymbol{\mathcal{X}} * \boldsymbol{\mathcal{X}}^\top - \boldsymbol{\mathcal{U}}_t * \boldsymbol{\mathcal{U}}_t^\top)\| \|\overline{U}_t^{(j)} \overline{W}_t^{(j)}\|^2 \|\overline{V}_{\boldsymbol{\mathcal{X}}^\perp}^{(j)\mathrm{H}} V_{\overline{U}_t^{(j)} \overline{W}_t^{(j)}}\|.$$

In the last inequality, we used the tensor norm as the maximum norm in each Fourier slice. Note that, similarly to one of the estimates above, we get

$$\|\mathcal{A}^* \mathcal{A}(\boldsymbol{\mathcal{X}} * \boldsymbol{\mathcal{X}}^\top - \boldsymbol{\mathcal{U}}_t * \boldsymbol{\mathcal{U}}_t^\top)\| \leq \|\boldsymbol{\mathcal{X}} * \boldsymbol{\mathcal{X}}^\top - \boldsymbol{\mathcal{U}}_t * \boldsymbol{\mathcal{U}}_t^\top\| + \|(\mathcal{A}^* \mathcal{A} - \mathcal{I})(\boldsymbol{\mathcal{X}} * \boldsymbol{\mathcal{X}}^\top - \boldsymbol{\mathcal{U}}_t * \boldsymbol{\mathcal{U}}_t^\top)\|$$
$$\leq \|\boldsymbol{\mathcal{X}}\|^2 + \|\boldsymbol{\mathcal{U}}_t\|^2 + c\sigma_{\min}^2(\overline{\boldsymbol{\mathcal{X}}}) \leq 11\|\boldsymbol{\mathcal{X}}\|^2 \tag{E.17}$$

where the last line holds due to the assumption $\|\boldsymbol{\mathcal{U}}_t\| \leq 3\|\boldsymbol{\mathcal{X}}\|$ and that $c$ is small enough.

Now, since $\mu \leq c\|\boldsymbol{\mathcal{X}}\|^{-2}\kappa^{-2}$, $\|\overline{U}_t^{(j)} \overline{W}_t^{(j)}\| \leq \|\boldsymbol{\mathcal{U}}_t\| \leq 3\|\boldsymbol{\mathcal{X}}\|$ and $\|\overline{U}_t^{(j)} \overline{W}_{t,\perp}^{(j)}\| \leq \|\boldsymbol{\mathcal{U}}_t\| \leq 3\|\boldsymbol{\mathcal{X}}\|$, constant $c > 0$ can be chosen so that $4\mu \cdot 11\|\boldsymbol{\mathcal{X}}\|^2 \leq \frac{1}{4800}\sigma_{\min}^2(\overline{\boldsymbol{\mathcal{X}}})$, together with (E.17) and (E.11) we can proceed with the estimation of $\overline{W}_{t,\perp}^{\mathrm{H}} \overline{W}_{t+1}^{(j)}$ as

$$\|\overline{W}_{t,\perp}^{(j)\mathrm{H}} \overline{W}_{t+1}^{(j)}\| \leq \mu \left( \tfrac{1}{4800}\sigma_{\min}^2(\overline{\boldsymbol{\mathcal{X}}}) + 9\|\boldsymbol{\mathcal{X}}\|^2 \right) \|\overline{V}_{\boldsymbol{\mathcal{X}}^\perp}^{(j)\mathrm{H}} V_{\overline{U}_t^{(j)} \overline{W}_t^{(j)}}\| + 4\mu c\sigma_{\min}^2(\overline{\boldsymbol{\mathcal{X}}}).$$

Using the assumption $\mu \leq c\|\boldsymbol{\mathcal{X}}\|^{-2}$ and choosing $c > 0$ small enough, we obtain that $\|\overline{W}_{t,\perp}^{(j)\mathrm{H}} \overline{W}_{t+1}^{(j)}\| \leq \frac{1}{2}$. Note that this implies that $\sigma_{\min}(\overline{\boldsymbol{\mathcal{W}}_t^\top * \boldsymbol{\mathcal{W}}_{t+1}}^{(j)}) = \sqrt{1 - \|\overline{W}_{t,\perp}^{(j)\mathrm{H}} \overline{W}_{t+1}^{(j)}\|^2} \geq \frac{1}{2}$, which finishes the proof. $\qquad\square$

**Lemma E.5.** *Assume that the following conditions hold*

$$\|\overline{\boldsymbol{\mathcal{U}}_t * \boldsymbol{\mathcal{W}}_{t,\perp}}^{(j)}\| \leq 2\sigma_{min}(\overline{\boldsymbol{\mathcal{U}}_t * \boldsymbol{\mathcal{W}}_t}^{(j)}), \tag{E.18}$$
$$\|\boldsymbol{\mathcal{U}}_t\| \leq 3\|\boldsymbol{\mathcal{X}}\|, \tag{E.19}$$
$$\|\boldsymbol{\mathcal{V}}_{\boldsymbol{\mathcal{X}}^\perp}^\top * \boldsymbol{\mathcal{V}}_{\boldsymbol{\mathcal{U}}_t * \boldsymbol{\mathcal{W}}_t}\| \leq \tilde{c} \tag{E.20}$$
$$\mu \leq c\|\boldsymbol{\mathcal{X}}\|^{-2}\kappa^{-2} \tag{E.21}$$
$$\|\boldsymbol{\mathcal{U}}_t * \boldsymbol{\mathcal{W}}_{t,\perp}\| \leq c\kappa^{-2}\|\boldsymbol{\mathcal{X}}\| \tag{E.22}$$
$$\|(\mathcal{A}^* \mathcal{A} - \mathcal{I})(\boldsymbol{\mathcal{X}} * \boldsymbol{\mathcal{X}}^\top - \boldsymbol{\mathcal{U}}_t * \boldsymbol{\mathcal{U}}_t^\top)\| \leq c\sigma_{min}^2(\overline{\boldsymbol{\mathcal{X}}}). \tag{E.23}$$

*Then the angle between the column space of the signal term $\boldsymbol{\mathcal{U}}_t * \boldsymbol{\mathcal{W}}_t$ and column space of $\boldsymbol{\mathcal{X}}$ stays sufficiently small from one iteration to another, namely*

$$\|\boldsymbol{\mathcal{V}}_{\boldsymbol{\mathcal{X}}^\perp}^\top * \boldsymbol{\mathcal{V}}_{\boldsymbol{\mathcal{U}}_{t+1} * \boldsymbol{\mathcal{W}}_{t+1}}\| \leq \left( 1 - \frac{\mu}{4}\sigma_{min}^2(\overline{\boldsymbol{\mathcal{X}}}) \right) \|\boldsymbol{\mathcal{V}}_{\boldsymbol{\mathcal{X}}^\perp}^\top * \boldsymbol{\mathcal{V}}_{\boldsymbol{\mathcal{U}}_t * \boldsymbol{\mathcal{W}}_t}\|$$
$$+ 150\mu\|(\mathcal{A}^* \mathcal{A} - \mathcal{I})(\boldsymbol{\mathcal{X}} * \boldsymbol{\mathcal{X}}^\top - \boldsymbol{\mathcal{U}}_t * \boldsymbol{\mathcal{U}}_t^\top)\| + 500\mu^2\|\boldsymbol{\mathcal{X}} * \boldsymbol{\mathcal{X}}^\top - \boldsymbol{\mathcal{U}}_t * \boldsymbol{\mathcal{U}}_t^\top\|^2.$$

*Proof.* To estimate the principal angle $\|\mathcal{V}_{\mathcal{X}^\perp}^\top * \mathcal{V}_{\mathcal{U}_{t+1}*\mathcal{W}_{t+1}}\|$, we first investigate the tensor-column subspace of $\mathcal{U}_{t+1} * \mathcal{W}_{t+1}$. By the definition of $\mathcal{U}_{t+1}$ and $\mathcal{W}_t * \mathcal{W}_t^\top + \mathcal{W}_{t,\perp} * \mathcal{W}_{t,\perp}^\top = \mathcal{I}$, we have

$$\mathcal{U}_{t+1} * \mathcal{W}_{t+1} = \Big(\mathcal{I} + \mu(\mathcal{A}^*\mathcal{A})(\mathcal{X} * \mathcal{X}^\top - \mathcal{U}_t * \mathcal{U}_t^\top)\Big) * \mathcal{U}_t * \mathcal{W}_{t+1}$$
$$= (\mathcal{I} + \mu\mathcal{Z}) * \mathcal{U}_t * \mathcal{W}_t * \mathcal{W}_t^\top * \mathcal{W}_{t+1} + (\mathcal{I} + \mu\mathcal{Z}) * \mathcal{U}_t * \mathcal{W}_{t,\perp} * \mathcal{W}_{t,\perp}^\top * \mathcal{W}_{t+1}.$$

where we use notation $\mathcal{Z} := (\mathcal{A}^*\mathcal{A})(\mathcal{X} * \mathcal{X}^\top - \mathcal{U}_t * \mathcal{U}_t^\top)$. This allows to represent $j$th slice of $\mathcal{U}_{t+1} * \mathcal{W}_{t+1}$ in the Fourier domain as

$$\overline{U}_{t+1}^{(j)} \overline{W}_{t+1}^{(j)} = (\mathrm{Id} + \mu\overline{Z}^{(j)})\overline{U}_t^{(j)} \overline{W}_t^{(j)} \overline{W}_t^{(j)\mathrm{H}} \overline{W}_{t+1}^{(j)} + (\mathrm{Id} + \mu\overline{Z}^{(j)})\overline{U}_t^{(j)} \overline{W}_{t,\perp}^{(j)} \overline{W}_{t,\perp}^{(j)\mathrm{H}} \overline{W}_{t+1}^{(j)}.$$

with $\overline{Z}^{(j)} = \overline{(\mathcal{A}^*\mathcal{A})(\mathcal{X} * \mathcal{X}^\top - \mathcal{U}_t * \mathcal{U}_t^\top)}^{(j)}$. Because of this representation and decomposition (E.12), to bound the principal angle between $\mathcal{U}_{t+1} * \mathcal{W}_{t+1}$ and $\mathcal{X}$, we want to apply inequality (H.5) from Lemma H.4, but for this we first need to check whether for

$$P^{(j)} := \overline{U}_t^{(j)} \overline{W}_{t,\perp}^{(j)} \overline{W}_{t,\perp}^{(j)\mathrm{H}} \overline{W}_{t+1}^{(j)} \Big(V_{\overline{U}_t^{(j)}\overline{W}_t^{(j)}}^{\mathrm{H}} \overline{U}_t^{(j)} \overline{W}_t^{(j)} \overline{W}_t^{(j)\mathrm{H}} \overline{W}_{t+1}^{(j)}\Big)^{-1} V_{\overline{U}_t^{(j)}\overline{W}_t^{(j)}}^{\mathrm{H}}$$

the following applies

$$\|\mu Z^{(j)} + P^{(j)} + \mu Z^{(j)} P^{(j)}\| \le 1.$$

For convenience, we denote $B^{(j)} := \mu Z^{(j)} + P^{(j)} + \mu Z^{(j)} P^{(j)}$. Using the triangular inequality and submultiplicativity of the norm, we bet the first simple bound on the norm of $B^{(j)}$

$$\|B^{(j)}\| \le \mu\|Z^{(j)}\| + (1 + \mu\|Z^{(j)}\|)\|P^{(j)}\| \tag{E.24}$$

Note that $P^{(j)}$ can be rewritten as

$$P^{(j)} = \overline{U}_t^{(j)} \overline{W}_{t,\perp}^{(j)} \overline{W}_{t,\perp}^{(j)\mathrm{H}} \overline{W}_{t+1}^{(j)} \Big(\overline{W}_t^{(j)\mathrm{H}} \overline{W}_{t+1}^{(j)}\Big)^{-1} \Big(V_{\overline{U}_t^{(j)}\overline{W}_t^{(j)}}^{\mathrm{H}} \overline{U}_t^{(j)} \overline{W}_t^{(j)}\Big)^{-1} V_{\overline{U}_t^{(j)}\overline{W}_t^{(j)}}^{\mathrm{H}},$$

which allows for the following estimate of its norm

$$\|P^{(j)}\| \le \|\overline{U}_t^{(j)} \overline{W}_{t,\perp}^{(j)}\|\|\overline{W}_{t,\perp}^{(j)\mathrm{H}} \overline{W}_{t+1}^{(j)}\|\Big\|\Big(\overline{W}_t^{(j)\mathrm{H}} \overline{W}_{t+1}^{(j)}\Big)^{-1}\Big\|\Big\|\Big(V_{\overline{U}_t^{(j)}\overline{W}_t^{(j)}}^{\mathrm{H}} \overline{U}_t^{(j)} \overline{W}_t^{(j)}\Big)^{-1}\Big\|\|V_{\overline{U}_t^{(j)}\overline{W}_t^{(j)}}^{\mathrm{H}}\|$$
$$\le \frac{\|\overline{U}_t^{(j)} \overline{W}_{t,\perp}^{(j)}\|\|\overline{W}_{t,\perp}^{(j)\mathrm{H}} \overline{W}_{t+1}^{(j)}\|}{\sigma_{\min}(\overline{W}_t^{(j)\mathrm{H}} \overline{W}_{t+1}^{(j)}) \cdot \sigma_{\min}(\overline{U}_t^{(j)} \overline{W}_t^{(j)})}.$$

From here, using assumption (E.18) and a lower bound on $\sigma_{\min}(\overline{W}_t^{(j)\mathrm{H}} \overline{W}_{t+1}^{(j)})$ from Lemma E.4, we get

$$\|P^{(j)}\| \le 4\|\overline{W}_{t,\perp}^{(j)\mathrm{H}} \overline{W}_{t+1}^{(j)}\|. \tag{E.25}$$

Using this and the definition of $Z^{(j)}$, we have

$$\|B^{(j)}\| \le \mu\|\overline{(\mathcal{A}^*\mathcal{A})(\mathcal{X} * \mathcal{X}^\top - \mathcal{U}_t * \mathcal{U}_t^\top)}^{(j)}\| + 4\Big(1 + \mu\|\overline{(\mathcal{A}^*\mathcal{A})(\mathcal{X} * \mathcal{X}^\top - \mathcal{U}_t * \mathcal{U}_t^\top)}^{(j)}\|\Big)\|\overline{W}_{t,\perp}^{(j)\mathrm{H}} \overline{W}_{t+1}^{(j)}\|. \tag{E.26}$$

Due to the assumption on $\mu$, we can bound $\mu\|\overline{(\mathcal{A}^*\mathcal{A})(\mathcal{X} * \mathcal{X}^\top - \mathcal{U}_t * \mathcal{U}_t^\top)}^{(j)}\|$ as follows

$$\mu\|\overline{(\mathcal{A}^*\mathcal{A})(\mathcal{X} * \mathcal{X}^\top - \mathcal{U}_t * \mathcal{U}_t^\top)}^{(j)}\| \le \mu\|\overline{(\mathcal{A}^*\mathcal{A})(\mathcal{X} * \mathcal{X}^\top - \mathcal{U}_t * \mathcal{U}_t^\top)}^{(j)}\|$$
$$\le \mu\|(\mathcal{I} - \mathcal{A}^*\mathcal{A})(\mathcal{X} * \mathcal{X}^\top - \mathcal{U}_t * \mathcal{U}_t^\top)\| + \mu\|\mathcal{X} * \mathcal{X}^\top - \mathcal{U}_t * \mathcal{U}_t^\top\|$$
$$\le \mu(c\sigma_{\min}^2(\overline{\mathcal{X}}) + 10\|\mathcal{X}\|^2) \le 1$$

where in the two last inequalities we use assumptions (E.23), (E.19) and (E.21) with the fact for the learning rate constant $c > 0$ can be chosen sufficiently small.

This, in turn, allows us to proceed with inequality (E.26) as

$$\|B^{(j)}\| \leq \mu\|\overline{(\mathcal{A}^*\mathcal{A})(\mathcal{X}*\mathcal{X}^\top - \mathcal{U}_t*\mathcal{U}_t^\top)^{(j)}}\| + 8\|\overline{W}_{t,\perp}^{(j)\mathrm{H}}\overline{W}_{t+1}^{(j)}\|. \tag{E.27}$$

Now, applying the bound on $\|\overline{W}_{t,\perp}^{(j)\mathrm{H}}\overline{W}_{t+1}^{(j)}\| \leq \|\mathcal{W}_{t,\perp}^\top * \mathcal{W}_{t+1}\|$ from Lemma E.4 and similar transformation for $\|\overline{(\mathcal{A}^*\mathcal{A})(\mathcal{X}*\mathcal{X}^\top - \mathcal{U}_t*\mathcal{U}_t^\top)^{(j)}}\|$ as above, we come the following result in (E.27)

$$\|B^{(j)}\| \leq \mu\|\mathcal{X}*\mathcal{X}^\top - \mathcal{U}_t*\mathcal{U}_t^\top\| + \mu\Big(\tfrac{1}{600}\sigma_{\min}(\overline{\mathcal{X}})^2 + 8\|\mathcal{U}_t*\mathcal{W}_t\|\|\mathcal{U}_t*\mathcal{W}_{t,\perp}\|\Big)\|\mathcal{V}_{\mathcal{X}^\perp}^\top * \mathcal{V}_{\mathcal{U}_t*\mathcal{W}_t}\|$$
$$+ 33\mu\|(\mathcal{A}^*\mathcal{A} - \mathcal{I})(\mathcal{X}*\mathcal{X}^\top - \mathcal{U}_t*\mathcal{U}_t^\top)\|$$

To show that this bound above can be made smaller than one, we use assumptions (E.22), (E.23) and that $\|\mathcal{U}_t*\mathcal{W}_t\| \leq \|\mathcal{U}\| \leq 2\|\mathcal{X}\|$, which leads to

$$\|B^{(j)}\| \leq \mu\|\mathcal{X}*\mathcal{X}^\top - \mathcal{U}_t*\mathcal{U}_t^\top\| + \mu\Big(\tfrac{1}{600}\sigma_{\min}(\overline{\mathcal{X}})^2 + 8c\frac{\sigma_{\min}(\overline{\mathcal{X}})}{\kappa^2}\cdot 3\|\mathcal{X}\|\Big)\|\mathcal{V}_{\mathcal{X}^\perp}^\top * \mathcal{V}_{\mathcal{U}_t*\mathcal{W}_t}\| + 33\mu c\sigma_{\min}^2(\overline{\mathcal{X}})$$
$$\leq \mu 10\|\mathcal{X}\|^2 + \mu c\frac{1}{300}\sigma_{\min}^2(\overline{\mathcal{X}}) + 33\mu c\sigma_{\min}^2(\overline{\mathcal{X}}) \leq 1,$$

with the last inequality following from the assumption on $\mu$. In such a way, we check the conditions of Lemma H.4 to be able to apply inequality (H.5). This gives

$$\|V_{\mathcal{X}^\perp}^{(j)\mathrm{H}}V_{\overline{U}_{t+1}^{(j)}\overline{W}_{t+1}^{(j)}}\| \leq \|V_{\mathcal{X}^\perp}^{(j)\mathrm{H}}V_{\overline{U}_t^{(j)}\overline{W}_t^{(j)}}\| \left(1 - \frac{\mu}{2}\sigma_{\min}^2(\overline{X}^{(j)}) + \mu\|\overline{U}_t^{(j)}\overline{W}_{t,\perp}^{(j)}\|\right)$$
$$+ \mu\|Z^{(j)} - (\overline{X}^{(j)}\overline{X}^{(j)\mathrm{H}} - \overline{U}_t^{(j)}\overline{U}_t^{(j)\mathrm{H}})\| + \left(1 + \mu\|Z^{(j)}\|\right)\frac{2\|\overline{W}_{t,\perp}^{(j)\mathrm{H}}\overline{W}_{t+1}^{(j)}\|\|\overline{U}_t^{(j)}\overline{W}_{t,\perp}^{(j)}\|}{\sigma_{\min}(\overline{W}_t^{(j)\mathrm{H}}\overline{W}_{t+1}^{(j)})\sigma_{\min}(\overline{U}_t^{(j)}\overline{W}_t^{(j)})}$$
$$+ 57\left(\mu\|Z^{(j)}\| + (1 + \mu\|Z^{(j)}\|)\frac{\|\overline{W}_{t,\perp}^{(j)\mathrm{H}}\overline{W}_{t+1}^{(j)}\|\|\overline{U}_t^{(j)}\overline{W}_{t,\perp}^{(j)}\|}{\sigma_{\min}(\overline{W}_t^{(j)\mathrm{H}}\overline{W}_{t+1}^{(j)})\sigma_{\min}(\overline{U}_t^{(j)}\overline{W}_t^{(j)})}\right)^2.$$

Applying again assumption (E.18) and a lower bound on $\sigma_{\min}(\overline{W}_t^{(j)\mathrm{H}}\overline{W}_{t+1}^{(j)})$ from Lemma E.4 as for (E.25), in addition to (E.22), we get

$$\|V_{\mathcal{X}^\perp}^{(j)\mathrm{H}}V_{\overline{U}_{t+1}^{(j)}\overline{W}_{t+1}^{(j)}}\| \leq \|V_{\mathcal{X}^\perp}^{(j)\mathrm{H}}V_{\overline{U}_t^{(j)}\overline{W}_t^{(j)}}\| \left(1 - \frac{\mu}{3}\sigma_{\min}^2(\overline{X}^{(j)})\right) + \mu\|Z^{(j)} - (\overline{X}^{(j)}\overline{X}^{(j)\mathrm{H}} - \overline{U}_t^{(j)}\overline{U}_t^{(j)\mathrm{H}})\|$$
$$+ 8(1 + \mu\|Z^{(j)}\|)\|\overline{W}_{t,\perp}^{(j)\mathrm{H}}\overline{W}_{t+1}^{(j)}\| + 57\left(\mu\|Z^{(j)}\| + 4(1 + \mu\|Z^{(j)}\|)\|\overline{W}_{t,\perp}^{(j)\mathrm{H}}\overline{W}_{t+1}^{(j)}\|\right)^2.$$

Now, making $\left(1 + \mu\|Z^{(j)}\|\right) \leq 3$ by choosing $c > 0$ small enough and using the properties of the terms involved, the above inequality gets the following view

$$\|V_{\mathcal{X}^\perp}^{(j)\mathrm{H}}V_{\overline{U}_{t+1}^{(j)}\overline{W}_{t+1}^{(j)}}\| \leq \|V_{\mathcal{X}^\perp}^{(j)\mathrm{H}}V_{\overline{U}_t^{(j)}\overline{W}_t^{(j)}}\| \left(1 - \frac{\mu}{3}\sigma_{\min}^2(\overline{\mathcal{X}})\right) + \mu\|(\mathcal{I} - \mathcal{A}^*\mathcal{A})(\mathcal{X}*\mathcal{X}^\top - \mathcal{U}_t*\mathcal{U}_t^\top)\|$$
$$+ 32\|\overline{W}_{t,\perp}^{(j)\mathrm{H}}\overline{W}_{t+1}^{(j)}\| + 57\left(\mu\|Z^{(j)}\| + 12\|\overline{W}_{t,\perp}^{(j)\mathrm{H}}\overline{W}_{t+1}^{(j)}\|\right)^2. \tag{E.28}$$

To proceed further with (E.28), we will first do several auxiliary estimates. We start by bounding the norm $\|\overline{W}_{t,\perp}^{(j)\mathrm{H}}\overline{W}_{t+1}^{(j)}\|$. Since it holds that $\|\overline{W}_{t,\perp}^{(j)\mathrm{H}}\overline{W}_{t+1}^{(j)}\| \leq \|\mathcal{W}_{t,\perp}^\top * \mathcal{W}_{t+1}\|$, from Lemma E.4, one gets

$$\|\overline{W}_{t,\perp}^{(j)\mathrm{H}}\overline{W}_{t+1}^{(j)}\| \leq \mu\Big(\tfrac{1}{4800}\sigma_{\min}^2(\overline{\mathcal{X}}) + \|\mathcal{U}_t*\mathcal{W}_t\|\|\mathcal{U}_t*\mathcal{W}_{t,\perp}\|\Big)\|\mathcal{V}_{\mathcal{X}^\perp}^\top * \mathcal{V}_{\mathcal{U}_t*\mathcal{W}_t}\|$$
$$+ 4\mu\|(\mathcal{A}^*\mathcal{A} - \mathcal{I})(\mathcal{X}*\mathcal{X}^\top - \mathcal{U}_t*\mathcal{U}_t^\top)\|$$
$$\leq \mu\Big(\tfrac{1}{4800}\sigma_{\min}^2(\overline{\mathcal{X}}) + 3c\sigma_{\min}^2(\overline{\mathcal{X}})\Big)\|\mathcal{V}_{\mathcal{X}^\perp}^\top * \mathcal{V}_{\mathcal{U}_t*\mathcal{W}_t}\| + 4\mu\|(\mathcal{A}^*\mathcal{A} - \mathcal{I})(\mathcal{X}*\mathcal{X}^\top - \mathcal{U}_t*\mathcal{U}_t^\top)\|$$
$$\leq \tfrac{1}{2400}\mu\sigma_{\min}^2(\overline{\mathcal{X}})\|\mathcal{V}_{\mathcal{X}^\perp}^\top * \mathcal{V}_{\mathcal{U}_t*\mathcal{W}_t}\| + 4\mu\|(\mathcal{A}^*\mathcal{A} - \mathcal{I})(\mathcal{X}*\mathcal{X}^\top - \mathcal{U}_t*\mathcal{U}_t^\top)\| \tag{E.29}$$

where we use in the second inequality that $\|\boldsymbol{\mathcal{U}}_t * \boldsymbol{\mathcal{W}}_t\| \le \|\boldsymbol{\mathcal{U}}_t\| \le 3\|\boldsymbol{\mathcal{X}}\|$ and $\|\boldsymbol{\mathcal{U}}_t * \boldsymbol{\mathcal{W}}_{t,\perp}\| \le c\kappa^{-2}\|\boldsymbol{\mathcal{X}}\|$ by assumption, and in the last line that $c > 0$ can be chosen small enough. Using this estimate, let us bound from above the squared term in (E.28) as follows

$$\mu\|Z^{(j)}\| + 12\|\overline{W}_{t,\perp}^{(j)\mathrm{H}}\overline{W}_{t+1}^{(j)}\| \le \mu\|Z^{(j)}\| + \mu\frac{\sigma_{\min}^2(\overline{\boldsymbol{\mathcal{X}}})}{200}\|\boldsymbol{\mathcal{V}}_{\boldsymbol{\mathcal{X}}^\perp}^\top * \boldsymbol{\mathcal{V}}_{\boldsymbol{\mathcal{U}}_t*\boldsymbol{\mathcal{W}}_t}\| + 48\mu\|(\mathcal{A}^*\mathcal{A} - \mathcal{I})(\boldsymbol{\mathcal{X}} * \boldsymbol{\mathcal{X}}^\top - \boldsymbol{\mathcal{U}}_t * \boldsymbol{\mathcal{U}}_t^\top)\|$$
$$\le \mu\|\overline{X}^{(j)}\overline{X}^{(j)\mathrm{H}} - \overline{U}_t^{(j)}\overline{U}_t^{(j)\mathrm{H}}\| + \mu\frac{\sigma_{\min}^2(\overline{\boldsymbol{\mathcal{X}}})}{200}\|\boldsymbol{\mathcal{V}}_{\boldsymbol{\mathcal{X}}^\perp}^\top * \boldsymbol{\mathcal{V}}_{\boldsymbol{\mathcal{U}}_t*\boldsymbol{\mathcal{W}}_t}\|$$
$$+ 49\mu\|(\mathcal{A}^*\mathcal{A} - \mathcal{I})(\boldsymbol{\mathcal{X}} * \boldsymbol{\mathcal{X}}^\top - \boldsymbol{\mathcal{U}}_t * \boldsymbol{\mathcal{U}}_t^\top)\|.$$

From here, using Jensen's inequality, we obtain

$$(\mu\|Z^{(j)}\| + 12\|\overline{W}_{t,\perp}^{(j)\mathrm{H}}\overline{W}_{t+1}^{(j)}\|)^2 \le 3\mu^2\|\overline{X}^{(j)}\overline{X}^{(j)\mathrm{H}} - \overline{U}_t^{(j)}\overline{U}_t^{(j)\mathrm{H}}\|^2 + 3\mu^2\frac{\sigma_{\min}^4(\overline{\boldsymbol{\mathcal{X}}})}{200^2}\|\boldsymbol{\mathcal{V}}_{\boldsymbol{\mathcal{X}}^\perp}^\top * \boldsymbol{\mathcal{V}}_{\boldsymbol{\mathcal{U}}_t*\boldsymbol{\mathcal{W}}_t}\|^2$$
$$+ 3 \cdot 49^2\mu^2\|(\mathcal{A}^*\mathcal{A} - \mathcal{I})(\boldsymbol{\mathcal{X}} * \boldsymbol{\mathcal{X}}^\top - \boldsymbol{\mathcal{U}}_t * \boldsymbol{\mathcal{U}}_t^\top)\|^2.$$

Now, we can come back to bounding (E.28) proceeding as follows

$$\|V_{\boldsymbol{\mathcal{X}}^\perp}^{(j)\mathrm{H}}V_{\overline{U}_{t+1}^{(j)}\overline{W}_{t+1}^{(j)}}\| \le \|\boldsymbol{\mathcal{V}}_{\boldsymbol{\mathcal{X}}^\perp}^\top * \boldsymbol{\mathcal{V}}_{\boldsymbol{\mathcal{U}}_t*\boldsymbol{\mathcal{W}}_t}\| \left(1 - \frac{\mu}{3}\sigma_{\min}^2(\overline{\boldsymbol{\mathcal{X}}}) + \frac{4\mu}{300}\sigma_{\min}^2(\overline{\boldsymbol{\mathcal{X}}})\right)$$
$$+ 129\mu\|(\mathcal{A}^*\mathcal{A} - \mathcal{I})(\boldsymbol{\mathcal{X}} * \boldsymbol{\mathcal{X}}^\top - \boldsymbol{\mathcal{U}}_t * \boldsymbol{\mathcal{U}}_t^\top)\|$$
$$+ 171\mu^2\|\overline{X}^{(j)}\overline{X}^{(j)\mathrm{H}} - \overline{U}_t^{(j)}\overline{U}_t^{(j)\mathrm{H}}\|^2 + \mu^2\frac{171\sigma_{\min}^4(\overline{\boldsymbol{\mathcal{X}}})}{200^2}\|\boldsymbol{\mathcal{V}}_{\boldsymbol{\mathcal{X}}^\perp}^\top * \boldsymbol{\mathcal{V}}_{\boldsymbol{\mathcal{U}}_t*\boldsymbol{\mathcal{W}}_t}\|^2$$
$$+ 171 \cdot 49^2\mu^2\|(\mathcal{A}^*\mathcal{A} - \mathcal{I})(\boldsymbol{\mathcal{X}} * \boldsymbol{\mathcal{X}}^\top - \boldsymbol{\mathcal{U}}_t * \boldsymbol{\mathcal{U}}_t^\top)\|^2$$
$$\le \|\boldsymbol{\mathcal{V}}_{\boldsymbol{\mathcal{X}}^\perp}^\top * \boldsymbol{\mathcal{V}}_{\boldsymbol{\mathcal{U}}_t*\boldsymbol{\mathcal{W}}_t}\| \left(1 - \frac{\mu}{3}\sigma_{\min}^2(\overline{\boldsymbol{\mathcal{X}}}) + \frac{4\mu}{300}\sigma_{\min}^2(\overline{\boldsymbol{\mathcal{X}}}) + \frac{171}{200^2}\kappa^{-4}\widetilde{c} \cdot c\mu\sigma_{\min}^2(\overline{\boldsymbol{\mathcal{X}}})\right)$$
$$+ 171\mu^2\|\boldsymbol{\mathcal{X}} * \boldsymbol{\mathcal{X}}^\top - \boldsymbol{\mathcal{U}}_t * \boldsymbol{\mathcal{U}}_t^\top\|^2$$
$$+ \mu(129 + 171 \cdot 49^2c^2\kappa^{-4})\|(\mathcal{A}^*\mathcal{A} - \mathcal{I})(\boldsymbol{\mathcal{X}} * \boldsymbol{\mathcal{X}}^\top - \boldsymbol{\mathcal{U}}_t * \boldsymbol{\mathcal{U}}_t^\top)\|,$$

where for the last inequality we used assumptions (E.23), (E.20) and (E.21), and the properties of the tubal tensor norm. Now choosing constant $c > 0$ sufficiently small, we obtain that

$$\|V_{\boldsymbol{\mathcal{X}}^\perp}^{(j)\mathrm{H}}V_{\overline{U}_{t+1}^{(j)}\overline{W}_{t+1}^{(j)}}\| \le \left(1 - \frac{\mu}{4}\sigma_{\min}^2(\overline{\boldsymbol{\mathcal{X}}})\right)\|\boldsymbol{\mathcal{V}}_{\boldsymbol{\mathcal{X}}^\perp}^\top * \boldsymbol{\mathcal{V}}_{\boldsymbol{\mathcal{U}}_t*\boldsymbol{\mathcal{W}}_t}\| + 200\mu^2\|\boldsymbol{\mathcal{X}} * \boldsymbol{\mathcal{X}}^\top - \boldsymbol{\mathcal{U}}_t * \boldsymbol{\mathcal{U}}_t^\top\|^2$$
$$+ 150\|(\mathcal{A}^*\mathcal{A} - \mathcal{I})(\boldsymbol{\mathcal{X}} * \boldsymbol{\mathcal{X}}^\top - \boldsymbol{\mathcal{U}}_t * \boldsymbol{\mathcal{U}}_t^\top)\|.$$

Since the right-hand side of the above inequality is independent of $j$, we obtain the lemma statement. $\square$

The following lemma shows that under a mild condition the technical assumption

$$\|\boldsymbol{\mathcal{U}}_{t+1}\| \le 3\|\boldsymbol{\mathcal{X}}\|$$

needed in the lemmas above holds.

**Lemma E.6.** *Assume that $\|\boldsymbol{\mathcal{U}}_t\| \le 3\|\boldsymbol{\mathcal{X}}\|$, $\mu \le \frac{1}{27}\|\boldsymbol{\mathcal{X}}\|^{-2}$ and that linear measurement operator $\mathcal{A}$ is such that*

$$\|(\mathcal{A}^*\mathcal{A} - \mathcal{I})(\boldsymbol{\mathcal{X}} * \boldsymbol{\mathcal{X}}^\top - \boldsymbol{\mathcal{U}}_t * \boldsymbol{\mathcal{U}}_t^\top)\| \le \|\boldsymbol{\mathcal{X}}\|^2$$

*Then for the iteration $t + 1$, it also holds $\|\boldsymbol{\mathcal{U}}_{t+1}\| \le 3\|\boldsymbol{\mathcal{X}}\|$.*

*Proof.* Consider the gradient iterate

$$\boldsymbol{\mathcal{U}}_{t+1} = \boldsymbol{\mathcal{U}}_t + \mu\mathcal{A}^*\mathcal{A}(\boldsymbol{\mathcal{X}} * \boldsymbol{\mathcal{X}}^\top - \boldsymbol{\mathcal{U}}_t * \boldsymbol{\mathcal{U}}_t^\top) * \boldsymbol{\mathcal{U}}_t$$
$$= \boldsymbol{\mathcal{U}}_t + \mu(\boldsymbol{\mathcal{X}} * \boldsymbol{\mathcal{X}}^\top - \boldsymbol{\mathcal{U}}_t * \boldsymbol{\mathcal{U}}_t^\top) * \boldsymbol{\mathcal{U}}_t + \mu(\mathcal{A}^*\mathcal{A} - \mathcal{I})(\boldsymbol{\mathcal{X}} * \boldsymbol{\mathcal{X}}^\top - \boldsymbol{\mathcal{U}}_t * \boldsymbol{\mathcal{U}}_t^\top) * \boldsymbol{\mathcal{U}}_t$$
$$= (\boldsymbol{\mathcal{I}} - \mu\boldsymbol{\mathcal{U}}_t * \boldsymbol{\mathcal{U}}_t^\top) * \boldsymbol{\mathcal{U}}_t + \mu\boldsymbol{\mathcal{X}} * \boldsymbol{\mathcal{X}}^\top * \boldsymbol{\mathcal{U}}_t + \mu(\mathcal{A}^*\mathcal{A} - \mathcal{I})(\boldsymbol{\mathcal{X}} * \boldsymbol{\mathcal{X}}^\top - \boldsymbol{\mathcal{U}}_t * \boldsymbol{\mathcal{U}}_t^\top) * \boldsymbol{\mathcal{U}}_t.$$

To estimate the norm of $\mathcal{U}_{t+1}$, we will bound each summand above separately. Due to the assumption on $\mu$ and the norm of $\mathcal{U}_t$, we have $\mu \leq \frac{1}{27}\|\mathcal{X}\|^{-2} \leq \frac{1}{3}\|\mathcal{U}_t\|^{-2}$. This allows us to estimate the tensor norm of $(\mathcal{I} - \mu\mathcal{U}_t * \mathcal{U}_t^\top) * \mathcal{U}_t$ via the norm of matrix block representation in the Fourier domain. Namely, assume that matrix $\overline{\mathcal{U}_t}$ has the SVD $\overline{\mathcal{U}_t} = V\Sigma W^H$. Then for matrix $\overline{(\mathcal{I} - \mu\mathcal{U}_t * \mathcal{U}_t^\top) * \mathcal{U}_t}$, we have

$$\overline{(\mathcal{I} - \mu\mathcal{U}_t * \mathcal{U}_t^\top) * \mathcal{U}_t} = V\Sigma W^H - \mu V\Sigma W^H W\Sigma V^H V\Sigma W^H = V\Sigma W^H - \mu V\Sigma^3 W^H = V(\Sigma - \mu\Sigma^3)W^H.$$

From here, since $\mu \leq \frac{1}{27}\|\mathcal{X}\|^{-2} \leq \frac{1}{3}\|\mathcal{U}\|^{-2}$ and $\|\mathcal{U}_t\| = \|\overline{\mathcal{U}_t}\|$, it holds that $\|\overline{(\mathcal{I} - \mu\mathcal{U}_t * \mathcal{U}_t^\top) * \mathcal{U}_t}\| = \|\overline{\mathcal{U}_t}\| - \mu\|\overline{\mathcal{U}_t}\|^3 = \|\mathcal{U}_t\|(1 - \mu\|\mathcal{U}_t\|^2)$. Besides, from the submultiplicativity of the tensor norm and the triangle inequality, we obtain that

$$\|\mathcal{U}_{t+1}\| \leq (1 - \mu\|\mathcal{U}_t\|^2 + \mu\|\mathcal{X}\|^2 + \mu\|(\mathcal{A}^*\mathcal{A} - \mathcal{I})(\mathcal{X} * \mathcal{X}^\top - \mathcal{U}_t * \mathcal{U}_t^\top)\|)\|\mathcal{U}_t\| \tag{E.30}$$

$$\leq (1 - \mu\|\mathcal{U}_t\|^2 + 2\mu\|\mathcal{X}\|^2)\|\mathcal{U}_t\|, \tag{E.31}$$

where in the last line we used the assumption on $\|(\mathcal{A}^*\mathcal{A} - \mathcal{I})(\mathcal{X} * \mathcal{X}^\top - \mathcal{U}_t * \mathcal{U}_t^\top)\|$. By combining inequality (E.31) with the assumption $\mu \leq \frac{1}{27\|\mathcal{X}\|^2} \leq \frac{1}{3\|\mathcal{U}\|^2}$, we obtain that $\|\mathcal{U}_{t+1}\| \leq 3\|\mathcal{X}\|$, which finishes the proof. $\qquad\square$

The following lemma shows that $\mathcal{U}_t * \mathcal{W}_t * \mathcal{W}_t^\top * \mathcal{U}_t^\top$ converges towards $\mathcal{X} * \mathcal{X}^T$, when projected onto the tensor column space of $\mathcal{X}$.

**Lemma E.7.** *Assume that the following conditions hold*

$$\|\mathcal{U}_t\| \leq 3\|\mathcal{X}\| \tag{E.32}$$

$$\mu \leq c \cdot \frac{1}{\sqrt{nk}} \cdot \kappa^{-2}\|\mathcal{X}\|^{-2} \tag{E.33}$$

$$\sigma_{min}(\overline{\mathcal{U}_t * \mathcal{W}_t}) \geq \frac{1}{\sqrt{10}}\sigma_{min}(\overline{\mathcal{X}}) \tag{E.34}$$

$$\|\mathcal{V}_{\mathcal{X}^\perp}^\top * \mathcal{V}_{\mathcal{U}_t * \mathcal{W}_t}\| \leq c\kappa^{-2} \tag{E.35}$$

*and*

$$\max\left\{\|\mathcal{V}_{\mathcal{X}}^\top * (\mathcal{A}^*\mathcal{A} - \mathcal{I})(\mathcal{Y}_t)\|_F, \ \|\mathcal{V}_{\mathcal{U}_t * \mathcal{W}_t}^\top * (\mathcal{A}^*\mathcal{A} - \mathcal{I})(\mathcal{Y}_t)\|_F, \ \|(\mathcal{A}^*\mathcal{A} - \mathcal{I})(\mathcal{Y}_t)\|\right\} \leq \kappa^{-2}\|\mathcal{Y}_t\|_F$$

*with $\mathcal{Y}_t := \mathcal{X} * \mathcal{X}^\top - \mathcal{U}_t * \mathcal{U}_t^\top$. Then it holds that*

$$\|\mathcal{V}_{\mathcal{X}^\perp}^\top * \mathcal{U}_t * \mathcal{U}_t^\top\|_F \leq 3\|\mathcal{V}_{\mathcal{X}^\perp}^\top * (\mathcal{X} * \mathcal{X}^\top - \mathcal{U}_t * \mathcal{U}_t^\top)\|_F + \|\mathcal{U}_t * \mathcal{W}_{t,\perp} * \mathcal{W}_{t,\perp}^\top * \mathcal{U}_t^\top\|_F \tag{E.36}$$

*as well as*

$$\|\mathcal{X} * \mathcal{X}^\top - \mathcal{U}_t * \mathcal{U}_t^\top\|_F \leq 4\|\mathcal{V}_{\mathcal{X}^\perp}^\top * (\mathcal{X} * \mathcal{X}^\top - \mathcal{U}_t * \mathcal{U}_t^\top)\|_F + \|\mathcal{U}_t * \mathcal{W}_{t,\perp} * \mathcal{W}_{t,\perp}^\top * \mathcal{U}_t^\top\|_F \tag{E.37}$$

*and*

$$\|\mathcal{V}_{\mathcal{X}^\perp}^\top(\mathcal{X} * \mathcal{X}^\top - \mathcal{U}_{t+1} * \mathcal{U}_{t+1}^\top)\|_F \leq \left(1 - \frac{\mu}{200}\sigma_{min}^2(\overline{\mathcal{X}})\right)\|\mathcal{V}_{\mathcal{X}^\perp}^\top * (\mathcal{X} * \mathcal{X}^\top - \mathcal{U}_t * \mathcal{U}_t^\top)\|_F$$

$$+ \mu\frac{\sigma_{min}^2(\overline{\mathcal{X}})}{100}\|\mathcal{U}_t * \mathcal{W}_{t,\perp} * \mathcal{W}_{t,\perp}^\top * \mathcal{U}_t^\top\|_F \tag{E.38}$$

*Proof.* We start by proving the first inequality (E.38). For this, let us decompose $\mathcal{V}_{\mathcal{X}^\perp}^\top * \mathcal{U}_t * \mathcal{U}_t^\top$ as follows

$$\mathcal{V}_{\mathcal{X}^\perp}^\top * \mathcal{U}_t * \mathcal{U}_t^\top = \mathcal{V}_{\mathcal{X}^\perp}^\top * \mathcal{U}_t * \mathcal{U}_t^\top * \mathcal{V}_{\mathcal{X}} * \mathcal{V}_{\mathcal{X}}^\top + \mathcal{V}_{\mathcal{X}^\perp}^\top * \mathcal{U}_t * \mathcal{U}_t^\top * \mathcal{V}_{\mathcal{X}^\perp} * \mathcal{V}_{\mathcal{X}^\perp}^\top,$$

then using the triangle inequality and submultiplicativity of the Frobenius and the spectral norm, we obtain

$$\|\mathcal{V}_{\mathcal{X}^\perp}^\top * \mathcal{U}_t * \mathcal{U}_t^\top\|_F \leq \|\mathcal{V}_{\mathcal{X}^\perp}^\top * \mathcal{U}_t * \mathcal{U}_t^\top * \mathcal{V}_{\mathcal{X}}\|_F + \|\mathcal{V}_{\mathcal{X}^\perp}^\top * \mathcal{U}_t * \mathcal{U}_t^\top * \mathcal{V}_{\mathcal{X}^\perp}\|_F$$

$$\leq \|\mathcal{V}_{\mathcal{X}^\perp}^\top * (\mathcal{X} * \mathcal{X}^\top - \mathcal{U}_t * \mathcal{U}_t^\top) * \mathcal{V}_{\mathcal{X}}\|_F + \|\mathcal{V}_{\mathcal{X}^\perp}^\top * \mathcal{U}_t * \mathcal{U}_t^\top * \mathcal{V}_{\mathcal{X}^\perp}\|_F$$

$$\leq \|\mathcal{V}_{\mathcal{X}}^\top * (\mathcal{X} * \mathcal{X}^\top - \mathcal{U}_t * \mathcal{U}_t^\top)\|_F + \|\mathcal{V}_{\mathcal{X}^\perp}^\top * \mathcal{U}_t * \mathcal{U}_t^\top * \mathcal{V}_{\mathcal{X}^\perp}\|_F, \tag{E.39}$$

where in the second line, we used the orthogonality of the decomposition. Now, we will work additionally on bounding the norm of $\mathcal{V}_{\mathcal{X}^\perp}^\top * \mathcal{U}_t * \mathcal{U}_t^\top * \mathcal{V}_{\mathcal{X}^\perp}$ to obtain (E.38). Here, we will use the orthogonal decomposition with respect to $\mathcal{W}_t$ and $\mathcal{W}_{t,\perp}$, which leads to

$$\|\mathcal{V}_{\mathcal{X}^\perp}^\top * \mathcal{U}_t * \mathcal{U}_t^\top * \mathcal{V}_{\mathcal{X}^\perp}\|_F \leq \|\mathcal{V}_{\mathcal{X}^\perp}^\top * \mathcal{U}_t * \mathcal{W}_t * \mathcal{W}_t^\top * \mathcal{U}_t^\top * \mathcal{V}_{\mathcal{X}^\perp}\|_F + \|\mathcal{V}_{\mathcal{X}^\perp}^\top * \mathcal{U}_t * \mathcal{W}_{t,\perp} * \mathcal{W}_{t,\perp}^\top * \mathcal{U}_t^\top * \mathcal{V}_{\mathcal{X}^\perp}\|_F$$
$$\leq \|\mathcal{V}_{\mathcal{X}^\perp}^\top * \mathcal{U}_t * \mathcal{W}_t * \mathcal{W}_t^\top * \mathcal{U}_t^\top * \mathcal{V}_{\mathcal{X}^\perp}\|_F + \|\mathcal{U}_t * \mathcal{W}_{t,\perp} * \mathcal{W}_{t,\perp}^\top * \mathcal{U}_t^\top\|_F$$

Now, for the first term above, we get

$$\|\mathcal{V}_{\mathcal{X}^\perp}^\top * \mathcal{U}_t * \mathcal{W}_t * \mathcal{W}_t^\top * \mathcal{U}_t^\top * \mathcal{V}_{\mathcal{X}^\perp}\|_F = \|\mathcal{V}_{\mathcal{X}^\perp}^\top * \mathcal{V}_{\mathcal{U}_t * \mathcal{W}_t} * \mathcal{V}_{\mathcal{U}_t * \mathcal{W}_t}^\top * \mathcal{U}_t * \mathcal{W}_t * \mathcal{W}_t^\top * \mathcal{U}_t^\top * \mathcal{V}_{\mathcal{X}^\perp}\|_F$$

$$= \sum_{j=1}^k \|\overline{\mathcal{V}_{\mathcal{X}^\perp}^\top * \mathcal{V}_{\mathcal{U}_t * \mathcal{W}_t} * \mathcal{V}_{\mathcal{U}_t * \mathcal{W}_t}^\top * \mathcal{U}_t * \mathcal{W}_t * \mathcal{W}_t^\top * \mathcal{U}_t^\top * \mathcal{V}_{\mathcal{X}^\perp}}^{(j)}\|_F$$

$$= \sum_{j=1}^k \|\overline{V}_{\mathcal{X}^\perp}^{(j)\mathrm{H}} \overline{V}_{\mathcal{U}_t * \mathcal{W}_t}^{(j)} \overline{V}_{\mathcal{U}_t * \mathcal{W}_t}^{(j)\mathrm{H}} \overline{U}_t^{(j)} \overline{W}_t^{(j)} \overline{W}_t^{(j)\mathrm{H}} \overline{U}_t^{(j)\mathrm{H}} \overline{V}_{\mathcal{X}^\perp}^{(j)}\|_F$$

$$= \sum_{j=1}^k \|\overline{V}_{\mathcal{X}^\perp}^{(j)\mathrm{H}} \overline{V}_{\mathcal{U}_t * \mathcal{W}_t}^{(j)} \left(\overline{V}_{\mathcal{X}^\perp}^{(j)\mathrm{H}} \overline{V}_{\mathcal{U}_t * \mathcal{W}_t}^{(j)}\right)^{-1} \overline{V}_{\mathcal{X}^\perp}^{(j)\mathrm{H}} \overline{V}_{\mathcal{U}_t * \mathcal{W}_t}^{(j)} \overline{V}_{\mathcal{U}_t * \mathcal{W}_t}^{(j)\mathrm{H}} \overline{U}_t^{(j)} \overline{W}_t^{(j)} \overline{W}_t^{(j)\mathrm{H}} \overline{U}_t^{(j)\mathrm{H}} \overline{V}_{\mathcal{X}^\perp}^{(j)}\|_F$$

$$\leq \max_{1 \leq j \leq k} \|\overline{V}_{\mathcal{X}^\perp}^{(j)\mathrm{H}} \overline{V}_{\mathcal{U}_t * \mathcal{W}_t}^{(j)}\| \max_{1 \leq j \leq k} \left\|\left(\overline{V}_{\mathcal{X}^\perp}^{(j)\mathrm{H}} \overline{V}_{\mathcal{U}_t * \mathcal{W}_t}^{(j)}\right)^{-1}\right\| \sum_{j=1}^k \|\overline{V}_{\mathcal{X}^\perp}^{(j)\mathrm{H}} \overline{V}_{\mathcal{U}_t * \mathcal{W}_t}^{(j)} \overline{V}_{\mathcal{U}_t * \mathcal{W}_t}^{(j)\mathrm{H}} \overline{U}_t^{(j)} \overline{W}_t^{(j)} \overline{W}_t^{(j)\mathrm{H}} \overline{U}_t^{(j)\mathrm{H}} \overline{V}_{\mathcal{X}^\perp}^{(j)}\|_F$$

$$= \frac{\|\mathcal{V}_{\mathcal{X}^\perp}^\top * \mathcal{V}_{\mathcal{U}_t * \mathcal{W}_t}\|}{\sigma_{\min}(\mathcal{V}_{\mathcal{X}^\perp}^\top * \mathcal{V}_{\mathcal{U}_t * \mathcal{W}_t})} \sum_{j=1}^k \|\overline{V}_{\mathcal{X}^\perp}^{(j)\mathrm{H}} \overline{V}_{\mathcal{U}_t * \mathcal{W}_t}^{(j)} \overline{V}_{\mathcal{U}_t * \mathcal{W}_t}^{(j)\mathrm{H}} \overline{U}_t^{(j)} \overline{W}_t^{(j)} \overline{W}_t^{(j)\mathrm{H}} \overline{U}_t^{(j)\mathrm{H}} \overline{V}_{\mathcal{X}^\perp}^{(j)}\|_F$$

$$= \frac{\|\mathcal{V}_{\mathcal{X}^\perp}^\top * \mathcal{V}_{\mathcal{U}_t * \mathcal{W}_t}\|}{\sigma_{\min}(\mathcal{V}_{\mathcal{X}^\perp}^\top * \mathcal{V}_{\mathcal{U}_t * \mathcal{W}_t})} \sum_{j=1}^k \|\overline{V}_{\mathcal{X}^\perp}^{(j)\mathrm{H}} \overline{U}_t^{(j)} \overline{W}_t^{(j)} \overline{W}_t^{(j)\mathrm{H}} \overline{U}_t^{(j)\mathrm{H}} \overline{V}_{\mathcal{X}^\perp}^{(j)}\|_F$$

$$= \frac{\|\mathcal{V}_{\mathcal{X}^\perp}^\top * \mathcal{V}_{\mathcal{U}_t * \mathcal{W}_t}\|}{\sigma_{\min}(\mathcal{V}_{\mathcal{X}^\perp}^\top * \mathcal{V}_{\mathcal{U}_t * \mathcal{W}_t})} \|\mathcal{V}_{\mathcal{X}^\perp} * \mathcal{U}_t * \mathcal{W}_t * \mathcal{W}_t^\top * \mathcal{U}_t^\top * \mathcal{V}_{\mathcal{X}^\perp}\|_F$$

$$= \frac{\|\mathcal{V}_{\mathcal{X}^\perp}^\top * \mathcal{V}_{\mathcal{U}_t * \mathcal{W}_t}\|}{\sigma_{\min}(\mathcal{V}_{\mathcal{X}^\perp}^\top * \mathcal{V}_{\mathcal{U}_t * \mathcal{W}_t})} \|\mathcal{V}_{\mathcal{X}^\perp} * \mathcal{U}_t * \mathcal{U}_t^\top * \mathcal{V}_{\mathcal{X}^\perp}\|_F$$

$$= \frac{\|\mathcal{V}_{\mathcal{X}^\perp}^\top * \mathcal{V}_{\mathcal{U}_t * \mathcal{W}_t}\|}{\sigma_{\min}(\mathcal{V}_{\mathcal{X}^\perp}^\top * \mathcal{V}_{\mathcal{U}_t * \mathcal{W}_t})} \|\mathcal{V}_{\mathcal{X}^\perp} * (\mathcal{X} * \mathcal{X}^\top - \mathcal{U}_t * \mathcal{U}_t^\top) * \mathcal{V}_{\mathcal{X}^\perp}\|_F$$

$$\leq \frac{\|\mathcal{V}_{\mathcal{X}^\perp}^\top * \mathcal{V}_{\mathcal{U}_t * \mathcal{W}_t}\|}{\sigma_{\min}(\mathcal{V}_{\mathcal{X}^\perp}^\top * \mathcal{V}_{\mathcal{U}_t * \mathcal{W}_t})} \|\mathcal{V}_{\mathcal{X}^\perp} * (\mathcal{X} * \mathcal{X}^\top - \mathcal{U}_t * \mathcal{U}_t^\top)\|_F \leq 2\|\mathcal{V}_{\mathcal{X}^\perp} * (\mathcal{X} * \mathcal{X}^\top - \mathcal{U}_t * \mathcal{U}_t^\top)\|_F$$

where in the last line we used the assumption (E.35). Them, using just established bound together with (E.39), we get

$$\|\mathcal{V}_{\mathcal{X}^\perp}^\top * \mathcal{U}_t * \mathcal{U}_t^\top\|_F \leq 3\|\mathcal{V}_{\mathcal{X}^\perp}^\top * (\mathcal{X} * \mathcal{X}^\top - \mathcal{U}_t * \mathcal{U}_t^\top)\|_F + \|\mathcal{U}_t * \mathcal{W}_{t,\perp} * \mathcal{W}_{t,\perp}^\top * \mathcal{U}_t^\top\|_F.$$

To get inequality (E.37), we use the orthogonal decomposition of $\mathcal{X} * \mathcal{X}^\top - \mathcal{U}_t * \mathcal{U}_t^\top$ with respect to $\mathcal{V}_{\mathcal{X}}$ and $\mathcal{V}_{\mathcal{X}^\perp}$, which leads to

$$\|\mathcal{X} * \mathcal{X}^\top - \mathcal{U}_t * \mathcal{U}_t^\top\|_F = \|\mathcal{V}_{\mathcal{X}}^\top * (\mathcal{X} * \mathcal{X}^\top - \mathcal{U}_t * \mathcal{U}_t^\top)\|_F + \|\mathcal{V}_{\mathcal{X}^\perp}^\top * (\mathcal{X} * \mathcal{X}^\top - \mathcal{U}_t * \mathcal{U}_t^\top)\|_F$$
$$= \|\mathcal{V}_{\mathcal{X}}^\top * (\mathcal{X} * \mathcal{X}^\top - \mathcal{U}_t * \mathcal{U}_t^\top)\|_F + \|\mathcal{V}_{\mathcal{X}^\perp}^\top * \mathcal{U}_t * \mathcal{U}_t^\top\|_F$$
$$\leq 4\|\mathcal{V}_{\mathcal{X}}^\top * (\mathcal{X} * \mathcal{X}^\top - \mathcal{U}_t * \mathcal{U}_t^\top)\|_F + \|\mathcal{U}_t * \mathcal{W}_{t,\perp} * \mathcal{W}_{t,\perp}^\top * \mathcal{U}_t^\top\|_F.$$

Inequality (E.38) follows from the two inequalities proved here and Lemma 9.5 in (Stöger & Soltanolkotabi, 2021). The building stones for this are the properties of the tubal tensor Frobenius norm. Namely, the Frobenius norm of any tubal

tensor $\mathcal{T}$ can be represented as the sum of Frobenius norms of each slice in the domain, that is

$$\|\mathcal{T}\|_F = \sum_{j=1}^{k} \|\overline{T}^{(j)}\|_F$$

and $\|\mathcal{T}\|_F \leq \sqrt{n \cdot k}\|\mathcal{T}\|$. Besides, the Frobenius norm of the product of two tensors $\mathcal{T}$ and $\mathcal{P}$ can be bounded as below

$$\|\mathcal{T} * \mathcal{P}\|_F = \sum_{j=1}^{k} \|\overline{T}^{(j)}\overline{P}^{(j)}\|_F \leq \max_{1 \leq j \leq k} \|\overline{T}^{(j)}\| \sum_{j=1}^{k} \|\overline{P}^{(j)}\|_F \leq \|\mathcal{T}\|\|\mathcal{P}\|_F.$$

$\square$

Now, we have collected all the necessary ingredients to prove the main result of this section, which shows that after a sufficient number of interactions, the relative error between $\mathcal{U}_t * \mathcal{U}_t^\top$ and $\mathcal{X} * \mathcal{X}^\top$ becomes small.

**Theorem E.1.** *Suppose that the stepsize satisfies $\mu \leq c_1 \sqrt{k}\kappa^{-4}\|\mathcal{X}\|^{-2}$ for some small $c_1 > 0$, and $\mathcal{A} : S^{n \times n \times k} \to \mathbb{R}^m$ satisfies $RIP(2r + 1, \delta)$ for some constant $0 < \delta \leq \dfrac{c_1}{\kappa^4\sqrt{r}}$. Set $\gamma \in (0, \frac{1}{2})$, and choose a number of iterations $t_*$ such that $\sigma_{min}(\mathcal{U}_{t_*} * \mathcal{W}_{t_*}) \geq \gamma$. Also, assume that $\|\mathcal{U}_{t_*} * \mathcal{W}_{t_*,\perp}\| \leq 2\gamma$, $\|\mathcal{U}_{t_*}\| \leq 3\|\mathcal{X}\|$, $\gamma \leq \dfrac{c_2\sigma_{min}(\mathcal{X})}{\kappa^2 \min\{n, R\}}$, and $\|\mathcal{V}_{\mathcal{X}^\perp}^\top * \mathcal{V}_{\mathcal{U}_{t_*}*\mathcal{W}_{t_*}}\| \leq c_2\kappa^{-2}$ for some small $c_2 > 0$. Then, after*

$$\widehat{t} - t_* \lesssim \frac{1}{\mu\sigma_{min}(\mathcal{X})^2} \ln\left(\min\left\{1, \frac{\kappa r}{k(\min\{n, R\} - r)}\right\} \frac{\|\mathcal{X}\|}{\gamma}\right)$$

*additional iterations, we have*

$$\frac{\|\mathcal{U}_{\widehat{t}} * \mathcal{U}_{\widehat{t}}^\top - \mathcal{X} * \mathcal{X}^\top\|_F}{\|\mathcal{X}\|^2} \lesssim k^{5/4}r^{1/8}\kappa^{-3/16}(\min\{n, R\} - r)^{3/8}\gamma^{21/16}\|\mathcal{X}\|^{-21/16}.$$

*Proof.* First, we set

$$t_1 = \min\left\{t \geq t_* : \sigma_{min}(\mathcal{V}_{\mathcal{X}}^\top * \mathcal{U}_t) \geq \tfrac{1}{\sqrt{10}}\sigma_{min}(\overline{\mathcal{X}})\right\},$$

and then aim to prove that over the iterations $t_* \leq t \leq t_1$, the following hold:

- $\sigma_{min}(\mathcal{V}_{\mathcal{X}}^\top * \mathcal{U}_t) \geq \frac{1}{2}\gamma\left(1 + \frac{1}{8}\mu\sigma_{min}(\mathcal{X})^2\right)^{t-t_*}$

- $\|\mathcal{U}_t * \mathcal{W}_{t,\perp}\| \leq 2\gamma\left(1 + 80\mu c_2\sqrt{k}\sigma_{min}(\mathcal{X})^2\right)^{t-t_*}$

- $\|\mathcal{U}_t\| \leq 3\|\mathcal{X}\|$

- $\|\mathcal{V}_{\mathcal{X}^\perp}^\top * \mathcal{V}_{\mathcal{U}_t*\mathcal{W}_t}\| \leq c_2\kappa^{-2}$.

Intuitively, this means that over the range $t_* \leq t \leq t_1$, the smallest singular value of the signal term $\mathcal{V}_{\mathcal{X}}^\top * \mathcal{U}_t$ grows at a faster rate than the largest singular value of the noise term $\mathcal{U}_t * \mathcal{W}_{t,\perp}$.

For $t = t_*$, these inequalities hold due to the assumptions of this theorem. Now, suppose they hold for some $t$ between $t_*$ and $t_1$. We'll show they also hold for $t + 1$.

First, note that we have:

$$
\begin{aligned}
&\|(\mathcal{A}^*\mathcal{A} - \mathcal{I})(\boldsymbol{\mathcal{X}} * \boldsymbol{\mathcal{X}}^\top - \boldsymbol{\mathcal{U}}_t * \boldsymbol{\mathcal{U}}_t^\top)\| \\
&= \|(\mathcal{A}^*\mathcal{A} - \mathcal{I})(\boldsymbol{\mathcal{X}} * \boldsymbol{\mathcal{X}}^\top - \boldsymbol{\mathcal{U}}_t * \boldsymbol{\mathcal{W}}_t * \boldsymbol{\mathcal{W}}_t^\top * \boldsymbol{\mathcal{U}}_t^\top - \boldsymbol{\mathcal{U}}_t * \boldsymbol{\mathcal{W}}_{t,\perp} * \boldsymbol{\mathcal{W}}_{t,\perp}^\top * \boldsymbol{\mathcal{U}}_t^\top)\| \\
&\leq \|(\mathcal{A}^*\mathcal{A} - \mathcal{I})(\boldsymbol{\mathcal{X}} * \boldsymbol{\mathcal{X}}^\top - \boldsymbol{\mathcal{U}}_t * \boldsymbol{\mathcal{W}}_t * \boldsymbol{\mathcal{W}}_t^\top * \boldsymbol{\mathcal{U}}_t^\top)\| + \|(\mathcal{A}^*\mathcal{A} - \mathcal{I})(\boldsymbol{\mathcal{U}}_t * \boldsymbol{\mathcal{W}}_{t,\perp} * \boldsymbol{\mathcal{W}}_{t,\perp} * \boldsymbol{\mathcal{U}}_t^\top)\| \\
(a) \quad &\leq \delta\sqrt{kr}\|\boldsymbol{\mathcal{X}} * \boldsymbol{\mathcal{X}}^\top - \boldsymbol{\mathcal{U}}_t * \boldsymbol{\mathcal{W}}_t * \boldsymbol{\mathcal{W}}_t^\top * \boldsymbol{\mathcal{U}}_t^\top\| + \delta\sqrt{k}\|\boldsymbol{\mathcal{U}}_t * \boldsymbol{\mathcal{W}}_{t,\perp} * \boldsymbol{\mathcal{W}}_{t,\perp} * \boldsymbol{\mathcal{U}}_t^\top\|_* \\
&\leq \delta\sqrt{kr}\left(\|\boldsymbol{\mathcal{X}} * \boldsymbol{\mathcal{X}}^\top\| + \|\boldsymbol{\mathcal{U}}_t * \boldsymbol{\mathcal{W}}_t * \boldsymbol{\mathcal{W}}_t^\top * \boldsymbol{\mathcal{U}}_t^\top\|\right) + \delta\sqrt{k}\|\boldsymbol{\mathcal{U}}_t * \boldsymbol{\mathcal{W}}_{t,\perp} * \boldsymbol{\mathcal{W}}_{t,\perp} * \boldsymbol{\mathcal{U}}_t^\top\|_* \\
&= \delta\sqrt{kr}\left(\|\boldsymbol{\mathcal{X}}\|^2 + \|\boldsymbol{\mathcal{U}}_t * \boldsymbol{\mathcal{W}}_t\|^2\right) + \delta\sqrt{k}\|\boldsymbol{\mathcal{U}}_t * \boldsymbol{\mathcal{W}}_{t,\perp} * \boldsymbol{\mathcal{W}}_{t,\perp} * \boldsymbol{\mathcal{U}}_t^\top\|_* \\
&\leq \delta\sqrt{kr}\left(\|\boldsymbol{\mathcal{X}}\|^2 + \|\boldsymbol{\mathcal{U}}_t\|^2\right) + \delta\sqrt{k}\|\boldsymbol{\mathcal{U}}_t * \boldsymbol{\mathcal{W}}_{t,\perp} * \boldsymbol{\mathcal{W}}_{t,\perp} * \boldsymbol{\mathcal{U}}_t^\top\|_* \\
(b) \quad &\leq \delta\sqrt{kr}\left(\|\boldsymbol{\mathcal{X}}\|^2 + 9\|\boldsymbol{\mathcal{X}}\|^2\right) + \delta\sqrt{k}(\min\{n,R\} - r)\|\boldsymbol{\mathcal{U}}_t * \boldsymbol{\mathcal{W}}_{t,\perp} * \boldsymbol{\mathcal{W}}_{t,\perp} * \boldsymbol{\mathcal{U}}_t^\top\| \\
&\leq 10\delta\sqrt{kr}\|\boldsymbol{\mathcal{X}}\|^2 + \delta\sqrt{k}(\min\{n,R\} - r)\|\boldsymbol{\mathcal{U}}_t * \boldsymbol{\mathcal{W}}_{t,\perp}\|^2 \\
&\leq 10\delta\sqrt{kr}\kappa^2\sigma_{\min}(\boldsymbol{\mathcal{X}})^2 + \delta\sqrt{k}(\min\{n,R\} - r)\|\boldsymbol{\mathcal{U}}_t * \boldsymbol{\mathcal{W}}_{t,\perp}\|^2 \\
(c) \quad &\leq 10c_1\sqrt{k}\kappa^{-2}\sigma_{\min}(\boldsymbol{\mathcal{X}})^2 + 4\delta\sqrt{k}(\min\{n,R\} - r)\gamma^2\left(1 + 80\mu c_2\sigma_{\min}(\boldsymbol{\mathcal{X}})^2\right)^{2(t-t_*)} \\
(d) \quad &\leq 10c_1\sqrt{k}\kappa^{-2}\sigma_{\min}(\boldsymbol{\mathcal{X}})^2 + 8\delta\sqrt{k}(\min\{n,R\} - r)\gamma^{7/4}\sigma_{\min}(\boldsymbol{\mathcal{X}})^{1/4} \\
(e) \quad &\leq 40c_1\sqrt{k}\kappa^{-2}\sigma_{\min}(\boldsymbol{\mathcal{X}})^2.
\end{aligned}
$$

In inequality (a), we used the fact that $\mathcal{A}$ satisfies RIP$(2r + 1, \delta)$ (and hence, RIP$(r + 1, \delta)$ and RIP$(2, \delta)$), and thus, by Lemmas G.2 and G.3, also satisfies S2SRIP$(r, \delta\sqrt{kr})$ and S2NRIP$(\delta\sqrt{k})$. Inequality (b) uses the assumption $\|\boldsymbol{\mathcal{U}}_t\| \leq 3\|\boldsymbol{\mathcal{X}}\|$ and the fact that $\boldsymbol{\mathcal{U}}_t * \boldsymbol{\mathcal{W}}_{t,\perp} * \boldsymbol{\mathcal{W}}_{t,\perp}^\top * \boldsymbol{\mathcal{U}}_t^\top$ has tubal rank at most $\min\{n,R\} - r$. In inequality (c), we used the assumption $\delta \leq \frac{c_1}{\kappa^4\sqrt{r}}$ along with the second bulleted inequality assumed by the inductive step. Inequality (d) holds due to the definitions of $t_1$ and $t_*$ and the fact that $t_* \leq t \leq t_1$. Finally, inequality (e) holds due to the assumption $\gamma \leq \frac{c_2\sigma_{\min}(\boldsymbol{\mathcal{X}})}{\kappa^2\min\{n,R\}}$.

If $c_1$ is chosen small enough, the above bound is less than $\|\boldsymbol{\mathcal{X}}\|$. Then, along with our other assumptions, we can use Lemma E.6 to obtain $\|\boldsymbol{\mathcal{U}}_{t+1}\| \leq 3\|\boldsymbol{\mathcal{X}}\|$.

Next, we can use Lemma E.1 along with the bound $\sigma_{\min}(\boldsymbol{\mathcal{V}}_{\boldsymbol{\mathcal{X}}}^\top * \boldsymbol{\mathcal{U}}_t) \leq \frac{1}{\sqrt{10}}\sigma_{\min}(\overline{\boldsymbol{\mathcal{X}}})$ to obtain

$$
\begin{aligned}
\sigma_{\min}(\boldsymbol{\mathcal{V}}_{\boldsymbol{\mathcal{X}}}^\top * \boldsymbol{\mathcal{U}}_{t+1}) &\geq \sigma_{\min}(\boldsymbol{\mathcal{V}}_{\boldsymbol{\mathcal{X}}}^\top * \boldsymbol{\mathcal{U}}_{t+1} * \boldsymbol{\mathcal{W}}_{t+1}) \\
&\geq \sigma_{\min}(\boldsymbol{\mathcal{V}}_{\boldsymbol{\mathcal{X}}}^\top * \boldsymbol{\mathcal{U}}_t)\left(1 + \frac{1}{4}\mu\sigma_{\min}(\boldsymbol{\mathcal{X}})^2 - \mu\sigma_{\min}(\boldsymbol{\mathcal{V}}_{\boldsymbol{\mathcal{X}}}^\top * \boldsymbol{\mathcal{U}}_t)^2\right) \\
&\geq \sigma_{\min}(\boldsymbol{\mathcal{V}}_{\boldsymbol{\mathcal{X}}}^\top * \boldsymbol{\mathcal{U}}_t)\left(1 + \frac{1}{4}\mu\sigma_{\min}(\boldsymbol{\mathcal{X}})^2 - \frac{1}{10}\mu\sigma_{\min}(\boldsymbol{\mathcal{X}})^2\right) \\
&\geq \sigma_{\min}(\boldsymbol{\mathcal{V}}_{\boldsymbol{\mathcal{X}}}^\top * \boldsymbol{\mathcal{U}}_t)\left(1 + \frac{1}{8}\mu\sigma_{\min}(\boldsymbol{\mathcal{X}})^2\right) \\
&\geq \frac{1}{2}\gamma\left(1 + \frac{1}{8}\mu\sigma_{\min}(\boldsymbol{\mathcal{X}})^2\right)^{t-t_*} \cdot \left(1 + \frac{1}{8}\mu\sigma_{\min}(\boldsymbol{\mathcal{X}})^2\right) \\
&= \frac{1}{2}\gamma\left(1 + \frac{1}{8}\mu\sigma_{\min}(\boldsymbol{\mathcal{X}})^2\right)^{t-t_*+1}
\end{aligned}
$$

Since $\sigma_{\min}(\boldsymbol{\mathcal{V}}_{\boldsymbol{\mathcal{X}}}^\top * \boldsymbol{\mathcal{U}}_{t+1} * \boldsymbol{\mathcal{W}}_{t+1}) = \sigma_{\min}(\boldsymbol{\mathcal{V}}_{\boldsymbol{\mathcal{X}}}^\top * \boldsymbol{\mathcal{U}}_{t+1})$, which is positive by the above bound, all the singular tubes of $\boldsymbol{\mathcal{V}}_{\boldsymbol{\mathcal{X}}}^\top * \boldsymbol{\mathcal{U}}_{t+1} * \boldsymbol{\mathcal{W}}_{t+1}$ are invertible. Hence, we can apply Lemma E.3 to obtain

$$\|\overline{\mathcal{U}_{t+1} * \mathcal{W}_{t+1,\perp}}^{(j)}\| \le \Big(1 - \frac{\mu}{2}\|\overline{\mathcal{U}_t * \mathcal{W}_{t,\perp}}^{(j)}\|^2 + 9\mu\|\overline{\mathcal{V}_{\mathcal{X}^\perp}^\top * \mathcal{V}_{\mathcal{U}_t * \mathcal{W}_t}}^{(j)}\|\|\mathcal{X}\|^2$$

$$+ 2\mu\|(\mathcal{A}^*\mathcal{A} - \mathcal{I})(\mathcal{X} * \mathcal{X}^\top - \mathcal{U}_t * \mathcal{U}_t^\top)\|\Big)\|\overline{\mathcal{U}_t * \mathcal{W}_{t,\perp}}^{(j)}\|$$

$$\le \Big(1 - \frac{\mu}{2} \cdot 4\gamma^2 \Big(1 + 80\mu c_2 \sqrt{k}\sigma_{\min}(\mathcal{X})^2\Big)^{2(t-t_*)} + 9\mu c_2 \kappa^{-2}\|\mathcal{X}\|^2$$

$$+ 2\mu \cdot 40 c_1 \sqrt{k}\kappa^{-2}\sigma_{\min}(\mathcal{X})^2\Big)\|\overline{\mathcal{U}_t * \mathcal{W}_{t,\perp}}^{(j)}\|$$

$$\le \Big(1 - \frac{\mu}{2} \cdot 4\gamma^2 \Big(1 + 80\mu c_2 \sqrt{k}\sigma_{\min}(\mathcal{X})^2\Big)^{2(t-t_*)} + 9\mu c_2 \sigma_{\min}(\mathcal{X})^2$$

$$+ 80 c_1 \mu \sqrt{k}\kappa^{-2}\sigma_{\min}(\mathcal{X})^2\Big)\|\overline{\mathcal{U}_t * \mathcal{W}_{t,\perp}}^{(j)}\|$$

$$\le \Big(1 + 80 c_1 \mu \sqrt{k}\kappa^{-2}\sigma_{\min}(\mathcal{X})^2\Big)\|\overline{\mathcal{U}_t * \mathcal{W}_{t,\perp}}^{(j)}\|$$

$$\le \Big(1 + 80 c_1 \mu \sqrt{k}\sigma_{\min}(\mathcal{X})^2\Big)\|\overline{\mathcal{U}_t * \mathcal{W}_{t,\perp}}^{(j)}\|$$

$$\le 2\gamma\Big(1 + 80 c_1 \mu \sqrt{k}\sigma_{\min}(\mathcal{X})^2\Big)^{t-t_*+1},$$

where we have used the inductive assumption that the inequalities hold for $t$ along with the fact that $\kappa = \|\mathcal{X}\|/\sigma_{\min}(\mathcal{X}) \ge 1$.

Next, we will bound the term using Lemma E.5

$$\|\mathcal{V}_{\mathcal{X}^\perp}^\top * \mathcal{V}_{\mathcal{U}_{t+1} * \mathcal{W}_{t+1}}\|$$

$$\le \Big(1 - \frac{\mu}{4}\sigma_{\min}^2(\mathcal{X})\Big)\|\mathcal{V}_{\mathcal{X}^\perp}^\top * \mathcal{V}_{\mathcal{U}_t * \mathcal{W}_t}\| + 150\mu\|(\mathcal{A}^*\mathcal{A} - \mathcal{I})(\mathcal{X} * \mathcal{X}^\top - \mathcal{U}_t * \mathcal{U}_t^\top)\| + 500\mu^2\|\mathcal{X} * \mathcal{X}^\top - \mathcal{U}_t * \mathcal{U}_t^\top\|^2$$

$$\le \Big(1 - \frac{\mu}{4}\sigma_{\min}^2(\mathcal{X})\Big)c_2\kappa^{-2} + 150\mu \cdot 40 c_1 \sqrt{k}\kappa^{-2}\sigma_{\min}(\mathcal{X})^2 + 500\mu^2 \cdot (\|\mathcal{X}\|^2 + \|\mathcal{U}_t\|^2)$$

$$\le \Big(1 - \frac{\mu}{4}\sigma_{\min}^2(\mathcal{X})\Big)c_2\kappa^{-2} + 6000\mu c_1 \sqrt{k}\kappa^{-2}\sigma_{\min}(\mathcal{X})^2 + 500\mu^2 \cdot (\|\mathcal{X}\|^2 + 9\|\mathcal{X}\|^2)^2$$

$$= \Big(1 - \frac{\mu}{4}\sigma_{\min}^2(\mathcal{X})\Big)c_2\kappa^{-2} + 6000\mu c_1 \sqrt{k}\kappa^{-2}\sigma_{\min}(\mathcal{X})^2 + 50000\mu^2\|\mathcal{X}\|^4$$

$$\le \Big(1 - \frac{\mu}{4}\sigma_{\min}^2(\mathcal{X})\Big)c_2\kappa^{-2} + 6000\mu c_1 \sqrt{k}\kappa^{-2}\sigma_{\min}(\mathcal{X})^2 + 50000\mu \cdot c_1\kappa^{-4}\|\mathcal{X}\|^{-2} \cdot \|\mathcal{X}\|^4$$

$$= \Big(1 - \frac{\mu}{4}\sigma_{\min}^2(\mathcal{X})\Big)c_2\kappa^{-2} + 6000\mu c_1 \sqrt{k}\kappa^{-2}\sigma_{\min}(\mathcal{X})^2 + 50000\mu \cdot c_1\kappa^{-4}\|\mathcal{X}\|^2$$

$$= \Big(1 - \frac{\mu}{4}\sigma_{\min}^2(\mathcal{X})\Big)c_2\kappa^{-2} + 6000\mu c_1 \sqrt{k}\kappa^{-2}\sigma_{\min}(\mathcal{X})^2 + 50000\mu \cdot c_1\kappa^{-4}\kappa^2\sigma_{\min}(\mathcal{X})^2$$

$$= \Big(1 - \frac{\mu}{4}\sigma_{\min}^2(\mathcal{X})\Big)c_2\kappa^{-2} + 56000\mu c_1 \sqrt{k}\kappa^{-2}\sigma_{\min}(\mathcal{X})^2$$

Here, we have again used the inductive assumptions along with the fact that $\kappa = \|\mathcal{X}\|/\sigma_{\min}(\mathcal{X})$. If we choose $c_1$ sufficiently small, we will have $\|\mathcal{V}_{\mathcal{X}^\perp}^\top * \mathcal{V}_{\mathcal{U}_{t+1} * \mathcal{W}_{t+1}}\| \le c_2\kappa^{-2}$.

Therefore, the four bullet points hold for $t+1$, and thus, the induction is complete.

With the above bullet points in mind, we note that

$$\frac{1}{\sqrt{10}}\sigma_{\min}(\mathcal{X}) \ge \sigma_{\min}(\mathcal{V}_{\mathcal{X}}^\top * \mathcal{U}_{t_1}) \ge \frac{1}{2}\gamma\Big(1 + \frac{1}{8}\mu\sigma_{\min}(\mathcal{X})^2\Big)^{t_1 - t_*},$$

and so,

$$t_1 - t_* \le \frac{\log\Big(\frac{2}{\gamma\sqrt{10}}\sigma_{\min}(\mathcal{X})\Big)}{\log\Big(1 + \frac{1}{8}\mu\sigma_{\min}(\mathcal{X})^2\Big)} \le \frac{16}{\mu\sigma_{\min}(\mathcal{X})^2}\log\Big(\frac{2}{\gamma\sqrt{10}}\sigma_{\min}(\mathcal{X})\Big),$$

where we have used the inequality $\frac{1}{\log(1+x)} \leq \frac{2}{x}$ for $0 < x < 1$. Furthermore, we can bound the norm of the signal term at iteration $t_1$ by

$$\|\mathcal{U}_{t_1} * \mathcal{W}_{t_1,\perp}\| \leq 2\gamma \left(1 + 80\mu c_2 \sqrt{k}\sigma_{\min}(\mathcal{X})^2\right)^{t_1 - t_*}$$

$$\leq 2\gamma \left(\frac{2}{\sqrt{10}} \cdot \frac{\sigma_{\min}(\mathcal{X})}{\gamma}\right)^{1280 c_2}$$

$$\leq 2\gamma \left(\frac{2}{\sqrt{10}} \cdot \frac{\sigma_{\min}(\mathcal{X})}{\gamma}\right)^{1/64}$$

$$\leq 3\gamma^{63/64}\sigma_{\min}(\mathcal{X})^{1/64}$$

$$\leq 3\gamma^{7/8}\sigma_{\min}(\mathcal{X})^{1/8},$$

where we have used the previous bound on $t_1 - t_*$, the fact that $c_2 > 0$ can be chosen to be sufficiently small, and the fact that $\sigma_{\min}(\mathcal{X}) \geq \gamma$.

Next, we set

$$t_2 = t_1 + \left\lfloor \frac{300}{\mu\sigma_{\min}(\mathcal{X})^2} \ln\left(\frac{5}{18}\kappa^{1/4}\sqrt{\frac{r}{k(\min\{n,R\}-r)}}\frac{\|\mathcal{X}\|^{7/4}}{\gamma^{7/4}}\right)\right\rfloor$$

$$t_3 = \min\left\{t \geq t_1 : \left(\sqrt{k(\min\{n,R\}-r)}+1\right)\left\|\mathcal{U}_t * \mathcal{W}_{t,\perp} * \mathcal{W}_{t,\perp}^\top * \mathcal{U}_t^\top\right\|_F \geq \|\mathcal{X} * \mathcal{X}^\top - \mathcal{U}_t * \mathcal{U}_t^\top\|_F\right\}$$

$$\widehat{t} = \min\{t_2, t_3\}.$$

We now aim to show that over the range $t_1 \leq t \leq \widehat{t}$, the following inequalities hold:

- $\sigma_{\min}(\mathcal{U}_t * \mathcal{W}_t) \geq \sigma_{\min}(\mathcal{V}_{\mathcal{X}}^\top * \mathcal{U}_t) \geq \frac{1}{\sqrt{10}}\sigma_{\min}(\mathcal{X})$

- $\|\mathcal{U}_t * \mathcal{W}_{t,\perp}\| \leq \left(1 + 80\mu c_2 \sqrt{k}\sigma_{\min}(\mathcal{X})^2\right)^{t-t_1}\|\mathcal{U}_{t_1} * \mathcal{W}_{t_1,\perp}\|$

- $\|\mathcal{U}_t\| \leq 3\|\mathcal{X}\|$

- $\|\mathcal{V}_{\mathcal{X}^\perp}^\top * \mathcal{V}_{\mathcal{U}_t * \mathcal{W}_t}\| \leq c_2\kappa^{-2}$

- $\|\mathcal{V}_{\mathcal{X}}^\top * (\mathcal{X} * \mathcal{X}^\top - \mathcal{U}_t * \mathcal{U}_t^\top)\|_F \leq 10\sqrt{kr}\left(1 - \frac{1}{400}\mu\sigma_{\min}(\mathcal{X})^2\right)^{t-t_1}\|\mathcal{X}\|^2$

For $t = t_1$, the first four bullet points follow from what we previously proved via induction. The last one holds since we trivially have

$$\|\mathcal{V}_{\mathcal{X}}^\top * (\mathcal{X} * \mathcal{X}^\top - \mathcal{U}_{t_1} * \mathcal{U}_{t_1}^\top)\|_F \leq \sqrt{kr}\|\mathcal{V}_{\mathcal{X}}^\top * (\mathcal{X} * \mathcal{X}^\top - \mathcal{U}_{t_1} * \mathcal{U}_{t_1}^\top)\|$$

$$\leq \sqrt{kr}\|\mathcal{X} * \mathcal{X}^\top - \mathcal{U}_{t_1} * \mathcal{U}_{t_1}^\top\|$$

$$\leq \sqrt{kr}\|\mathcal{X} * \mathcal{X}^\top\| + \sqrt{kr}\|\mathcal{U}_{t_1} * \mathcal{U}_{t_1}^\top\|$$

$$\leq \sqrt{kr}\|\mathcal{X}\|^2 + \sqrt{kr}\|\mathcal{U}_{t_1}\|^2$$

$$\leq 10\sqrt{kr}\|\mathcal{X}\|^2.$$

Now suppose all the bullet points hold for some integer $t \in [t_1, \widehat{t}-1]$. Again, we aim to show they all hold for $t+1$. In a similar manner as done before, we can bound $\|(\mathcal{A}^*\mathcal{A} - \mathcal{I})(\mathcal{X} * \mathcal{X}^\top - \mathcal{U}_t * \mathcal{U}_t^\top)\| \leq 10\delta\sqrt{kr}\|\mathcal{X}\|^2 + \delta\sqrt{k}(\min\{n,R\} - r)\|\mathcal{U}_t * \mathcal{W}_{t,\perp}\|^2$, and then continue as follows

$$\|(\mathcal{A}^*\mathcal{A} - \mathcal{I})(\boldsymbol{\mathcal{X}} * \boldsymbol{\mathcal{X}}^\top - \boldsymbol{\mathcal{U}}_t * \boldsymbol{\mathcal{U}}_t^\top)\|$$

$$\leq 10\delta\sqrt{kr}\|\boldsymbol{\mathcal{X}}\|^2 + \delta\sqrt{k}(\min\{n, R\} - r)\|\boldsymbol{\mathcal{U}}_t * \boldsymbol{\mathcal{W}}_{t,\perp}\|^2$$

$$\leq 10 \cdot \frac{c_1}{\kappa^4\sqrt{r}} \cdot \sqrt{kr} \cdot \kappa^2\sigma_{\min}(\boldsymbol{\mathcal{X}})^2 + \delta\sqrt{k}(\min\{n, R\} - r)\left(1 + 80\mu c_2\sqrt{k}\sigma_{\min}(\boldsymbol{\mathcal{X}})^2\right)^{2(t-t_1)}\|\boldsymbol{\mathcal{U}}_{t_1} * \boldsymbol{\mathcal{W}}_{t_1,\perp}\|^2$$

$$\leq 10c_1\sqrt{k}\kappa^{-2}\sigma_{\min}(\boldsymbol{\mathcal{X}})^2 + \delta\sqrt{k}(\min\{n, R\} - r)\left(1 + 80\mu c_2\sqrt{k}\sigma_{\min}(\boldsymbol{\mathcal{X}})^2\right)^{2(t-t_1)} \cdot 9\gamma^{7/4}\sigma_{\min}(\boldsymbol{\mathcal{X}})^{1/4}$$

$$\leq 10c_1\sqrt{k}\kappa^{-2}\sigma_{\min}(\boldsymbol{\mathcal{X}})^2 + 9\delta\sqrt{k}(\min\{n, R\} - r)\left(1 + 80\mu c_2\sqrt{k}\sigma_{\min}(\boldsymbol{\mathcal{X}})^2\right)^{2(t_2-t_1)}\gamma^{7/4}\sigma_{\min}(\boldsymbol{\mathcal{X}})^{1/4}$$

$$\leq 10c_1\sqrt{k}\kappa^{-2}\sigma_{\min}(\boldsymbol{\mathcal{X}})^2 + 9\delta\sqrt{k}(\min\{n, R\} - r)\left(\frac{5}{18}\kappa^{1/4}\sqrt{\frac{r}{k(\min\{n, R\} - r)}}\frac{\|\boldsymbol{\mathcal{X}}\|^{7/4}}{\gamma^{7/4}}\right)^{O(c_2)}\gamma^{7/4}\sigma_{\min}(\boldsymbol{\mathcal{X}})^{1/4}$$

$$\leq 40c_1\sqrt{k}\kappa^{-2}\sigma_{\min}(\boldsymbol{\mathcal{X}})^2$$

where we have used the bounds $\delta \leq \frac{c_1}{\kappa^4\sqrt{r}}$, $\|\boldsymbol{\mathcal{X}}\| = \kappa\sigma_{\min}(\boldsymbol{\mathcal{X}})$, $\|\boldsymbol{\mathcal{U}}_{t_1} * \boldsymbol{\mathcal{W}}_{t_1,\perp}\| \leq 3\gamma^{7/8}\sigma_{\min}(\boldsymbol{\mathcal{X}})^{1/8}$, along with the inductive assumptions and the definition of $t_1$.

Next, we note that if $\sigma_{\min}(\boldsymbol{\mathcal{V}}_{\boldsymbol{\mathcal{X}}}^\top * \boldsymbol{\mathcal{U}}_t) \leq \frac{1}{2}\sigma_{\min}(\boldsymbol{\mathcal{X}})$, then we can use Lemma E.1 along with the inductive assumptions to obtain

$$\sigma_{\min}(\boldsymbol{\mathcal{U}}_{t+1} * \boldsymbol{\mathcal{W}}_{t+1}) \geq \sigma_{\min}(\boldsymbol{\mathcal{V}}_{\boldsymbol{\mathcal{X}}}^\top * \boldsymbol{\mathcal{U}}_{t+1})$$

$$\geq \sigma_{\min}(\boldsymbol{\mathcal{V}}_{\boldsymbol{\mathcal{X}}}^\top * \boldsymbol{\mathcal{U}}_{t+1} * \boldsymbol{\mathcal{W}}_t)$$

$$\geq \sigma_{\min}(\boldsymbol{\mathcal{V}}_{\boldsymbol{\mathcal{X}}}^\top * \boldsymbol{\mathcal{U}}_t)\left(1 + \frac{1}{4}\mu\sigma_{\min}(\boldsymbol{\mathcal{X}})^2 - \mu\sigma_{\min}(\boldsymbol{\mathcal{V}}_{\boldsymbol{\mathcal{X}}}^\top * \boldsymbol{\mathcal{U}}_t)^2\right)$$

$$\geq \sigma_{\min}(\boldsymbol{\mathcal{V}}_{\boldsymbol{\mathcal{X}}}^\top * \boldsymbol{\mathcal{U}}_t)\left(1 + \frac{1}{4}\mu\sigma_{\min}(\boldsymbol{\mathcal{X}})^2 - \mu \cdot \frac{1}{4}\sigma_{\min}(\boldsymbol{\mathcal{X}})^2\right)$$

$$= \sigma_{\min}(\boldsymbol{\mathcal{V}}_{\boldsymbol{\mathcal{X}}}^\top * \boldsymbol{\mathcal{U}}_t)$$

$$\geq \frac{1}{\sqrt{10}}\sigma_{\min}(\boldsymbol{\mathcal{X}})$$

Alternatively, if $\sigma_{\min}(\boldsymbol{\mathcal{V}}_{\boldsymbol{\mathcal{X}}}^\top * \boldsymbol{\mathcal{U}}_t) \geq \frac{1}{2}\sigma_{\min}(\boldsymbol{\mathcal{X}})$, then we can again use Lemma E.1 along with the inductive assumptions and the fact that $\mu \leq c_1\kappa^{-2}\|\boldsymbol{\mathcal{X}}\|^2$ for sufficiently small $c_1$ to obtain

$$\sigma_{\min}(\boldsymbol{\mathcal{U}}_{t+1} * \boldsymbol{\mathcal{W}}_{t+1}) \geq \sigma_{\min}(\boldsymbol{\mathcal{V}}_{\boldsymbol{\mathcal{X}}}^\top * \boldsymbol{\mathcal{U}}_{t+1})$$

$$\geq \sigma_{\min}(\boldsymbol{\mathcal{V}}_{\boldsymbol{\mathcal{X}}}^\top * \boldsymbol{\mathcal{U}}_{t+1} * \boldsymbol{\mathcal{W}}_t)$$

$$\geq \sigma_{\min}(\boldsymbol{\mathcal{V}}_{\boldsymbol{\mathcal{X}}}^\top * \boldsymbol{\mathcal{U}}_t)\left(1 + \frac{1}{4}\mu\sigma_{\min}(\boldsymbol{\mathcal{X}})^2 - \mu\sigma_{\min}(\boldsymbol{\mathcal{V}}_{\boldsymbol{\mathcal{X}}}^\top * \boldsymbol{\mathcal{U}}_t)^2\right)$$

$$\geq \frac{1}{2}\sigma_{\min}(\boldsymbol{\mathcal{X}})\left(1 - \mu\sigma_{\min}(\boldsymbol{\mathcal{U}}_t)^2\right)$$

$$\geq \frac{1}{2}\sigma_{\min}(\boldsymbol{\mathcal{X}})\left(1 - \mu\|\boldsymbol{\mathcal{U}}_t\|^2\right)$$

$$\geq \frac{1}{2}\sigma_{\min}(\boldsymbol{\mathcal{X}})\left(1 - 9\mu\|\boldsymbol{\mathcal{X}}\|^2\right)$$

$$\geq \frac{1}{2}\sigma_{\min}(\boldsymbol{\mathcal{X}})\left(1 - 9c_1\kappa^{-2}\right)$$

$$\geq \frac{1}{\sqrt{10}}\sigma_{\min}(\boldsymbol{\mathcal{X}})$$

In either case, we have $\sigma_{\min}(\boldsymbol{\mathcal{U}}_{t+1} * \boldsymbol{\mathcal{W}}_{t+1}) \geq \sigma_{\min}(\boldsymbol{\mathcal{V}}_{\boldsymbol{\mathcal{X}}}^\top * \boldsymbol{\mathcal{U}}_{t+1}) \geq \frac{1}{\sqrt{10}}\sigma_{\min}(\boldsymbol{\mathcal{X}})$.

Again, since $\sigma_{\min}(\mathcal{V}_{\mathcal{X}}^\top * \mathcal{U}_{t+1} * \mathcal{W}_t) \geq \frac{1}{\sqrt{10}}\sigma_{\min}(\mathcal{X}) > 0$, we have that $\mathcal{V}_{\mathcal{X}}^\top * \mathcal{U}_{t+1} * \mathcal{W}_t$ has full tubal rank with all invertible t-SVD singular tubes. Hence, by Lemma E.3, we again can bound

$$\|\mathcal{U}_{t+1} * \mathcal{W}_{t+1,\perp}\| \leq \left(1 + 80\mu c_2 \sqrt{k}\sigma_{\min}(\mathcal{X})^2\right)^{t+1-t_1} \|\mathcal{U}_{t_1} * \mathcal{W}_{t_1,\perp}\|.$$

In the exact same way as before, we can use Lemma E.6 to establish $\|\mathcal{U}_{t+1}\| \leq 3\|\mathcal{X}\|$, and use Lemma E.7 to establish $\|\mathcal{V}_{\mathcal{X}\perp}^\top * \mathcal{V}_{\mathcal{U}_{t+1}*\mathcal{W}_{t+1}}\| \leq c_2\kappa^{-2}$.

To bound $\|\mathcal{V}_{\mathcal{X}}^\top * (\mathcal{X} * \mathcal{X}^\top - \mathcal{U}_{t+1} * \mathcal{U}_{t+1}^\top)\|_F$, we will aim to use Lemma E.7. By the inductive assumptions, we already have $\|\mathcal{U}_t\| \leq 3\|\mathcal{X}\|$, $\sigma_{\min}(\mathcal{U}_t * \mathcal{W}_t) \geq \frac{1}{\sqrt{10}}\sigma_{\min}(\mathcal{X})$, and $\|\mathcal{V}_{\mathcal{X}\perp}^\top * \mathcal{V}_{\mathcal{U}_t*\mathcal{W}_t}\| \leq c_2\kappa^{-2}$. To derive the remaining condition of Lemma E.7, we first split

$$
\begin{aligned}
&\|\mathcal{V}_{\mathcal{X}}^\top * (\mathcal{I} - \mathcal{A}^*\mathcal{A})(\mathcal{X} * \mathcal{X}^\top - \mathcal{U} * \mathcal{U}^\top)\|_F \\
=&\|\mathcal{V}_{\mathcal{X}}^\top * (\mathcal{I} - \mathcal{A}^*\mathcal{A})(\mathcal{X} * \mathcal{X}^\top - \mathcal{U}_t * \mathcal{W}_t\mathcal{W}_t^\top * \mathcal{U}_t^\top - \mathcal{U}_t * \mathcal{W}_{t,\perp}\mathcal{W}_{t,\perp}^\top * \mathcal{U}_t^\top)\|_F \\
\leq&\|\mathcal{V}_{\mathcal{X}}^\top * (\mathcal{I} - \mathcal{A}^*\mathcal{A})(\mathcal{X} * \mathcal{X}^\top - \mathcal{U}_t * \mathcal{W}_t * \mathcal{W}_t^\top * \mathcal{U}_t^\top)\|_F + \|\mathcal{V}_{\mathcal{X}}^\top * (\mathcal{I} - \mathcal{A}^*\mathcal{A})(\mathcal{U}_t * \mathcal{W}_{t,\perp} * \mathcal{W}_{t,\perp}^\top * \mathcal{U}_t^\top)\|_F.
\end{aligned}
$$

To bound the first term, we note that $\mathcal{X} * \mathcal{X}^\top - \mathcal{U}_t * \mathcal{W}_t * \mathcal{W}_t^\top * \mathcal{U}_t^\top$ is tubal-symmetric with tubal rank at most $2r$, so we can write it as the sum of two tubal-symmetric tensors $\mathcal{Z}_1, \mathcal{Z}_2 \in S^{n \times n \times k}$ with tubal rank at most $r$, and then apply Lemma G.4 to obtain

$$
\begin{aligned}
\|\mathcal{V}_{\mathcal{X}}^\top * (\mathcal{I} - \mathcal{A}^*\mathcal{A})(\mathcal{X} * \mathcal{X}^\top - \mathcal{U}_t * \mathcal{W}_t * \mathcal{W}_t^\top * \mathcal{U}_t^\top)\|_F &= \|\mathcal{V}_{\mathcal{X}}^\top * (\mathcal{I} - \mathcal{A}^*\mathcal{A})(\mathcal{Z}_1 + \mathcal{Z}_2)\|_F \\
&\leq \|\mathcal{V}_{\mathcal{X}}^\top * (\mathcal{I} - \mathcal{A}^*\mathcal{A})(\mathcal{Z}_1)\|_F + \|\mathcal{V}_{\mathcal{X}}^\top * (\mathcal{I} - \mathcal{A}^*\mathcal{A})(\mathcal{Z}_2)\|_F \\
&\leq \delta(\|\mathcal{Z}_1\|_F + \|\mathcal{Z}_2\|_F) \\
&\leq \delta\sqrt{2}\|\mathcal{Z}_1 + \mathcal{Z}_2\|_F \\
&= \delta\sqrt{2}\|\mathcal{X} * \mathcal{X}^\top - \mathcal{U}_t * \mathcal{W}_t * \mathcal{W}_t^\top * \mathcal{U}_t^\top\|_F \\
&\leq \delta\sqrt{2}\|\mathcal{X} * \mathcal{X}^\top - \mathcal{U}_t * \mathcal{U}_t^\top\|_F
\end{aligned}
$$

For the second piece, we use the symmetric t-SVD to write $\mathcal{U}_t * \mathcal{W}_{t,\perp} * \mathcal{W}_{t,\perp}^\top * \mathcal{U}_t^\top = \sum_i \mathcal{V}_i * s_i * \mathcal{V}_i^\top$. Then, we can bound

$$
\begin{aligned}
\|\mathcal{V}_{\mathcal{X}}^\top * (\mathcal{I} - \mathcal{A}^*\mathcal{A})(\mathcal{U}_t * \mathcal{W}_{t,\perp} * \mathcal{W}_{t,\perp}^\top * \mathcal{U}_t^\top)\|_F &= \left\|\mathcal{V}_{\mathcal{X}}^\top * (\mathcal{I} - \mathcal{A}^*\mathcal{A})\left(\sum_i \mathcal{V}_i * s_i * \mathcal{V}_i^\top\right)\right\|_F \\
&\leq \sum_i \left\|\mathcal{V}_{\mathcal{X}}^\top * (\mathcal{I} - \mathcal{A}^*\mathcal{A})\left(\mathcal{V}_i * s_i * \mathcal{V}_i^\top\right)\right\|_F \\
&\leq \sum_i \delta\left\|\mathcal{V}_i * s_i * \mathcal{V}_i^\top\right\|_F \\
&= \sum_i \delta\|s_i\|_2 \\
&= \delta\left\|\mathcal{U}_t * \mathcal{W}_{t,\perp} * \mathcal{W}_{t,\perp}^\top * \mathcal{U}_t^\top\right\|_* \\
&\leq \delta\sqrt{k(\min\{n, R\} - r)}\left\|\mathcal{U}_t * \mathcal{W}_{t,\perp} * \mathcal{W}_{t,\perp}^\top * \mathcal{U}_t^\top\right\|_F \\
&\leq \|\mathcal{X} * \mathcal{X}^\top - \mathcal{U}_t * \mathcal{U}_t^\top\|_F,
\end{aligned}
$$

where we have used the fact that $\mathcal{U}_t * \mathcal{W}_{t,\perp} * \mathcal{W}_{t,\perp}^\top * \mathcal{U}_t^\top$ has tubal rank $\leq \min\{n, R\} - r$ along with the definition of $t_3$.

Hence,

$$\|\mathcal{V}_{\mathcal{X}}^\top * (\mathcal{I} - \mathcal{A}^*\mathcal{A})(\mathcal{X} * \mathcal{X}^\top - \mathcal{U} * \mathcal{U}^\top)\|_F$$
$$\leq \|\mathcal{V}_{\mathcal{X}}^\top * (\mathcal{I} - \mathcal{A}^*\mathcal{A})(\mathcal{X} * \mathcal{X}^\top - \mathcal{U}_t * \mathcal{W}_t * \mathcal{W}_t^\top * \mathcal{U}_t^\top)\|_F + \|\mathcal{V}_{\mathcal{X}}^\top * (\mathcal{I} - \mathcal{A}^*\mathcal{A})(\mathcal{U}_t * \mathcal{W}_{t,\perp} * \mathcal{W}_{t,\perp}^\top * \mathcal{U}_t^\top)\|_F$$
$$\leq \delta\sqrt{2}\|\mathcal{X} * \mathcal{X}^\top - \mathcal{U}_t * \mathcal{U}_t^\top\|_F + \delta\|\mathcal{X} * \mathcal{X}^\top - \mathcal{U}_t * \mathcal{U}_t^\top\|_F$$
$$\leq c\kappa^{-2}\|\mathcal{X} * \mathcal{X}^\top - \mathcal{U}_t * \mathcal{U}_t^\top\|_F,$$

where we have used the assumption that $\delta \leq \frac{c_1}{\kappa^4\sqrt{r}} \leq c\kappa^{-2}$.

Similarly, we can bound

$$\|\mathcal{V}_{\mathcal{U}_t * \mathcal{W}_t}^\top * (\mathcal{I} - \mathcal{A}^*\mathcal{A})(\mathcal{X} * \mathcal{X}^\top - \mathcal{U}_t * \mathcal{U}_t)\|_F \leq c\kappa^{-2}\|\mathcal{X} * \mathcal{X}^\top - \mathcal{U}_t * \mathcal{U}_t^\top\|_F,$$

and

$$\|(\mathcal{I} - \mathcal{A}^*\mathcal{A})(\mathcal{X} * \mathcal{X}^\top - \mathcal{U}_t * \mathcal{U}_t)\| \leq c\kappa^{-2}\|\mathcal{X} * \mathcal{X}^\top - \mathcal{U}_t * \mathcal{U}_t^\top\|_F.$$

Then, by Lemma E.7, we have

$$\|\mathcal{V}_{\mathcal{X}^\perp}^\top (\mathcal{X} * \mathcal{X}^\top - \mathcal{U}_{t+1} * \mathcal{U}_{t+1}^\top)\|_F \leq \left(1 - \frac{\mu}{200}\sigma_{\min}^2(\mathcal{X})\right)\|\mathcal{V}_{\mathcal{X}^\perp}^\top * (\mathcal{X} * \mathcal{X}^\top - \mathcal{U}_t * \mathcal{U}_t^\top)\|_F$$
$$+ \mu\frac{\sigma_{\min}^2(\mathcal{X})}{100}\|\mathcal{U}_t * \mathcal{W}_{t,\perp} * \mathcal{W}_{t,\perp}^\top * \mathcal{U}_t^\top\|_F$$

By the inductive assumption,

$$\|\mathcal{V}_{\mathcal{X}^\perp}^\top * (\mathcal{X} * \mathcal{X}^\top - \mathcal{U}_t * \mathcal{U}_t^\top)\|_F \leq 10\sqrt{kr}\left(1 - \frac{1}{400}\mu\sigma_{\min}(\mathcal{X})^2\right)^{t-t_1}\|\mathcal{X}\|^2.$$

Also, using the inductive assumption and the bound from the previous part, we can bound

$$\|\mathcal{U}_t * \mathcal{W}_{t,\perp} * \mathcal{W}_{t,\perp}^\top * \mathcal{U}_t^\top\|_F \leq \sqrt{k(\min\{n,R\} - r)}\|\mathcal{U}_t * \mathcal{W}_{t,\perp} * \mathcal{W}_{t,\perp}^\top * \mathcal{U}_t^\top\|$$
$$\leq \sqrt{k(\min\{n,R\} - r)}\|\mathcal{U}_t * \mathcal{W}_{t,\perp}\|^2$$
$$\leq \sqrt{k(\min\{n,R\} - r)}\left(1 + 80\mu c_2\sqrt{k}\sigma_{\min}(\mathcal{X})^2\right)^{2(t-t_1)}\|\mathcal{U}_{t_1} * \mathcal{W}_{t_1,\perp}\|^2$$
$$\leq \sqrt{k(\min\{n,R\} - r)}\left(1 + 80\mu c_2\sqrt{k}\sigma_{\min}(\mathcal{X})^2\right)^{2(t-t_1)} \cdot 9\gamma^{7/4}\sigma_{\min}(\mathcal{X})^{1/4}$$

Since $t \leq t_2$, we have

$$t - t_1 \leq t_2 - t_1 \leq \frac{300}{\mu\sqrt{k}\sigma_{\min}(\mathcal{X})^2}\ln\left(\frac{5}{18}\kappa^{1/4}\sqrt{\frac{r}{\min\{n,R\} - r}}\frac{\|\mathcal{X}\|^{7/4}}{\gamma^{7/4}}\right),$$

and thus,

$$\|\mathcal{U}_t * \mathcal{W}_{t,\perp} * \mathcal{W}_{t,\perp}^\top * \mathcal{U}_t^\top\|_F \leq \sqrt{k(\min\{n,R\} - r)}\left(1 + 80\mu c_2\sqrt{k}\sigma_{\min}(\mathcal{X})^2\right)^{2(t-t_1)} \cdot 9\gamma^{7/4}\sigma_{\min}(\mathcal{X})^{1/4}$$
$$\leq \frac{5}{2}\sqrt{kr}\left(1 - \frac{\mu}{400}\sigma_{\min}(\mathcal{X})^2\right)^{t-t_1}\|\mathcal{X}\|^2.$$

Combining these inequalities yields

$$\|\mathcal{V}_{\mathcal{X}^\perp}^\top(\mathcal{X} * \mathcal{X}^\top - \mathcal{U}_{t+1} * \mathcal{U}_{t+1}^\top)\|_F \leq \left(1 - \frac{\mu}{200}\sigma_{\min}^2(\mathcal{X})\right)\|\mathcal{V}_{\mathcal{X}^\perp}^\top * (\mathcal{X} * \mathcal{X}^\top - \mathcal{U}_t * \mathcal{U}_t^\top)\|_F$$
$$+ \mu\frac{\sigma_{\min}^2(\mathcal{X})}{100}\|\mathcal{U}_t * \mathcal{W}_{t,\perp} * \mathcal{W}_{t,\perp}^\top * \mathcal{U}_t^\top\|_F$$
$$\leq \left(1 - \frac{\mu}{200}\sigma_{\min}^2(\mathcal{X})\right) \cdot 10\sqrt{kr}\left(1 - \frac{1}{400}\mu\sigma_{\min}(\mathcal{X})^2\right)^{t-t_1}\|\mathcal{X}\|^2$$
$$+ \mu\frac{\sigma_{\min}^2(\mathcal{X})}{100} \cdot \frac{5}{2}\sqrt{kr}\left(1 - \frac{\mu}{400}\sigma_{\min}(\mathcal{X})^2\right)^{t-t_1}\|\mathcal{X}\|^2$$
$$\leq 10\sqrt{kr}\left(1 - \frac{1}{400}\mu\sigma_{\min}(\mathcal{X})^2\right)^{t+1-t_1}\|\mathcal{X}\|^2$$

Hence, by induction, the five bullet points hold for $t + 1$.

If $\widehat{t} = t_2$, then, we can use Lemma E.7, the previous bullet points, and the definition of $t_2$ to bound

$$\|\boldsymbol{\mathcal{X}} * \boldsymbol{\mathcal{X}}^\top - \boldsymbol{\mathcal{U}}_{\widehat{t}} * \boldsymbol{\mathcal{U}}_{\widehat{t}}^\top\|_F \leq 4\|\boldsymbol{\mathcal{V}}_{\boldsymbol{\mathcal{X}}^\perp}^\top * (\boldsymbol{\mathcal{X}} * \boldsymbol{\mathcal{X}}^\top - \boldsymbol{\mathcal{U}}_{\widehat{t}} * \boldsymbol{\mathcal{U}}_{\widehat{t}}^\top)\|_F + \|\boldsymbol{\mathcal{U}}_{\widehat{t}} * \boldsymbol{\mathcal{W}}_{\widehat{t},\perp} * \boldsymbol{\mathcal{W}}_{\widehat{t},\perp}^\top * \boldsymbol{\mathcal{U}}_{\widehat{t}}^\top\|_F$$

$$\leq 40\sqrt{kr}\left(1 - \frac{1}{400}\mu\sigma_{\min}(\boldsymbol{\mathcal{X}})^2\right)^{\widehat{t}-t_1}\|\boldsymbol{\mathcal{X}}\|^2 + \frac{5}{2}\sqrt{kr}\left(1 - \frac{1}{400}\mu\sigma_{\min}(\boldsymbol{\mathcal{X}})^2\right)^{\widehat{t}-t_1}\|\boldsymbol{\mathcal{X}}\|^2$$

$$= \frac{85}{2}\sqrt{kr}\left(1 - \frac{1}{400}\mu\sigma_{\min}(\boldsymbol{\mathcal{X}})^2\right)^{\widehat{t}-t_1}\|\boldsymbol{\mathcal{X}}\|^2$$

$$\lesssim \sqrt{kr}\left(\frac{5}{18}\kappa^{1/4}\sqrt{\frac{r}{k(\min\{n,R\}-r)}}\frac{\|\boldsymbol{\mathcal{X}}\|^{7/4}}{\gamma^{7/4}}\right)^{-3/4}\|\boldsymbol{\mathcal{X}}\|^2$$

$$\lesssim k^{5/4}r^{1/8}\kappa^{-3/16}(\min\{n,R\}-r)^{3/8}\gamma^{21/16}\|\boldsymbol{\mathcal{X}}\|^{11/16}$$

If instead we have $\widehat{t} = t_3$, then

$$\|\boldsymbol{\mathcal{X}} * \boldsymbol{\mathcal{X}}^\top - \boldsymbol{\mathcal{U}}_{\widehat{t}} * \boldsymbol{\mathcal{U}}_{\widehat{t}}^\top\|_F$$

$$\leq 4\|\boldsymbol{\mathcal{V}}_{\boldsymbol{\mathcal{X}}^\perp}^\top * (\boldsymbol{\mathcal{X}} * \boldsymbol{\mathcal{X}}^\top - \boldsymbol{\mathcal{U}}_{\widehat{t}} * \boldsymbol{\mathcal{U}}_{\widehat{t}}^\top)\|_F + \|\boldsymbol{\mathcal{U}}_{\widehat{t}} * \boldsymbol{\mathcal{W}}_{\widehat{t},\perp} * \boldsymbol{\mathcal{W}}_{\widehat{t},\perp}^\top * \boldsymbol{\mathcal{U}}_{\widehat{t}}^\top\|_F$$

$$\leq 4\|\boldsymbol{\mathcal{X}} * \boldsymbol{\mathcal{X}}^\top - \boldsymbol{\mathcal{U}}_{\widehat{t}} * \boldsymbol{\mathcal{U}}_{\widehat{t}}^\top\|_F + \|\boldsymbol{\mathcal{U}}_{\widehat{t}} * \boldsymbol{\mathcal{W}}_{\widehat{t},\perp} * \boldsymbol{\mathcal{W}}_{\widehat{t},\perp}^\top * \boldsymbol{\mathcal{U}}_{\widehat{t}}^\top\|_F$$

$$\leq 4(\sqrt{k(\min\{n,R\}-r)}+1)\|\boldsymbol{\mathcal{U}}_{\widehat{t}} * \boldsymbol{\mathcal{W}}_{\widehat{t},\perp} * \boldsymbol{\mathcal{W}}_{\widehat{t},\perp}^\top * \boldsymbol{\mathcal{U}}_{\widehat{t}}^\top\|_F + \|\boldsymbol{\mathcal{U}}_{\widehat{t}} * \boldsymbol{\mathcal{W}}_{\widehat{t},\perp} * \boldsymbol{\mathcal{W}}_{\widehat{t},\perp}^\top * \boldsymbol{\mathcal{U}}_{\widehat{t}}^\top\|_F$$

$$= 4(\sqrt{k(\min\{n,R\}-r)}+5)\|\boldsymbol{\mathcal{U}}_{\widehat{t}} * \boldsymbol{\mathcal{W}}_{\widehat{t},\perp} * \boldsymbol{\mathcal{W}}_{\widehat{t},\perp}^\top * \boldsymbol{\mathcal{U}}_{\widehat{t}}^\top\|_F$$

$$\leq 4(\sqrt{k(\min\{n,R\}-r)}+5)\sqrt{\min\{n,R\}-r}\|\boldsymbol{\mathcal{U}}_{\widehat{t}} * \boldsymbol{\mathcal{W}}_{\widehat{t},\perp} * \boldsymbol{\mathcal{W}}_{\widehat{t},\perp}^\top * \boldsymbol{\mathcal{U}}_{\widehat{t}}^\top\|$$

$$\leq 4(\sqrt{k(\min\{n,R\}-r)}+5)\sqrt{\min\{n,R\}-r}\|\boldsymbol{\mathcal{U}}_{\widehat{t}} * \boldsymbol{\mathcal{W}}_{\widehat{t},\perp}\|^2$$

$$\leq 4(\sqrt{k(\min\{n,R\}-r)}+5)\sqrt{k(\min\{n,R\}-r)}\left(1 + 80\mu c_2\sqrt{k}\sigma_{\min}(\boldsymbol{\mathcal{X}})^2\right)^{2(\widehat{t}-t_1)}\|\boldsymbol{\mathcal{U}}_{t_1} * \boldsymbol{\mathcal{W}}_{t_1,\perp}\|^2$$

$$\leq 4(\sqrt{k(\min\{n,R\}-r)}+5)\sqrt{k(\min\{n,R\}-r)}\left(1 + 80\mu c_2\sqrt{k}\sigma_{\min}(\boldsymbol{\mathcal{X}})^2\right)^{2(\widehat{t}-t_1)} \cdot 9\gamma^{63/32}\sigma_{\min}(\boldsymbol{\mathcal{X}})^{1/32}$$

$$\lesssim k(\min\{n,R\}-r)\left(\frac{5}{18}\kappa^{1/4}\sqrt{\frac{r}{k(\min\{n,R\}-r)}}\frac{\|\boldsymbol{\mathcal{X}}\|^{7/4}}{\gamma^{7/4}}\right)^{O(c_2)}\gamma^{63/32}\sigma_{\min}(\boldsymbol{\mathcal{X}})^{1/32}$$

$$\lesssim k(\min\{n,R\}-r)\left(\frac{5}{18}\kappa^{1/4}\sqrt{\frac{r}{k(\min\{n,R\}-r)}}\frac{\|\boldsymbol{\mathcal{X}}\|^{7/4}}{\gamma^{7/4}}\right)^{O(c_2)}\gamma^{21/16}\gamma^{21/32}\frac{\|\boldsymbol{\mathcal{X}}\|^{1/32}}{\kappa^{1/32}}$$

$$\lesssim k(\min\{n,R\}-r)\left(\frac{5}{18}\kappa^{1/4}\sqrt{\frac{r}{k(\min\{n,R\}-r)}}\frac{\|\boldsymbol{\mathcal{X}}\|^{7/4}}{\gamma^{7/4}}\right)^{O(c_2)}\gamma^{21/16}\left(\frac{\|\boldsymbol{\mathcal{X}}\|}{\min\{n,R\}\kappa^3}\right)^{21/32}\frac{\|\boldsymbol{\mathcal{X}}\|^{1/32}}{\kappa^{1/32}}$$

$$\lesssim \frac{k(\min\{n,R\}-r)}{\min\{n,R\}^{21/32}}\left(\frac{5}{18}\kappa^{1/4}\sqrt{\frac{r}{k(\min\{n,R\}-r)}}\frac{\|\boldsymbol{\mathcal{X}}\|^{7/4}}{\gamma^{7/4}}\right)^{O(c_2)}\gamma^{21/16}\kappa^{-2}\|\boldsymbol{\mathcal{X}}\|^{11/16}$$

$$\lesssim k^{5/4}r^{1/8}\kappa^{-3/16}(\min\{n,R\}-r)^{3/8}\gamma^{21/16}\|\boldsymbol{\mathcal{X}}\|^{11/16}.$$

So in either case, we have

$$\|\boldsymbol{\mathcal{X}} * \boldsymbol{\mathcal{X}}^\top - \boldsymbol{\mathcal{U}}_{\widehat{t}} * \boldsymbol{\mathcal{U}}_{\widehat{t}}^\top\|_F \lesssim k^{5/4}r^{1/8}\kappa^{-3/16}(\min\{n,R\}-r)^{3/8}\gamma^{21/16}\|\boldsymbol{\mathcal{X}}\|^{11/16},$$

and thus,

$$\frac{\|\boldsymbol{\mathcal{X}} * \boldsymbol{\mathcal{X}}^\top - \boldsymbol{\mathcal{U}}_{\widehat{t}} * \boldsymbol{\mathcal{U}}_{\widehat{t}}^\top\|_F}{\|\boldsymbol{\mathcal{X}}\|^2} \lesssim k^{5/4}r^{1/8}\kappa^{-3/16}(\min\{n,R\}-r)^{3/8}\gamma^{21/16}\|\boldsymbol{\mathcal{X}}\|^{-21/16}.$$

Finally, by the definition of $\widehat{t}$, we have that

$$
\begin{aligned}
\widehat{t} - t_* &\leq t_2 - t_* \\
&\leq (t_2 - t_1) + (t_1 - t_*) \\
&\leq \frac{300}{\mu\sqrt{k}\sigma_{\min}(\boldsymbol{\mathcal{X}})^2} \ln\left(\frac{5}{18}\kappa^{1/4}\sqrt{\frac{r}{k(\min\{n,R\}-r)}}\frac{\|\boldsymbol{\mathcal{X}}\|^{7/4}}{\gamma^{7/4}}\right) + \frac{16}{\mu\sigma_{\min}(\boldsymbol{\mathcal{X}})^2}\log\left(\frac{2}{\gamma\sqrt{10}}\sigma_{\min}(\boldsymbol{\mathcal{X}})\right) \\
&\lesssim \frac{1}{\mu\sigma_{\min}(\boldsymbol{\mathcal{X}})^2}\ln\left(\min\left\{1,\frac{\kappa r}{k(\min\{n,R\}-r)}\right\}\frac{\|\boldsymbol{\mathcal{X}}\|}{\gamma}\right)
\end{aligned}
$$

$\square$

## F. Proof of Main Result

Now that our analyses of the spectral stage and the convergence stage are complete, we are ready to combine these pieces to obtain the proof of our main result. Since $\mathcal{A}$ satisfies RIP$(2r+1,\delta)$, by Lemma G.2, $\mathcal{A}$ also satisfies S2SRIP$(2r,\sqrt{2kr}\delta)$. Hence, $\boldsymbol{\mathcal{E}} := (\mathcal{I} - \mathcal{A}^*\mathcal{A})(\boldsymbol{\mathcal{X}} * \boldsymbol{\mathcal{X}}^\top)$ satisfies

$$
\|\boldsymbol{\mathcal{E}}\| = \|(\mathcal{I} - \mathcal{A}^*\mathcal{A})(\boldsymbol{\mathcal{X}} * \boldsymbol{\mathcal{X}}^\top)\| \leq \sqrt{2kr}\delta\|\boldsymbol{\mathcal{X}} * \boldsymbol{\mathcal{X}}^\top\| \leq \sqrt{2kr}\cdot c\kappa^{-4}r^{-1/2}\cdot\|\boldsymbol{\mathcal{X}}\|^2 = c\sqrt{k}\kappa^{-2}\sigma_{\min}(\boldsymbol{\mathcal{X}})^2.
$$

Then, by applying Lemma D.9, with $\epsilon = \frac{1}{\tilde{C}}e^{-3\tilde{c}}$, we have that with probability at least $1 - k(\tilde{C}\epsilon)^{R-2r+1} - ke^{-\tilde{c}R} = 1 - ke^{-3\tilde{c}(R-2r+1)} - ke^{-\tilde{c}R} \geq 1 - ke^{-3\tilde{c}\cdot\frac{1}{3}R} - ke^{-\tilde{c}R} = 1 - O(ke^{-\tilde{c}R})$, after

$$
t_* \lesssim \frac{1}{\mu\sigma_{\min}(\boldsymbol{\mathcal{X}})^2}\ln\left(\frac{2\kappa^2\sqrt{n}}{\tilde{c}_3\sqrt{\min\{n;R\}}}\right)
$$

iterations, we have

$$
\|\boldsymbol{\mathcal{U}}_{t_\star}\| \leq 3\|\boldsymbol{\mathcal{X}}\| \tag{F.1}
$$

$$
\|\boldsymbol{\mathcal{V}}_{\boldsymbol{\mathcal{X}}^\perp} * \boldsymbol{\mathcal{V}}_{\boldsymbol{\mathcal{U}}_{t_\star}*\boldsymbol{\mathcal{W}}_{t_\star}}\| \leq c\kappa^{-2}. \tag{F.2}
$$

and for each $1 \leq j \leq k$, we have

$$
\sigma_r\left(\overline{\boldsymbol{\mathcal{U}}_{t_\star} * \boldsymbol{\mathcal{W}}_{t_\star}}^{(j)}\right) \geq \frac{1}{4}\alpha\beta \tag{F.3}
$$

$$
\sigma_1\left(\overline{\boldsymbol{\mathcal{U}}_{t_\star} * \boldsymbol{\mathcal{W}}_{t_\star,\perp}}^{(j)}\right) \leq \frac{\kappa^{-2}}{8}\alpha\beta \tag{F.4}
$$

$$
\tag{F.5}
$$

where (since $R \geq 3r$ and $\epsilon$ is a constant),

$$
\sqrt{k} \lesssim \beta \lesssim \sqrt{k}\left(\frac{2\kappa^2\sqrt{n}}{\tilde{c}_3\sqrt{\min\{n;R\}}}\right)^{16\kappa^2}.
$$

By choosing

$$
\alpha \lesssim \frac{4c_2\sigma_{\min}(\boldsymbol{\mathcal{X}})}{\kappa^2\min\{n,R\}\sqrt{k}}\left(\frac{2\kappa^2\sqrt{n}}{\tilde{c}_3\sqrt{\min\{n,R\}}}\right)^{-16\kappa^2},
$$

we have

$$
\gamma = \frac{1}{4}\alpha\beta \lesssim \frac{c_2\sigma_{\min}(\boldsymbol{\mathcal{X}})}{\kappa^2\min\{n,R\}}.
$$

Also, $\frac{\kappa^{-2}}{8}\alpha\beta = \frac{1}{2\kappa^2}\gamma \leq 2\gamma$ holds. Therefore, we can apply Theorem E.1, which gives us that after

$$
\widehat{t} - t_* \lesssim \frac{1}{\mu\sigma_{\min}(\boldsymbol{\mathcal{X}})^2}\ln\left(\min\left\{1,\frac{\kappa r}{k(\min\{n,R\}-r)}\right\}\frac{\|\boldsymbol{\mathcal{X}}\|}{\gamma}\right)
$$

iterations beyond the first phase, we have

$$\frac{\|\boldsymbol{\mathcal{U}}_{\widehat{t}} * \boldsymbol{\mathcal{U}}_{\widehat{t}}^\top - \boldsymbol{\mathcal{X}} * \boldsymbol{\mathcal{X}}^\top\|_F}{\|\boldsymbol{\mathcal{X}}\|^2} \lesssim k^{5/4} r^{1/8} \kappa^{-3/16} (\min\{n, R\} - r)^{3/8} \gamma^{21/16} \|\boldsymbol{\mathcal{X}}\|^{-21/16}.$$

The total amount of iterations is then bounded by

$$
\begin{aligned}
\widehat{t} &= t_* + (\widehat{t} - t_*) \\
&\lesssim \frac{1}{\mu \sigma_{\min}(\boldsymbol{\mathcal{X}})^2} \ln\left(\frac{2\kappa^2 \sqrt{n}}{\tilde{c}_3 \sqrt{\min\{n, R\}}}\right) + \frac{1}{\mu \sigma_{\min}(\boldsymbol{\mathcal{X}})^2} \ln\left(\min\left\{1, \frac{\kappa r}{k(\min\{n, R\} - r)}\right\} \frac{\|\boldsymbol{\mathcal{X}}\|}{\gamma}\right) \\
&\lesssim \frac{1}{\mu \sigma_{\min}(\boldsymbol{\mathcal{X}})^2} \ln\left(\frac{2\kappa^2 \sqrt{n}}{\tilde{c}_3 \sqrt{\min\{n, R\}}} \cdot \min\left\{1, \frac{\kappa r}{k(\min\{n, R\} - r)}\right\} \frac{\|\boldsymbol{\mathcal{X}}\|}{\gamma}\right) \\
&\lesssim \frac{1}{\mu \sigma_{\min}(\boldsymbol{\mathcal{X}})^2} \ln\left(\frac{2\kappa^2 \sqrt{n}}{\tilde{c}_3 \sqrt{\min\{n, R\}}} \cdot \min\left\{1, \frac{\kappa r}{k(\min\{n, R\} - r)}\right\} \frac{4\|\boldsymbol{\mathcal{X}}\|}{\alpha\beta}\right) \\
&\lesssim \frac{1}{\mu \sigma_{\min}(\boldsymbol{\mathcal{X}})^2} \ln\left(\frac{C_1 \kappa n}{\min\{n, R\}} \cdot \min\left\{1, \frac{\kappa r}{k(\min\{n, R\} - r)}\right\} \frac{\|\boldsymbol{\mathcal{X}}\|}{k\alpha}\right),
\end{aligned}
$$

where we have used the choice of $\gamma = \frac{1}{4}\alpha\beta$ and the fact that $\beta \gtrsim \sqrt{k}$. Finally, the error is bounded by

$$
\begin{aligned}
\frac{\|\boldsymbol{\mathcal{U}}_{\widehat{t}} * \boldsymbol{\mathcal{U}}_{\widehat{t}}^\top - \boldsymbol{\mathcal{X}} * \boldsymbol{\mathcal{X}}^\top\|_F}{\|\boldsymbol{\mathcal{X}}\|^2} &\lesssim k^{5/4} r^{1/8} \kappa^{-3/16} (\min\{n, R\} - r)^{3/8} \gamma^{21/16} \|\boldsymbol{\mathcal{X}}\|^{-21/16} \\
&\lesssim k^{5/4} r^{1/8} \kappa^{-3/16} (\min\{n, R\} - r)^{3/8} (\alpha\beta)^{21/16} \|\boldsymbol{\mathcal{X}}\|^{-21/16} \\
&\lesssim k^{5/4} r^{1/8} \kappa^{-3/16} (\min\{n, R\} - r)^{3/8} k^{21/32} \left(\frac{2\kappa^2 \sqrt{n}}{\tilde{c}_3 \sqrt{\min\{n, R\}}}\right)^{21\kappa^2} \left(\frac{\alpha}{\|\boldsymbol{\mathcal{X}}\|}\right)^{21/16} \\
&\lesssim k^{61/32} r^{1/8} \kappa^{-3/16} (\min\{n, R\} - r)^{3/8} \left(\frac{C_2 \kappa^2 \sqrt{n}}{\sqrt{\min\{n, R\}}}\right)^{21\kappa^2} \left(\frac{\alpha}{\|\boldsymbol{\mathcal{X}}\|}\right)^{21/16},
\end{aligned}
$$

as desired.

Remark: One could obtain similar results for the cases where $r \leq R < 2r$ and $2r \leq R < 3r$ by choosing the parameter $\epsilon \in (0, 1)$ appropriately.

## G. Restricted Isometry Property

In this section, we show that a measurement operator which satisfies the standard restricted isometry property also satisfies two other variants of the restricted isometry property - a fact which we used in our analysis of the convergence stage.

We say that a measurement operator $\mathcal{A} : S^{n \times n \times k} \to \mathbb{R}^m$ satisfies the spectral-to-spectral Restricted Isometry Property of rank-$r$ with constant $\delta > 0$ (abbreviated S2SRIP($r, \delta$)) if for all tensors $\boldsymbol{\mathcal{Z}} \in S^{n \times n \times k}$ with tubal-rank $\leq r$,

$$\|(\mathcal{I} - \mathcal{A}^* \mathcal{A})(\boldsymbol{\mathcal{Z}})\| \leq \delta \|\boldsymbol{\mathcal{Z}}\|.$$

We say that a measurement operator $\mathcal{A} : S^{n \times n \times k} \to \mathbb{R}^m$ satisfies the spectral-to-nuclear Restricted Isometry Property with constant $\delta > 0$ (abbreviated S2NRIP($\delta$)) if for all tensors $\boldsymbol{\mathcal{Z}} \in S^{n \times n \times k}$ with tubal-rank $\leq r$,

$$\|(\mathcal{I} - \mathcal{A}^* \mathcal{A})(\boldsymbol{\mathcal{Z}})\| \leq \delta \|\boldsymbol{\mathcal{Z}}\|_*.$$

**Lemma G.1.** *Suppose that* $\mathcal{A} : S^{n \times n \times k} \to \mathbb{R}^m$ *satisfies RIP$(r + r', \delta)$ with $0 < \delta < 1$. Then, for any $\boldsymbol{\mathcal{Z}}, \boldsymbol{\mathcal{Y}} \in S^{n \times n \times k}$ with $\mathrm{rank}(\boldsymbol{\mathcal{Z}}) \leq r$ and $\mathrm{rank}(\boldsymbol{\mathcal{Y}}) \leq r'$, we have*

$$|\langle (\mathcal{I} - \mathcal{A}^* \mathcal{A})(\boldsymbol{\mathcal{Z}}), \boldsymbol{\mathcal{Y}} \rangle| \leq \delta \|\boldsymbol{\mathcal{Z}}\|_F \|\boldsymbol{\mathcal{Y}}\|_F.$$

*Proof.* Let $\boldsymbol{\mathcal{Y}}' = \frac{\|\boldsymbol{\mathcal{Z}}\|_F}{\|\boldsymbol{\mathcal{Y}}\|_F}\boldsymbol{\mathcal{Y}}$ so that $\|\boldsymbol{\mathcal{Y}}'\|_F = \|\boldsymbol{\mathcal{Z}}\|_F$. Note that $\boldsymbol{\mathcal{Z}} + \boldsymbol{\mathcal{Y}}' \in S^{n \times n \times k}$ and $\boldsymbol{\mathcal{Z}} - \boldsymbol{\mathcal{Y}}' \in S^{n \times n \times k}$ both have tubal rank $\leq r + r'$. Then, by using the identities $\|\boldsymbol{a} + \boldsymbol{b}\|^2 - \|\boldsymbol{a} - \boldsymbol{b}\|^2 = 4\langle \boldsymbol{a}, \boldsymbol{b}\rangle$ and $\|\boldsymbol{a} + \boldsymbol{b}\|^2 + \|\boldsymbol{a} - \boldsymbol{b}\|^2 = 2\|\boldsymbol{a}\|^2 + 2\|\boldsymbol{b}\|^2$ (which both hold over any inner product space) along with the fact that $\mathcal{A}$ satisfies RIP$(r + r', \delta)$, we have:

$$
\begin{aligned}
\langle (\mathcal{I} - \mathcal{A}^*\mathcal{A})(\boldsymbol{\mathcal{Z}}), \boldsymbol{\mathcal{Y}}'\rangle &= \langle \boldsymbol{\mathcal{Z}}, \boldsymbol{\mathcal{Y}}'\rangle - \langle \mathcal{A}^*\mathcal{A}(\boldsymbol{\mathcal{Z}}), \boldsymbol{\mathcal{Y}}'\rangle \\
&= \langle \boldsymbol{\mathcal{Z}}, \boldsymbol{\mathcal{Y}}'\rangle - \langle \mathcal{A}(\boldsymbol{\mathcal{Z}}), \mathcal{A}(\boldsymbol{\mathcal{Y}}')\rangle \\
&= \langle \boldsymbol{\mathcal{Z}}, \boldsymbol{\mathcal{Y}}'\rangle - \frac{1}{4}\|\mathcal{A}(\boldsymbol{\mathcal{Z}} + \boldsymbol{\mathcal{Y}}')\|_2^2 + \frac{1}{4}\|\mathcal{A}(\boldsymbol{\mathcal{Z}} - \boldsymbol{\mathcal{Y}}')\|_2^2 \\
&\leq \langle \boldsymbol{\mathcal{Z}}, \boldsymbol{\mathcal{Y}}'\rangle - \frac{1}{4}(1 - \delta)\|\boldsymbol{\mathcal{Z}} + \boldsymbol{\mathcal{Y}}'\|_F^2 + \frac{1}{4}(1 + \delta)\|\boldsymbol{\mathcal{Z}} - \boldsymbol{\mathcal{Y}}'\|_F^2 \\
&= \langle \boldsymbol{\mathcal{Z}}, \boldsymbol{\mathcal{Y}}'\rangle - \frac{1}{4}\left(\|\boldsymbol{\mathcal{Z}} + \boldsymbol{\mathcal{Y}}'\|_F^2 - \|\boldsymbol{\mathcal{Z}} - \boldsymbol{\mathcal{Y}}'\|_F^2\right) + \frac{1}{4}\delta\left(\|\boldsymbol{\mathcal{Z}} + \boldsymbol{\mathcal{Y}}'\|_F^2 + \|\boldsymbol{\mathcal{Z}} - \boldsymbol{\mathcal{Y}}'\|_F^2\right) \\
&= \frac{1}{2}\delta\left(\|\boldsymbol{\mathcal{Z}}\|_F^2 + \|\boldsymbol{\mathcal{Y}}'\|_F^2\right) \\
&= \delta\|\boldsymbol{\mathcal{Z}}\|_F\|\boldsymbol{\mathcal{Y}}'\|_F
\end{aligned}
$$

In a similar manner, $\langle (\mathcal{I} - \mathcal{A}^*\mathcal{A})(\boldsymbol{\mathcal{Z}}), \boldsymbol{\mathcal{Y}}'\rangle \geq -\delta\|\boldsymbol{\mathcal{Z}}\|_F\|\boldsymbol{\mathcal{Y}}'\|_F$. Hence, $\left|\langle (\mathcal{I} - \mathcal{A}^*\mathcal{A})(\boldsymbol{\mathcal{Z}}), \boldsymbol{\mathcal{Y}}'\rangle\right| \leq \delta\|\boldsymbol{\mathcal{Z}}\|_F\|\boldsymbol{\mathcal{Y}}'\|_F$. Then, since $\boldsymbol{\mathcal{Y}}$ is a scalar multiple of $\boldsymbol{\mathcal{Y}}'$, we have

$$
|\langle (\mathcal{I} - \mathcal{A}^*\mathcal{A})(\boldsymbol{\mathcal{Z}}), \boldsymbol{\mathcal{Y}}\rangle| = \frac{\|\boldsymbol{\mathcal{Y}}\|_F}{\|\boldsymbol{\mathcal{Y}}'\|_F}\left|\langle (\mathcal{I} - \mathcal{A}^*\mathcal{A})(\boldsymbol{\mathcal{Z}}), \boldsymbol{\mathcal{Y}}'\rangle\right| \leq \frac{\|\boldsymbol{\mathcal{Y}}\|_F}{\|\boldsymbol{\mathcal{Y}}'\|_F}\delta\|\boldsymbol{\mathcal{Z}}\|_F\|\boldsymbol{\mathcal{Y}}'\|_F = \delta\|\boldsymbol{\mathcal{Z}}\|_F\|\boldsymbol{\mathcal{Y}}\|_F.
$$

$\square$

**Lemma G.2.** *Suppose that $\mathcal{A} : S^{n \times n \times k} \to \mathbb{R}^m$ satisfies RIP$(r + 1, \delta_1)$, where $0 < \delta_1 < 1$. Then, $\mathcal{A}$ also satisfies S2SRIP$(r, \sqrt{kr}\delta_1)$.*

*Proof.* Suppose $\boldsymbol{\mathcal{Z}} \in S^{n \times n \times k}$ has tubal-rank $r$. Since $(\mathcal{I} - \mathcal{A}^*\mathcal{A})(\boldsymbol{\mathcal{Z}})$ is symmetric, its t-SVD is of the form

$$
(\mathcal{I} - \mathcal{A}^*\mathcal{A})(\boldsymbol{\mathcal{Z}}) = \boldsymbol{\mathcal{V}}_{(\mathcal{I} - \mathcal{A}^*\mathcal{A})(\boldsymbol{\mathcal{Z}})} * \boldsymbol{\Sigma}_{(\mathcal{I} - \mathcal{A}^*\mathcal{A})(\boldsymbol{\mathcal{Z}})} * \boldsymbol{\mathcal{V}}_{(\mathcal{I} - \mathcal{A}^*\mathcal{A})(\boldsymbol{\mathcal{Z}})}^\top.
$$

Now, define $\boldsymbol{\mathcal{V}} = \boldsymbol{\mathcal{V}}_{(\mathcal{I} - \mathcal{A}^*\mathcal{A})(\boldsymbol{\mathcal{Z}})}(:, 1, :) \in \mathbb{R}^{n \times 1 \times k}$ and let $\boldsymbol{s} \in \mathbb{R}^{1 \times 1 \times k}$ be defined by $\boldsymbol{s}(1, 1, \ell) = \frac{1}{\sqrt{k}}e^{\sqrt{-1}2\pi j\ell}$ where $j = \arg\max_{j'} |\widehat{\boldsymbol{\Sigma}}(1, 1, j')|$. With this definition, one can check that $\left|\langle (\mathcal{I} - \mathcal{A}^*\mathcal{A})(\boldsymbol{\mathcal{Z}}), \boldsymbol{\mathcal{V}} * \boldsymbol{s} * \boldsymbol{\mathcal{V}}^\top\rangle\right| = \|(\mathcal{I} - \mathcal{A}^*\mathcal{A})(\boldsymbol{\mathcal{Z}})\|$. Then, since $\mathcal{A}$ satisfies RIP$(r + 1, \delta_1)$ and rank$(\boldsymbol{\mathcal{Z}}) \leq r$ and rank$(\boldsymbol{\mathcal{V}} * \boldsymbol{s} * \boldsymbol{\mathcal{V}}^\top) = 1$, by Lemma G.1, we have

$$
\begin{aligned}
\|(\mathcal{I} - \mathcal{A}^*\mathcal{A})(\boldsymbol{\mathcal{Z}})\| &= \left|\langle (\mathcal{I} - \mathcal{A}^*\mathcal{A})(\boldsymbol{\mathcal{Z}}), \boldsymbol{\mathcal{V}} * \boldsymbol{s} * \boldsymbol{\mathcal{V}}^\top\rangle\right| \\
&\leq \delta_1\|\boldsymbol{\mathcal{V}} * \boldsymbol{s} * \boldsymbol{\mathcal{V}}^\top\|_F\|\boldsymbol{\mathcal{Z}}\|_F \\
&= \delta_1\|\boldsymbol{\mathcal{Z}}\|_F \\
&\leq \delta_1\sqrt{kr}\|\boldsymbol{\mathcal{Z}}\|.
\end{aligned}
$$

Since the bound $\|(\mathcal{I} - \mathcal{A}^*\mathcal{A})(\boldsymbol{\mathcal{Z}})\| \leq \delta_1\sqrt{kr}\|\boldsymbol{\mathcal{Z}}\|$ holds for any $\boldsymbol{\mathcal{Z}} \in S^{n \times n \times k}$ with tubal rank $\leq r$, we have that $\mathcal{A}$ satisfies S2SRIP$(r, \sqrt{kr}\delta_1)$. $\square$

**Lemma G.3.** *Suppose that $\mathcal{A} : S^{n \times n \times k} \to \mathbb{R}^m$ satisfies RIP$(2, \delta_2)$ where $0 < \delta_2 < 1$. Then, $\mathcal{A}$ also satisfies S2NRIP$(\sqrt{k}\delta_2)$.*

*Proof.* Since $\mathcal{A}$ satisfies RIP$(2, \delta_2)$, by Lemma G.2 for $r = 1$, $\mathcal{A}$ satisfies S2SRIP$(1, \sqrt{k}\delta_2)$. Now, suppose that $\boldsymbol{\mathcal{Z}} \in S^{n \times n \times k}$. Since $\boldsymbol{\mathcal{Z}}$ is symmetric, it has a t-SVD in the form

$$
\boldsymbol{\mathcal{Z}} = \sum_{i=1}^n \boldsymbol{\mathcal{V}}_i * \boldsymbol{s}_i * \boldsymbol{\mathcal{V}}_i^\top.
$$

Then, since each term $\boldsymbol{\mathcal{V}}_i * \boldsymbol{s}_i * \boldsymbol{\mathcal{V}}_i^\top$ is symmetric with tubal rank 1, we have

$$
\begin{aligned}
\|(\mathcal{I} - \mathcal{A}^*\mathcal{A})(\boldsymbol{\mathcal{Z}})\| &= \left\| (\mathcal{I} - \mathcal{A}^*\mathcal{A}) \left( \sum_{i=1}^n \boldsymbol{\mathcal{V}}_i * \boldsymbol{s}_i * \boldsymbol{\mathcal{V}}_i^\top \right) \right\| \\
&= \left\| \sum_{i=1}^n (\mathcal{I} - \mathcal{A}^*\mathcal{A}) \left( \boldsymbol{\mathcal{V}}_i * \boldsymbol{s}_i * \boldsymbol{\mathcal{V}}_i^\top \right) \right\| \\
&\leq \sum_{i=1}^n \left\| (\mathcal{I} - \mathcal{A}^*\mathcal{A}) \left( \boldsymbol{\mathcal{V}}_i * \boldsymbol{s}_i * \boldsymbol{\mathcal{V}}_i^\top \right) \right\| \\
&\leq \sum_{i=1}^n \sqrt{k} \delta_2 \left\| \boldsymbol{\mathcal{V}}_i * \boldsymbol{s}_i * \boldsymbol{\mathcal{V}}_i^\top \right\| \\
&= \sum_{i=1}^n \sqrt{k} \delta_2 \left\| \boldsymbol{s}_i \right\| \\
&\leq \sqrt{k} \delta_2 \|\boldsymbol{\mathcal{Z}}\|_*
\end{aligned}
$$

Since the bound $\|(\mathcal{I} - \mathcal{A}^*\mathcal{A})(\boldsymbol{\mathcal{Z}})\| \leq \sqrt{k}\delta_2\|\boldsymbol{\mathcal{Z}}\|_*$ holds for any $\boldsymbol{\mathcal{Z}} \in S^{n \times n \times k}$, we have that $\mathcal{A}$ satisfies S2NRIP($\sqrt{k}\delta_2$). $\qquad\square$

**Lemma G.4.** *Suppose $\mathcal{A} : S^{n \times n \times k} \to \mathbb{R}^m$ satisfies RIP($2r, \delta_3$), where $0 < \delta_3 < 1$, and $\boldsymbol{\mathcal{V}} \in \mathbb{R}^{n \times r \times k}$ satisfies $\boldsymbol{\mathcal{V}}^\top * \boldsymbol{\mathcal{V}} = \boldsymbol{\mathcal{I}}$. Then, for any $\boldsymbol{\mathcal{Z}} \in S^{n \times n \times k}$ with $\mathrm{rank}(\boldsymbol{\mathcal{Z}}) \leq r$, we have*

$$
\left\| \boldsymbol{\mathcal{V}}^\top * [(\mathcal{I} - \mathcal{A}^*\mathcal{A})(\boldsymbol{\mathcal{Z}})] \right\|_F \leq \delta_3 \|\boldsymbol{\mathcal{Z}}\|_F.
$$

*Proof.* Let $\boldsymbol{\mathcal{Z}} \in S^{n \times n \times k}$, and let $\boldsymbol{\mathcal{Y}} = \frac{\boldsymbol{\mathcal{V}}^\top * [(\mathcal{I} - \mathcal{A}^*\mathcal{A})(\boldsymbol{\mathcal{Z}})]}{\|\boldsymbol{\mathcal{V}}^\top * [(\mathcal{I} - \mathcal{A}^*\mathcal{A})(\boldsymbol{\mathcal{Z}})]\|_F} \in \mathbb{R}^{r \times n \times k}$. Trivially, $\|\boldsymbol{\mathcal{Y}}\|_F = 1$, and so, $\|\boldsymbol{\mathcal{V}} * \boldsymbol{\mathcal{Y}}\|_F^2 = \langle \boldsymbol{\mathcal{V}} * \boldsymbol{\mathcal{Y}}, \boldsymbol{\mathcal{V}} * \boldsymbol{\mathcal{Y}} \rangle = \left\langle \boldsymbol{\mathcal{Y}}, \boldsymbol{\mathcal{V}}^\top * \boldsymbol{\mathcal{V}} * \boldsymbol{\mathcal{Y}} \right\rangle = \langle \boldsymbol{\mathcal{Y}}, \boldsymbol{\mathcal{Y}} \rangle = \|\boldsymbol{\mathcal{Y}}\|_F^2 = 1$. Then, by using Lemma G.1, we have that

$$
\begin{aligned}
\left\| \boldsymbol{\mathcal{V}}^\top * [(\mathcal{I} - \mathcal{A}^*\mathcal{A})(\boldsymbol{\mathcal{Z}})] \right\|_F &= \left\langle \boldsymbol{\mathcal{V}}^\top * [(\mathcal{I} - \mathcal{A}^*\mathcal{A})(\boldsymbol{\mathcal{Z}})], \boldsymbol{\mathcal{Y}} \right\rangle \\
&= \langle [(\mathcal{I} - \mathcal{A}^*\mathcal{A})(\boldsymbol{\mathcal{Z}})], \boldsymbol{\mathcal{V}} * \boldsymbol{\mathcal{Y}} \rangle \\
&\leq \delta_3 \|\boldsymbol{\mathcal{Z}}\|_F \|\boldsymbol{\mathcal{V}} * \boldsymbol{\mathcal{Y}}\|_F \\
&= \delta_3 \|\boldsymbol{\mathcal{Z}}\|_F
\end{aligned}
$$

$\qquad\square$

# H. Properties of Aligned Matrix Subspaces

In this section, we collect some properties of matrices and their subspaces, useful for the proof of the results in the tensor Fourier domain.

**Lemma H.1.** *((Stöger & Soltanolkotabi, 2021)) For some orthogonal matrix $X \in \mathbb{C}^{n \times r}$ and some full-rank matrix $Y \in \mathbb{C}^{n \times R}$ consider $X^{\mathrm{H}}Y = V\Sigma W^{\mathrm{H}}$, and the following decomposition of $Y$*

$$
Y = YWW^{\mathrm{H}} + YW_\perp W_\perp^{\mathrm{H}} \tag{H.1}
$$

*with its SVD decomposition $Y = \sum_{i=1}^R \sigma_i u_i v_i^{\mathrm{H}}$ and the best rank-r approximation $Y_r = \sum_{i=1}^r \sigma_i u_i v_i^{\mathrm{H}}$. Then if the distance between the column subspace of $Y_r$ and the subspace spanned by the columns of $X$ is small enough, that is $\|X_\perp^{\mathrm{H}} V_{Y_r}\| \leq \frac{1}{8}$, then the decomposition* (H.1) *follows some low-rank approximation properties, namely*

$$
\|X_\perp^{\mathrm{H}} V_{YW}\| \leq 7\|X_\perp^{\mathrm{H}} V_{Y_r}\| \tag{H.2}
$$

$$
\|YW_\perp\| \leq 2\sigma_{r+1}(Y). \tag{H.3}
$$

**Lemma H.2.** *For a matrix $X \in \mathbb{C}^{n \times r}$, $r \leq n$, with its SVD-decomposition $X = V_X \Sigma_X W_X^{\mathrm{H}}$ and some a full-rank matrix $Y \in \mathbb{C}^{n \times R}$, consider $V_X^{\mathrm{H}} Y = V \Sigma W^{\mathrm{H}}$, and the following decomposition of $Y$*

$$Y = YWW^{\mathrm{H}} + YW_{\perp}W_{\perp}^{\mathrm{H}}. \tag{H.4}$$

*Let matrix $H \in \mathbb{C}^{r \times r}$ be defined as*

$$H = V_X^{\mathrm{H}}(\mathrm{Id} + \mu Z)YW$$

*with some $Z \in \mathbb{C}^{n \times n}$, parameter $\mu \leq \frac{1}{\sqrt{3}}\|V^{\mathrm{H}}Y\|^{-2}$ and $\|V_{\perp}^{\mathrm{H}}V_{YW}\| \leq c_2$ with sufficiently small constants $c_1, c_2 > 0$. Then $H$ can be represented as follows*

$$H = (\mathrm{Id} + \mu \Sigma_X^2 - \mu P_1 + \mu P_2 + \mu^2 P_3)V_X YW(\mathrm{Id} - \mu W^{\mathrm{H}} Y^{\mathrm{H}} V_X V_X^{\mathrm{H}} YW)$$

*with matrices $P_1, P_2, P_3 \in \mathbb{C}^{r \times r}$ such that*

$$P_1 := V_X^{\mathrm{H}} YY^{\mathrm{H}} V_{X^{\perp}} V_{X^{\perp}}^{\mathrm{H}} V_{YW}(VV_{YW})^{-1}(\mathrm{Id} - \mu V_X^{\mathrm{H}} YY^{\mathrm{H}} V_X)^{-1}$$
$$P_2 := V_X^{\mathrm{H}}(Z - XX^{\mathrm{H}} + YY^{\mathrm{H}})V_{YW}(V_X^{\mathrm{H}} V_{YW})^{-1}(\mathrm{Id} - \mu V_X^{\mathrm{H}} YWW^{\mathrm{H}} Y^{\mathrm{H}} V_X)^{-1}$$
$$P_3 := \Sigma_X^2 V_X^{\mathrm{H}} YW(\mathrm{Id} - \mu W^{\mathrm{H}} Y^{\mathrm{H}} V_X V_X^{\mathrm{H}} YW)^{-1} W^{\mathrm{H}} Y^{\mathrm{H}} V_X$$

*with*

$$\|P_1\| \leq 4\|YW\|^2 \|V_{X^{\perp}} V_{YW}\|^2$$
$$\|P_2\| \leq 4\|Z - XX^{\mathrm{H}} + YY^{\mathrm{H}}\|$$
$$\|P_3\| \leq 2\|X\|^2 \|YW\|^2.$$

*Moreover, it holds that*

$$\sigma_{min}(H) \geq \left(1 + \mu \sigma_{min}^2(X) - \mu\|P_1\| - \mu\|P_2\| - \mu^2\|P_3\|\right)\sigma_{min}(V_X^{\mathrm{H}} Y)\left(1 - \mu \sigma_{min}^2(V_X^{\mathrm{H}} Y)\right).$$

*Proof.* The proof of this Lemma follows from Lemma 9.1 in (Stöger & Soltanolkotabi, 2021) by using an independent matrix $Z \in \mathbb{C}^{n \times n}$ instead of the matrix $\mathcal{A}^* \mathcal{A}(XX^{\mathrm{H}} - YY^{\mathrm{H}})$, omitting the assumption $\|Y\| \leq 3\|X\|$ and updating respectively the transformation steps. $\square$

**Lemma H.3.** *For a matrix $X \in \mathbb{C}^{n \times r}$, $r \leq n$ with its SVD-decomposition $X = V_X \Sigma_X W_X^{\mathrm{H}}$ and some full-rank matrix $Y \in \mathbb{C}^{n \times R}$ and $Y_1 = (\mathrm{Id} + \mu Z)Y$ consider $V_X^{\mathrm{H}} Y = V \Sigma W^{\mathrm{H}}$, $V_X^{\mathrm{H}} Y_1 = V_1 \Sigma_1 W_1^{\mathrm{H}}$, and the following decomposition of $Y$ and $Y_1$*

$$Y = YWW^{\mathrm{H}} + YW_{\perp}W_{\perp}^{\mathrm{H}}.$$
$$Y_1 = Y_1 W_1 W_1^{\mathrm{H}} + Y_1 W_{1,\perp} W_{1,\perp}^{\mathrm{H}}.$$

*Assume that $V_X^{\mathrm{H}} Y_1 W$ is invertible, which also implies that $Y_1 W$ is has full-rank, and that $\|V_{X^{\perp}}^{\mathrm{H}} V_{Y_1 W}\| \leq \frac{1}{50}$ and $\mu \leq \min\left\{\frac{1}{\sqrt{3}}\|V_{X^{\perp}}^{\mathrm{H}} YW_{\perp}\|^{-2}, \frac{1}{9}\|X\|^{-2}\right\}$ and moreover, $\mu$ is small enough so that $0 \preceq \mathrm{Id} - \mu V_{X^{\perp}}^{\mathrm{H}} YWW^{\mathrm{H}} Y^{\mathrm{H}} V_{X^{\perp}} \preceq \mathrm{Id}$. Consider two matrices*

$$G_1 := -V_{X^{\perp}}^{\mathrm{H}} Y_1 W(V_X^{\mathrm{H}} Y_1 W)^{-1} V_X^{\mathrm{H}} Y_1 W_{\perp} W_{\perp}^{\mathrm{H}} W_{1,\perp}$$
$$G_2 := V_{X^{\perp}}^{\mathrm{H}} Y_1 W_{\perp} W_{\perp}^{\mathrm{H}} W_{1,\perp}.$$

*Then these matrices can be represented as*

$$G_1 = \mu V_{X^{\perp}}^{\mathrm{H}} V_{Y_1 W}(V_X^{\mathrm{H}} V_{Y_1 W})^{-1} M_1 V_{X^{\perp}}^{\mathrm{H}} YW_{\perp} W_{\perp}^{\mathrm{H}} W_{1,\perp}$$

*with $M_1 := V_X^{\mathrm{H}}(ZV_{X^{\perp}} - XX^{\mathrm{H}} V_{X^{\perp}})$ and*

$$G_2 = \left(\mathrm{Id} - \mu M_2 + \mu M_3)V_{X^{\perp}}^{\mathrm{H}} YW_{\perp}(\mathrm{Id} - \mu W_{\perp}^{\mathrm{H}} Y^{\mathrm{H}} YW_{\perp}) - \mu^2(M_2 - M_3)V_{X^{\perp}}^{\mathrm{H}} YW_{\perp} W_{\perp}^{\mathrm{H}} Y^{\mathrm{H}} YW_{\perp}\right) \cdot$$
$$\cdot W_{\perp}^{\mathrm{H}} W_{1,\perp}$$

with $M_2 = V_{X^\perp}^{\mathrm{H}} Y W W^{\mathrm{H}} Y^{\mathrm{H}} V_{X^\perp}$ and $M_3 := V_{X^\perp}^{\mathrm{H}} (Z - (XX^{\mathrm{H}} - YY^{\mathrm{H}})) V_{X^\perp}$. *Moreover, the norm of $G_1$ and $G_2$ can be bounded respectively as*

$$\|G_1\| \leq 2\mu(\|V_{X^\perp}^{\mathrm{H}} V_{YW}\|\|YW\|^2 + \|Z - (XX^{\mathrm{H}} - YY^{\mathrm{H}})\|)\|V_{X^\perp}^{\mathrm{H}} V_{Y_1 W}\|\|YW_\perp\|,$$

$$\|G_2\| \leq \|YW_\perp\|\Big(1 - \mu\|YW_\perp\|^2 + \mu\|Z - (XX^{\mathrm{H}} - YY^{\mathrm{H}})\|\Big)$$
$$+ \mu^2\Big(\|YW\|^2 + \|Z - (XX^{\mathrm{H}} - YY^{\mathrm{H}})\|\Big)\|YW_\perp\|^3.$$

*Proof.* The proof of this Lemma follows from Lemma 9.2 in (Stöger & Soltanolkotabi, 2021) by changing the matrix $\mathcal{A}^* \mathcal{A}(XX^{\mathrm{H}} - YY^{\mathrm{H}})$ to the independent matrix $Z \in \mathbb{C}^{n \times n}$ and taking into account the respective changes without having the condition $\|Y\| \leq 3\|X\|$. □

**Lemma H.4.** *For a matrix $X \in \mathbb{C}^{n \times r}$, $r \leq n$ with its SVD-decomposition $X = V_X \Sigma_X W_X^{\mathrm{H}}$ and some full-rank matrix $Y \in \mathbb{C}^{n \times R}$ and $Y_1 := (\mathrm{Id} + \mu Z)Y$ consider $V_X^{\mathrm{H}} Y = V\Sigma W^{\mathrm{H}}$, $V_X^{\mathrm{H}} Y_1 = V_1 \Sigma_1 W_1^{\mathrm{H}}$, and the following decomposition of $Y$ and $Y_1$*

$$Y = YWW^{\mathrm{H}} + YW_\perp W_\perp^{\mathrm{H}},$$
$$Y_1 = Y_1 W_1 W_1^{\mathrm{H}} + Y_1 W_{1,\perp} W_{1,\perp}^{\mathrm{H}}.$$

*Then it holds that*

$$\|W_\perp^{\mathrm{H}} W_1\| \leq \mu\left(1 + \mu\frac{\|Z\|\|YW\|}{\sigma_{min}(V_X^{\mathrm{H}}Y)}\right)\|YW\|\|YW_\perp\|\|V_{X^\perp}^{\mathrm{H}} V_{YW}\| + \mu\frac{\|Z - (XX^{\mathrm{H}} - YY^{\mathrm{H}})\|}{\sigma_{min}(V_X^{\mathrm{H}}Y)}\|YW_\perp\| \quad \text{(H.5)}$$

*Moreover, if for $P := YW_\perp W_\perp^{\mathrm{H}} W_1 (V_{YW}^{\mathrm{H}} YWW^{\mathrm{H}} W_1)^{-1} V_{YW}^{\mathrm{H}}$ the following applies*

$$\|\mu Z + P + \mu ZP\| \leq 1,$$

*then it holds that*

$$\|V_{X^\perp}^{\mathrm{H}} V_{Y_1 W_1}\| \leq \|V_{X^\perp}^{\mathrm{H}} V_{YW}\|\left(1 - \frac{\mu}{2}\sigma_{min}^2(X) + \mu\|YW_\perp\|\right) + \mu\|Z - (XX^{\mathrm{H}} - YY^{\mathrm{H}})\|$$
$$+ (1 + \mu\|Z\|)\frac{2\|W_\perp^{\mathrm{H}} W_1\|\|YW_\perp\|}{\sigma_{min}(W^{\mathrm{H}} W_1)\sigma_{min}(YW)} \quad \text{(H.6)}$$
$$+ 57\left(\mu\|Z\| + (1 + \mu\|Z\|)\frac{\|W_\perp^{\mathrm{H}} W_1\|\|YW_\perp\|}{\sigma_{min}(W^{\mathrm{H}} W_1)\sigma_{min}(YW)}\right)^2$$

*Proof.* The proof of inequality (H.5) follows from the first part of the proof of Lemma B.3 in (Stöger & Soltanolkotabi, 2021). For this one needs to change the matrix $\mathcal{A}^* \mathcal{A}(XX^{\mathrm{H}} - YY^{\mathrm{H}})$ in (Stöger & Soltanolkotabi, 2021) to an independent matrix $Z \in \mathbb{C}^{n \times n}$ and take into account the above-given decomposition of matrices $Y$ and $Y_1$ and lack of assumptions on $\mu$ and the norm of matrix $Z$. Inequality (H.6) follows from the proof of Lemma 9.3 in (Stöger & Soltanolkotabi, 2021). □

## I. Random Tubal Tensors

In this section, we derive bounds on the minimum and maximum singular values as well as the Frobenius norm of a random tubal tensor with i.i.d. Gaussian random entries. In our analysis of the spectral stage, we applied these lemmas to the small random initialization.

We start with the following proposition from Rudelson and Vershynin (2009), which bounds the smallest singular value of an $r \times R$ random real Gaussian matrix.

**Proposition I.1** ((Rudelson & Vershynin, 2009)). *Let $G \in \mathbb{R}^{r \times R}$ with $r \leq R$ have i.i.d. $\mathcal{N}(0,1)$ entries. Then, for every $\epsilon > 0$, we have*

$$\sigma_{min}(G) \geq \epsilon(\sqrt{R} - \sqrt{r-1})$$

*with probability at least $1 - (C\epsilon)^{R-r+1} - e^{-cR}$. The constants $C, c > 0$ are universal.*

Also, the following proposition from Tao and Vu (2010) bounds the smallest singular value of an $r \times r$ random complex Gaussian matrix.

**Proposition I.2** ((Tao & Vu, 2010)). *Let $\boldsymbol{G} \in \mathbb{R}^{r \times r}$ have i.i.d. $\mathcal{CN}(0, 1)$ entries. Then, for every $\epsilon > 0$, we have*

$$\sigma_{min}(\boldsymbol{G}) \geq \frac{\epsilon}{\sqrt{r}}$$

*with probability at least $1 - \epsilon^2$.*

Using these propositions, we can obtain a bound on the smallest singular value of an $r \times R$ random complex Gaussian matrix, provided that $r \leq R$.

**Lemma I.1.** *Let $\boldsymbol{G} \in \mathbb{C}^{r \times R}$ with $r \leq R$ have i.i.d. $\mathcal{CN}(0, 1)$ entries. Then, for every $\epsilon > 0$, we have*

$$\sigma_{min}(\boldsymbol{G}) \geq \begin{cases} \epsilon(\sqrt{R} - \sqrt{2r - 1}) & \text{if } R > 2r \\ \dfrac{\epsilon}{\sqrt{r}} & \text{if } r \leq R \leq 2r \end{cases}$$

*with probability at least*

$$\begin{cases} 1 - (C\epsilon)^{R - 2r + 1} - e^{-cR} & \text{if } R > 2r \\ 1 - \epsilon^2 & \text{if } r \leq R \leq 2r \end{cases}.$$

*The constants $C, c > 0$ are universal.*

*Proof.* First, suppose $R > 2r$. Let $\boldsymbol{G} = \boldsymbol{U\Sigma V}^H$ be the SVD of $\boldsymbol{G}$ where $\boldsymbol{U} \in \mathbb{C}^{r \times r}$ and $\boldsymbol{V} \in \mathbb{C}^{R \times R}$ are unitary and $\boldsymbol{\Sigma} \in \mathbb{R}^{r \times R}$. Then, the following real $2r \times 2R$ matrix has a real SVD of

$$\begin{bmatrix} \operatorname{Re}\{\boldsymbol{G}\} & -\operatorname{Im}\{\boldsymbol{G}\} \\ \operatorname{Im}\{\boldsymbol{G}\} & \operatorname{Re}\{\boldsymbol{G}\} \end{bmatrix} = \begin{bmatrix} \operatorname{Re}\{\boldsymbol{U}\} & -\operatorname{Im}\{\boldsymbol{U}\} \\ \operatorname{Im}\{\boldsymbol{U}\} & \operatorname{Re}\{\boldsymbol{U}\} \end{bmatrix} \begin{bmatrix} \boldsymbol{\Sigma} & 0 \\ 0 & \boldsymbol{\Sigma} \end{bmatrix} \begin{bmatrix} \operatorname{Re}\{\boldsymbol{V}\} & -\operatorname{Im}\{\boldsymbol{V}\} \\ \operatorname{Im}\{\boldsymbol{V}\} & \operatorname{Re}\{\boldsymbol{V}\} \end{bmatrix}^T.$$

By using the fact that for any $\boldsymbol{A} \in \mathbb{R}^{p \times q}$ with $p \leq q$, $\sigma_{\min}(\boldsymbol{A})^2 = \min\limits_{\substack{\boldsymbol{x} \in \mathbb{R}^p \\ \|\boldsymbol{x}\|_2 = 1}} \|\boldsymbol{A}^T \boldsymbol{x}\|_2^2$, we have

$$\begin{aligned}
\sigma_{\min}(\boldsymbol{G})^2 &= \sigma_{\min}\left( \begin{bmatrix} \operatorname{Re}\{\boldsymbol{G}\} & -\operatorname{Im}\{\boldsymbol{G}\} \\ \operatorname{Im}\{\boldsymbol{G}\} & \operatorname{Re}\{\boldsymbol{G}\} \end{bmatrix} \right)^2 \\
&= \min_{\substack{\boldsymbol{x} \in \mathbb{R}^{2r} \\ \|\boldsymbol{x}\|_2 = 1}} \left\| \begin{bmatrix} \operatorname{Re}\{\boldsymbol{G}\}^T & \operatorname{Im}\{\boldsymbol{G}\}^T \\ -\operatorname{Im}\{\boldsymbol{G}\}^T & \operatorname{Re}\{\boldsymbol{G}\}^T \end{bmatrix} \boldsymbol{x} \right\|_2^2 \\
&= \min_{\substack{\boldsymbol{x} \in \mathbb{R}^{2r} \\ \|\boldsymbol{x}\|_2 = 1}} \left[ \left\| \begin{bmatrix} \operatorname{Re}\{\boldsymbol{G}\}^T & \operatorname{Im}\{\boldsymbol{G}\}^T \end{bmatrix} \boldsymbol{x} \right\|_2^2 + \left\| \begin{bmatrix} -\operatorname{Im}\{\boldsymbol{G}\}^T & \operatorname{Re}\{\boldsymbol{G}\}^T \end{bmatrix} \boldsymbol{x} \right\|_2^2 \right] \\
&\geq \min_{\substack{\boldsymbol{x} \in \mathbb{R}^{2r} \\ \|\boldsymbol{x}\|_2 = 1}} \left\| \begin{bmatrix} \operatorname{Re}\{\boldsymbol{G}\}^T & \operatorname{Im}\{\boldsymbol{G}\}^T \end{bmatrix} \boldsymbol{x} \right\|_2^2 + \min_{\substack{\boldsymbol{x} \in \mathbb{R}^{2r} \\ \|\boldsymbol{x}\|_2 = 1}} \left\| \begin{bmatrix} \operatorname{Im}\{\boldsymbol{G}\}^T & \operatorname{Re}\{\boldsymbol{G}\}^T \end{bmatrix} \boldsymbol{x} \right\|_2^2 \\
&= \sigma_{\min}\left( \begin{bmatrix} \operatorname{Re}\{\boldsymbol{G}\} \\ \operatorname{Im}\{\boldsymbol{G}\} \end{bmatrix} \right)^2 + \sigma_{\min}\left( \begin{bmatrix} -\operatorname{Im}\{\boldsymbol{G}\} \\ \operatorname{Re}\{\boldsymbol{G}\} \end{bmatrix} \right)^2 \\
&= 2\sigma_{\min}\left( \begin{bmatrix} \operatorname{Re}\{\boldsymbol{G}\} \\ \operatorname{Im}\{\boldsymbol{G}\} \end{bmatrix} \right)^2,
\end{aligned}$$

where the last line follows since reordering the rows of a matrix or flipping the sign of some rows doesn't change the singular values.

Since $\boldsymbol{G} \in \mathbb{C}^{r \times R}$ has i.i.d. $\mathcal{CN}(0, 1)$ entries, $\sqrt{2} \begin{bmatrix} \operatorname{Re}\{\boldsymbol{G}\} \\ \operatorname{Im}\{\boldsymbol{G}\} \end{bmatrix} \in \mathbb{R}^{2r \times R}$ has i.i.d. $\mathcal{N}(0, 1)$ entries. Therefore, by Proposition I.1, we have that

$$\sigma_{\min}(\boldsymbol{G}) \geq \sigma_{\min}\left( \sqrt{2} \begin{bmatrix} \operatorname{Re}\{\boldsymbol{G}\} \\ \operatorname{Im}\{\boldsymbol{G}\} \end{bmatrix} \right) \geq \epsilon(\sqrt{R} - \sqrt{2r - 1})$$

with probability at least $1 - (C\epsilon)^{R-2r+1} - e^{-cR}$, as desired.

Next, suppose $r \leq R \leq 2r$. Let $\boldsymbol{G}_{r \times r}$ be an $r \times r$ submatrix of $\boldsymbol{G}$. Then,

$$\sigma_{\min}(\boldsymbol{G})^2 = \min_{\substack{\boldsymbol{x} \in \mathbb{C}^r \\ \|\boldsymbol{x}\|_2 = 1}} \|\boldsymbol{G}^H \boldsymbol{x}\|_2^2 \geq \min_{\substack{\boldsymbol{x} \in \mathbb{C}^r \\ \|\boldsymbol{x}\|_2 = 1}} \|\boldsymbol{G}_{r \times r}^H \boldsymbol{x}\|_2^2 = \sigma_{\min}(\boldsymbol{G}_{r \times r})^2.$$

Hence, by Proposition I.2, we have

$$\sigma_{\min}(\boldsymbol{G}) \geq \sigma_{\min}(\boldsymbol{G}_{r \times r}) \geq \frac{\epsilon}{\sqrt{r}}$$

with probability at least $1 - \epsilon^2$. $\qquad\square$

Using the above lemma, we can bound the smallest singular value of an $r \times R \times k$ tubal tensor.

**Lemma I.2.** *Let $\mathcal{G} \in \mathbb{R}^{r \times R \times k}$ with $r \leq R$ have i.i.d. $\mathcal{N}(0, \frac{1}{R})$ entries. Then, for every $\epsilon > 0$, we have*

$$\sigma_{min}(\boldsymbol{G}) \geq \begin{cases} \dfrac{\epsilon \sqrt{k}(\sqrt{R} - \sqrt{2r-1})}{\sqrt{R}} & \text{if } R > 2r \\[2mm] \dfrac{\epsilon\sqrt{k}}{\sqrt{rR}} & \text{if } r \leq R \leq 2r \end{cases}$$

*with probability at least*

$$\begin{cases} 1 - k(C\epsilon)^{R-2r+1} - ke^{-cR} & \text{if } R > 2r \\ 1 - k\epsilon^2 & \text{if } r \leq R \leq 2r \end{cases}.$$

*Proof.* Since the entries of $\mathcal{G}$ are i.i.d. $\mathcal{N}(0, \frac{1}{R})$, the entries of $\widetilde{\mathcal{G}}$ are i.i.d. $\mathcal{CN}(0, \frac{k}{R})$. Hence, each scaled slice $\sqrt{\frac{R}{k}}\widetilde{\mathcal{G}}^{(j)} \in \mathbb{C}^{r \times R}$ for $j = 1, \dots, k$ has i.i.d. $\mathcal{CN}(0, 1)$ entries. By Lemma I.1, each scaled slice satisfies

$$\sigma_{\min}\left(\sqrt{\tfrac{R}{k}}\widetilde{\mathcal{G}}^{(j)}\right) \geq \begin{cases} \epsilon(\sqrt{R} - \sqrt{2r-1}) & \text{if } R > 2r \\ \dfrac{\epsilon}{\sqrt{r}} & \text{if } r \leq R \leq 2r \end{cases}$$

with probability at least

$$\begin{cases} 1 - (C\epsilon)^{R-2r+1} - e^{-cR} & \text{if } R > 2r \\ 1 - \epsilon^2 & \text{if } r \leq R \leq 2r \end{cases}.$$

Then, by taking a union bound, we have that

$$\sigma_{\min}(\mathcal{G}) = \min_{1 \leq j \leq k} \sigma_{\min}\left(\widetilde{\mathcal{G}}^{(j)}\right) \geq \begin{cases} \dfrac{\epsilon\sqrt{k}(\sqrt{R} - \sqrt{2r-1})}{\sqrt{R}} & \text{if } R > 2r \\[2mm] \dfrac{\epsilon\sqrt{k}}{\sqrt{rR}} & \text{if } r \leq R \leq 2r \end{cases}$$

with probability at least

$$\begin{cases} 1 - k(C\epsilon)^{R-2r+1} - ke^{-cR} & \text{if } R > 2r \\ 1 - k\epsilon^2 & \text{if } r \leq R \leq 2r \end{cases}.$$

$\qquad\square$

The following proposition bounds the operator norm of an $r \times R$ random Gaussian matrix.

**Proposition I.3** ((Vershynin, 2018))**.** *Let $\boldsymbol{U} \in \mathbb{C}^{n \times R}$ have i.i.d. $\mathcal{CN}(0, 1)$ entries. Then, with probability at least $1 - O(e^{-c\max\{n, R\}})$, we have*

$$\|\boldsymbol{U}\| \lesssim \sqrt{\max\{n, R\}}.$$

Using the above proposition, we can bound the norm of an $n \times R \times k$ random Gaussian tubal tensor.

**Lemma I.3.** *Let $\mathcal{U} \in \mathbb{R}^{n \times R \times k}$ have i.i.d. $\mathcal{N}(0, \frac{1}{R})$ entries. Then, with probability at least $1 - O(ke^{-c\max\{n,R\}})$, we have*

$$\|\mathcal{U}\| \lesssim \sqrt{\frac{k \max\{n, R\}}{R}}.$$

*Proof.* Since the entries of $\mathcal{U}$ are i.i.d. $\mathcal{N}(0, \frac{1}{R})$, the entries of $\widetilde{\mathcal{U}}$ are i.i.d. $\mathcal{CN}(0, \frac{k}{R})$. Hence, each scaled slice $\sqrt{\frac{R}{k}} \widetilde{\mathcal{U}}^{(j)} \in \mathbb{C}^{r \times R}$ for $j = 1, \ldots, k$ has i.i.d $\mathcal{CN}(0, 1)$ entries. By Proposition I.3, each scaled slice satisfies

$$\left\| \sqrt{\tfrac{R}{k}} \widetilde{\mathcal{U}}^{(j)} \right\| \lesssim \sqrt{\max\{n, R\}}$$

with probability at least $1 - O(e^{-c\max\{n,R\}})$. Then, by taking a union bound, we have that

$$\|\mathcal{U}\| = \max_{1 \leq j \leq k} \left\| \widetilde{\mathcal{U}}^{(j)} \right\| \lesssim \sqrt{\frac{k \max\{n, R\}}{R}}$$

with probability at least $1 - O(ke^{-c\max\{n,R\}})$. $\qquad\square$

**Lemma I.4.** *Let $\mathcal{U} \in \mathbb{R}^{n \times R \times k}$ have i.i.d. $\mathcal{N}(0, \frac{1}{R})$ entries. Then, for any fixed $\mathcal{V}_1 \in \mathbb{R}^{n \times 1 \times k}$ with $\|\mathcal{V}_1\| = 1$, we have*

$$\|\mathcal{U}^\top * \mathcal{V}_1\|_F \asymp \sqrt{k}$$

*with probability at least $1 - O(ke^{-cR})$.*

*Proof.* Since the entries of $\mathcal{U}$ are i.i.d. $\mathcal{N}(0, \frac{1}{R})$, the entries of $\widetilde{\mathcal{U}}$ are i.i.d. $\mathcal{CN}(0, \frac{k}{R})$, and thus, the entries of $\widetilde{\mathcal{U}}^\top$ are also i.i.d. $\mathcal{CN}(0, \frac{k}{R})$. Then, for each slice $j = 1, \ldots, k$, each entry of the matrix-vector product $\widetilde{\mathcal{U}^\top}^{(j)} \widetilde{\mathcal{V}}_1^{(j)} \in \mathbb{C}^R$ is i.i.d. $\mathcal{CN}(0, \frac{k}{R} \|\widetilde{\mathcal{V}}_1^{(j)}\|_F^2)$. Hence, the quantity

$$\frac{2R}{k} \frac{\left\| \widetilde{\mathcal{U}^\top}^{(j)} \widetilde{\mathcal{V}}_1^{(j)} \right\|_F^2}{\left\| \widetilde{\mathcal{V}}_1^{(j)} \right\|_F^2}$$

has a $\chi^2(2R)$ distribution. It follows that

$$\left\| \widetilde{\mathcal{U}^\top}^{(j)} \widetilde{\mathcal{V}}_1^{(j)} \right\|_F^2 \asymp k \left\| \widetilde{\mathcal{V}}_1^{(j)} \right\|_F^2$$

holds with probability at least $1 - O(e^{-cR})$. By taking a union bound over all $j = 1, \ldots, k$, we get that

$$\left\| \mathcal{U}^\top * \mathcal{V}_1 \right\|_F^2 = \frac{1}{k} \left\| \widetilde{\mathcal{U}^\top} \odot \widetilde{\mathcal{V}}_1 \right\|_F^2 = \frac{1}{k} \sum_{j=1}^k \left\| \widetilde{\mathcal{U}^\top}^{(j)} \widetilde{\mathcal{V}}_1^{(j)} \right\|_F^2 \asymp \sum_{j=1}^k \left\| \widetilde{\mathcal{V}}_1^{(j)} \right\|_F^2 = \left\| \widetilde{\mathcal{V}}_1 \right\|_F^2 = k \|\mathcal{V}_1\|_F^2 = k,$$

i.e., $\|\mathcal{U}^\top * \mathcal{V}_1\|_F \asymp \sqrt{k}$ with probability at least $1 - O(ke^{-cR})$. $\qquad\square$

