# OpenReview forum: "Implicit Regularization for Tubal Tensor Factorizations via Gradient Descent"
_ICML.cc/2025/Conference — ICML 2025 oral_

### Official Review · Reviewer_m6Sc · 2025-03-14

**Overall Recommendation:** 3

**Summary:**

In this paper, the authors study tubal tensor factorization via gradient descent. Assuming the ground truth tensor is
of low tubal rank, they show that gradient descent from a small initialization will converge to the ground truth tensor,
and the error can be made arbitrarily small by scaling down the initialization scale.

More specifically, they show that when the learner tensor is still small, the dynamics can be approximated by the tensor
power method, whence the learner tensor will have a structure similar to the spectral initialization at the end of this
stage. Then, they show that GD will converge to the ground truth from this warm start.

**Claims And Evidence:**

This is a theory paper and the authors provide proofs (and numerical experiments) to support their claims.

**Essential References Not Discussed:**

No

**Experimental Designs Or Analyses:**

This is a theory paper and the simulations provided by the authors are serviceable.

**Methods And Evaluation Criteria:**

NA.

**Other Comments Or Suggestions:**

I don't think tubal rank and tubal-SVD are widely-used notions/techniques in this area. It might be good to have a section in the
appendix explaining them in more detail.

**Other Strengths And Weaknesses:**

The usage of tubal rank is not well-motivated. Matrix factorization can be viewed as training linear networks, and
the usual CP-rank based tensor decomposition can be related to two-layer networks with Gaussian inputs via the Hermite
analysis [1], and the low-rank condition is equivalent to saying the teacher network has a small number of neurons. On
the other hand, I do not see the connection between tubal tensor factorization and other tasks. Moreover, it seems that
the notion of tubal rank is restricted to order $3$ tensors and cannot be easily generalized to tensors of higher orders.


[1] R. Ge, J. D. Lee, and T. Ma, “Learning One-hidden-layer Neural Networks with Landscape Design,” in International Conference on Learning Representations, 2018. [Online]. Available: https://openreview.net/forum?id=BkwHObbRZ

**Questions For Authors:**

* Could you explain the role of the low tubal rank condition? See the "Theoretical Claims" section.
* Could you justify the usage of tubal rank (cf. the "Other Strengths And Weaknesses" section)?
* Could you obtain results similar to the one in [2]? Namely, is it true that GD will gradually increase the tubal rank
  of the leaner tensor until it can fit the target. It seems that you have a fine characterization on the training
  dynamics. It would be more interesting if you can obtain results like this based your analysis on the training dynamics.

I'm willing to raise my score if you could answer the first two questions.

[2] Z. Li, Y. Luo, and K. Lyu, “Towards Resolving the Implicit Bias of Gradient Descent for Matrix Factorization: Greedy Low-Rank Learning,” arXiv:2012.09839 [cs, stat], Apr. 2021, Accessed: Jan. 30, 2022. [Online]. Available: http://arxiv.org/abs/2012.09839

**Relation To Broader Scientific Literature:**

This paper seems to be a more technical challenging version of the corresponding results in matrix factorization
and (near-)orthogonal tensor decomposition. It is not clear to me what new insights we can draw from this paper.

**Theoretical Claims:**

Unfortunately, I am not familiar with tubal tensor factorization, so I couldn't check the correctness of the proof.
However, based on my understanding of matrix factorization and orthogonal tensor decomposition, the results and proof
strategy look reasonable.

I have one concern regarding the role of the low-tubal-rank condition. It is not clear to me where this low rank
condition is used and how it is related to the *implicit* regularization.

In [Li et al., 2018], the number of measurements needed is $\tilde{O}(d r^2)$ (instead of $d^3$), as
this many measurements is enough to ensure the (matrix) RIP condition they need. In this paper, however, the authors
assume the RIP condition for tubal-rank $r$ tensors. It is unclear whether this low tubal rank condition leads to
a better sample complexity. In addition, in [Li et al., 2018], the number of steps needed also depends on $r$ instead
of $d$. It is not clear whether Theorem 3.1 of this paper implies a similar result.

Meanwhile, in [Li et al., 2018], the shapes of the learner matrices are $d \times d$ instead of $d \times R$. In other
word, they do not explicitly encode a low-rank structure in the learner model, but the sample/iteration complexity
still depends on $r$ instead of purely on $d$. In this paper, the shape of the learner tensor is $n \times R \times k$,
instead of, say, $n \times d \times k$, and many parts of the results depend on $R$ instead of $r$. I don't think we
can call it *implicit* regularization if we need to have a good estimation on the rank a priori and encode it in the
learner model.

---

> ### Author Rebuttal · Authors · 2025-04-01
>
> Thank you very much for taking the time to evaluate our paper and provide valuable feedback.
>
> **RIP Condition:**
>
> [Zhang et. al. 2021] shows that sub-Gaussian measurements $\mathcal{A}:\mathbb{R}^{n\times n\times k}\to\mathbb{R}^m$ have the RIP for tubal-rank $r$ tensors w.h.p. if $m\ge O(\delta^{-2}rnk)$. Since we desire $\delta=O(\kappa^{-4}r^{-1/2})$, we need $m\ge O(\kappa^{-8}r^2nk)$ measurements. For $k=1$, the matrix case, this is very much comparable to the results in [Li et al. 2018]. In our revision, we will add a discussion on the number of measurements necessary for the RIP to hold.
>
> **Role of Low Tubal Rank Condition in Implicit Regularization:**
>
> While Thm 3.1 holds for $R$ as small as $3r$, it is more interesting for $R\ge n$ so that our learner model, like [Li et. al. 2018], also does not explicitly encode low tubal rank structure, and no bound on the ground truth tubal rank is needed. In this case, we can substitute $\min\\{n,R\\}=n$ so there is no dependence on $R$.
>
> When $R\ge n$ and the number of measurements satisfies $m<k(n^2+n)/2$ (but is still large enough for $\mathcal{A}$ to satisfy the RIP), there are infinitely many $\mathcal{U}\in\mathbb{R}^{n\times R\times k}$ with $\mathcal{A}(\mathcal{U}*\mathcal{U}^T) = \mathcal{A}(\mathcal{X} * \mathcal{X}^T)$, and thus, $\ell(\mathcal{U})=0$. However, the gradient descent iterates converge to a tensor of low tubal rank instead of one of the many high tubal rank tensors with zero loss. This suggests gradient descent is implicitly biased towards "simpler" tensors with low tubal rank.
>
> We hope this clarifies the implicit bias phenomena, as well as the role of the low tubal rank condition. Also, we will add a remark after the main theorem discussing how for $R\ge n$ the theorem describes an implicit regularization phenomenon.
>
> **Motivation for our work and the use of tubal rank:**
>
> The study of implicit regularization was originally motivated by trying to understand how using gradient descent to train neural networks results in good generalization. However, implicit regularization has become its own field of study. Specifically, it has been proposed to design linear network architectures in such a way that an implicit bias towards a desired structure arises - as an alternative to explicit regularization. This viewpoint has been taken for the sparse recovery problem [Vaskevicius et. al. 2019] and the low-rank matrix completion problem [Li. et. al. 2018], but to the best of our knowledge, all previous works generalizing this approach to tensors either provide only a partial analysis or only analyze the gradient flow problem, not gradient descent, so we see our work as the first to establish this kind of approach for a tensor recovery problem. (In particular, we are not sure, which work about implicit bias of gradient descent for near-orthogonal tensor factorization you are referring to, so it would be great if you could provide a reference so that we can specifically comment.) In this context, we find the study of low tubal rank recovery interesting as this model has been shown to be useful in video representation.
>
> We believe that this is an important line of research with many open questions. See our response to Reviewer phi4 for more details.
>
> We agree other tensor ranks are also important, but we feel that our work is an important starting point for further analysis. This paper is not the end of the story, but rather one part of a larger line of research into implicit regularization for tensor recovery problems.
>
> We agree that the additional motivational thread of using implicit regularization as an algorithm for structured recovery is currently not stressed enough in the paper. We will be sure to emphasize it in our revision.
>
> **Other:**
>
> Thank you for pointing out that tubal rank/t-SVD are not widely-used notions. We will expand the details we currently have in the Notations and Preliminaries section in the main body of our paper, as well as add a section on tubal rank/t-SVD in the Appendix.
>
> Thank you for suggesting the question whether the tubal rank of the learner tensor gradually increases in the course of the gradient descent iterations. Indeed, during the spectral stage where the gradient descent iterates are small, gradient descent behaves approximately like a power method. As such, when the gradient descent iterates are expressed in the basis of the slices of the t-SVD of the tubal-symmetric tensor $\mathcal{I}+\mu(\mathcal{A}^*\mathcal{A})(\mathcal{X}*\mathcal{X}^T)$, the coefficients grow approximately exponentially at rates determined by the corresponding eigentubes. As such, it seems like one should be able to obtain similar results as in [2], however, we could not check the details carefully in the short revision period. We aim for a detailed investigation in future work.
>
> See Response to Reviewer phi4 for the citations for [Vaskevicius et. al. 2019] and [Zhang et. al. 2019].

---

> > ### Comment · Reviewer_m6Sc · 2025-04-04
> >
> > Thank you for your response. I understand your setting and results much better now. I have raised the score to 3

---

> > > ### Author Response · Authors · 2025-04-09
> > >
> > > Thank you very much for reading our rebuttal and adjusting our score. We are glad you understand our settings and results better after these clarifications. We will be sure to include the above points in our revision to improve and clarify the motivation of our paper.

---

### Official Review · Reviewer_THrp · 2025-03-14

**Overall Recommendation:** 3

**Summary:**

This paper studies the factorization of third-order tubal tensors using gradient descent (GD) with small initialization in the overparameterized setting. Under the standard restricted isometry property (RIP) condition, the authors show that GD recovers the ground truth tensor in a finite number of steps, provided the initialization is sufficiently small. The analysis builds on the insight that, with small initialization, GD initially behaves like a spectral method, effectively providing a good spectral "initialization" for the second phase, during which GD converges linearly. The results extend prior analyses of GD with small initialization from matrix factorization to tubal tensor factorization.

**Claims And Evidence:**

See below.

**Essential References Not Discussed:**

See below.

**Experimental Designs Or Analyses:**

See below.

**Methods And Evaluation Criteria:**

See below.

**Other Comments Or Suggestions:**

Overall, I find the paper interesting and well-written. Below are my suggestions for improvement.

**Clarifying the Connection to Matrix Factorization:**
The proof outline in Section 4 resembles that of matrix factorization—GD first finds a spectral initialization and then converges linearly. Of course extending these results to tensor factorization is nontrivial. Some of the differences are briefly discussed at the end of Section 4. It would strengthen the paper if the authors could further elaborate on the technical innovations (for example, compared to Stoger & Soltanolkotabi 2021) for addressing these challenges mentioned at the end of Section 4.

**Parameter Dependence in Theorem 3.1:**
The parameter dependence in Theorem 3.1 feels suboptimal. Notably, there is an exponential dependence on the condition number $\kappa$. Is this dependence unavoidable, or could it be improved? Additionally, while the other parameters appear polynomially in the bound, the degree of these polynomials seems artificial. It would be helpful if the authors could clarify whether these dependencies are inherent to the problem or merely technical artifacts, and if the latter, provide an explanation for their origin.

Minor Typos:

Line 192: An extraneous “`” character.

Line 205 (right column): A repeated sentence.


Lastly, I did not verify the proofs in detail.

**Other Strengths And Weaknesses:**

See below.

**Questions For Authors:**

See above.

**Relation To Broader Scientific Literature:**

See below.

**Theoretical Claims:**

See below.

---

> ### Author Rebuttal · Authors · 2025-04-01
>
> Thank you very much for your positive evaluation of our paper. We are glad you found it to be interesting and well-written.
>
> **Clarifying the Connection to Matrix Factorization:**
>
> Thank you for bringing the connection to matrix factorization to the discussion.  Yes, while aiming to leverage matrix techniques as much as possible (no need to reinvent the wheel!), we encountered at lot of the technical complexities of these adaptations, which are briefly mentioned in Section 4.
>
> To describe these technicalities in more details, we will extend Section 4. For example, the tubal-tensor iterates $\mathcal{U}\_t \in \mathbb{R}^{n \times R \times k}$ produced by gradient descent exhibit significant interactions between their slices when transitioning from step $t$ to $t+1$. This necessitated nontrivial estimations, such as those presented in Lemmas E.4 and E.5, to control these interactions and provide the respective bounds. Another intriguing point is that, one need to choose the learning rate $\mu$ and the initialization scale $\alpha$ very carefully for the noise term $\mathcal{U}\_{t} * \mathcal{W}\_{\perp, t}$ to grow slowly enough in each of the tensor slices in order to not allow overtaking the signal term $\mathcal{U}\_{t} * \mathcal{W}\_{t}$ in the norm.
>
> **Parameter Dependence in Theorem 3.1:**
>
> We agree that the parameter dependence in Theorem 3.1 looks unfamiliar at first sight. However,  we find it not surprising that if the tensor is ill-conditioned, i.e., has a very small tubal singular value, gradient descent without regularization will have a hard time discovering that rank-one component unless the initialization is very small (in fact, our bound on the required number of iterations could also be expressed as a polynomial in the initialization parameter, making the exponential dependence implicit).
>
> This is in line with prior results in the matrix sensing case, which also require an initialization of size depending exponentially on the condition number, see [Stöger and Soltanolkotabi 2021]. In the matrix case, however, this dependence is not as apparent, as the term with the exponential dependence is absorbed in one of the absolute constants, see the range for parameter $\beta$ in Lemma 8.6. of [Stöger and Soltanolkotabi 2021] with its proof and the corresponding results for tensors in our case. It is worth to mention that of course it is not clear in either case if the dependence needs to be exponential. We consider it an interesting question for follow-up work to investigate the precise dependence on the condition number.
>
> Yes, some of the parameters appear polynomially in the bound, and as our numerical experiments, see Figure 4, show, indeed, the test error depends polynomially on the initialization parameter $\alpha$. It might be that the degree of the true polynomial dependence slightly differs from the one obtained theoretically; however, we strongly believe that the power $\frac{21}{16}$ approximates this dependence well.
>
> **Other:**
>
> Thank you for catching those typos. We will be sure to fix those in our revision.

---

### Official Review · Reviewer_phi4 · 2025-03-21

**Overall Recommendation:** 4

**Summary:**

To gain insight on initialization methods, the authors consider its effect on a gradient descent method for calculating a low rank tubal tensor tensor factorization, given linear measurements of the tensor. Tensor tubal rank of a $m\times n\times k$ tensor is the the rank of the SVD of the front $m\times n$ slices of a tensor, after applying the discrete Fourier Transform in the last dimension.

**Claims And Evidence:**

The theoretical claims are correctly proved and well supported by numerical evidence.

**Essential References Not Discussed:**

I believe the authors should cite [Ge and Ma, *On the Optimization Landscape of Tensor Decompositions*]. Although it is not very related, and the results and proof techniques are quite different, I think the authors should acknowledge one of the first works on trying to obtain guarantees regarding tensor decomposition.

**Experimental Designs Or Analyses:**

Does not apply.

**Methods And Evaluation Criteria:**

Does not apply

**Other Comments Or Suggestions:**

- The bibliography has several words written in wrong case (some words that are lowercase should be or start with an uppercase letter).

**Other Strengths And Weaknesses:**

- The paper is well-written, easy to follow, and the mathematical notation is consistent with what is used in the literature.
- I find that the numerical experiments are very elucidating, and help understand better the theoretical results that the authors show.
- The theoretical results are very interesting! This is one of the few papers I have seen that provides a full analysis of the gradient descent method, from initialization to (linear) convergence to the optima. The authors are also able to characterize an interesting transition from power-method behavior to near-the-maxima behavior, and observe that transition in numerical experiments. Finally, they relate their results to parameters of initialization. In that regard, this is a novel paper that deserves acceptance.
- Nevertheless, the analysis has some weaknesses. First, the authors consider tubal tensor rank factorization. This is factorization to can be calculated efficiently using matrix SVD, and in that regard it differs from other tensor decompositions that are NP-Hard to calculate. Therefore, even though the function that is minimized is not convex, its analysis is surely easier than other tensor decompositions. Furthermore, the condition of RIP on $\mathcal{A}$ seems to be too stringent: Since this is an RIP condition over low rank decomposition, and not over sparsity conditions, I wonder if this only includes maps that are very close to being orthogonal.

**Questions For Authors:**

- There is a generalization of tubal rank (also by Kilmer) called the M-rank, where M is the DFT matrix for tubal rank (but it can be any invertible matrix in general for M-rank). Do your results generalize well to that setting?

**Relation To Broader Scientific Literature:**

The paper analyses a gradient descent method, and in that sense it fits within other works in the literature that have considered this. In the topic of tensor decomposition and most relevant to this paper, some works include [Ge and Ma, *On the Optimization Landscape of Tensor Decompositions*], and [Wang et al, *Beyond lazy training for over-parameterized tensor decomposition*].

On the other hand, the authors focus on methods for tubal rank decompositions. Although this decomposition can be calculated efficiently through matrix SVD, some recent works have studied gradient descent type methods in this context [Liu et al, 2019, Liu et al, 2020, Liu et al, 2024].

**Theoretical Claims:**

The theoretical claims sound correct, but I was not able to check all details of the proof.

---

> ### Author Rebuttal · Authors · 2025-04-01
>
> Thank you very much for your positive evaluation of our paper and for recognizing our work as one of the few papers that fully analyzes gradient descent.
>
> **Motivation for our work and the use of tubal rank:**
>
> The study of implicit regularization was originally motivated by trying to understand how using gradient descent to train neural networks results in good generalization. However, implicit regularization has become its own field of study. Specifically, it has been proposed to design linear network architectures in such a way that an implicit bias towards a desired structure arises - as an alternative to explicit regularization. This viewpoint has been taken for the sparse recovery problem [Vaskevicius et. al. 2019] and the low-rank matrix completion problem [Li. et. al. 2018], but to the best of our knowledge, all previous works generalizing this approach to tensors either provide only a partial analysis or only analyze the gradient flow problem, not gradient descent, so we see our work as the first to establish this kind of approach for a tensor recovery problem. (In particular, we are not sure, which work about implicit bias of gradient descent for near-orthogonal tensor factorization you are referring to, so it would be great if you could provide a reference so that we can specifically comment.) In this context, we find the study of low tubal rank recovery interesting as this model has been shown to be useful in video representation.
>
> We believe that it is an important line of research to answer questions like:
> - What types of structured tensor recovery problems can be solved with implicit regularization?
> - How to design the learner architecture to bias gradient descent towards solutions with the appropriate structure?
> - Under what conditions and how quickly does convergence occur?
> - Can one characterize the trajectory of the algorithm?
>
> We do agree that other types of low rank tensor structures are also important, and we also agree that the corresponding analysis can be more challenging, but we feel that our work is an important starting point for further analysis. This paper is not the end of the story, but rather one part of a larger line of research into implicit regularization for tensor recovery problems.
>
> We agree that the additional motivational thread of using implicit regularization as an algorithm for structured recovery is currently not stressed enough in the paper. We will be sure to emphasize it in our revision.
>
> **RIP Condition:**
>
> We would like to note that the RIP condition is a standard condition in the literature, and is used in similar works [Li et. al. 2018] and [Stöger and Soltanolkotabi 2021]. Essentially, this is necessary to ensure that there is one low tubal rank tensor for which the loss function is zero (although there are still infinitely many high tubal rank tensors for which the loss is also zero). Furthermore, [Zhang et. al. 2019] shows that a random sub-Gaussian measurement operator $\mathcal{A} : \mathbb{R}^{n \times n \times k} \to \mathbb{R}^m$ will satisfy the RIP for tubal-rank $r$ tensors with restricted isometry constant $\delta$ with high probability if $m \ge O(\delta^{-2}rnk)$. Since we desire $\delta = O(\kappa^{-4}r^{-1/2})$, we need $m \ge O(\kappa^{-8}r^2nk)$ measurements. Hence, many random measurement operators (with far fewer measurements than the number of entries in the tensor) will satisfy the RIP.
>
> **Other:**
>
> Thank you for bringing the papers [Ge and Ma 2017] and [Wang et. al. 2020] to our attention. We agree that they are related to our paper, and we will be happy to cite and include them in our paper's discussion.
>
> Attempting to generalize our results to the M-rank setting (instead of tubal-rank) is an interesting idea for future work. In the case where the matrix $M$ is a scalar multiple of a unitary matrix, we believe that our results will generalize in a straightforward manner, although we have not checked all the details carefully. However, in the case where $M$ is not a scalar multiple of a unitary matrix, generalizing these results will be more difficult as our proof makes use of the fact that transforming a tensor into the Fourier domain (equivalently the $M$ domain) scales the Frobenius norm by a constant factor.
>
> Thank you for pointing out the issue with lowercase/uppercase in our bibliography. We will be sure to fix this in our revision.
>
> T. Vaskevicius, V. Kanade, and P. Rebeschini. "Implicit regularization for optimal sparse recovery." Advances in Neural Information Processing Systems. 2019.
>
> F. Zhang, et al. "Tensor restricted isometry property analysis for a large class of random measurement ensembles." Science China. Information Sciences 64.1 (2021): 119101. https://arxiv.org/pdf/1906.01198

---

> > ### Comment · Reviewer_phi4 · 2025-04-04
> >
> > I am satisfied with the rebuttal. I particularly liked the authors' statement in the rebuttal regarding RIP over a random measurement model, which I think they should include in the revision.

---

> > > ### Author Response · Authors · 2025-04-09
> > >
> > > Thank you very much for reading our rebuttal. We are pleased to hear that you are satisfied with it. We will certainly include our statement regarding the RIP over a random measurement model in the revised version. Thank you for this helpful suggestion.

---

### Decision · Program_Chairs · 2025-05-01

**Decision:**

Accept (oral)

**Comment:**

Reviewers were unanimous in voting for acceptance of this submission, commending its theoretical contributions and supporting experiments.  Several (relatively minor) suggestions were given, some of which were already addressed by the authors during the discussion period.  I of course support the reviewers' recommendation.